# Agnostic Active Learning Is Always Better Than Passive Learning

**Steve Hanneke**
Department of Computer Science
Purdue University
`steve.hanneke@gmail.com`

## Abstract

This work resolves a long-standing open question of central importance to the theory of active learning, closing a qualitative and quantitative gap in our understanding of active learning in the non-realizable case. We provide the first sharp characterization of the optimal first-order query complexity of agnostic active learning, and propose a new general active learning algorithm which achieves it. Remarkably, the optimal query complexity admits a leading term which is *always* strictly smaller than the sample complexity of passive supervised learning (by a factor proportional to the best-in-class error rate). This was not previously known to be possible. For comparison, in all previous general analyses, the leading term exhibits an additional factor, such as the disagreement coefficient or related complexity measures, and therefore only provides improvements over passive learning in restricted cases. The present work completely removes such factors from the leading term, implying that *every concept class benefits from active learning in the non-realizable case*. Whether such benefits are possible has been the driving question underlying the past two decades of research on the theory of agnostic active learning. This work finally settles this fundamental question.

## 1 Introduction

Active learning is a well-known powerful variant of supervised learning, in which the learning algorithm interactively participates in the process of labeling the training examples. In this setting, there is a pool (or stream) of unlabeled examples, and the learning algorithm selects individual examples and queries an oracle (typically a human labeler) to observe their labels. This happens sequentially, so that the learner has observed previously-queried labels before deciding which example to query next. The intended purpose of active learning is to reduce the overall number of labels necessary for learning to a given accuracy, called the *query complexity*. We are therefore particularly interested in using active learning in scenarios where its query complexity is significantly smaller than the number of randomly-sampled training examples which would be needed to achieve the same accuracy, called the *sample complexity* of *passive* supervised learning.

Active learning has not only been incredibly useful for many practical machine learning problems (e.g., Cohn et al., 1996; Tong and Koller, 2001; Zhu et al., 2003; Olsson, 2009; Settles, 2012; Ren et al., 2021; Mosqueira-Rey et al., 2023) but has also given rise to a rich and nuanced theoretical literature (e.g., Dasgupta, 2005, 2011; Balcan et al., 2009; Hanneke, 2007b, 2014; Zhang and Chaudhuri, 2014; Hanneke and Yang, 2015; see Appendix A for a detailed survey). Moreover, the insights and techniques discovered in this literature have had tremendous influence on other branches of the learning theory literature (e.g., Awasthi et al., 2014; Foster et al., 2021; Hanneke, 2009b, 2016a,b, 2024; Zhivotovskiy and Hanneke, 2018; Simon, 2015; Balcan and Long, 2013; El-Yaniv and Wiener, 2010; Balcan et al., 2022).

39th Conference on Neural Information Processing Systems (NeurIPS 2025).

Within the literature on the theory of active learning, a central topic which has garnered by-far the most interest is that of *agnostic* active learning: that is, the study of active learning algorithms capable of providing performance guarantees even in noisy or otherwise non-realizable learning problems, without assumptions on the form of the noise. This line of work was initiated by the groundbreaking $A^2$ algorithm (Agnostic Active) of Balcan, Beygelzimer, and Langford (2005, 2006, 2009) (with its general analysis later given by Hanneke, 2007b) and concurrently a lower bound analysis of Kääriäinen (2005, 2006) (later strengthened by Beygelzimer, Dasgupta, and Langford, 2009). These results were later refined and extended in numerous ways. However, throughout this two-decades long history, there has persisted a significant gap between the sharpest known upper and lower bounds on the optimal query complexity. Moreover, this gap represents an important *qualitative* distinction: while the lower bound is always smaller than the sample complexity of passive learning, the existing upper bounds only reflect such improvements under further restrictive conditions (e.g., bounded disagreement coefficient). Thus, the issue of resolving this gap is of central importance to this subject, since it has implications for answering the question:

> *Does every concept class admit benefits from using active learning instead of passive learning?*

The main contribution of the present work is to establish that this is indeed true, and in fact the known lower bound is always attainable. To achieve this, we introduce new algorithmic principles for active learning (the AVID principle), improving concentration of error estimates via adaptively isolating regions where the error estimates have high variance and allocating more queries to such regions.

## 2 Background and Summary of the Main Result

Let $\mathbb{C}$ be any concept class[1] (a set of functions $\mathcal{X} \to \{0, 1\}$ on a set $\mathcal{X}$ called the *instance space*) and denote by $\mathsf{d} = \mathrm{VC}(\mathbb{C})$ the VC dimension of $\mathbb{C}$ (Vapnik and Chervonenkis, 1971; see Definition 4). Let $P$ be an (unknown) joint distribution on $\mathcal{X} \times \{0, 1\}$, and define the *error rate* of any *classifier* $h : \mathcal{X} \to \{0, 1\}$ as $\mathrm{er}_P(h) := P((x, y) : h(x) \neq y)$. In the *active learning* problem, there is a sequence $(X_1, Y_1), \ldots, (X_m, Y_m)$ of i.i.d. samples from $P$, but the learner initially only observes the $X_i$ values (the *unlabeled* examples). It then has the capability to *query* any example $X_i$, which reveals the corresponding true label $Y_i$, in a *sequential* manner (i.e., it chooses its next query $X_{i'}$ after observing the label $Y_i$ of its previous query point $X_i$). After a number of such queries, the learner returns a classifier $\hat{h}$. The goal is to achieve a small *excess* error rate $\mathrm{er}_P(\hat{h}) \leq \inf_{h \in \mathbb{C}} \mathrm{er}_P(h) + \varepsilon$ while making as few queries as possible. We are particularly interested in quantifying the number of queries sufficient to achieve this, as a function of $\varepsilon$ and the value of the *best-in-class* error rate $\inf_{h \in \mathbb{C}} \mathrm{er}_P(h)$, known as a *first-order* query complexity bound.

Specifically, for any $\varepsilon, \delta, \beta \in (0, 1)$, the *optimal query complexity*, $\mathrm{QC}_a(\varepsilon, \delta; \beta, \mathbb{C})$, is defined as the minimal $Q \in \mathbb{N}$ for which there exists an active learner $\mathbb{A}_a$ such that (for a sufficiently large number $m$ of unlabeled examples), for every $P$ with $\inf_{h \in \mathbb{C}} \mathrm{er}_P(h) \leq \beta$, with probability at least $1 - \delta$, $\mathbb{A}_a$ makes at most $Q$ queries and returns $\hat{h}$ satisfying $\mathrm{er}_P(\hat{h}) \leq \inf_{h \in \mathbb{C}} \mathrm{er}_P(h) + \varepsilon$. The main quantity for comparison is the *sample complexity* of supervised *passive learning*. A passive learner $\mathbb{A}_p$ simply trains on $n$ *labeled* training examples $(X_1, Y_1), \ldots, (X_n, Y_n)$ sampled i.i.d. from $P$ to produce a classifier $\hat{h}$. For $\varepsilon, \delta, \beta \in (0, 1)$, the *optimal sample complexity* of passive learning, $\mathcal{M}_p(\varepsilon, \delta; \beta, \mathbb{C})$, is defined as the minimal size $n \in \mathbb{N}$ of such a training sample for which there exists a passive learner $\mathbb{A}_p$ such that, for every $P$ with $\inf_{h \in \mathbb{C}} \mathrm{er}_P(h) \leq \beta$, with probability at least $1 - \delta$, $\mathbb{A}_p$ returns $\hat{h}$ satisfying $\mathrm{er}_P(\hat{h}) \leq \inf_{h \in \mathbb{C}} \mathrm{er}_P(h) + \varepsilon$. We remark that, in both the active and passive cases, these definitions place no restrictions on the computational efficiency of the learning algorithms, but rather focus on the *data efficiency*, which is our primary interest in this work (see Section G).

Since both the query complexity and sample complexity concern the number of *labels* sufficient for learning, it is natural to compare $\mathrm{QC}_a(\varepsilon, \delta; \beta, \mathbb{C})$ with $\mathcal{M}_p(\varepsilon, \delta; \beta, \mathbb{C})$ to quantify the benefits of active learning. Thus, the primary interest in the theory of agnostic active learning is quantifying how much smaller $\mathrm{QC}_a(\varepsilon, \delta; \beta, \mathbb{C})$ is compared to $\mathcal{M}_p(\varepsilon, \delta; \beta, \mathbb{C})$. Since our interest is *agnostic* learning, it is most interesting to focus on the regime where $P$ is far-from-realizable: that is, where $\beta$ is much larger than $\varepsilon$. In this regime, it is well known from the works of Vapnik and Chervonenkis (1974); Devroye and Lugosi (1995); Hanneke, Larsen, and Zhivotovskiy (2024b) that the optimal sample

---

[1]To focus on non-trivial cases, we suppose $|\mathbb{C}| \geq 3$. We also suppose $\mathcal{X}$ is equipped with a $\sigma$-algebra specifying its measurable subsets, and we adopt the standard mild measure-theoretic restrictions on the $\sigma$-algebra and the class $\mathbb{C}$ from empirical process theory: namely, the image-admissible Suslin property (Dudley, 1999).

complexity of passive learning satisfies $\mathcal{M}_p(\varepsilon, \delta; \beta, \mathbb{C}) = \Theta\left(\frac{\beta}{\varepsilon^2}\left(\mathsf{d} + \log\left(\frac{1}{\delta}\right)\right)\right)$. In comparison, the known lower bound for active learning is $\mathrm{QC}_a(\varepsilon, \delta; \beta, \mathbb{C}) = \Omega\left(\frac{\beta^2}{\varepsilon^2}\left(\mathsf{d} + \log\left(\frac{1}{\delta}\right)\right)\right)$ (Kääriäinen, 2006; Beygelzimer, Dasgupta, and Langford, 2009). Thus, the strongest improvement we might hope from active learning is a factor of $\beta$ (representing the *best-in-class* error rate).

However, in the prior literature, this $\beta$-factor improvement has only been demonstrated in upper bounds under *restrictions* to $\mathbb{C}$ or $P$. Specifically, every general upper bound on $\mathrm{QC}_a(\varepsilon, \delta; \beta, \mathbb{C})$ in the literature has the form $c(\beta)\mathsf{d}\frac{\beta^2}{\varepsilon^2}$ (ignoring logs), where $c(\beta)$ is a $(\mathbb{C}, P)$-dependent quantity. For instance, one commonly appearing such quantity $c(\beta)$ is the *disagreement coefficient* $\theta(\beta)$ of Hanneke (2007b). We refer the reader to Appendix A for a detailed survey of such quantities $c(\beta)$ which have appeared in the literature. Importantly, for all such upper bounds in the literature, the corresponding factor $c(\beta)$ has the property that there exist simple classes $\mathbb{C}$ and distributions $P$ for which $c(\beta) \geq \frac{1}{\beta}$ (see Hanneke and Yang, 2015; Hanneke, 2016b, 2024): for instance, even for linear classifiers on $\mathbb{R}^2$ or singletons on $\mathbb{N}$. Note that when $c(\beta) \geq \frac{1}{\beta}$, a query complexity $c(\beta)\mathsf{d}\frac{\beta^2}{\varepsilon^2}$ becomes *no smaller* than $\mathsf{d}\frac{\beta}{\varepsilon^2}$, the sample complexity of *passive* learning. Moreover, one can show that avoiding such $\mathsf{d}\frac{\beta}{\varepsilon^2}$ query complexities would require new algorithmic techniques (see Appendix A).

Naturally, the question of refining such $c(\beta)$ factors has been a subject of much interest for many years. In particular, it has remained open whether such factors might even be *avoided entirely*, so that the $\beta$-factor improvement might *always* be achievable. In a series of talks, I conjectured that the lower bound $\Omega\left(\frac{\beta^2}{\varepsilon^2}\left(\mathsf{d} + \log\left(\frac{1}{\delta}\right)\right)\right)$ is *always* sharp (in the far-from-realizable regime), and even offered a sizable prize for a solution (along with lower-order terms) (e.g., Hanneke and Nowak, 2019).

**Contributions of this Work:**   In the present work, we completely resolve this question. We prove that (in the above regime) $\mathrm{QC}_a(\varepsilon, \delta; \beta, \mathbb{C}) = \Theta\left(\frac{\beta^2}{\varepsilon^2}\left(\mathsf{d} + \log\left(\frac{1}{\delta}\right)\right)\right)$. In other words, the $\beta$-factor improvement is *always* achievable, the known lower bound is *sharp*, and there is *no need* for restrictions on $(\mathbb{C}, P)$ or additional factors $c(\beta)$ as appear in all prior works.

Extending to the *full range* of $\beta$, the more-general form of the bound we prove also includes an additive *lower-order* term to account for the small-$\beta$ regime. In the simplest such bound (Theorem 1), this lower-order term is simply $\tilde{O}\left(\frac{\mathsf{d}}{\varepsilon}\right)$, so that the general form is $\mathrm{QC}_a(\varepsilon, \delta; \beta, \mathbb{C}) = \tilde{O}\left(\mathsf{d}\frac{\beta^2}{\varepsilon^2} + \frac{\mathsf{d}}{\varepsilon}\right)$ (Theorem 3 and Appendix F refine this lower-order term for some classes). For comparison, the general form of the passive sample complexity is $\mathcal{M}_p(\varepsilon, \delta; \beta, \mathbb{C}) = \tilde{\Theta}\left(\mathsf{d}\frac{\beta}{\varepsilon^2} + \frac{\mathsf{d}}{\varepsilon}\right)$. We note that, even in the *nearly-realizable* regime ($\beta = \tilde{O}(\varepsilon)$), it is known that $\frac{\mathsf{d}}{\varepsilon}$ is a lower bound on the query complexity for many classes $\mathbb{C}$ (Dasgupta, 2005; Hanneke, 2014; see Appendix D of Hanneke and Yang, 2015), so that this term is sometimes unavoidable, and hence the benefits of active learning can wane in the nearly-realizable regime. Likewise, the lower bound $\mathsf{d}\frac{\beta^2}{\varepsilon^2}$ implies the benefits can also diminish in the very-high-noise regime ($\beta = \Omega(1)$). In contrast, as discussed above, in the *far-from-realizable* regime ($\sqrt{\varepsilon} \leq \beta \ll 1$), the bound is of order $\mathsf{d}\frac{\beta^2}{\varepsilon^2}$, reflecting a $\beta$-factor improvement over the sample complexity of passive learning $\mathsf{d}\frac{\beta}{\varepsilon^2}$. Additionally, the intermediate regime of *moderate-size* $\beta$ (i.e., $\varepsilon \ll \beta < \sqrt{\varepsilon}$) *also* exhibits improvements over passive learning for all $\mathbb{C}$: in this regime, $\mathcal{M}_p(\varepsilon, \delta; \beta, \mathbb{C}) = \Omega\left(\mathsf{d}\frac{\beta}{\varepsilon^2}\right)$, whereas $\mathrm{QC}_a(\varepsilon, \delta; \beta, \mathbb{C}) = \tilde{O}\left(\frac{\mathsf{d}}{\varepsilon}\right) \ll \mathsf{d}\frac{\beta}{\varepsilon^2}$, reflecting an improvement by a factor $\tilde{O}(\frac{\varepsilon}{\beta})$. Altogether, this result reveals a previously-unknown and truly remarkable fact: $\mathrm{QC}_a(\varepsilon, \delta; \beta, \mathbb{C}) \ll \mathcal{M}_p(\varepsilon, \delta; \beta, \mathbb{C})$ in all regimes $\varepsilon \ll \beta \ll 1$, or in other words, in all regimes outside the nearly-realizable and very-high-noise cases, the following is true:

*For every concept class $\mathbb{C}$, the optimal query complexity of agnostic active learning is strictly smaller than the optimal sample complexity of agnostic passive learning.*

This result resolves an important long-standing open question central to the past two decades of research on the theory of agnostic active learning.

# 3 Main Results

Formally, the following theorem expresses the new upper bound, together with known lower bounds for comparison (Kääriäinen, 2006; Beygelzimer, Dasgupta, and Langford, 2009; Hanneke, 2014; Hanneke and Yang, 2015). A more-detailed version of the result appears in Theorem 5 (Appendix C).

**Theorem 1.** *For every concept class* $\mathbb{C}$*, letting* $\mathsf{d} = \mathrm{VC}(\mathbb{C})$*,* $\forall \varepsilon, \delta \in (0, 1/8)$*,* $\forall \beta \in [0, 1]$*,*

$$\mathrm{QC}_a(\varepsilon, \delta; \beta, \mathbb{C}) = O\left( \frac{\beta^2}{\varepsilon^2} \left( \mathsf{d} + \log\left(\frac{1}{\delta}\right) \right) \right) + \tilde{O}\left( \frac{\mathsf{d}}{\varepsilon} \right)$$

*and* $\mathrm{QC}_a(\varepsilon, \delta; \beta, \mathbb{C}) = \Omega\left( \frac{\beta^2}{\varepsilon^2} \left( \mathsf{d} + \log\left(\frac{1}{\delta}\right) \right) \right)$*. Moreover, for every* $\mathsf{d} \in \mathbb{N}$ *there exists* $\mathbb{C}$ *with* $\mathrm{VC}(\mathbb{C}) = \mathsf{d}$ *such that* $\mathrm{QC}_a(\varepsilon, \delta; \beta, \mathbb{C}) = \Omega\left( \frac{\beta^2}{\varepsilon^2} \left( \mathsf{d} + \log\left(\frac{1}{\delta}\right) \right) + \frac{\mathsf{d}}{\varepsilon} \right)$*.*

We provide a new general active learning algorithm $\mathbb{A}_{\mathrm{avid}}$ achieving this upper bound in Section 4. Importantly, the algorithm *does not need to know* $\beta$ (or anything else about $P$) to achieve this guarantee: i.e., it is completely *adaptive* to the value $\beta$. Moreover, the number of *unlabeled* examples the algorithm requires is only $\tilde{\Theta}\left( \mathsf{d}\frac{\beta}{\varepsilon^2} + \frac{\mathsf{d}}{\varepsilon} \right)$, of the same order as the sample complexity of passive learning; it can also adaptively determine how many unlabeled examples to use without knowing $\beta$.

**The AVID Principle:** The main innovation underlying the algorithm, which enables it to achieve this query complexity, represents a new principle for the design of active learning learning algorithms, which we call *Adaptive Variance Isolation by Disagreements* (AVID). The algorithm adaptively partitions the instance space $\mathcal{X}$ into *regions*, with the aim of *isolating* a region $\Delta \subseteq \mathcal{X}$ where it is most challenging to learn, due to exceptionally high *variance* in the error estimation problem in the $\Delta$ region (where $\Delta$ will be defined as a union of pairwise *disagreement* regions witnessing the high variance, carefully selected to ensure $P_X(\Delta) = O(\beta)$). It then allocates disproportionately *more* queries to this challenging region $\Delta$ compared to the (considerably-easier) remaining region $\mathcal{X} \setminus \Delta$. This idea has interesting connections to techniques explored in other branches of the literature (e.g., Hanneke, Larsen, and Zhivotovskiy, 2024b; Bousquet and Zhivotovskiy, 2021; Puchkin and Zhivotovskiy, 2022), discussed in Appendix A.

## 3.1 Refinement of the Lower-order Term for Some Classes

The AVID principle already suffices to achieve the query complexity bound in Theorem 1. Moreover, for *most* concept classes of interest, the query complexity bound in Theorem 1 is already *optimal*, matching a lower bound (up to log factors in the lower-order term): e.g., linear classifiers in $\mathbb{R}^k$, $k \geq 2$ (Dasgupta, 2005; Hanneke, 2014; Hanneke and Yang, 2015). However, while the lead term $\frac{\beta^2}{\varepsilon^2} \left( \mathsf{d} + \log\left(\frac{1}{\delta}\right) \right)$ is already optimal for every concept class $\mathbb{C}$, there do exist some special classes $\mathbb{C}$ for which a further refinement of the *lower-order* term $\frac{\mathsf{d}}{\varepsilon}$ is possible (e.g., threshold classifiers $\mathbb{1}_{[a,\infty)}$ on $\mathbb{R}$). As our second main result, we provide a refinement of the upper bound in Theorem 1 to capture such special classes, thereby establishing a query complexity bound which is nearly optimal for *every* concept class.

Since such refinements are only possible for some concept classes, the expression of this refinement necessarily depends on an additional complexity measure of the class $\mathbb{C}$. We prove that the *optimal* lower-order term in the query complexity is well-captured by a quantity known as the *star number* of $\mathbb{C}$, introduced by Hanneke and Yang (2015). In particular, Hanneke and Yang (2015) showed that the star number precisely characterizes the optimal query complexity in the *realizable case* ($\beta = 0$); since this is a limiting case of agnostic learning, it is natural that this quantity plays a crucial role in characterizing the optimal lower-order term. The formal definition is as follows.

**Definition 2.** *For any concept class* $\mathbb{C}$*, the* star number $\mathfrak{s} = \mathfrak{s}(\mathbb{C})$ *is the supremum* $n \in \mathbb{N}$ *for which* $\exists x_1, \ldots, x_n \in \mathcal{X}$ *and* $h_0, h_1, \ldots, h_n \in \mathbb{C}$ *such that* $\forall i, j \in \{1, \ldots, n\}$*,* $h_i(x_j) \neq h_0(x_j) \Leftrightarrow i = j$*.*

The star number essentially describes a scenario which is intuitively challenging for active learners in the realizable case, wherein there is a set of instances $x_j$ and a *default* labeling $h_0(x_j)$, but the *target* concept is some $h_i$ which differs from $h_0$ at just *one* instance $x_i$, unknown to the learner (which

must therefore query nearly all of these $x_j$ instances, searching for the special point $x_i$, in order to identify the target concept $h_i$). Hanneke and Yang (2015) provide numerous examples calculating $\mathfrak{s}$ for various concept classes. For instance, thresholds on $\mathbb{R}$ have $\mathfrak{s} = 2$ and decision stumps on $\mathbb{R}^k$ have $\mathfrak{s} = 2k$. However, it is worth noting that $\mathfrak{s}$ is typically large (or infinite) for most concept classes of interest in learning theory (e.g., $\mathfrak{s} = \infty$ for linear classifiers on $\mathbb{R}^k$, $k \geq 2$). This fact is important to the present work, since Hanneke and Yang (2015); Hanneke (2016b, 2024) have shown that the $c(\beta)$ factors (discussed in Section 2 above) appearing in all previous general upper bounds *all* become no smaller than $\mathfrak{s} \wedge \frac{1}{\beta}$ in the worst case over distributions (subject to the $\beta$ constraint). Thus, all general upper bounds $c(\beta)\mathsf{d}\frac{\beta^2}{\varepsilon^2}$ from the prior literature become no smaller than $\mathsf{d}\frac{\beta}{\varepsilon^2}$ in the worst case when $\mathfrak{s} = \infty$. In a sense, this means Theorem 1 is actually *most* interesting in the (typical) case of $\mathfrak{s} = \infty$, since *no* previously known upper bounds offer any improvements over passive learning in this case (without further restrictions to $P$), in stark contrast to Theorem 1 which has *no dependence* on $\mathfrak{s}$ and provides improvements over passive learning in the lead term for *every* concept class.

Nonetheless, the special structure of classes with $\mathfrak{s} < \infty$ turns out to provide some additional advantages for active learning, so that in order to state a general query complexity bound which is optimal for *every* concept class $\mathbb{C}$, we need to account for this structure, via a dependence on $\mathfrak{s}$ in the lower-order term. Specifically, by combining the AVID principle with existing principles for active learning (namely, *disagreement-based* queries), we can take further advantage of the power of active learning, thereby enabling a refinement of the lower-order term for classes with $\mathfrak{s} < \infty$. The following result presents a new general query complexity bound reflecting such refinements, together with a known lower bound for comparison (due to Kääriäinen, 2006; Beygelzimer, Dasgupta, and Langford, 2009; Hanneke and Yang, 2015). The implication is that this new upper bound is nearly optimal for *every* concept class $\mathbb{C}$ (including the lower-order term, up to a factor of $\mathsf{d}$, which we discuss below). A more-detailed version of the result appears in Theorem 5 of Appendix C (and distribution-dependent variants are presented in Appendix F, replacing $\mathfrak{s}$ with variants of the *disagreement coefficient*).

**Theorem 3.** *For every* $\mathbb{C}$, *letting* $\mathsf{d} = \mathrm{VC}(\mathbb{C})$ *and* $\mathfrak{s} = \mathfrak{s}(\mathbb{C})$, $\forall \varepsilon, \delta \in (0, 1/8)$, $\forall \beta \in [0, 1]$,

$$\mathrm{QC}_a(\varepsilon, \delta; \beta, \mathbb{C}) = O\left(\frac{\beta^2}{\varepsilon^2}\left(\mathsf{d} + \log\left(\frac{1}{\delta}\right)\right)\right) + \tilde{O}\left(\left(\mathfrak{s} \wedge \frac{1}{\varepsilon}\right)\mathsf{d}\right),$$

*and* $\mathrm{QC}_a(\varepsilon, \delta; \beta, \mathbb{C}) = \Omega\left(\frac{\beta^2}{\varepsilon^2}\left(\mathsf{d} + \log\left(\frac{1}{\delta}\right)\right)\right) + \mathfrak{s} \wedge \frac{1}{\varepsilon}.$

We may note that the upper bound in Theorem 1 is an immediate implication of Theorem 3 (we have stated Theorem 1 separately merely to emphasize that the improvements over passive learning are available without any special properties of $\mathbb{C}$ such as finite star number). Theorem 3 provides a refinement in the lower-order term compared to Theorem 1 when $\mathfrak{s} < \frac{1}{\varepsilon}$. In particular, for $\mathfrak{s} < \infty$, the asymptotic dependence on $\varepsilon$ in the lower-order term is $\log^2\left(\frac{1}{\varepsilon}\right)$. We leave open the question of whether this can be further refined to $\log\left(\frac{1}{\varepsilon}\right)$, which would match a known lower bound on this dependence for all infinite classes (Kulkarni, Mitter, and Tsitsiklis, 1993; Hanneke and Yang, 2015). The only significant difference between the upper and lower bounds in Theorem 3 is the factor of $\mathsf{d}$ in the lower-order term. I conjecture this term can be further refined to $\tilde{O}\left(\mathfrak{s} \wedge \frac{\mathsf{d}}{\varepsilon}\right)$, which is known to be sharp for some classes (Hanneke and Yang, 2015), and would fully answer a question posed by Hanneke and Nowak (2019). Beyond this, it is known that a gap between such lower-order terms in general upper and lower bounds is unavoidable if the only dependence on $\mathbb{C}$ is via $\mathsf{d}$ and $\mathfrak{s}$. Specifically, it follows from arguments in Appendix D of Hanneke and Yang (2015) that for some classes $\mathbb{C}$ this term should be $\tilde{\Theta}\left(\mathfrak{s} \wedge \frac{\mathsf{d}}{\varepsilon}\right)$ while for other classes $\mathbb{C}$ the term should be $\tilde{\Theta}\left(\mathfrak{s} \wedge \frac{1}{\varepsilon} + \mathsf{d}\right)$. Thus, obtaining matching (big-$\Theta$) upper and lower bounds would require introducing a new complexity measure reflecting the distinctions between these types of classes, which we leave as an open question.

## 4 Algorithm and Outline of the Analysis

We next present the algorithm achieving Theorems 1 and 3 and a sketch of its analysis (the complete formal proof is given in Appendix E). Before stating the algorithm, we first introduce a few additional definitions and convenient notational conventions.

**Error and disagreement regions:** For any function $h : \mathcal{X} \to \{0, 1\}$, define its *error region*

$$\mathrm{ER}(h) := \{(x, y) \in \mathcal{X} \times \{0, 1\} : h(x) \neq y\}.$$

In particular, note that $\mathrm{er}_P(h) = P(\mathrm{ER}(h))$. For any set $V \subseteq \mathbb{C}$ define the *region of disagreement*: $\mathrm{DIS}(V) := \{x \in \mathcal{X} : \exists f, g \in V, f(x) \neq g(x)\}$. For any two functions $f, g : \mathcal{X} \to \{0, 1\}$, abbreviate by $\{f \neq g\} := \{x \in \mathcal{X} : f(x) \neq g(x)\}$ their *pairwise disagreement region*.

**Overloaded set notation:** For convenience, we adopt a convention of treating sets $A \subseteq \mathcal{X}$ as notationally interchangeable with their *labeled extension* $A \times \{0, 1\} \subseteq \mathcal{X} \times \{0, 1\}$. For instance, for functions $f, g, h : \mathcal{X} \to \{0, 1\}$, we may write $\mathrm{ER}(h) \cap \{f \neq g\}$, which, by the above convention, is interpreted as $\mathrm{ER}(h) \cap (\{f \neq g\} \times \{0, 1\})$. We also overload notation for set intersections to allow for intersections of sets with *sequences*: that is, for any set $\mathcal{Z}$, sequence $S = \{z_1, \ldots, z_m\} \in \mathcal{Z}^m$, and set $A \subseteq \mathcal{Z}$, we define $S \cap A$ as the *subsequence* $\{z_i : i \leq m, z_i \in A\}$, and likewise $S \setminus A := S \cap (\mathcal{Z} \setminus A)$. We also apply these conventions in combination: i.e., for a sequence $S \in (\mathcal{X} \times \{0, 1\})^m$ and a set $\Delta \subseteq \mathcal{X}$, we define $S \cap \Delta := S \cap (\Delta \times \{0, 1\})$ and $S \setminus \Delta := S \cap ((\mathcal{X} \setminus \Delta) \times \{0, 1\})$.

**Empirical estimates:** We will make use of *empirical estimates* of quantities such as $\mathrm{er}_P(h)$ and $P_X(f \neq g)$. For any set $\mathcal{Z}$ and sequence $S = \{z_1, \ldots, z_m\} \in \mathcal{Z}^m$, for any set $A \subseteq \mathcal{Z}$, define the *empirical measure*: $\hat{P}_S(A) := \frac{1}{m}|S \cap A| = \frac{1}{m}\sum_{i=1}^m \mathbb{1}[z_i \in A]$. Again, we also apply these conventions in combination: i.e., for $S \in (\mathcal{X} \times \{0, 1\})^*$ and $\Delta \subseteq \mathcal{X}$, we define $\hat{P}_S(\Delta) := \hat{P}_S(\Delta \times \{0, 1\})$. For any sequence $S \in (\mathcal{X} \times \{0, 1\})^*$ and function $h : \mathcal{X} \to \{0, 1\}$, define its *empirical error rate* (or *empirical risk*): $\hat{\mathrm{er}}_S(h) := \hat{P}_S(\mathrm{ER}(h))$.

**Decision lists:** We will often express *decision-list* aggregations of functions $f, g : \mathcal{X} \to \{0, 1\}$. For instance, for any set $\Delta \subseteq \mathcal{X}$, we may write $h = f\mathbb{1}_{\mathcal{X} \setminus \Delta} + g\mathbb{1}_\Delta$ to express a function $h$ with $h(x) = f(x)$ for $x \notin \Delta$ and $h(x) = g(x)$ for $x \in \Delta$.

## 4.1 The AVID Agnostic Algorithm: Adaptive Variance Isolation by Disagreements

We are now ready to describe the algorithm achieving the upper bounds in Theorems 1 and 3 (for full formality, some additional technical minutiae for the definition are given in Section C). Fix any values $\varepsilon, \delta \in (0, 1)$ (the error and confidence parameters input to the learner). Fix any distribution $P$ (unknown to the learner) and let $(X_1, Y_1), \ldots, (X_m, Y_m)$ be independent $P$-distributed random variables (for any sufficiently large $m$, quantified explicitly in Theorem 5). The algorithm is stated in Figure 1, expressed in terms of certain quantities and data subsets defined as follows.[2] Let $C := \frac{11}{10}$, $N := \lceil \log_C\left(\frac{2}{\varepsilon}\right)\rceil$, and for each $k \in \mathbb{N}$ define $\varepsilon_k := C^{1-k}$ and $m_k := \Theta\left(\frac{1}{\varepsilon_k}\left(\mathrm{d}\log\left(\frac{1}{\varepsilon_k}\right) + \log\left(\frac{1}{\delta}\right)\right)\right)$ (see Section C for the precise constants). In Step 3, $C'$ denotes an appropriate universal constant (see Section C). As defined in Figure 1, the algorithm makes use of different portions of the data ($S_k^1, S_k^2, S_{k,i}^3, S_k^4$) for different purposes, and to complete the definition of the algorithm we next specify how these data subsets are defined in the algorithm. We first split the initial $2M_1 := 2\sum_{k=1}^{N+1} m_k$ examples $\{(X_1, Y_1), \ldots, (X_{2M_1}, Y_{2M_1})\}$ into consecutive disjoint contiguous segments $S_1^1, \ldots, S_{N+1}^1, S_1^4, \ldots, S_{N+1}^4$, with the segments $S_k^1$ and $S_k^4$ being of size $m_k$. The algorithm also allocates disjoint segments ($S_k^2, S_{k,i}^3$) of the remaining data $\{(X_i, Y_i) : 2M_1 < i \leq m\}$, but does so *adaptively* during its execution. Specifically, if and when the algorithm reaches Step 2 with a value $k$, or reaches Step 9 (in which case let $k = N + 1$), for the value $i_k$ and the set $\Delta_{i_k}$ as defined at that time in the algorithm, it constructs a data subset $S_k^2$, allocating to $S_k^2$ the next $m_k'$ consecutive examples which have not yet been allocated to any data subset $S_{k'}^1$, $S_{k'}^2, S_{k',i'}^3, S_{k'}^4$ (i.e., *fresh*, previously-unused, examples), where, letting $\hat{p}_k := 2\hat{P}_{S_k^4}(\Delta_{i_k})$, we define $m_k' := \Theta\left(\frac{\hat{p}_k}{\varepsilon_k^2}\left(\mathrm{d} + \log\left(\frac{3+N-k}{\delta}\right)\right)\right)$ (see Section C for the precise constants). Similarly, if and when the algorithm reaches Step 5 with some values of $(k, i)$, it constructs a data subset $S_{k,i}^3$, allocating to $S_{k,i}^3$ the next $m_k$ consecutive examples which have not yet been allocated.

---

[2] For simplicity, we have expressed the algorithm as representing a set of surviving concepts $V_k \subseteq \mathbb{C}$. However, it should be clear from the definition that running the algorithm does not require explicitly storing $V_k$. Rather, the various uses of this set can be implemented as constrained optimization problems (in Steps 4-6 and $\hat{h}_k$), where the constraints are merely the inequalities which would define the sets $V_{k'}$, $k' \leq k$, and Step 3 is then replaced by simply adding one more constraint to the constraint set.

---
**Algorithm** $\mathbb{A}_{\mathrm{avid}}$
Input: Error parameter $\varepsilon$, Confidence parameter $\delta$, Unlabeled data $X_1, \ldots, X_m$
Output: Classifier $\hat{h}$
0. Initialize $i = i_1 = 0$, $\Delta_0 = \emptyset$, $V_0 = \mathbb{C}$
1. For $k = 1, \ldots, N$
2.      Query all examples in $S_k^1 \cap D_{k-1} \setminus \Delta_{i_k}$ and $S_k^2 \cap \Delta_{i_k}$
3.      $V_k \leftarrow \left\{ h \in V_{k-1} : \hat{\mathrm{er}}_k^{1,2}(h) \leq \hat{\mathrm{er}}_k^{1,2}(\hat{h}_k) + \frac{\varepsilon_k}{C'} \right\}$
4.      If $V_k = \emptyset$ or $\hat{\mathrm{er}}_k^{1,2}(\hat{h}_k) < \min_{h \in V_k} \hat{\mathrm{er}}_k^{1,2}(h) - \frac{\varepsilon_k}{4C'}$, Then Return $\hat{h} := \hat{h}_k$
5.      While $\max_{f,g \in V_k} \hat{P}_{S_{k,i}^3}(\{f \neq g\} \setminus \Delta_i) > \varepsilon_{k+2}$
6.          $(f, g) \leftarrow \mathrm{argmax}_{(f',g') \in V_k^2} \hat{P}_{S_{k,i}^3}(\{f' \neq g'\} \setminus \Delta_i)$
7.          $\Delta_{i+1} \leftarrow \Delta_i \cup \{f \neq g\}$, and update $i \leftarrow i + 1$
8.      $i_{k+1} \leftarrow i$
9. Query all examples in $S_{N+1}^1 \cap D_N \setminus \Delta_{i_{N+1}}$ and $S_{N+1}^2 \cap \Delta_{i_{N+1}}$ and Return $\hat{h} := \hat{h}_{N+1}$
---

Figure 1: The AVID Agnostic algorithm. Notations $N$, $D_{k-1}$, $\varepsilon_k$, $\hat{h}_k$, $S_k^1$, $S_k^2$, $S_{k,i}^3$, $\hat{\mathrm{er}}_k^{1,2}$ defined in the text.

To complete the definition of the algorithm, we define $D_{k-1}$, $\hat{\mathrm{er}}_k^{1,2}$, and $\hat{h}_k$, appearing in the algorithm, as follows. For each value of $k$ encountered in the '*For*' loop, as well as for $k = N + 1$ in the case the algorithm reaches Step 9, define (where $V_{k-1}$ and $\Delta_{i_k}$ are as defined in the algorithm):

$$D_{k-1} := \mathrm{DIS}(V_{k-1}),$$

$$\forall h, \ \hat{\mathrm{er}}_k^{1,2}(h) := \hat{P}_{S_k^1}(\mathrm{ER}(h) \cap D_{k-1} \setminus \Delta_{i_k}) + \hat{P}_{S_k^2}(\mathrm{ER}(h) \cap \Delta_{i_k}), \tag{1}$$

$$V_{k-1}^{(4)} := \{f \mathbb{1}_{\{f=g\} \setminus \Delta_{i_k}} + h_1 \mathbb{1}_{\{f \neq g\} \setminus \Delta_{i_k}} + h_2 \mathbb{1}_{\Delta_{i_k}} : f, g \in V_{k-1}, h_1, h_2 \in \mathbb{C}\}, \tag{2}$$

$$\text{and} \quad \hat{h}_k := \underset{h \in V_{k-1}^{(4)}}{\mathrm{argmin}} \ \hat{\mathrm{er}}_k^{1,2}(h). \tag{3}$$

This completes the definition of the $\mathbb{A}_{\mathrm{avid}}$ algorithm.

We remark that the examples in $S_{k,i}^3$ and $S_k^4$ are *never queried* in the algorithm, and thus the algorithm (necessarily) only uses the unlabeled $X_i$ values in these data subsets (to estimate certain marginal $P_X$ probabilities), so in fact these can be regarded as *unlabeled* data subsets. Similarly, the algorithm only queries a *portion* of $S_k^1$ and $S_k^2$, and the remaining unqueried portions are in fact *never used* by the algorithm. For notational simplicity, we do not make these facts explicit in the notation.

**Description of the algorithm:** We briefly summarize the behavior of the algorithm (with explanations following in Section 4.2). As the algorithm iterates over rounds $k$ of the '*For*' loop, it maintains a partition of the space into a region $\Delta_{i_k}$ and its complement $\mathcal{X} \setminus \Delta_{i_k}$. In each round, the algorithm refines a set $V_k$ of surviving concepts from $\mathbb{C}$, aiming to prune out suboptimal concepts (Step 3). There are two crucial aspects of this, both in how the estimates of $\mathrm{er}_P(h)$ are defined, and in the choice of function $\hat{h}_k$ to which we compare. For the purpose of error estimation, in Step 2 it queries a number of random examples in $\mathcal{X} \setminus \Delta_{i_k}$ (or rather, the slightly smaller region $D_{k-1} \setminus \Delta_{i_k}$, since examples in $\mathcal{X} \setminus D_{k-1}$ are uninformative for estimating error *differences*) and a number of random examples in $\Delta_{i_k}$. It uses the examples from each of the two regions to estimate the error rate of each $h$ in that region, and combines these two estimates into an overall error estimate $\hat{\mathrm{er}}_k^{1,2}(h)$ as in (1). It then prunes suboptimal concepts from $V_{k-1}$, removing all $h \in V_{k-1}$ having estimated error $\hat{\mathrm{er}}_k^{1,2}(h) > \hat{\mathrm{er}}_k^{1,2}(\hat{h}_k) + \frac{\varepsilon_k}{C'}$. The reason $\hat{\mathrm{er}}_k^{1,2}(h)$ estimates error rates in the two regions separately is that, as it will turn out, we require a disproportionately larger number of samples to accurately estimate the error rates in the region $\Delta_{i_k}$ compared to the complement $\mathcal{X} \setminus \Delta_{i_k}$: for the latter, we use the samples in $S_k^1 \cap D_{k-1} \setminus \Delta_{i_k}$ (queried in Step 2), where $S_k^1$ has a modest size $m_k = \tilde{\Theta}\left(\frac{\mathsf{d}}{\varepsilon_k}\right)$, while for the former we use the samples in $S_k^2 \cap \Delta_{i_k}$ (also queried in Step 2), where $S_k^2$ has a potentially larger size $m_k'$ roughly $\tilde{\Theta}\left(\frac{P_X(\Delta_{i_k})\mathsf{d}}{\varepsilon_k^2}\right)$. The other crucial aspect in Step 3 is how we define the function $\hat{h}_k$ to which we compare. For this, rather than (the seemingly-natural idea of) simply comparing to the smallest $\hat{\mathrm{er}}_k^{1,2}(h)$ among $h \in V_{k-1}$, we instead compare to an even smaller value: the smallest $\hat{\mathrm{er}}_k^{1,2}(h)$ among a more-complex class $V_{k-1}^{(4)}$ defined in (2), comprised of *decision list*

functions which use one concept $h_2$ for predictions in $\Delta_{i_k}$, and use (equivalently) a majority vote of three concepts $f, g, h_1$ for predictions in $\mathcal{X} \setminus \Delta_{i_k}$. $\hat{h}_k$ is defined as a minimizer of $\hat{\mathrm{er}}_k^{1,2}$ in $V_{k-1}^{(4)}$, as in (3). This use of a more-complex comparator function is critical for certain parts of the proof (namely, keeping $P_X(\Delta_{i_k})$ small). However, given that $\hat{h}_k$ is chosen from a more-complex class, it becomes possible that $\hat{h}_k$ may be *substantially better* than all $h \in V_k$. In this event, the algorithm terminates early and returns $\hat{h}_k$ (Step 4). Otherwise, if it makes it past this early-stopping case, its next objective is to define the region $\Delta_{i_{k+1}}$ for use in the next iteration. This occurs in the '*While*' loop (Steps 5-7). On each round of this loop, it uses a fresh data set $S_{k,i}^3$ of size $m_k = \tilde{\Theta}\left(\frac{\mathrm{d}}{\varepsilon_k}\right)$ to check whether there exist $f, g \in V_k$ significantly *distant* from each other in the region $\mathcal{X} \setminus \Delta_i$ (Step 5). If so, it adds their pairwise disagreement region $\{f \neq g\}$ to the $\Delta_i$ region to define $\Delta_{i+1}$ and increments $i$ (Step 7). It repeats this until no such pair $f, g$ exists, at which time it defines $i_{k+1} = i$ (Step 8) and proceeds to the next iteration of the '*For*' loop. After $N = O(\log(1/\varepsilon))$ such iterations, it returns $\hat{h}_{N+1}$ (Step 9).

We note that the algorithm's returned classifier $\hat{h}$ might *not* be an element of $\mathbb{C}$ (known as an *improper* learner), but rather can be represented as a (shallow) *decision list* of concepts from $\mathbb{C}$. This aspect is quite important to certain parts of the proof, and we leave open the question of whether Theorems 1 and 3 are achievable by a proper learner (see Appendix G). We also remark that the $D_{k-1}$ set is *only* needed for establishing Theorem 3: the algorithm achieves the query complexity bound in Theorem 1 even if we replace $D_{k-1}$ with the full space $\mathcal{X}$ everywhere.

## 4.2 Principles and Outline of the Proof

Next we explain the high-level principles underlying the design of the algorithm, highlighting the *two key innovations* compared to previous approaches, which enable the improved query complexity guarantee (namely, separating out the $\Delta_{i_k}$ regions, and the definition of $\hat{h}_k$).

**Empirical localization:** The principles underlying the design of the algorithm begin with a familiar principle from statistical learning: *empirical localization* (Koltchinskii, 2006; Bartlett, Bousquet, and Mendelson, 2005). Specifically, the uniform Bernstein inequality (Lemma 7) implies that for an i.i.d. data set $S$, the sample complexity of uniform concentration of differences $|(\hat{\mathrm{er}}_S(f) - \hat{\mathrm{er}}_S(g)) - (\mathrm{er}_P(f) - \mathrm{er}_P(g))|$ becomes smaller when the *diameter* $\mathrm{diam}(\mathbb{C}) = \sup_{f,g \in \mathbb{C}} P_X(f \neq g)$ of the concept class is small, noting that $P_X(f \neq g)$ bounds the *variance* of loss differences $\mathbb{1}[f(x) \neq y] - \mathbb{1}[g(x) \neq y]$. Quantitatively, for any $0 < \varepsilon' < \mathrm{diam}(\mathbb{C})$, $\tilde{\Theta}\left(\mathrm{d}\frac{\mathrm{diam}(\mathbb{C})}{(\varepsilon')^2}\right)$ samples $S$ suffice to guarantee $|(\hat{\mathrm{er}}_S(f) - \hat{\mathrm{er}}_S(g)) - (\mathrm{er}_P(f) - \mathrm{er}_P(g))| \leq \varepsilon'$. This fact leads to a natural well-known algorithmic principle, wherein we can *prune* from $\mathbb{C}$ concepts $h$ having $\hat{\mathrm{er}}_S(h) - \min_{h' \in \mathbb{C}} \hat{\mathrm{er}}_S(h') > \varepsilon'$ (as the above inequality implies these verifiably have suboptimal error rates), leaving a subset $V_1'$ of surviving concepts, while preserving $h^\star \in V_1'$, where $h^\star := \mathrm{argmin}_{h \in \mathbb{C}} \mathrm{er}_P(h)$. Moreover, if these surviving concepts $V_1'$ have $\mathrm{diam}(V_1') < \mathrm{diam}(\mathbb{C})$, we get an *improved* concentration guarantee for $\hat{\mathrm{er}}_S(f) - \hat{\mathrm{er}}_S(g)$ among $f, g \in V_1'$ from the uniform Bernstein inequality, which enables us to prune *even more* concepts from $V_1'$, leaving a set $V_2'$ of surviving concepts, and so on for $V_3', V_4', \ldots$. Quantitatively, we can combine this with a schedule of resolutions $\varepsilon_k$, so that as long as $h^\star \in V_{k-1}'$ and $\mathrm{diam}(V_{k-1}') \leq \varepsilon_k$, an i.i.d. data set $S_k^1$ of size $m_k = \tilde{\Theta}\left(\frac{\mathrm{d}}{\varepsilon_k}\right) = \tilde{\Omega}\left(\mathrm{d}\frac{\mathrm{diam}(V_{k-1}')}{\varepsilon_k^2}\right)$ suffices to guarantee $\left|(\hat{\mathrm{er}}_{S_k^1}(f) - \hat{\mathrm{er}}_{S_k^1}(g)) - (\mathrm{er}_P(f) - \mathrm{er}_P(g))\right| \leq \frac{\varepsilon_k}{C'}$, enabling us to further reduce to a subset $V_k' = \left\{h \in V_{k-1}' : \hat{\mathrm{er}}_{S_k^1}(h) \leq \min_{h' \in V_{k-1}'} \hat{\mathrm{er}}_{S_k^1}(h') + \frac{\varepsilon_k}{C'}\right\}$ for which all $h \in V_k'$ have $\mathrm{er}_P(h) - \mathrm{er}_P(h^\star) \leq 2\frac{\varepsilon_k}{C'}$, while preserving $h^\star \in V_k'$. Iterating this $N = \Theta\left(\log_C\left(\frac{1}{\varepsilon}\right)\right)$ times (recalling $\varepsilon_k = C^{1-k}$) results in a subset $V_N'$ of concepts $h$ with $\mathrm{er}_P(h) - \mathrm{er}_P(h^\star) \leq \varepsilon$.

**Disagreement-based active learning:** An additional observation, underlying many active learning algorithms (*disagreement-based* methods), is that the above argument still holds while replacing $\hat{\mathrm{er}}_{S_k^1}(h)$ with $\hat{P}_{S_k^1}(\mathrm{ER}(h) \cap D_{k-1}')$, where $D_{k-1}' := \mathrm{DIS}(V_{k-1}')$. To see this, note that $\forall h, h' \in V_{k-1}'$, $\hat{P}_{S_k^1}(\mathrm{ER}(h) \cap D_{k-1}') - \hat{P}_{S_k^1}(\mathrm{ER}(h') \cap D_{k-1}') = \hat{\mathrm{er}}_{S_k^1}(h) - \hat{\mathrm{er}}_{S_k^1}(h')$. Thus, we may equivalently define $V_k' = \left\{h \in V_{k-1}' : \hat{P}_{S_k^1}(\mathrm{ER}(h) \cap D_{k-1}') \leq \min_{h' \in V_{k-1}'} \hat{P}_{S_k^1}(\mathrm{ER}(h') \cap D_{k-1}') + \frac{\varepsilon_k}{C'}\right\}$. Moreover, as long as $\mathrm{diam}(V_{k-1}') \leq \varepsilon_k$, we have $P_X(D_{k-1}') \leq \mathfrak{s}\varepsilon_k$ (Hanneke and Yang, 2015). Since the quantities in $V_k'$ only rely on the labels of examples in $D_{k-1}' \cap S_k^1$, constructing $V_k'$ only requires

a number of queries $O(\mathfrak{s}\varepsilon_k m_k) \wedge m_k$. Summing over $k$, these queries total to at most the claimed lower-order term in Theorem 3 (though note that even without this $D'_{k-1}$ refinement we still recover the lower-order term from Theorem 1). So far, this is all essentially standard reasoning commonly followed in the prior literature on active learning (e.g., Hanneke, 2009b, 2014; Koltchinskii, 2010).

**Handling non-shrinking diameter:** However, the above algorithmic principle breaks down if we reach a $k$ with $\operatorname{diam}(V'_{k-1}) \neq O(\varepsilon_k)$. This failure can easily occur in the agnostic setting, where it is possible for the set $V'_{k-1}$ above to contain multiple relatively-good functions $f, g$ which are nevertheless *far* from each other.[3] This is the motivation for the *first key innovation* in $\mathbb{A}_{\mathrm{avid}}$: namely, if we ever reach such a $k$, where the $V_k$ set does not naturally have $\operatorname{diam}(V_k) \leq \varepsilon_{k+1}$ (as tested in Step 5), the algorithm *removes* a portion of the space $\mathcal{X}$ to *artificially* reduce the diameter. Specifically, it identifies a pair $f, g \in V_k$ with $P_X(f \neq g) > \varepsilon_{k+1}$ (intuitively, an *obstruction* to having low diameter) and *separates out* their pairwise disagreement region $\{f \neq g\}$ from the region of focus of the algorithm (Steps 5-7).[4] Having set aside this region, the algorithm continues, focusing on the remaining set $\mathcal{X} \setminus \{f \neq g\}$. This step is repeated, and these set-aside regions $\{f \neq g\}$ are altogether captured in the set $\Delta_i$ (Step 7). Thus, we repeatedly find pairs $f, g \in V_k$ with $P_X(\{f \neq g\} \setminus \Delta_i) > \varepsilon_{k+1}$ (Steps 5-6) and add $\{f \neq g\}$ to $\Delta_i$ (Step 7) until the diameter of $V_k$ on $\mathcal{X} \setminus \Delta_i$ is reduced below $\varepsilon_{k+1}$. At that point, the algorithm proceeds to the next round ($k \leftarrow k + 1$). On the next round $k$, since we have (artificially) ensured the diameter of $V_{k-1}$ is at most $\varepsilon_k$ in the region $\mathcal{X} \setminus \Delta_{i_k}$, the uniform Bernstein argument implies $m_k$ examples $S^1_k$ suffice to guarantee every $f, g \in V_{k-1}$ have $\hat{P}_{S^1_k}(\mathrm{ER}(f) \cap D_{k-1} \setminus \Delta_{i_k}) - \hat{P}_{S^1_k}(\mathrm{ER}(g) \cap D_{k-1} \setminus \Delta_{i_k})$ within $\pm \frac{\varepsilon_k}{2C'}$ of $P(\mathrm{ER}(f) \setminus \Delta_{i_k}) - P(\mathrm{ER}(g) \setminus \Delta_{i_k}))$.

**Error in the $\Delta_{i_k}$ region:** There remains the issue of estimating error rates in the $\Delta_{i_k}$ *isolated* region. For this, the algorithm uses a data set $S^2_k$ of size $m'_k \approx \mathsf{d}\frac{P_X(\Delta_{i_k})}{\varepsilon_k^2}$, queries all examples in $S^2_k \cap \Delta_{i_k}$, and uses these to estimate the error rates $P(\mathrm{ER}(h) \cap \Delta_{i_k})$ in the $\Delta_{i_k}$ region. By a refinement of the uniform convergence bound of Talagrand (1994) accounting for an *envelope* set $\Delta_{i_k}$ (Lemma 8), this number $m'_k$ of examples suffices to ensure $\left|\hat{P}_{S^2_k}(\mathrm{ER}(h) \cap \Delta_{i_k}) - P(\mathrm{ER}(h) \cap \Delta_{i_k})\right| \leq \frac{\varepsilon_k}{4C'}$ for every $h \in \mathbb{C}$. Combining this with the above error-differences estimates in the $\mathcal{X} \setminus \Delta_{i_k}$ region, we can guarantee that the functions $f, g \in V_{k-1}$ have $\left|\left(\hat{\mathrm{er}}^{1,2}_k(f) - \hat{\mathrm{er}}^{1,2}_k(g)\right) - (\mathrm{er}_P(f) - \mathrm{er}_P(g))\right| \leq \frac{\varepsilon_k}{C'}$, recalling the definition of $\hat{\mathrm{er}}^{1,2}_k$ from (1). Altogether, we conclude that, as long as $h^\star \in V_{k-1}$, a set $V''_k := \left\{h \in V_{k-1} : \hat{\mathrm{er}}^{1,2}_k(h) \leq \min_{h' \in V_{k-1}} \hat{\mathrm{er}}^{1,2}_k(h') + \frac{\varepsilon_k}{C'}\right\}$ would contain only functions $h$ satisfying $\mathrm{er}_P(h) - \mathrm{er}_P(h^\star) \leq 2\frac{\varepsilon_k}{C'}$ while preserving $h^\star \in V''_k$. The actual definition of $V_k$ in Step 3 is only slightly different from this, for reasons we discuss next.

**Bounding the size of $\Delta_{i_k}$:** Since the number of queries in $S^2_k \cap \Delta_{i_k}$ is $\approx \mathsf{d}P_X(\Delta_{i_k})^2/\varepsilon_k^2$, if we hope to achieve a query complexity with lead term $\tilde{O}\left(\mathsf{d}\frac{\beta^2}{\varepsilon^2}\right)$ it is crucial to guarantee $P_X(\Delta_{i_k}) = O(\beta)$. This is the motivation for the *second key innovation* in $\mathbb{A}_{\mathrm{avid}}$: defining the update in $V_k$ by comparison to the function $\hat{h}_k$ in (3), rather than the best $h' \in V_{k-1}$. This turns out to be the most subtle part of the argument, requiring precise choices in the design of the algorithm. The essential argument is as follows. Suppose the algorithm reaches Step 6 for some $(k, i)$, so that it will add $\{f \neq g\}$ to the $\Delta_i$ region. We then want to argue that $P(\mathrm{ER}(h^\star) \cap \{f \neq g\} \setminus \Delta_i) = \Omega(P(\{f \neq g\} \setminus \Delta_i))$: that is, each time we add to $\Delta_i$, we *chop off* a portion of $\mathrm{ER}(h^\star)$ of size (under $P$) proportional to the increase in $P_X(\Delta_i)$. Clearly if we can show this is always the case, we will inductively maintain $P_X(\Delta_i) = O(\beta)$, resulting in the claimed leading term in the query complexity. Now, to show this indeed occurs, we first note that one of $f, g$ must err on at least *half* of $\{f \neq g\} \setminus \Delta_{i_k}$; w.l.o.g. suppose it is $f$: that is, $P(\mathrm{ER}(f) \cap \{f \neq g\} \setminus \Delta_{i_k}) \geq \frac{1}{2}P_X(\{f \neq g\} \setminus \Delta_{i_k})$. Now consider a function $f^\star = f\mathbb{1}_{\{f=g\} \setminus \Delta_{i_k}} + h^\star\mathbb{1}_{\{f \neq g\} \setminus \Delta_{i_k}} + f\mathbb{1}_{\Delta_{i_k}}$ which replaces $f$ by $h^\star$ in the region $\{f \neq g\} \setminus \Delta_{i_k}$. Note that, if $h^\star \in V_{k-1}$, then $f^\star \in V^{(4)}_{k-1}$ defined in (2). Since $\hat{h}_k$ has minimal $\hat{\mathrm{er}}^{1,2}_k$

---

[3]For instance, for $\mathbb{C}$ the class of *intervals* $\mathbb{1}_{[a,b]}$ on $\mathbb{R}$, with $P_X = \mathrm{Uniform}([0,1])$ and $P(Y = 1|X) = \mathbb{1}_{[0,1/4] \cup [3/4,1]}(X)$, the concepts $\mathbb{1}_{[0,1/4]}$ and $\mathbb{1}_{[3/4,1]}$ are both optimal among $\mathbb{C}$, yet distance $1/2$ apart.

[4]This reasoning is somewhat reminiscent of the motivation for the *splitting* approach to active learning (Dasgupta, 2005), differing only in how we resolve the obstruction: whereas splitting would resolve it with queries to eliminate one element from each obstructing pair, here we resolve it by subtracting the pairwise disagreement region from the region of focus $\mathcal{X} \setminus \Delta_i$ (see Appendix A.2.3). This idea is also related to a technique of Hanneke, Larsen, and Zhivotovskiy (2024b) for agnostic passive learning, discussed in Appendix A.3.

among $V_{k-1}^{(4)}$, and $f \in V_k$ implies $\hat{\mathrm{er}}_k^{1,2}(f) \leq \hat{\mathrm{er}}_k^{1,2}(\hat{h}_k) + \frac{\varepsilon_k}{C'}$, extending the above concentration of $\hat{\mathrm{er}}_k^{1,2}$ differences to functions in $V_{k-1}^{(4)}$ (with appropriate adjustment of constants in $m_k, m_k'$) implies $\mathrm{er}_P(f) - \mathrm{er}_P(f^\star) \leq 2\frac{\varepsilon_k}{C'}$. Thus, since $f^\star$ and $f$ only disagree on $\{f \neq g\} \setminus \Delta_{i_k}$, we have

$$\frac{1}{2} P_X(\{f \neq g\} \setminus \Delta_{i_k}) - P(\mathrm{ER}(h^\star) \cap \{f \neq g\} \setminus \Delta_{i_k})$$

$$\leq P(\mathrm{ER}(f) \cap \{f \neq g\} \setminus \Delta_{i_k}) - P(\mathrm{ER}(h^\star) \cap \{f \neq g\} \setminus \Delta_{i_k}) = \mathrm{er}_P(f) - \mathrm{er}_P(f^\star) \leq 2\frac{\varepsilon_k}{C'}.$$

In other words, $P(\mathrm{ER}(h^\star) \cap \{f \neq g\} \setminus \Delta_{i_k}) \geq \frac{1}{2} P_X(\{f \neq g\} \setminus \Delta_{i_k}) - 2\frac{\varepsilon_k}{C'}$. This is almost what we wanted, aside from having $\Delta_{i_k}$ in place of $\Delta_i$. We then argue $P(\mathrm{ER}(h^\star) \cap \{f \neq g\} \setminus \Delta_i) \geq P(\mathrm{ER}(h^\star) \cap \{f \neq g\} \setminus \Delta_{i_k}) - P_X(\{f \neq g\} \setminus \Delta_{i_k}) + P_X(\{f \neq g\} \setminus \Delta_i)$, which (by the above) is at least $P_X(\{f \neq g\} \setminus \Delta_i) - \frac{1}{2} P_X(\{f \neq g\} \setminus \Delta_{i_k}) - 2\frac{\varepsilon_k}{C'}$. Since both $f, g \in V_{k-1}$, we know $P_X(\{f \neq g\} \setminus \Delta_{i_k}) \leq \varepsilon_k$, so that this lower-bound is at least $P_X(\{f \neq g\} \setminus \Delta_i) - \frac{\varepsilon_k}{2} - 2\frac{\varepsilon_k}{C'}$. On the other hand, for appropriate constants in $m_k$, the condition in Step 5 allows us to upper-bound $\varepsilon_k$ in terms of $P_X(\{f \neq g\} \setminus \Delta_i)$: namely, $P_X(\{f \neq g\} \setminus \Delta_i) \geq \frac{\varepsilon_k}{c}$ for $c$ with $C^2 < c \leq \frac{3}{2} \wedge \frac{C'}{9}$. Thus, we have $P(\mathrm{ER}(h^\star) \cap \{f \neq g\} \setminus \Delta_i) \geq \left(1 - \frac{c}{2} - 2\frac{c}{C'}\right) P_X(\{f \neq g\} \setminus \Delta_i)$. Since each $\Delta_{i_k}$ is a union of such (disjoint) $\{f \neq g\} \setminus \Delta_i$ regions ($i < i_k$), $\beta \geq P(\mathrm{ER}(h^\star) \cap \Delta_{i_k}) \geq \left(1 - \frac{c}{2} - 2\frac{c}{C'}\right) P_X(\Delta_{i_k})$.

**The early stopping case:** The above argument for $P_X(\Delta_{i_k}) = O(\beta)$ hinges on having $h^\star \in V_{k-1}$. However, since $\hat{h}_k$ is a more-complex function than $h^\star$, there is a chance that $h^\star \notin V_k$ after Step 3. For this reason, we have added the early stopping case in Step 4. By using slightly tighter concentration inequalities than used to update $V_k$, this step effectively *tests* that $\hat{h}_k$ is not *so* much better than all concepts in $V_{k-1}$ that $h^\star$ might have been removed. Thus, if we make it past Step 4, we maintain $h^\star \in V_k$ so that the above argument applies on the next round. On the other hand, in the event that this test fails, we have effectively verified that $\hat{h}_k$ is at least *slightly better* than all concepts in $V_{k-1}$ (including $h^\star$), and we can safely return $\hat{h}_k$ in this case.

**Overall behavior:** The effective overall behavior of the algorithm is to *isolate* in the region $\Delta_{i_k}$ the most-challenging part of the error estimation problem, due to the high variance (diameter) of the error differences in that region. It then allocates a disproportionately larger number of queries $S_k^2 \cap \Delta_{i_k}$ to this region, toward estimating the error rates there. By comparing with the function $\hat{h}_k$ (which separately optimizes errors in pairwise difference regions $\{f \neq g\} \setminus \Delta_{i_k}$) in the definition of $V_k$, we can maintain that $\Delta_{i_k}$ never grows larger than $O(\beta)$, so that the number of queries in $S_k^2 \cap \Delta_{i_k}$ does not grow excessively large. The remaining region $\mathcal{X} \setminus \Delta_{i_k}$ enjoys the property that the set $V_{k-1}$ has diameter $\leq \varepsilon_k$, so that we can easily estimate error differences in this region by a uniform Bernstein inequality. Altogether, after at most $N = O\left(\log\left(\frac{1}{\varepsilon}\right)\right)$ rounds, this achieves the objective of $\varepsilon$ excess error rate, while using a number of queries as stated in the query complexity bound in Theorem 3. The formal proof is given in Appendix E.

## 5 Conclusions and Summary of the Appendices

This work resolves a long-standing open question of central importance to the theory of active learning, proving that *every* concept class benefits from active learning in the non-realizable case. Quantitatively, we establish a new sharp upper bound on the optimal query complexity, with leading term that is smaller than that of passive learning by a factor proportional to the best-in-class error rate.

The appendices include the formal proofs, along with additional contents. Appendix A presents a thorough summary of related work and background on the theory of active learning, as well as other works with techniques related to those used here. Appendix C presents remaining minutiae for the definition of $\mathbb{A}_{\mathrm{avid}}$, along with a more-detailed version of Theorem 3, including formal claims regarding the number of *unlabeled* examples. Appendix E presents the formal proof of Theorem 3. Appendix F presents distribution-dependent refinements of Theorem 3, which replace the star number $\mathfrak{s}$ with certain $P$-dependent complexity measures: variants of the disagreement coefficient. We further argue that the disagreement coefficient $\theta_P(\varepsilon)$, as originally defined by Hanneke (2007b), provably *cannot* be attained as a replacement for $\mathfrak{s}$ in the lower-order term (by any algorithm), while on the other hand $\mathbb{A}_{\mathrm{avid}}$ *does* achieve a lower-order term $\tilde{O}(\theta_P(\beta + \varepsilon)^2 \mathrm{d})$. We also present subregion-based refinements of the algorithm and analysis, based on techniques of Zhang and Chaudhuri (2014). Appendix G presents extensions (*multiclass* classification, *stream-based* active learning), along with several open questions and future directions.

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

# A  Survey of the Theory of Active Learning and Other Related Work

There is at this time quite an extensive literature on the theory of active learning. We refer the interested reader to the surveys of Hanneke (2014), Dasgupta (2011), and the 2019 ICML tutorial of Hanneke and Nowak (2019) for detailed discussions of classic works in this literature. In this section, we present a brief survey of the subject, with particular emphasis on the parts most-closely related to the present work.

## A.1  A Brief Historical Overview

The literature on active learning has a long history, dating back at least to the classical works on *experiment design* in statistics (Peirce, 1879; Fisher, 1935), wherein the analogous setting to active learning is referred to as *sequential design* (e.g., Wald, 1947; DeGroot, 1962; Fedorov, 1972; Atkinson and Donev, 1992; Efromovich, 2007; Zhang and Oles, 2000; Paninski, 2005; Lewi, Butera, and Paninski, 2009; Bull, 2013; Naghshvar and Javidi, 2013; Chaudhuri, Kakade, Netrapalli, and Sanghavi, 2015). Active learning has also been an important subject within the machine learning literature from the very beginning (e.g., Popplestone, 1969; Simon and Lea, 1974; Buchanan, 1976; Smith, Mitchell, Chestek, and Buchanan, 1977; Mitchell, 1979). Below we briefly mention some of the background of the subject in the *learning theory* literature, before giving detailed background of the literature on agnostic active learning.

**Membership Queries:**  In the learning theory literature, the idea of active learning also appeared as a natural variant of the problem of *Exact learning with queries*. Specifically, in this setting, supposing there is an unknown *target concept* $h^\star \in \mathbb{C}$, the objective of the learner is to *exactly identify* $h^\star$. To achieve this goal, the learner has access to an oracle (who knows $h^\star$), to which it may pose queries of a given type. The most relevant such queries (to the present work) are *membership queries*: namely, it may construct any $x \in \mathcal{X}$ and query for the value $h^\star(x)$ (in later works in machine learning, this is sometimes known as *query synthesis*). Early discussion of this framework and corresponding algorithmic principles appear in the seminal work of Mitchell (1979). General analyses of the number of queries necessary and sufficient to identify $h^\star$ (i.e., the *query complexity* of Exact learning) were developed in the works of Angluin (1987); Hegedüs (1995); Hellerstein, Pillaipakkamnatt, Raghavan, and Wilkins (1996); Nowak (2008, 2011); Hopkins, Kane, Lovett, and Mahajan (2020), and a related average-case analysis was developed by Dasgupta (2004).

Closer to the setting considered in the present work, the idea of learning with membership queries has also been extensively studied in the context of PAC learning in the realizable case. In that setting, the learner observes i.i.d. samples $(X_i, Y_i)$ with unknown distribution $P$, under the assumption that there exists an unknown target concept $h^\star \in \mathbb{C}$ with $\mathrm{er}_P(h^\star) = 0$. The learner is additionally permitted to make membership queries for this concept $h^\star$, with the goal of producing a predictor $\hat{h}$ having $\mathrm{er}_P(\hat{h}) \leq \varepsilon$ with high probability $1 - \delta$. While most of the literature on PAC learning with membership queries has focused on the benefits of such queries for the *computational* complexity of learning (e.g., Valiant, 1984; Baum, 1991; Jackson, 1997), the literature also contains several works on the number of samples and queries for learning in this setting (e.g., Eisenberg and Rivest, 1990; Eisenberg, 1992; Seung, Opper, and Sompolinsky, 1992; Turán, 1993; Kulkarni, Mitter, and Tsitsiklis, 1993; Diakonikolas, Kane, and Ma, 2024).

**Modern Active Learning with Label Queries:**  While the early literature on PAC learning with membership queries included several strong positive results (exhibiting advantages in both query complexity and computational complexity compared to learning from i.i.d. samples alone), when researchers implemented these algorithms and tried to use them for practical machine learning with a human labeler as the oracle, they found that the instances $x \in \mathcal{X}$ queried by the learner often turned out to be rather nonsensical, unnatural, or borderline cases between two labels (e.g., Baum and Lang, 1992). As such, human labelers were unable to provide useful answers to the queries, leading to poor performance of the learning algorithm. To address this issue, researchers turned to studying algorithms whose queries are restricted to only *natural* instances $x \in \mathcal{X}$, which in most works (with a few notable exceptions, e.g., Awasthi, Feldman, and Kanade, 2013) essentially means $x$ in the support of the marginal distribution $P_X$: i.e., the types of examples that might occur naturally in the population. To actualize this restriction, researchers proposed a simple variant of active learning (which has become the standard framework in the literature, and is now essentially synonymous

with the term *active learning*), in which there are i.i.d. samples $(X_1, Y_1), \ldots, (X_m, Y_m)$ from an unknown distribution $P$, but the learner initially only observes the *unlabeled* examples $X_i$, and can *query* to observe individual labels $Y_i$ (in a *sequential* fashion, so that it observes the label $Y_i$ of its previous query before selecting the next query $X_{i'}$) (Cohn, Atlas, and Ladner, 1994; Freund, Seung, Shamir, and Tishby, 1997; Tong and Koller, 2001). Such queries can typically be answered by human experts, being of the same type as used for data annotation in standard supervised machine learning. In this scenario, the unlabeled examples $X_i$ are typically assumed to be available in abundance, while obtaining the labels $Y_i$ is considered comparably more *expensive* (relying on the effort of a human expert), so that the primary objective is to minimize the number of *label queries* needed to achieve a given accuracy of a learned predictor $\hat{h}$. This is the setting studied in the present work.

The theoretical literature on this subject has origins in early works discussing algorithmic principles based on version spaces (Mitchell, 1979; Cohn, Atlas, and Ladner, 1994). Many of the early works providing actual bounds on the query complexity focused on showing improvements over passive learning for special scenarios, such as linear classifiers under distribution assumptions (e.g., Freund, Seung, Shamir, and Tishby, 1997; Dasgupta, Kalai, and Monteleoni, 2005; Har-Peled, Roth, and Zimak, 2007; Balcan, Beygelzimer, and Langford, 2006; Balcan, Broder, and Zhang, 2007; Balcan and Long, 2013; Gonen, Sabato, and Shalev-Shwartz, 2013; Wang and Singh, 2016; Cavallanti, Cesa-Bianchi, and Gentile, 2011; Dekel, Gentile, and Sridharan, 2012). This was followed by a boom of general-case analyses, providing general theories analyzing the query complexity for any concept class (e.g., Dasgupta, 2005; Hanneke, 2007a,b, 2009b,a, 2011, 2012, 2014; Dasgupta, Hsu, and Monteleoni, 2007; Balcan, Hanneke, and Vaughan, 2010; Beygelzimer, Dasgupta, and Langford, 2009; Koltchinskii, 2010; Zhang and Chaudhuri, 2014; El-Yaniv and Wiener, 2012; Wiener, Hanneke, and El-Yaniv, 2015; Hanneke and Yang, 2015; Hanneke, Karbasi, Moran, and Velegkas, 2024a), some of which are discussed in more detail below.

**Agnostic PAC Learning:**   The PAC learning framework has also been extended to allow *non-realizable* distributions $P$, that is, removing the restriction that $\inf_{h \in \mathbb{C}} \mathrm{er}_P(h) = 0$. This framework was abstractly formulated in the classic work of Vapnik and Chervonenkis (1974), with interest in the computer science literature initiated by the works of Haussler (1992); Kearns, Schapire, and Sellie (1994). Since such non-realizable distributions $P$ might not allow for predictors $\hat{h}$ with $\mathrm{er}_P(\hat{h}) \le \varepsilon$, the objective in this framework changes to merely achieving a *relatively* low error rate compared to the best error rate achievable by concepts in the class $\mathbb{C}$. More precisely, we aim to produce a predictor $\hat{h}$ which, with probability at least $1 - \delta$, satisfies $\mathrm{er}_P(\hat{h}) \le \inf_{h \in \mathbb{C}} \mathrm{er}_P(h) + \varepsilon$. The goal is to achieve this objective, for *every* distribution $P$, without *any* restrictions. This framework is termed *agnostic PAC learning*, to emphasize that we do not assume any special knowledge of $P$ when designing such a learning algorithm (Kearns, Schapire, and Sellie, 1994).

While an agnostic learning algorithm should achieve this objective for *every* distribution $P$, this need not restrict the *analysis* of such learners to consider only the worst case over all $P$. In particular, in the present work, we are primarily interested in analyzing the number of queries necessary and sufficient for agnostic active learning, as a function of the *best-in-class* error rate $\inf_{h \in \mathbb{C}} \mathrm{er}_P(h)$, known as a *first-order* query complexity bound. Precisely, as introduced in Section 2, for every $\varepsilon, \delta, \beta \in (0, 1)$, we denote by $\mathrm{QC}_a(\varepsilon, \delta; \beta, \mathbb{C})$ the minimax optimal first-order query complexity: that is, the minimal $Q \in \mathbb{N}$ for which there exists an active learning algorithm $\mathbb{A}_a$ such that (for a sufficiently large number $m$ of unlabeled examples), for every distribution $P$ with $\inf_{h \in \mathbb{C}} \mathrm{er}_P(h) \le \beta$, with probability at least $1 - \delta$, $\mathbb{A}_a$ makes at most $Q$ queries and returns a predictor $\hat{h}$ satisfying $\mathrm{er}_P(\hat{h}) \le \inf_{h \in \mathbb{C}} \mathrm{er}_P(h) + \varepsilon$. While, in principle, this definition of $\mathrm{QC}_a(\varepsilon, \delta; \beta, \mathbb{C})$ admits learners which explicitly depend on knowledge of $\beta$, we will find that the optimal query complexity is achievable (up to constant factors and lower-order terms) simultaneously for all $\beta$ by an active learner which does *not* require knowledge of $\beta$. Such a learner is said to be *adaptive* to $\beta$. In particular, such a learner is therefore an agnostic PAC learner, and the $\beta$ restriction only enters in its analysis.

**The Passive Learning Baseline:**   Since the predictor $\hat{h}$ produced by an active learning algorithm is based on its queried subset of a given set of i.i.d. examples $(X_i, Y_i)$, the natural quantity for comparison is the number of *i.i.d. labeled* examples necessary to obtain the same accuracy: i.e., the sample complexity of standard supervised learning, which in this literature is termed *passive*

*learning.*[5] Recall from Section 2 that we denote by $\mathcal{M}_p(\varepsilon, \delta; \beta, \mathbb{C})$ the minimax optimal sample complexity of passive learning: i.e., the minimal $n$ such that there exists a passive learning algorithm $\mathbb{A}_p$ that, for every $P$ with $\inf_{h \in \mathbb{C}} \mathrm{er}_P(h) \leq \beta$, for $S \sim P^n$ and $\hat{h}_n = \mathbb{A}_p(S)$, guarantees with probability at least $1 - \delta$ that $\mathrm{er}_P(\hat{h}_n) \leq \inf_{h \in \mathbb{C}} \mathrm{er}_P(h) + \varepsilon$. Since we can always design an active learner that simply queries the first $n$ examples and runs a passive learner $\mathbb{A}_p$, we clearly always have $\mathrm{QC}_a(\varepsilon, \delta; \beta, \mathbb{C}) \leq \mathcal{M}_p(\varepsilon, \delta; \beta, \mathbb{C})$. Thus, the main question of interest is whether $\mathrm{QC}_a(\varepsilon, \delta; \beta, \mathbb{C})$ is *strictly smaller* than $\mathcal{M}_p(\varepsilon, \delta; \beta, \mathbb{C})$, and if so, by how much. Lower bounds of Vapnik and Chervonenkis (1974); Devroye and Lugosi (1995) establish that

$$\mathcal{M}_p(\varepsilon, \delta; \beta, \mathbb{C}) = \Omega\left(\frac{\beta}{\varepsilon^2}\left(\mathsf{d} + \log\left(\frac{1}{\delta}\right)\right) + \frac{1}{\varepsilon}\left(\mathsf{d} + \log\left(\frac{1}{\delta}\right)\right)\right), \tag{4}$$

recalling that d denotes the *VC dimension* of $\mathbb{C}$ (Vapnik and Chervonenkis, 1971; see Definition 4 of Appendix B). The classic analysis of Vapnik and Chervonenkis (1974) further established this lower bound can nearly be achieved by the simple method of *empirical risk minimization*, i.e., $\hat{h}_n = \operatorname{argmin}_{h \in \mathbb{C}} \hat{\mathrm{er}}_S(h)$, providing an upper bound $\mathcal{M}_p(\varepsilon, \delta; \beta, \mathbb{C}) = O\left(\frac{\beta}{\varepsilon^2}\left(\mathsf{d}\log\left(\frac{1}{\varepsilon}\right) + \log\left(\frac{1}{\delta}\right)\right) + \frac{1}{\varepsilon}\left(\mathsf{d}\log\left(\frac{1}{\varepsilon}\right) + \log\left(\frac{1}{\delta}\right)\right)\right)$. This has since been refined in various ways, such as via localized chaining arguments (e.g., Giné and Koltchinskii, 2006). Most recently, Hanneke, Larsen, and Zhivotovskiy (2024b) proved an upper bound $\mathcal{M}_p(\varepsilon, \delta; \beta, \mathbb{C}) = O\left(\frac{\beta}{\varepsilon^2}\left(\mathsf{d} + \log\left(\frac{1}{\delta}\right)\right)\right) + \tilde{O}\left(\frac{1}{\varepsilon}\left(\mathsf{d} + \log\left(\frac{1}{\delta}\right)\right)\right)$, matching the lower bound (4) up to log factors in the lower-order term (the problem of removing these remaining log factors remains open at this time). The algorithm achieving this is *improper*, meaning its returned $\hat{h}_n$ is not necessarily an element of $\mathbb{C}$, and Hanneke, Larsen, and Zhivotovskiy (2024b) in fact show that for some concept classes $\mathbb{C}$ improperness is *necessary* to match the lower bound (4) in the lead term, as all proper learners incur an extra $\log\left(\frac{1}{\beta}\right)$ factor. In the special case of $\beta = 0$ (the *realizable case*), the lower bound (4) was shown to be achievable by Hanneke (2016a) (also necessarily via an improper learner), so that $\mathcal{M}_p(\varepsilon, \delta; 0, \mathbb{C}) = \Theta\left(\frac{1}{\varepsilon}\left(\mathsf{d} + \log\left(\frac{1}{\delta}\right)\right)\right)$. The lower bound (4) will therefore serve as a suitable baseline for gauging whether the query complexity $\mathrm{QC}_a(\varepsilon, \delta; \beta, \mathbb{C})$ of active learning is smaller than the sample complexity $\mathcal{M}_p(\varepsilon, \delta; \beta, \mathbb{C})$ of passive learning.

**The Need for Distribution-dependent Analysis in Realizable Active Learning:** Much of the early work on active learning focused on the *realizable case*, i.e., the special case $\beta = 0$. In this special case, it was quickly observed by Dasgupta (2004, 2005) that there are some concept classes (e.g., *thresholds* $\mathbb{1}_{[a,\infty)}$ on $\mathbb{R}$) where active learning offers strong improvements over passive learning, and other concept classes (e.g., *intervals* $\mathbb{1}_{[a,b]}$ on $\mathbb{R}$) where the (distribution-free) minimax query complexity $\mathrm{QC}_a(\varepsilon, \delta; 0, \mathbb{C})$ offers *no significant improvements* over passive learning. The essential advantage in the former case arises from a kind of "binary search" behavior, where the "uncertainty" is being sequentially reduced by a careful choice of queries. In contrast, the essential challenge in the latter case is the problem of "searching in the dark" for a small-but-important region: e.g., the optimal concept is 1 for a single unknown $x_i$ among some $x_1, \ldots, x_{1/\varepsilon}$, and $P_X = \mathrm{Uniform}(\{x_1, \ldots, x_{1/\varepsilon}\})$. It turns out this *hard* scenario is embedded in many concept classes of interest, a fact which was formalized and quantified by Hanneke and Yang (2015) in the *star number* complexity measure (Definition 2) discussed below. Such concept classes $\mathbb{C}$ naturally exhibit a lower bound $\mathrm{QC}_a(\varepsilon, \delta; 0, \mathbb{C}) = \Omega\left(\frac{1}{\varepsilon}\right)$. Even worse, consider a scenarios where the optimal concept can be 1 for any d points $x_i$ among $x_1, \ldots, x_{\mathsf{d}/(2\varepsilon)}$, and $P_X = \mathrm{Uniform}(\{x_1, \ldots, x_{\mathsf{d}/(2\varepsilon)}\})$. Hanneke and Yang (2015) show this scenario has $\mathrm{QC}_a(\varepsilon, \delta; 0, \mathbb{C}) = \Omega\left(\frac{\mathsf{d}}{\varepsilon}\right)$, so that $\mathrm{QC}_a(\varepsilon, \delta; 0, \mathbb{C})$ has the same joint dependence on $(\mathsf{d}, \varepsilon)$ as passive learning $\mathcal{M}_p(\varepsilon, \delta; 0, \mathbb{C}) = \Theta\left(\frac{1}{\varepsilon}\left(\mathsf{d} + \log\left(\frac{1}{\delta}\right)\right)\right)$, only offering

---

[5]Since the active learner also has access to the remaining (unqueried) i.i.d. *unlabeled examples* $X_i$, it is also natural to compare to the related framework of *semi-supervised* learning, in which a learner has access to some number $n$ of i.i.d. labeled examples with distribution $P$ and additionally some larger number $m$ of i.i.d. *unlabeled* examples with distribution $P_X$ (Chapelle, Scholkopf, and Zien, 2006). While, under some favorable conditions, the labeled sample complexity $n$ of semi-supervised learning can be smaller than that of strictly-supervised passive learning (see Balcan and Blum, 2010), the lower bounds on the (distribution-free) sample complexity of passive learning discussed in this work remain valid for the labeled sample complexity of semi-supervised learning (regardless of how many unlabeled examples are available), so that for the purpose of comparison in the present work, the distinction between supervised and semi-supervised passive learning as a baseline is not important, and we will simply compare to passive supervised learning for simplicity.

an improvement in the dependence on $\delta$ (where $\delta$ thus only affects the *unlabeled* sample complexity). Moreover, as noted by Hanneke (2014), similar scenarios[6] are embedded (with $\mathsf{d} = \Theta(\mathrm{VC}(\mathbb{C}))$) in most concept classes $\mathbb{C}$ of interest in learning theory (e.g., linear classifiers in $\mathbb{R}^{3\mathsf{d}}$, and axis-aligned rectangles in $\mathbb{R}^{2\mathsf{d}}$), so that such classes also exhibit no significant improvements over passive learning in their (distribution-free minimax) realizable-case query complexity $\mathrm{QC}_a(\varepsilon, \delta; 0, \mathbb{C})$.

Motivated by the fact that such *hard* scenarios are embedded in many concept classes of interest, Dasgupta (2005) suggested that, for such concept classes, the only viable way to understand the potential advantages of active learning is to focus on *distribution-dependent* analysis, toward identifying special scenarios where active learning algorithms offer improvements over passive learning, by formulating appropriate assumptions on the distribution $P$. This narrative quickly caught on in the literature, with a variety of distribution-dependent analyses and general $P$-dependent complexity measures being proposed to analyze certain active learning strategies under various restrictions on the realizable distribution $P$ (Dasgupta, 2005; Hanneke, 2007a,b, 2014; Balcan, Broder, and Zhang, 2007; Balcan and Long, 2013; Zhang and Chaudhuri, 2014; El-Yaniv and Wiener, 2012; Wiener, Hanneke, and El-Yaniv, 2015). We discuss several of these in detail below.

**Active Learning in the Non-realizable Case:**  Given the above narrative, when approaching the analysis of active learning in the *non-realizable* case ($\beta > 0$), it might at first seem perfectly reasonable to expect that for many concept classes $\mathbb{C}$ the query complexity $\mathrm{QC}_a(\varepsilon, \delta; \beta, \mathbb{C})$ might *not* be much smaller than the sample complexity of passive learning $\mathcal{M}_p(\varepsilon, \delta; \beta, \mathbb{C})$. As such, the literature on agnostic active learning has largely focused on *extending* the distribution-dependent analyses from the realizable case to the agnostic setting (e.g., Balcan, Beygelzimer, and Langford, 2006, 2009; Hanneke, 2007b,a, 2009b, 2011, 2014; Balcan and Hanneke, 2012; Zhang and Chaudhuri, 2014; Wiener, Hanneke, and El-Yaniv, 2015).  These upper bounds conformed to the accepted narrative, in that they offer improvements under some distributions, but for most classes $\mathbb{C}$, in the worst case over distributions $P$ (with $\inf_{h \in \mathbb{C}} \mathrm{er}_P(h) \leq \beta$) they essentially revert to the passive sample complexity $\mathcal{M}_p(\varepsilon, \delta; \beta, \mathbb{C})$.

However, since this narrative was born from analysis of the *realizable case*, there was no actual reason to believe it should remain valid in the non-realizable case. In particular, there remained an intriguing possibility that there could perhaps be *other advantages* of active learning *specific to the non-realizable case*: that is, beyond the "binary search" type advantages known from the realizable case (as captured by the complexity measures proposed for realizable-case analysis). Some hints that such additional advantages may exist appear in the works of Efromovich (2007) and Hanneke and Yang (2015) studying certain special scenarios (noise models more-restrictive than the agnostic setting), which found that active learning can also be useful for *adaptively identifying noisy regions* (i.e., regions of $X$'s where $P(Y|X)$ is close to $\frac{1}{2}$) and allocating queries appropriately to compensate for this noisiness without wasting excessive queries in less-noisy regions (as passive learning would). This additional advantage, specific to the non-realizable case, offered quantitative advantages over passive learning under the specific conditions studied in those works (e.g., Hanneke and Yang, 2015 showed improvements in query complexity under certain noise models, namely Tsybakov noise and Benign noise, for all classes $\mathbb{C}$, including those with the "searching in the dark" scenario embedded in them). However, these works left open the question of whether such advantages can be observed also in the more-challenging *agnostic* setting. The extension to the agnostic case is not at all clear, since in this setting (unlike the special noise models in the works above) the source of non-realizability is not only the noisiness of the $P(Y|X)$ label distribution, but also *model misspecification*: i.e., it is possible to have $\beta > 0$ even when $P(Y|X) \in \{0, 1\}$, if the *Bayes classifier* is not in $\mathbb{C}$, in which

---

[6]The hard scenario embedded in these classes is slightly more structured. Namely, for any $\mathsf{d} \in \mathbb{N}$ and infinite $\mathcal{X}$, partition $\mathcal{X}$ into disjoint infinite subsets $\mathcal{X}_1, \ldots, \mathcal{X}_\mathsf{d}$, and define $\mathbb{C}_\mathsf{d} = \{h : \forall i \leq \mathsf{d}, \sum_{x \in \mathcal{X}_i} h(x) \leq 1\}$, which has $\mathrm{VC}(\mathbb{C}_\mathsf{d}) = \mathsf{d}$. For such classes $\mathbb{C}_\mathsf{d}$, Hanneke and Yang (2015) show that $\mathrm{QC}_a(\varepsilon, \delta; 0, \mathbb{C}_\mathsf{d}) = \Omega\left(\frac{\mathsf{d}}{\varepsilon}\right)$ (as an aside, it is a straightforward exercise to show a matching upper bound for this class). For homogeneous linear classifiers in $\mathbb{R}^{3\mathsf{d}}$, we can construct the $\mathcal{X}_i$ sets as *circles* in disjoint subspaces: i.e., $\mathcal{X}_i$ is all $(z_1, \ldots, z_{3\mathsf{d}}) \in \mathbb{R}^{3\mathsf{d}}$ s.t. all coordinates $z_j = 0$ except $z_{3(i-1)+1}^2 + z_{3(i-1)+2}^2 = 1$ and $z_{3(i-1)+3} = 1$ (to allow for non-homogeneous linear classifiers on these circles, each controlled using 3 distinct weights in the 3d-dimensional linear classifier); the classifiers with boundary tangent to these circles witness the $\mathsf{d}$ singleton problems in $\mathbb{C}_\mathsf{d}$. For axis-aligned rectangles in $\mathbb{R}^{2\mathsf{d}}$, we can construct the $\mathcal{X}_i$ sets as diagonal *lines* in disjoint 2-dimensional subspaces: i.e., $\mathcal{X}_i$ is all $(z_1, \ldots, z_{2\mathsf{d}}) \in \mathbb{R}^{2\mathsf{d}}$ with all $z_j = 0$ except $z_{2(i-1)+1} \in [0, 1]$ and $z_{2(i-1)+2} = 1 - z_{2(i-1)+1}$; as each $\mathcal{X}_i$ can be classified by a 2-dimensional rectangle based on 2 distinct coordinates, the classifiers with a single *corner* intersecting each of these diagonal lines then witness the $\mathsf{d}$ singleton problems in $\mathbb{C}_\mathsf{d}$.

case the idea of adapting to "noisiness" of labels is no longer a useful framing of the problem. (The present work finds the appropriate re-framing of this capability, replacing the notion of "noisiness" by the *variance* of the excess-error estimation problem, and identifies algorithmic principles for isolating such high-variance regions, constituting the AVID principle).

**Quantifying the Query Complexity (Realizable Case):**    As discussed above, motivated by negative results for distribution-free analysis in the realizable case, numerous works have studied the query complexity of active learning under restrictions on the distribution $P$. These distribution-dependent analyses are often expressed in terms of abstract $P$-dependent complexity measures intended to capture favorable conditions for active learning, compared to passive learning. Quantitatively, in the realizable case ($\beta = 0$), such $P$-dependent query complexity bounds are typically expressed in the form $c_P(\varepsilon) \cdot \mathsf{d} \cdot \text{polylog}\left(\frac{1}{\varepsilon\delta}\right)$ for some $P$-dependent complexity measure $c_P(\varepsilon)$. Examples of such complexity measures include the splitting index (Dasgupta, 2005), disagreement coefficient (Hanneke, 2007b, 2009b, 2011), empirical extended teaching dimension (Hanneke, 2007a), and a subregion variant of the disagreement coefficient (Zhang and Chaudhuri, 2014), among others (e.g., El-Yaniv and Wiener, 2012; Hanneke, 2012, 2014; Wiener, Hanneke, and El-Yaniv, 2015; Hanneke and Yang, 2015). Some of these were accompanied by related minimax lower bounds holding for any fixed $P_X$ marginal distribution (Dasgupta, 2005; Hanneke, 2007a; Balcan and Hanneke, 2012). We discuss several of these $P$-dependent analyses and $c_P(\varepsilon)$ complexity measures in detail in Appendix A.2.

The works of Hanneke and Yang (2015); Hanneke (2016b, 2024) later showed that all of these proposed complexity measures $c_P(\varepsilon)$ have worst-case values (i.e., $\sup_P c_P(\varepsilon)$ over realizable $P$) *precisely equal* a complexity measure $\mathfrak{s}$ therein termed the *star number* of $\mathbb{C}$ (Definition 2), a quantity which abstractly formalizes and quantifies the extent to which the aforementioned "searching in the dark" scenario is embedded in a given concept class $\mathbb{C}$. Thus, the star number *unifies* all of these complexity measures in the case of distribution-free analysis. Moreover, Hanneke and Yang (2015) also show the star number sharply characterizes the optimal distribution-free query complexity in the realizable case: namely,

$$\mathfrak{s} \wedge \left(\frac{1}{\varepsilon} + \mathsf{d}\right) \lesssim \text{QC}_a(\varepsilon, \delta; 0, \mathbb{C}) \lesssim \left(\mathfrak{s} \wedge \frac{\mathsf{d}}{\varepsilon}\right) \log\left(\frac{1}{\varepsilon}\right) \tag{5}$$

(see Hanneke and Yang, 2015, for more-detailed bounds). Hanneke and Yang (2015) also show the upper and lower bounds in (5) represent a *nearly-sharp* dependence on $(\mathsf{d}, \mathfrak{s})$: that is, there exist concept classes $\overline{\mathbb{C}}_{\mathsf{d},\mathfrak{s}}$ and $\underline{\mathbb{C}}_{\mathsf{d},\mathfrak{s}}$ of any given VC dimension $\mathsf{d}$ and star number $\mathfrak{s} \geq \mathsf{d}$ for which $\text{QC}_a(\varepsilon, \delta; 0, \underline{\mathbb{C}}_{\mathsf{d},\mathfrak{s}}) = \Theta\left(\mathfrak{s} \wedge \left(\frac{1}{\varepsilon} + \mathsf{d}\right)\right)$ and $\text{QC}_a(\varepsilon, \delta; 0, \overline{\mathbb{C}}_{\mathsf{d},\mathfrak{s}}) = \Theta\left(\mathfrak{s} \wedge \frac{\mathsf{d}}{\varepsilon}\right)$. Thus, the bounds in (5) are essentially *unimprovable* (up to a log factor) without introducing additional complexity measures for the class $\mathbb{C}$. In particular, this also means any upper bound on $\text{QC}_a(\varepsilon, \delta; 0, \mathbb{C})$ depending on $\mathbb{C}$ only via $\mathsf{d}$ and $\mathfrak{s}$ can be no smaller than $\Omega\left(\mathfrak{s} \wedge \frac{\mathsf{d}}{\varepsilon}\right)$.

The bounds in (5) imply that $\text{QC}_a(\varepsilon, \delta; 0, \mathbb{C})$ admits an improved dependence on $\varepsilon$ compared to $\mathcal{M}_p(\varepsilon, \delta; 0, \mathbb{C}) = \Theta\left(\frac{1}{\varepsilon}\left(\mathsf{d} + \log\left(\frac{1}{\delta}\right)\right)\right)$ *if and only if* $\mathfrak{s} < \infty$. While there exist some interesting concept classes $\mathbb{C}$ with finite star number (e.g., threshold classifiers on $\mathbb{R}$, decision stumps on $\mathbb{R}^p$), it turns out *most* concept classes of interest in learning theory have infinite (or very large) star number (e.g., $\mathfrak{s} = \infty$ for linear classifiers on $\mathbb{R}^p$, $p \geq 2$). Thus, the general bounds in (5) quantitatively reflect the fact (already observed in several cases by Dasgupta, 2004, 2005) that we should typically *not* expect any significant benefits of active learning in the realizable case to be reflected in the (distribution-free minimax) query complexity $\text{QC}_a(\varepsilon, \delta; 0, \mathbb{C})$. Moreover, as mentioned above, the concept classes $\overline{\mathbb{C}}_{\mathsf{d},\mathfrak{s}}$ with $\mathfrak{s} = \infty$ witnessing near-sharpness of the upper bound in (5) are also embedded in many concept classes of interest in learning theory (see footnote 6), further strengthening the lower bound to $\Omega\left(\frac{\mathsf{d}}{\varepsilon}\right)$ for such classes, thus matching the sample complexity $\mathcal{M}_p(\varepsilon, \delta; 0, \mathbb{C})$ of passive learning in all dependencies (except $\delta$).

**Quantifying the Query Complexity (Agnostic Case):**    Turning to the agnostic case ($\beta \geq 0$), Kääriäinen (2006) established a general lower bound $\text{QC}_a(\varepsilon, \delta; \beta, \mathbb{C}) = \Omega\left(\frac{\beta^2}{\varepsilon^2} \log\left(\frac{1}{\delta}\right)\right)$, later strengthened by Beygelzimer, Dasgupta, and Langford (2009) to

$$\text{QC}_a(\varepsilon, \delta; \beta, \mathbb{C}) = \Omega\left(\frac{\beta^2}{\varepsilon^2}\left(\mathsf{d} + \log\left(\frac{1}{\delta}\right)\right)\right). \tag{6}$$

Comparing this to the sample complexity of passive learning (discussed above), namely $\mathcal{M}_p(\varepsilon, \delta; \beta, \mathbb{C}) = \Theta\left(\frac{\beta}{\varepsilon^2}\left(\mathsf{d} + \log\left(\frac{1}{\delta}\right)\right)\right) + \tilde{\Theta}\left(\frac{1}{\varepsilon}\left(\mathsf{d} + \log\left(\frac{1}{\delta}\right)\right)\right)$, we see that in the regime $\beta \gg \sqrt{\varepsilon}$, the best improvement we can hope for from active learning would be to replace the factor $\beta$ with $\beta^2$: i.e., squaring the dependence on the best-in-class error rate $\inf_{h \in \mathbb{C}} \mathrm{er}_P(h)$. In the regime $\beta \lesssim \sqrt{\varepsilon}$, the realizable-case lower bound from (5) becomes relevant (the realizable case being a special case, since clearly $\inf_{h \in \mathbb{C}} \mathrm{er}_P = 0$ satisfies the condition $\inf_{h \in \mathbb{C}} \mathrm{er}_P(h) \leq \beta$), which may be thought of as a *lower-order* additive term.

The work of Balcan, Beygelzimer, and Langford (2006) initiated the study of upper bounds on the query complexity in the agnostic case, showing that the lower bound (6) can be matched in the special cases of *threshold* classifiers (concepts $\mathbb{1}_{[a,\infty)}$ on $\mathcal{X} = \mathbb{R}$), and (in the regime $\beta \lesssim \varepsilon/\sqrt{\mathsf{d}}$) matched up to a factor $\mathsf{d}$ for homogeneous linear classifiers under $P_X$ uniform in an origin-centered ball, extending these well-known examples from the realizable case. This analysis was generalized to *all* concept classes by Hanneke (2007b), expressing a query complexity bound of the form $\tilde{O}\left(c_P(\beta + \varepsilon)\mathsf{d}\left(\frac{\beta^2}{\varepsilon^2} + 1\right)\right)$, where the factor $c_P(\beta + \varepsilon)$ is based on a $P$-dependent complexity measure $\theta_P(\beta + \varepsilon)$ therein termed the *disagreement coefficient* (Definition 25). In particular, the bound of Hanneke (2007b) matches the lower bound of Kääriäinen (2006); Beygelzimer, Dasgupta, and Langford (2009) up to logs only when $\theta_P(\beta + \varepsilon) = \tilde{O}(1)$. The latter holds for threshold classifiers, and for other classes under restrictions on $P$, but in many other cases $\theta_P(\beta + \varepsilon)$ can be as large as $\frac{1}{\beta+\varepsilon}$ due to the "searching in the dark" problem discussed above; in such cases, the query complexity upper bound of Hanneke (2007b) is no smaller than the sample complexity of passive learning $\mathcal{M}_p(\varepsilon, \delta; \beta, \mathbb{C})$. Numerous later works (some discussed in detail below) discovered refinements and alternative $P$-dependent complexity measures used to express upper bounds on the query complexity (Hanneke, 2007a; Dasgupta, Hsu, and Monteleoni, 2007; Hanneke, 2009b, 2011, 2014; Zhang and Chaudhuri, 2014; Wiener, Hanneke, and El-Yaniv, 2015). However, like the bound of Hanneke (2007b), all of these results establish query complexity upper bounds of the form $\tilde{O}\left(c_P(\beta + \varepsilon)\mathsf{d}\left(\frac{\beta^2}{\varepsilon^2} + 1\right)\right)$ for some $P$-dependent complexity measure $c_P(\beta + \varepsilon)$, all of which have the property that, in the "searching in the dark" type scenarios discussed above, the value $c_P(\beta + \varepsilon) \geq \frac{1}{\beta+\varepsilon}$, so that in such scenarios these upper bounds are all no smaller than the sample complexity of passive learning $\mathcal{M}_p(\varepsilon, \delta; \beta, \mathbb{C})$.

As in the realizable case, these various analysis were later all unified under worst-case analysis over $P$ by the *star number* in the work of Hanneke and Yang (2015). Indeed, these complexity measures $c_P(\beta + \varepsilon)$ are in fact the same family of complexity measures alluded to above for the realizable case. As such, by the aforementioned result of Hanneke and Yang (2015), they all satisfy $\sup_P c_P(\beta + \varepsilon) = \mathfrak{s} \wedge \frac{1}{\beta+\varepsilon}$. Thus, the upper bounds established by these works, all being of the form $\tilde{O}\left(c_P(\beta + \varepsilon)\mathsf{d}\left(\frac{\beta^2}{\varepsilon^2} + 1\right)\right)$, unify to a single upper bound of the form $\tilde{O}\left(\left(\mathfrak{s} \wedge \frac{1}{\beta+\varepsilon}\right)\mathsf{d}\left(\frac{\beta^2}{\varepsilon^2} + 1\right)\right)$ in the worst case over distributions $P$ (satisfying $\inf_{h \in \mathbb{C}} \mathrm{er}_P(h) \leq \beta$). In particular, this also means they all fail to imply any improvements over the sample complexity of passive learning $\mathcal{M}_p(\varepsilon, \delta; \beta, \mathbb{C})$ in the worst case over such distributions $P$, for any concept class $\mathbb{C}$ with $\mathfrak{s} = \infty$.[7] This is significant, since (as discussed above) most commonly-studied concept classes have $\mathfrak{s} = \infty$, including, for instance, linear classifiers in $\mathbb{R}^p$, $p \geq 2$. On the other hand, the lower bound (6), of the form $\Omega\left(\frac{\beta^2}{\varepsilon^2}\left(\mathsf{d} + \log\left(\frac{1}{\delta}\right)\right)\right)$, has no such factor $\mathfrak{s} \wedge \frac{1}{\beta+\varepsilon}$. The natural question is therefore which of these can be strengthened: the upper bound or lower bound.

The above gap has a qualitative significance. If the lower bound could be strengthened to match the upper bound, it would mean that (as in the realizable case) there are classes where active learning offers no advantage in its minimax query complexity compared to passive learning. On the other hand, if the upper bound can be strengthened to match the lower bound, it would mean that (unlike the realizable case) the query complexity of active learning is *always* smaller than the sample complexity of passive learning in the agnostic setting: i.e., for *every* concept class. The problem of resolving this gap has remained open until now. In the present work, we completely resolve this question,

---

[7]One can also show that this is not merely a result of loose analysis. The algorithms (prior to the present work) can be made to behave similarly to passive learners (meaning they query almost indiscriminately) in some scenarios constructed on large star sets, resulting in a number of queries $\frac{\beta}{\varepsilon^2}$.

strengthening the upper bound to match the lower bound (6), thereby establishing that active learning is *always* better than passive learning in the agnostic case, providing an improvement by squaring the dependence on the best-in-class error rate in the leading term: i.e., replacing $\beta$ with $\beta^2$. Establishing this upper bound requires a new principle for active learning, specific to the agnostic setting, which we develop in this work (termed AVID, for *Adaptive Variance Isolation by Disagreements*).

Before proceeding with the presentation of our results, we first provide, in the next subsection, a detailed survey of several of the prior works mentioned in the above brief historical summary.

## A.2   Detailed Description of Relevant Techniques in the Prior Literature

In this subsection, we provide further details of relevant works in the literature. Due to the vastness and diversity of the literature on the theory of active learning, we will not provide an exhaustive survey here, instead focusing on the techniques and results most-relevant to the present work.

### A.2.1   Disagreement-based Active Learning

By-far the most well-studied technique in the literature on the theory of active learning is *disagreement-based* active learning. A disagreement-based active learner is given as input the sequence $X_1, X_2, \ldots, X_m$ of unlabeled examples. It maintains (either explicitly or implicitly) a set $V \subseteq \mathbb{C}$ of *surviving concepts* (known as a *version space*), with a guarantee that the *best-in-class* concept[8] $h^\star$ is always retained in $V$. To choose its query points, it finds the next unlabeled example $X_i$ in the sequence for which $\exists f, g \in V$ with $f(X_i) \neq g(X_i)$, and queries for the label $Y_i$: or more succinctly, it queries the next $X_i \in \mathrm{DIS}(V)$, where

$$\mathrm{DIS}(V) := \{x \in \mathcal{X} : \exists f, g \in V, f(x) \neq g(x)\},$$

denotes the *region of disagreement* of $V$. It then updates the set $V$ of surviving concepts based on this new information (or, in some variants, it performs this update only periodically, rather than after every query). This is abstractly summarized in the following outline (where Step 4 can be instantiated in various ways, as discussed below).

---

**Algorithm Outline:** Disagreement-based Active Learning
Input: Unlabeled data $X_1, \ldots, X_m$
Output: Classifier $\hat{h}$
0. Initialize $V = \mathbb{C}$
1. For $i = 1, 2, \ldots, m$
2.    If $X_i \in \mathrm{DIS}(V)$
3.      Query for label $Y_i$
4.    Update $V$
5. Return any $\hat{h} \in V$

---

The idea is that, if we seek to return a concept $\hat{h} \in V$ with small $\mathrm{er}_P(\hat{h}) - \mathrm{er}_P(h^\star)$, then for any $X_i \notin \mathrm{DIS}(V)$, since all surviving concepts *agree* on the classification of $X_i$, the label $Y_i$ would provide no information that would help with this goal, so we do not bother querying for this label: that is, such a $Y_i$ cannot help to estimate the *relative* performances $\mathrm{er}_P(f) - \mathrm{er}_P(g)$ of concepts $f, g \in V$, since regardless of $Y_i$, we have $\mathbb{1}[f(X_i) \neq Y_i] - \mathbb{1}[g(X_i) \neq Y_i] = 0$. In contrast, the next $X_i \in \mathrm{DIS}(V)$ in the sequence is a random sample from $P_X(\cdot|\mathrm{DIS}(V))$, so that for any $f, g \in V$, $\mathbb{1}[f(X_i) \neq Y_i] - \mathbb{1}[g(X_i) \neq Y_i]$ is an unbiased estimate of the difference of error rates under the conditional distribution $P(\cdot|\mathrm{DIS}(V) \times \{0, 1\})$, which (again since $f, g$ agree outside $\mathrm{DIS}(V)$) is proportional to $\mathrm{er}_P(f) - \mathrm{er}_P(g)$. By reasoning about uniform concentration of these estimates, we can define an update rule for $V$ in Step 4 that never removes the best-in-class concept $h^\star$ while pruning sub-optimal concepts from $V$ (where the resolution of this pruning improves as $i$ grows).

In the *realizable case*, since we always have $h^\star(X_i) = Y_i$, the updates to $V$ in Step 4 can simply remove all concepts incorrect on a queried example $(X_i, Y_i)$: that is, $V \leftarrow \{h \in V : h(X_i) = Y_i\}$ (called the *version space*), which always retains $h^\star \in V$. The algorithmic principle of disagreement-based queries, and corresponding reasoning about correctness and potential advantages, was already

---

[8]The theory easily generalizes to cases where the infimum $\inf_{h \in \mathbb{C}} \mathrm{er}_P(h)$ is not attained, in which case define $h^\star \in \mathbb{C}$ to have $\mathrm{er}_P(h^\star)$ sufficiently close to the infimum, e.g., as in (10) below.

identified in the early work of Mitchell (1979) (for the membership queries model).[9] The precise form expressed above (sequentially checking unlabeled examples for disagreements) was first explicitly studied by Cohn, Atlas, and Ladner (1994), and in their honor, this realizable-case technique is referred to as CAL in the literature. As for its theoretical analysis, the original works of Mitchell (1979); Cohn, Atlas, and Ladner (1994) include the observation that $h^\star$ is retained in $V$, and Cohn, Atlas, and Ladner (1994) additionally include some discussion of generalization. However, the formal analysis of the query complexity of this technique in the PAC framework only began with the later work of Balcan, Beygelzimer, and Langford (2006) (bounding the query complexity for some specific concept classes), and the general analysis of the technique (applicable to any concept class) began with the works of Hanneke (2007b, 2009b, 2011); Dasgupta, Hsu, and Monteleoni (2007).

The idea of disagreement-based active learning was first extended to the agnostic setting ($\beta \geq 0$) by Balcan, Beygelzimer, and Langford (2006), with an instantiation of the above outline they called the $A^2$ algorithm (for *Agnostic Active*). The main idea in $A^2$ is to instantiate the update to $V$ in Step 4 using uniform concentration inequalities. In their original version, they specifically define $\mathrm{UB}(h)$ and $\mathrm{LB}(h)$ as high-probability uniform upper and lower bounds on $\mathrm{er}_{P(\cdot|\mathrm{DIS}(V)\times\{0,1\})}(h)$ based on the queries from $\mathrm{DIS}(V)$ since the last update to $V$ (where they only update $V$ periodically in their algorithm, so that the queried examples since the last update are i.i.d. samples from $P(\cdot|\mathrm{DIS}(V)\times\{0,1\})$). They then define the update as $V \leftarrow \{h \in V : \mathrm{LB}(h) \leq \min_{h' \in V} \mathrm{UB}(h')\}$. The idea is that they wish to remove a concept $h$ from $V$ if there is another concept $h' \in V$ whose *upper bound* $\mathrm{UB}(h')$ on its error rate is smaller than the *lower bound* $\mathrm{LB}(h)$ on the error rate of $h$. In particular, since $h, h'$ agree on all $x \notin \mathrm{DIS}(V)$, we have

$$\mathrm{er}_P(h) - \mathrm{er}_P(h') \propto \mathrm{er}_{P(\cdot|\mathrm{DIS}(V)\times\{0,1\})}(h) - \mathrm{er}_{P(\cdot|\mathrm{DIS}(V)\times\{0,1\})}(h') \geq \mathrm{LB}(h) - \mathrm{UB}(h'),$$

and hence a concept $h$ can be removed from $V$ *only if* $\mathrm{er}_P(h) - \mathrm{er}_P(h') > 0$ for some $h' \in V$, meaning that $h$ is verifiably suboptimal, guaranteeing that the algorithm always retains $h^\star \in V$. Conversely, by querying the examples in $\mathrm{DIS}(V)$, we are improving the concentration inequalities $\mathrm{UB}(h), \mathrm{LB}(h)$, so that suboptimal concepts are removed from $V$, which has two benefits: (1) we are converging to a set of relatively low-error concepts (important for the final error guarantee), and (2) by reducing $V$ we are potentially also reducing $\mathrm{DIS}(V)$, focusing the algorithm's queries to more-informative samples and decreasing the query complexity.

The original analysis of Balcan, Beygelzimer, and Langford (2006) included the above correctness guarantee (i.e., the algorithm maintains $h^\star \in V$), along with a general guarantee that the $A^2$ algorithm returns an $\hat{h}$ with $\mathrm{er}_P(\hat{h}) \leq \mathrm{er}_P(h^\star) + \varepsilon$, with a number of queries never significantly worse than that of passive learning. Also, as a sort of *proof of concept* illustrating the potential benefits of $A^2$ in a simple example, they also quantified the query complexity advantages in the special case of *threshold classifiers* (concepts $\mathbb{1}_{[a,\infty)}$ on $\mathcal{X} = \mathbb{R}$), showing a bound $\tilde{O}\left(\frac{\beta^2}{\varepsilon^2} + 1\right)$ for that class (matching the lower bound of Kääriäinen, 2006). They also studied the special case of learning homogeneous linear classifiers under a uniform distribution in an origin-centered ball in $\mathbb{R}^{\mathsf{d}}$, focusing on the regime $\beta \lesssim \varepsilon/\sqrt{\mathsf{d}}$, for which they showed the query complexity is $\tilde{O}\left(\mathsf{d}^2 \log\left(\frac{1}{\varepsilon}\right) \log\left(\frac{1}{\delta}\right)\right)$.

The first *general* analyses (i.e., applicable to *any* concept class) of the query complexity of active learning in the agnostic setting were given in the works of Hanneke (2007b,a). In particular, Hanneke (2007b) analyzed the $A^2$ disagreement-based active learning algorithm, providing a general query complexity bound expressed in terms of a new complexity measure therein termed the *disagreement coefficient*. Specifically, for $r > 0$, denoting by $\mathrm{B}_{P_X}(h^\star, r) = \{h \in \mathbb{C} : P_X(x : h(x) \neq h^\star(x)) \leq r\}$ the $h^\star$-centered $r$-ball (under $\mathrm{L}_1(P_X)$), the disagreement coefficient is defined as

$$\theta_P(\beta + \varepsilon) := \sup_{r > \beta + \varepsilon} \frac{P_X(\mathrm{DIS}(\mathrm{B}_{P_X}(h^\star, r)))}{r} \vee 1.$$

The intuitive interpretation of the relevance of this quantity is that, as the algorithm progresses, the set $V$ of surviving concepts will become closer and closer to $h^\star$ (up until a distance $O(\beta + \varepsilon)$), so that the probability of querying decreases as $P_X(\mathrm{DIS}(V)) \leq P_X(\mathrm{DIS}(\mathrm{B}_{P_X}(h^\star, r)))$ for an appropriate $r$ decreasing as the number of queries grows.

Hanneke (2007b) proves that, for any $\mathbb{C}$ and $P$, for $\beta = \mathrm{er}_P(h^\star)$, the $A^2$ algorithm succeeds after a number of queries $\tilde{O}\left(\theta_P(\beta + \varepsilon)^2 \mathsf{d}\left(\frac{\beta^2}{\varepsilon^2} + 1\right)\right)$. This matches the lower bound (6) of Kääriäinen

---

[9]Mitchell (1979) also discusses some reasonable extensions of version spaces to the non-realizable case.

(2006); Beygelzimer, Dasgupta, and Langford (2009) up to logs whenever $\theta_P(\beta + \varepsilon) = \tilde{O}(1)$. In particular, Hanneke (2007b) bounds $\theta_P(\beta + \varepsilon)$ for a number of scenarios $(\mathbb{C}, P)$, including showing that this general query complexity upper bound recovers the examples of Balcan, Beygelzimer, and Langford (2006): $\theta_P(\beta + \varepsilon) \leq 2$ for threshold classifiers, and $\theta_P(\beta + \varepsilon) = O(\sqrt{d})$ for homogeneous linear classifiers under a uniform distribution on an origin-centered sphere (thus also removing the constraints on $\beta, \varepsilon$ from the result of Balcan, Beygelzimer, and Langford, 2006). However, Hanneke (2007b) also found $\theta_P(\beta + \varepsilon)$ can sometimes be as large as $\frac{1}{\beta + \varepsilon}$, particularly for the "searching in the dark" scenarios discussed above, in which case the query complexity bound is no smaller than the sample complexity of passive learning.

Subsequently, Dasgupta, Hsu, and Monteleoni (2007) refined the dependence on $\theta_P(\beta + \varepsilon)$ in this bound (analyzing a different disagreement-based algorithm), replacing $\theta_P(\beta + \varepsilon)^2$ with $\theta_P(\beta + \varepsilon)$. Their technique also identifies a principle enabling more-practical implementation of disagreement-based active learning, expressing the algorithm as a *reduction to empirical risk minimization* (ERM). In general (even with $A^2$), we can always maintain the set $V$ *implicitly* (i.e., without storing this large object $V$ explicitly), by simply maintaining the set of *constraints* that define it, which then enable us to perform the various operations (e.g., checking whether $X_i \in \mathrm{DIS}(V)$, or computing $\min_{h' \in V} \mathrm{UB}(h)$) via constraint satisfaction or constrained optimization problems. The algorithm of Dasgupta, Hsu, and Monteleoni (2007) takes this a step further, expressing such operations as (effectively) unconstrained optimization problems, or in other words, calls to an ERM subroutine (i.e., an algorithm which returns a concept in $\mathbb{C}$ minimizing the number of mistakes on any given labeled data set). Specifically, they store two labeled data sets $Q_i, L_i$, where $Q_i$ are the queried examples so far (up to round $i$) and $L_i$ are the unqueried examples so far (with *inferred* labels). On round $i$, they consider the concepts $h^1, h^0 \in \mathbb{C}$ of minimal $\hat{\mathrm{er}}_{Q_{i-1}}(h)$ subject to $\hat{\mathrm{er}}_{L_{i-1}}(h) = 0$ and $h^1(X_i) = 1$, $h^0(X_i) = 0$, if they exist (noting that such concepts can each be obtained by a single call to an ERM oracle with appropriately high weight, or repetition, of the $L_{i-1}$ and $(X_i, y)$ examples). If $\hat{\mathrm{er}}_{Q_{i-1}}(h^1)$ and $\hat{\mathrm{er}}_{Q_{i-1}}(h^0)$ are of similar sizes, they query for $Y_i$ and add it to $Q_{i-1}$ to get $Q_i$ (letting $L_i = L_{i-1}$), and otherwise they take an *inferred* label $\hat{y}_i = \operatorname{argmin}_y \hat{\mathrm{er}}_{Q_{i-1}}(h^y)$ and add $(X_i, \hat{y}_i)$ to $L_{i-1}$ to get $L_i$ (letting $Q_i = Q_{i-1}$). Note that this is equivalent to maintaining a set $V$ of all concepts $h \in \mathbb{C}$ having $\hat{\mathrm{er}}_{L_{i-1}}(h) = 0$ and $\hat{\mathrm{er}}_{Q_{i-1}}(h)$ of similar size to $\min_{h'} \hat{\mathrm{er}}_{Q_{i-1}}(h')$ among all $h' \in \mathbb{C}$ with $\hat{\mathrm{er}}_{L_{i-1}}(h') = 0$, and querying $X_i$ iff $X_i \in \mathrm{DIS}(V)$. Thus, this algorithm can also be viewed as a disagreement-based active learner (and this connection is made explicit in the analysis of Dasgupta, Hsu, and Monteleoni, 2007). (Notably, subsequent works of Beygelzimer, Hsu, Langford, and Zhang, 2010; Hsu, 2010 even further simplified this technique by dropping the $L_{i-1}$ constraints, obtaining similar query complexity bounds).

The specific quantification of the "similar sizes" criterion for the difference of empirical error rates comes from uniform Bernstein-style concentration inequalities (related to the uniform Bernstein inequality stated in Lemma 7 of Appendix D below). In particular, letting $S_{i-1} = \{(X_j, Y_j) : j < i\}$, among all pairs of concepts $h, h' \in \mathbb{C}$ with $\hat{\mathrm{er}}_{L_{i-1}}(h) = \hat{\mathrm{er}}_{L_{i-1}}(h') = 0$, since they *agree* on examples in $L_{i-1}$, we have

$$\hat{\mathrm{er}}_{Q_{i-1}}(h) - \hat{\mathrm{er}}_{Q_{i-1}}(h') \propto \hat{\mathrm{er}}_{Q_{i-1} \cup L_{i-1}}(h) - \hat{\mathrm{er}}_{Q_{i-1} \cup L_{i-1}}(h') = \hat{\mathrm{er}}_{S_{i-1}}(h) - \hat{\mathrm{er}}_{S_{i-1}}(h'),$$

so that we can make use of concentration inequalities for estimating differences of error rates from the i.i.d. data set $S_{i-1}$ (e.g., as in Lemma 7). Reasoning inductively, if $\hat{\mathrm{er}}_{L_{i-1}}(h^\star) = 0$, such a concentration inequality guarantees that among all $h \in \mathbb{C}$ with $\hat{\mathrm{er}}_{L_{i-1}}(h) = 0$, $\hat{\mathrm{er}}_{Q_{i-1}}(h^\star)$ can never be too much larger than $\hat{\mathrm{er}}_{Q_{i-1}}(h)$, so that if the algorithm does not query $X_i$ (meaning $\hat{\mathrm{er}}_{Q_{i-1}}(h^{1-\hat{y}_i})$ is much larger than $\hat{\mathrm{er}}_{Q_{i-1}}(h^{\hat{y}_i})$), the corresponding label $\hat{y}_i$ must be $h^\star(X_i)$, so that adding $(X_i, \hat{y}_i)$ to $L_i$ retains that $\hat{\mathrm{er}}_{L_i}(h^\star) = 0$. Conversely, if $\hat{\mathrm{er}}_{Q_{i-1}}(h^0)$ and $\hat{\mathrm{er}}_{Q_{i-1}}(h^1)$ are of similar sizes, then the algorithm has effectively verified that there exist concepts $h, h' \in \mathbb{C}$ with $\mathrm{er}_P(h)$ and $\mathrm{er}_P(h')$ rather close to $\mathrm{er}_P(h^\star)$ which nevertheless disagree on $X_i$ ($h(X_i) \neq h'(X_i)$), and querying for $Y_i$ and adding $(X_i, Y_i)$ to $Q_i$ then strengthens the concentration of the $\hat{\mathrm{er}}_{Q_i}$ estimates, to help further distinguish among such small-error concepts in subsequent rounds.

Dasgupta, Hsu, and Monteleoni (2007) analyzed the query complexity of this algorithm, showing that it guarantees $\mathrm{er}_P(\hat{h}) \leq \mathrm{er}_P(h^\star) + \varepsilon$ after a number of queries $\tilde{O}\left(\theta_P(\beta + \varepsilon)d\left(\frac{\beta^2}{\varepsilon^2} + 1\right)\right)$. Compared to the original analysis of Hanneke (2007b), this improves the bound in its dependence on $\theta_P(\beta + \varepsilon)$, reducing from quadratic $\theta_P(\beta + \varepsilon)^2$ to linear $\theta_P(\beta + \varepsilon)$. Again, the conclusion is that the algorithm's query complexity matches the lower bound (6) of Kääriäinen (2006); Beygelzimer, Dasgupta, and

Langford (2009) up to logs whenever $\theta_P(\beta + \varepsilon) = \tilde{O}(1)$; however, again, in scenarios $(\mathbb{C}, P)$ with $\theta_P(\beta + \varepsilon) = \Omega\left(\frac{1}{\beta+\varepsilon}\right)$, the bound offers no improvements over the sample complexity of passive learning.

The above techniques, and corresponding analysis in terms of the disagreement coefficient, seeded a vast literature, with many variations on the technique, analysis, and complexity measures, and many examples of scenarios $(\mathbb{C}, P)$ for which $\theta_P(\beta + \varepsilon)$ can be favorably bounded. This branch of the literature is collectively referred to as *disagreement-based active learning* (see e.g., the works of Hanneke, 2009b, 2011, 2012, 2014, 2016b; Balcan, Hanneke, and Vaughan, 2010; Hsu, 2010; El-Yaniv and Wiener, 2012; Friedman, 2009; Mahalanabis, 2011; Koltchinskii, 2010; Wang, 2011; Beygelzimer, Dasgupta, and Langford, 2009; Beygelzimer, Hsu, Langford, and Zhang, 2010; Raginsky and Rakhlin, 2011; Ailon, Begleiter, and Ezra, 2014; Huang, Agarwal, Hsu, Langford, and Schapire, 2015; Wiener, Hanneke, and El-Yaniv, 2015; Hanneke and Yang, 2010, 2015, 2019; Yan, Chaudhuri, and Javidi, 2018, 2019; Gelbhart and El-Yaniv, 2019; Cortes, DeSalvo, Gentile, Mohri, and Zhang, 2019a; Cortes, DeSalvo, Gentile, Mohri, and Zhang, 2019b,c, 2020; DeSalvo, Gentile, and Thune, 2021; Shayestehmanesh, 2020; Puchkin and Zhivotovskiy, 2022). A detailed summary of this line of work is presented in the survey of Hanneke (2014).

In the context of distribution-free analysis, Hanneke and Yang (2015) showed that $\sup_P \theta_P(\beta + \varepsilon) = \mathfrak{s} \wedge \frac{1}{\beta+\varepsilon}$, where $\mathfrak{s}$ is the *star number* of $\mathbb{C}$ (Definition 2), and where the sup is over realizable distributions $P$ (so that, in particular, they satisfy the condition $\mathrm{er}_P(h^\star) \leq \beta$). Thus, in terms of their implications for the distribution-free query complexity $\mathrm{QC}_a(\varepsilon, \delta; \beta, \mathbb{C})$, these $P$-dependent analyses of disagreement-based active learning simplify to a bound of the form $\mathrm{QC}_a(\varepsilon, \delta; \beta, \mathbb{C}) = \tilde{O}\left(\left(\mathfrak{s} \wedge \frac{1}{\beta+\varepsilon}\right)\mathsf{d}\left(\frac{\beta^2}{\varepsilon^2} + 1\right)\right)$. In particular, such bounds are capable of providing improvements in distribution-free query complexity over the sample complexity of passive learning $\mathcal{M}_p(\varepsilon, \delta; \beta, \mathbb{C})$ if and only if $\mathfrak{s} < \infty$ (which, as discussed above, is a rather strong restriction). This contrasts with Theorems 1, 3, which provide improvements for *all* concept classes $\mathbb{C}$, regardless of whether $\mathfrak{s}$ is finite or infinite. (The role of $\mathfrak{s}$ in Theorem 3 is merely in refining the *lower-order* term in the special case that $\mathfrak{s} < \infty$). In this sense, Theorems 1, 3 are *most* interesting for classes $\mathbb{C}$ with $\mathfrak{s} = \infty$, since no previous techniques provide improvements over passive learning for such classes.

We note that, while the AVID Agnostic algorithm (Figure 1) itself should *not* be regarded as a disagreement-based active learner (as its primary advantage over passive learning is *not* based on the restriction of queries to $\mathrm{DIS}(V)$), elements of disagreement-based learning have been incorporated into it for the purpose of the refined *lower-order* term in the upper bound in Theorem 3. Specifically, the choice to query examples in $S_k^1 \cap D_{k-1} \setminus \Delta_{i_k}$ in Step 2 (and similarly Step 9) restricts to queries in $D_{k-1} = \mathrm{DIS}(V_{k-1})$. This restriction is directly responsible for the lower-order term in Theorem 3 being of the form $\tilde{O}\left(\left(\mathfrak{s} \wedge \frac{1}{\varepsilon}\right)\mathsf{d}\right)$ rather than $\tilde{O}\left(\frac{\mathsf{d}}{\varepsilon}\right)$ as in Theorem 1. On the other hand, for the purpose of the lead term, this incorporation of disagreement-based queries is unnecessary, and indeed Theorem 1 remains valid *without* this aspect of the algorithm: that is, in Steps 2 and 9, if we simply query all of $S_k^1 \setminus \Delta_{i_k}$, the algorithm still achieves the query complexity bound stated in Theorem 1 with its lower-order term $\tilde{O}\left(\frac{\mathsf{d}}{\varepsilon}\right)$.

The argument leading to the refined lower-order term in Theorem 3 makes use of reasoning directly rooted in the analysis of disagreement-based methods via the disagreement coefficient (Lemma 22), and indeed we present $P$-dependent refinements of this lower-order term directly expressed in terms of $\theta_P(\beta + \varepsilon)$ in Appendix F.2. In particular, we show (Corollary 28) the lower-order term $\tilde{O}\left(\left(\mathfrak{s} \wedge \frac{1}{\varepsilon}\right)\mathsf{d}\right)$ can be replaced by $\tilde{O}\left(\theta_P(\beta + \varepsilon)^2\mathsf{d}\right)$, yielding an overall $P$-dependent query complexity bound $O\left(\frac{\beta^2}{\varepsilon^2}\left(\mathsf{d} + \log\left(\frac{1}{\delta}\right)\right)\right) + \tilde{O}\left(\theta_P(\beta + \varepsilon)^2\mathsf{d}\right)$. we further argue, in Appendix F.1, that it is *not* possible (by any algorithm) to reduce this lower-order term to $\tilde{O}(\theta_P(\beta + \varepsilon)\mathsf{d})$ or even $\tilde{O}(\theta_P(0)\mathsf{d})$, though we do show that other intermediate forms of the term are achievable, such as $\tilde{O}\left(\theta_P(\beta + \varepsilon)\mathsf{d}\left(\frac{\beta+\varepsilon}{\varepsilon}\right)\right)$.

### A.2.2 Subregion-based (Margin-based) Active Learning

Shortly after the work of Balcan, Beygelzimer, and Langford (2006), which included an analysis of homogeneous linear classifiers under a uniform distribution, Balcan, Broder, and Zhang (2007) proposed a refinement of disagreement-based active learning specific to linear classifiers. Rather than querying every example in the region of disagreement $\mathrm{DIS}(V)$, they identified a *subregion*

$R \subseteq \mathrm{DIS}(V)$ which suffices for the purpose of estimating differences of error rates $\mathrm{er}_P(f) - \mathrm{er}_P(g)$ among $f, g \in V$. The key idea is to choose $R$ so that any $f, g \in V$ have $P_X(\{f \neq g\} \setminus R)$ small, so that $R$ captures *most* of the disagreements between concepts $f, g \in V$ that are far apart. In their case, since they were specifically focusing on homogeneous linear classifiers (i.e., concepts $h_w(x) = \mathbb{1}[\langle w, x \rangle \geq 0]$ on $\mathcal{X} = \mathbb{R}^{\mathsf{d}}$) under $P_X$ uniform in an origin-centered ball, they could describe this region $R$ as a *slab* around the boundary of a current hypothesis $h_{\hat{w}}$: that is, $R = \left\{ x \in \mathbb{R}^{\mathsf{d}} : |\langle \hat{w}, x \rangle| \leq b \right\}$ for an appropriate width $b$ (which decreases over time as the algorithm progresses). In other words, the algorithm queries examples $X_i$ with *low margin* under the current hypothesis $\hat{w}$. As such, this technique is referred to as *margin-based active learning*. They analyzed this technique for the realizable case and under a specialized noise condition (Tsybakov noise), and found it provides advantages over disagreement-based learning: in the realizable case, improving the query complexity from $\mathsf{d}^{3/2} \cdot \mathrm{polylog}\left(\frac{1}{\varepsilon\delta}\right)$ to $\mathsf{d} \cdot \mathrm{polylog}\left(\frac{1}{\varepsilon\delta}\right)$ (matching the query complexities achieved by earlier works Freund, Seung, Shamir, and Tishby, 1997; Dasgupta, Kalai, and Monteleoni, 2005; Dasgupta, 2005), while allowing for some robustness to non-realizable distributions $P$ (albeit not fully agnostic). The technique was later extended in various ways, including studying adaptivity to certain noise parameters (Wang and Singh, 2016) and generalizing beyond the uniform distribution, to general isotropic log-concave or s-concave distributions (Balcan and Long, 2013; Balcan and Zhang, 2017).

This idea was extended to general concept classes $\mathbb{C}$ and distributions $P$, including the agnostic setting, in the work of Zhang and Chaudhuri (2014). Again the idea is to identify a region $R \subseteq \mathrm{DIS}(V)$ for which concepts $f, g \in V$ have only small disagreements outside $R$: $P_X(\{f \neq g\} \setminus R) \leq \eta$, for a small $\eta$. Rather than an explicit region $R$ (as in margin-based active learning), they simply choose a subset of the unlabeled examples via a linear program, which they show (in the analysis) can be related to an optimal choice of such a region. We discuss this technique in detail in Appendix F.3.

The implication of this refinement of disagreement-based learning is a $P$-dependent query complexity bound, stated in terms of a subregion-based refinement of the disagreement coefficient, defined as follows (adopting some simplifications from Hanneke, 2016b). As above, define the $r$-ball $\mathrm{B}_{P_X}(h^\star, r) = \{h \in \mathbb{C} : P_X(x : h(x) \neq h^\star(x)) \leq r\}$ for $r > 0$. Also, for $\eta \geq 0$, define

$$\Phi_{P_X}(\mathrm{B}_{P_X}(h^\star, r), \eta) := \inf\left\{ P_X(R) : \sup_{h \in \mathrm{B}_{P_X}(h^\star, r)} P_X(\{h \neq f\} \setminus R) \leq \eta, R \subseteq \mathcal{X}, f : \mathcal{X} \to \{0, 1\} \right\},$$

where $R$ and $f$ are restricted to be measurable. Finally, for $\varepsilon \geq 0$, define the subregion disagreement coefficient (Definition 31) as

$$\varphi_P(\varepsilon, \eta) := \sup_{r > \eta + \varepsilon} \frac{\Phi_{P_X}(\mathrm{B}_{P_X}(h^\star, r), (r - \eta)/c)}{r} \vee 1,$$

for an appropriate universal constant $c > 1$. The technique of Zhang and Chaudhuri (2014) provides a $P$-dependent query complexity bound of the form $\tilde{O}\left(\varphi_P(\varepsilon, 2\beta)\mathsf{d}\left(\frac{\beta^2}{\varepsilon^2} + 1\right)\right)$. In particular, it follows from the definitions that $\varphi_P(\varepsilon, 2\beta) \leq \theta_P(2\beta + \varepsilon)$ (see Appendix F.3), and Zhang and Chaudhuri (2014) discuss some examples where the gap is large. Thus, this represents a refinement of the query complexity bounds for disagreement-based active learning discussed above.

As a primary example where $\varphi_P(\varepsilon, 2\beta) \ll \theta_P(\beta + \varepsilon)$, consider again the scenario of homogeneous linear classifiers on $\mathbb{R}^{\mathsf{d}}$ under $P_X$ an isotropic log-concave distribution (as considered in the margin-based active learning works of Balcan, Broder, and Zhang, 2007; Balcan and Long, 2013 discussed above). In this scenario, Zhang and Chaudhuri (2014) show that $\varphi_P(\varepsilon, 2\beta) = O\left(\log\left(\frac{\beta}{\varepsilon}\right)\right)$ (based on concentration arguments from Balcan and Long, 2013). Thus, in this scenario, the query complexity bound of Zhang and Chaudhuri (2014) is $\tilde{O}\left(\mathsf{d}\left(\frac{\beta^2}{\varepsilon^2} + 1\right)\right)$. In contrast, Hanneke (2007b) showed $\theta_P(\beta + \varepsilon) = \Omega\left(\sqrt{\mathsf{d}} \wedge \frac{1}{\beta + \varepsilon}\right)$ for $P_X$ the uniform distribution on an origin-centered sphere (a special case of isotropic log-concave), so that the query complexity bounds for disagreement-based active learning are roughly $\mathsf{d}^{3/2}\left(\frac{\beta^2}{\varepsilon^2} + 1\right)$, hence are suboptimal by a factor $\sqrt{\mathsf{d}}$.

That said, in the context of distribution-free analysis, it is unclear whether there are advantages from this subregion technique. Specifically, Hanneke (2016b) showed that $\sup_P \varphi_P(\varepsilon, 0) = \mathfrak{s} \wedge \frac{1}{\varepsilon}$ (where the sup is restricted to realizable distributions $P$), which matches the worst-case value of $\theta_P(\varepsilon)$

(established by Hanneke and Yang, 2015). In (54) of Appendix F.3, we further extend this to $\varphi_P(\varepsilon, \eta)$ (using the fact that $\varphi_P(\varepsilon, \eta) \geq \varphi_P(\eta + \varepsilon, 0)$), establishing that (for $\varepsilon, \eta \geq 0$ with $\eta + \varepsilon \leq 1$)

$$\sup_P \varphi_P(\varepsilon, \eta) = \mathfrak{s} \wedge \frac{1}{\eta + \varepsilon},$$

where again the $\sup$ is restricted realizable distributions $P$. Thus, the implication of the $P$-dependent query complexity bound of Zhang and Chaudhuri (2014) for bounding the distribution-free query complexity $\mathrm{QC}_a(\varepsilon, \delta; \beta, \mathbb{C})$ is merely to recover the same query complexity bound $\tilde{O}\left(\left(\mathfrak{s} \wedge \frac{1}{\beta + \varepsilon}\right) \mathsf{d}\left(\frac{\beta^2}{\varepsilon^2} + 1\right)\right)$ already known to hold for disagreement-based active learning. In particular, this means that the above query complexity bound of Zhang and Chaudhuri (2014) is capable of providing improvements in the distribution-free query complexity of active learning, compared to the sample complexity $\mathcal{M}_p(\varepsilon, \delta; \beta, \mathbb{C})$ of passive learning, *if and only if* $\mathfrak{s} < \infty$ (again, a rather strong restriction). Again, this contrasts with Theorems 1, 3, which provide improvements for *all* concept classes $\mathbb{C}$, regardless of $\mathfrak{s}$, with $\mathfrak{s}$ merely influencing refinements in the lower-order term in Theorem 3.

In Appendix F.3, we give a refinement of the AVID Agnostic algorithm, which adopts this subregion technique (in combination with the AVID principle). We show this Subregion-AVID Agnostic algorithm achieves a $P$-dependent refinement of the lower-order term compared to the original AVID Agnostic algorithm. For instance, one implication of this refinement is replacing the term $\tilde{O}\left(\left(\mathfrak{s} \wedge \frac{1}{\varepsilon}\right) \mathsf{d}\right)$ in Theorem 3 with $\tilde{O}\left(\varphi_P(\varepsilon, 5\beta)^2 \mathsf{d}\right)$, yielding a $P$-dependent query complexity bound $O\left(\frac{\beta^2}{\varepsilon^2}\left(\mathsf{d} + \log\left(\frac{1}{\delta}\right)\right)\right) + \tilde{O}\left(\varphi_P(\varepsilon, 5\beta)^2 \mathsf{d}\right)$. It follows from an example in Appendix F.1 that the above quadratic dependence $\varphi_P(\varepsilon, 5\beta)^2$ cannot be reduced to $\varphi_P(\varepsilon, 5\beta)$ (or even $\varphi_P(0, 0)$) without introducing additional factors, though we also establish intermediate forms of the term, such as $\tilde{O}\left(\varphi_P(\varepsilon, 5\beta)\mathsf{d}\left(\frac{\beta + \varepsilon}{\varepsilon}\right)\right)$.

### A.2.3 Other Topics and Techniques in the Theory of Active Learning

In addition to disagreement-based active learning and its subregion-based refinement, and query complexity bounds for the agnostic setting, the active learning literature also contains numerous other techniques and topics. Though these ideas are not directly used in the present work, we briefly survey them here for completeness, and in some cases, to discuss connections to the results and techniques of the present work. For brevity, we omit most of the formal definitions, algorithms, and precise statements of the results, rather summarizing the essential ideas, and referring the interested reader to the original works for the precise results and details (some of which are also surveyed by Hanneke, 2014 in detail).

**The Splitting Index:** The earliest general theory of active learning, providing query complexity bounds applicable to any concept class and realizable-case distribution, was proposed by Dasgupta (2005). That work proposes a $(\mathbb{C}, P)$-dependent complexity measure called the *splitting index* $\rho \in [0, 1]$, based on the property that, for any $\gamma > \varepsilon$ and any finite set of pairs $(f, g) \in \mathbb{C}^2$ with $P_X(f \neq g) > \gamma$, there will likely be an unlabeled example $X_i$ for which, regardless of whether $Y_i$ is 0 or 1, we are guaranteed that at least a $\rho$ fraction of the pairs $(f, g)$ will have at least one function incorrect on $(X_i, Y_i)$ (see Dasgupta, 2005 for the precise definition). The idea is that $\rho$ measures a notion of *progress*, from querying such an $X_i$, toward reducing the *diameter* of the version space below $\gamma$. The pairs $(f, g)$ in the version space having $P_X(f \neq g) > \gamma$ are the *obstructions* to reducing the diameter of the version space. The above definition guarantees there will be an example $X_i$ we can query such that, regardless of which label $Y_i$ is returned, discarding all inconsistent concepts from the version space results in a reduction in the set of such obstructing pairs, leaving at most a $1 - \rho$ fraction of them. If we start with an $\alpha$-cover $V$ of the concept class $\mathbb{C}$ of size roughly $\alpha^{-\mathsf{d}}$ (with $\alpha < \varepsilon$ small enough to guarantee all queried labels agree with some $h \in V$), we would require at most $O\left(\frac{\mathsf{d}}{\rho} \log\left(\frac{1}{\alpha}\right)\right)$ such queries to eliminate *all* obstructing pairs, and thus reduce the diameter of the version space below $\gamma$. We can then decrease $\gamma$ by a constant factor and repeat, until the diameter is below $\varepsilon$, at which time we can return any surviving concept, yielding a query complexity roughly $O\left(\frac{\mathsf{d}}{\rho} \log\left(\frac{1}{\alpha}\right) \log\left(\frac{1}{\varepsilon}\right)\right)$.

The splitting index also provides a lower bound $\Omega\left(\frac{1}{\rho}\right)$ on the realizable-case query complexity (where, in this case, $\rho$ can be $P_X$-dependent, but should be $h^\star$-independent; see Dasgupta, 2005; Balcan

and Hanneke, 2012; Hanneke, 2014 for precise statements). This is particularly interesting due to being $P_X$-dependent, and yet still providing near-matching upper and lower bounds (in contrast, other quantities such as the disagreement coefficient and subregion-based refinement only provide upper bounds, and provably cannot yield general $P_X$-dependent lower bounds). Since the above results reveal the $P_X$-dependent query complexity can be well-captured by the splitting index, whereas Hanneke and Yang (2015) have shown the optimal *distribution-free* realizable-case query complexity is characterized by the *star number* (Definition 2), it is also natural to study the relation between these quantities. Toward this end, Hanneke and Yang (2015) have in fact shown that these quantities are *equivalent* in the context of distribution-free analysis: namely, $\sup_P \left\lfloor \frac{1}{\rho} \right\rfloor = \min\left\{ \mathfrak{s}, \left\lfloor \frac{1}{\varepsilon} \right\rfloor \right\}$.

The splitting index analysis also provides another interesting feature, which is perhaps missing from other works on active learning in the realizable case: it quantifies a *trade-off* between the query complexity and the number of *unlabeled* examples. This is reflected in the above (imprecise) definition in the part that requires that such a $\rho$-splitting example $X_i$ is likely to exist in the unlabeled data (this is made precise in Dasgupta, 2005 by another parameter $\tau$ reflecting the probability of obtaining such an example). Using a larger number of unlabeled examples can increase the likelihood of including an example $X_i$ that eliminates a larger fraction of pairs, so that the splitting index $\rho$ can grow larger (hence decreasing the query complexity) for larger unlabeled sample sizes. This improvement from having larger unlabeled sample sizes is not reflected in other complexity measures proposed in the literature, and in such cases the splitting-based query complexity bounds can be substantially smaller than those based on these other complexities, such as the disagreement coefficient (see Hanneke, 2014 for explicit comparisons). It is worth noting that such trade-offs are not known to be possible in the agnostic setting.

While the original work of Dasgupta (2005) was developed for the realizable case, subsequent works have explored extensions to non-realizable settings under restrictions on the types of non-realizability. Specifically, Balcan and Hanneke (2012); Hanneke (2014); Tosh and Hsu (2020) have extended the theory to allow for so-called *Massart noise*, wherein it is assumed $P(Y = h^\star(X)|X) - \frac{1}{2}$ is everywhere positive and bounded away from $0$. The extension of the splitting technique to that setting merely requires that we only remove a concept from consideration upon having sufficiently many errors on queried examples. The resulting query complexity bounds are then similar to the above.

To date, the splitting technique has not been extended to the *agnostic* setting in any meaningful way (e.g., to obtain query complexity bounds which could not also be obtained by, say, running disagreement-based active learning with an $(\varepsilon/2)$-cover of the concept class). The agnostic setting presents significant challenges for this technique, due to the $\rho$-splitting examples $X_i$ being possibly in $\mathrm{ER}(h^\star)$ for the best-in-class concept $h^\star$, meaning such examples cannot be trusted as the sole source of information for pruning suboptimal concepts; see the scenario in Appendix F.1 (which is constructed therein for a different reason, but also illustrates this issue).

We may remark that the AVID principle, developed in the present work and employed in $\mathbb{A}_{\mathrm{avid}}$, has certain aspects that are intriguingly reminiscent of the splitting technique of Dasgupta (2005). As in splitting, $\mathbb{A}_{\mathrm{avid}}$ aims to reduce the *diameter* of a set $V$ of surviving concepts. Toward this end, again as in splitting, it identifies *obstructing pairs*: $f, g \in V$ with $P_X(f \neq g) > \varepsilon_k$, where $\varepsilon_k$ is the desired diameter guarantee at that stage. However, the main difference is in how such obstructions are addressed in the algorithm. While the splitting technique would attempt to resolve this obstruction by *querying* to eliminate at least one of $f, g$ for many such obstructing pairs, $\mathbb{A}_{\mathrm{avid}}$ instead simply *removes* (isolates) the region $\{f \neq g\}$ from the space $\mathcal{X}$ (adding it to the $\Delta$ region), and estimates error rates separately in $\Delta$ and $\mathcal{X} \setminus \Delta$. Thus, in this aspect, the algorithmic principle underlying $\mathbb{A}_{\mathrm{avid}}$ is considerably different from splitting. Nevertheless, this common focus on addressing pairs $(f, g)$ obstructing the reduction of the diameter presents an intriguing connection, which might potentially warrant further exploration.

**Empirical Teaching Dimension:** Another approach to agnostic active learning, based on principles seemingly distinct from disagreement-based methods, was proposed in the work of Hanneke (2007a). The technique there is inspired by early work on Exact learning with membership queries in the realizable case by Hegedüs (1995); Hellerstein, Pillaipakkamnatt, Raghavan, and Wilkins (1996), which found interesting connections between active learning and the complexity of *machine teaching* (Goldman and Kearns, 1995). Hanneke (2007a) extends those ideas to the PAC setting, starting with an upper bound for the realizable case, based on a $P_X$-dependent complexity measure $\tau(\varepsilon)$ therein

termed the *extended teaching dimension growth function*: for an i.i.d.-$P_X$ data set $S$ of size $\frac{1}{\varepsilon}$, $\tau(\varepsilon)$ (roughly) represents the minimal size of a subsample which induces the same version space (for any fixed target concept $h^\star$). The main technique is to find sets of $\tau(\varepsilon)$ unlabeled examples for which the labels are guaranteed to significantly reduce the *number* of concepts in (a finite cover of) the version space. The work also presents a realizable-case lower bound based on a modified variant of this complexity measure.

Hanneke (2007a) further extends the upper bound to the non-realizable case, establishing an upper bound $\tilde{O}\left(\tau(\beta + \varepsilon)\mathsf{d}\left(\frac{\beta^2}{\varepsilon^2} + 1\right)\right)$. In particular, this matches the lower bound (6) of Kääriäinen (2006); Beygelzimer, Dasgupta, and Langford (2009) up to logs when $\tau(\beta + \varepsilon) = \tilde{O}(1)$. Hanneke (2007a) provides examples where $\tau(\beta + \varepsilon)$ is bounded, including the class of thresholds (concepts $\mathbb{1}_{[a,\infty)}$ on $\mathcal{X} = \mathbb{R}$) and axis-aligned rectangles (of at least some volume) under restrictions on $P_X$; however, as with the previously discussed complexity measures, $\tau(\beta + \varepsilon)$ can be as large as $\frac{1}{\beta+\varepsilon}$ for the "searching in the dark" type scenarios discussed above, in which case the above query complexity bound is no smaller than the sample complexity of passive learning. Indeed, as with the complexity measures discussed above, results of Hanneke and Yang (2015) imply that, taking the worst case over distributions, $\tau(\varepsilon)$ becomes equivalent to the *star number* (Definition 2): $\sup_{P_X} \tau(\varepsilon) \approx \min\left\{\mathfrak{s}, \frac{1}{\varepsilon}\right\}$.

Variants of this $\tau(\varepsilon)$ complexity measure were later further analyzed (for several example scenarios, and more-generally, in relation to the disagreement coefficient), under the name *version space compression set size*, and (interestingly) have also been found useful for studying *disagreement-based* active learning, by El-Yaniv and Wiener (2010, 2012); Wiener, Hanneke, and El-Yaniv (2015); Hanneke and Yang (2015); Hanneke (2016b); Hanneke and Kontorovich (2021).

**Restricted Noise Models:** Besides the study of first-order agnostic query complexity guarantees $\mathrm{QC}_a(\varepsilon, \delta; \beta, \mathbb{C})$ (the subject of the present work), the theory of active learning additionally includes many works on query complexity guarantees holding under other conditions or parameterizations, or in other words, under various *noise models*. Here we briefly survey some of this literature.

The most-similar noise model to that studied in the present work is the *benign noise* setting (Hanneke, 2009b), which differs from the agnostic setting only in that it makes the additional assumption that $\inf_{h \in \mathbb{C}} \mathrm{er}_P(h) = \inf_h \mathrm{er}_P(h)$, where the infimum on the right hand side is over *all* measurable functions $h : \mathcal{X} \to \{0, 1\}$ (not necessarily in $\mathbb{C}$). In other words, the benign noise setting assumes the best-in-class error $\beta = \inf_{h \in \mathbb{C}} \mathrm{er}_P(h)$ is also the *Bayes risk* of $P$: i.e., the error rate of the function $x \mapsto \mathbb{1}\left[P(Y = 1|X = x) \geq \frac{1}{2}\right]$. Since the distributions used to establish the lower bound (4) for passive learning satisfy the benign noise condition, this still serves as a suitable comparison point for the query complexity of active learning. Similarly, the distributions used to establish the lower bound (6) for active learning also satisfy benign noise, and therefore the lower bound (6) also holds in the benign noise setting. Notably, in the special case of benign noise, Hanneke and Yang (2015) have shown a result analogous to the present work: the optimal first-order query complexity of active learning is $\tilde{O}\left(\mathsf{d}\frac{\beta^2}{\varepsilon^2} + \min\left\{\mathfrak{s}, \frac{\mathsf{d}}{\varepsilon}\right\}\right)$. In particular, comparing to (4), this means, under the benign noise assumption, the query complexity of active learning is always better than the sample complexity of passive learning. That work posed the question of whether such improvements are also attainable in the more-challenging *agnostic* setting, a question which the present work answers positively. Notably, the above result for benign noise is even slightly sharper in the lower-order term, compared to our Theorem 3 (which has $\mathsf{ds}$ rather than $\mathfrak{s}$); I conjecture this $\mathsf{ds}$ can also be reduced to $\mathfrak{s}$ in the agnostic setting. There are interesting connections or analogies between the algorithm used by Hanneke and Yang (2015) and the AVID principle developed in the present work, and we discuss these connections in Appendix A.3 below. However, one noteworthy point is that $\mathbb{A}_{\mathrm{avid}}$ requires vastly fewer *unlabeled* examples to obtain the query complexity guarantee, compared to the method of Hanneke and Yang (2015), so that the present work also offers some benefits over the known techniques for the benign noise setting as well.

### A.3 Background of the AVID Principle

Having surveyed much of the related work on agnostic active learning above, we conclude our discussion of related work by discussing previous works in the learning theory literature containing ideas related to our main technique (the AVID principle).

Arguably the main innovation involved in this work is the decomposition of the space $\mathcal{X}$ into regions $\mathcal{X} \setminus \hat{\Delta}_{i_k}$ and $\Delta_{i_k}$, and augmenting the predictor $\hat{h}_k$ to be a (shallow) *decision list* of concepts from $\mathbb{C}$. One key inspiration for the main idea underlying the technique is rooted in the works of Bousquet and Zhivotovskiy (2021); Puchkin and Zhivotovskiy (2022) on prediction with an *abstention* option (evaluated with the *Chow loss*). Interestingly, this continues a long precedent of finding useful connections and cross-inspirations between active learning and prediction with abstentions (Mitchell, 1979; El-Yaniv and Wiener, 2010, 2012; Zhang and Chaudhuri, 2014, e.g.,). Specifically, Bousquet and Zhivotovskiy (2021); Puchkin and Zhivotovskiy (2022) consider methods exhibiting a kind of *transition time*, in which they determine that, for some $f, g \in \mathbb{C}$, *abstaining* in a the pairwise disagreement region $\{x : f(x) \neq g(x)\}$, and predicting with $f$ in its complement, comes out to have smaller Chow loss than the overall loss of the best $h \in \mathbb{C}$. Some reasoning very much analogous to this (and directly inspired by it) can be found in one of the base cases of the arguments in the present paper (namely, concerning the "early stopping" case in the algorithm), in which we find that in the case of early stopping (Step 4), we can find $f, g \in V_{k-1}$ and $h_1, h_2 \in \mathbb{C}$, such that predicting with $h_1$ in $\{f \neq g\} \setminus \Delta_{i_k}$ (rather than abstaining), with $f$ in $\{f = g\} \setminus \Delta_{i_k}$, and with $h_2$ in $\Delta_{i_k}$, produces a *smaller* overall error rate in compared to the best concept $h^\star \in \mathbb{C}$. Of course, the algorithm and analysis here contain many additional pieces on top of this, but it is interesting that this connection to learning with abstentions still remains present at the core (though it is noteworthy that this connection is qualitatively different from the usual one, in that here we are not replacing abstentions with queries, but rather that a part of the analysis inspires part of our analysis). We remark that this analysis of learning with abstentions by Bousquet and Zhivotovskiy (2021); Puchkin and Zhivotovskiy (2022) was also inspirational for an active learning method in the work of Zhu and Nowak (2022) (though the aim in that work is different from the present work, and the setting is generally not comparable to ours).

At a high level, we can view the technique as also analogous to an idea of Hanneke and Yang (2015) developed for the *benign noise* model: namely, the restriction of the agnostic setting to the case the *Bayes classifier* $h^\star_{\mathrm{Bayes}}(x) \mapsto \mathbb{1}[P(Y = 1|X = x) \geq 1/2]$ is in the concept class $\mathbb{C}$. Hanneke and Yang (2015) prove a query complexity bound for this special case which matches Theorem 3 (and indeed, refines the lower-order term's $\mathfrak{s}d$ dependence to simply $\mathfrak{s}$). In that context, since the $h^\star_{\mathrm{Bayes}} \in \mathbb{C}$, the only source of non-realizability is in the *noisiness* of the conditional label distribution $Y|X$. Thus, if an active learner could *repeatedly query* a given $X_t$ to receive *multiple* conditionally independent samples of $Y_t$ given $X_t$, it could use the majority vote of these samples to effectively *de-noise* the label of $X_t$, thereby identifying $h^\star_{\mathrm{Bayes}}(X_t)$. This strategy only fails if $P(Y = 1|X = X_t)$ is very close to $\frac{1}{2}$, in which case this de-noising would require too many queries to be worthwhile, particularly since such noisy examples have very little effect on the *excess* error rate $\mathrm{er}_P(\hat{h}) - \mathrm{er}_P(h^\star_{\mathrm{Bayes}})$. As such, if the active learner cannot identify the optimal label within some number of queries, it should *abandon* the example $X_t$ and move on. Of course, in the model of active learning studied in this work, and in the work of Hanneke and Yang (2015), an active learner cannot actually obtain multiple conditionally independent copies of the label $Y_t$. However, by appropriate discretization of the space $\mathcal{X}$ based on the structure of the concept class $\mathbb{C}$, Hanneke and Yang (2015) are able to *approximate* this idealized behavior. The resulting algorithm effectively *adapts* to the noisiness of the labels of examples $X_t$ within the equivalence classes induced by this discretization, allocating more queries to the noisier (high-label-variance) regions (and abandoning the regions it finds to be too noisy). In that sense, the high-level idea behind the AVID principle is similar in nature. The goal is to isolate the regions where learning is more challenging, due to higher variance in error difference estimation, and allocate disproportionately more queries to these regions. Of course, in the agnostic case, this is made much more challenging, since the source of non-realizabilityy is not merely label noise, but also model misspecification (i.e., $h^\star_{\mathrm{Bayes}} \notin \mathbb{C}$) so that de-noising the examples may sometimes have little benefit (e.g., it is even possible to have $\beta > 0$ while $P(Y = 1|X) \in \{0, 1\}$). As such, the AVID principle necessarily makes greater use of the structure of the concept class to isolate such regions of high variance in error difference estimation.

It is worth mentioning that other works on active learning have also considered decomposing the space $\mathcal{X}$ into subregions and learning separately in each region (e.g., Cortes, DeSalvo, Gentile, Mohri, and Zhang, 2019a; Cortes, DeSalvo, Gentile, Mohri, and Zhang, 2019b, 2020). However, we note that these works retain the above issue of having a query complexity of the form $c(\beta)\mathsf{d}\frac{\beta^2}{\varepsilon^2}$ for a complexity measure $c(\beta)$ (as discussed in Section 2) such that, in the worst case over distributions $P$

(respecting the $\beta$ constraint) the results become ultimately no smaller than the sample complexity of passive learning.

The idea of decomposing a predictor into a decision list based on pairwise disagreement regions has an even closer parallel in the recent work of Hanneke, Larsen, and Zhivotovskiy (2024b), which removes a log factor from the lead term in the (first-order) sample complexity of *passive* learning, thereby obtaining an optimal lead term of $\Theta\!\left(\frac{\beta}{\varepsilon^2}\left(\mathsf{d}+\log\!\left(\frac{1}{\delta}\right)\right)\right)$. The overall approach in that work is in many ways similar to the technique in the present work, though with some important differences in the actual algorithms. In particular, since the interest in that work is merely removing a factor $\log\!\left(\frac{1}{\beta}\right)$, it essentially suffices for the algorithm to reduce the best-in-class *error rate* in a region $\mathcal{X}\setminus\Delta$ down to $\frac{\beta}{\log(1/\beta)}$ (for $P_X(\Delta)=O(\beta)$), so that a uniform Bernstein inequality for the error rate of ERM implies the desired result in that region $\mathcal{X}\setminus\Delta$, and a uniform convergence analysis of ERM under the conditional distribution given $\Delta$ implies the desired result in the region $\Delta$. In contrast, our interest in the present work is a factor of $\beta$ in the lead term, with a lower-order term of size $\tilde{O}\!\left(\frac{\mathsf{d}}{\varepsilon}\right)$, and to achieve this our algorithm aims to reduce (below $\varepsilon$) the *diameter* of a set $V_k$ of surviving concepts, in a region $\mathcal{X}\setminus\Delta$ (with $P_X(\Delta)=O(\beta)$). We achieve this via uniform estimation of error differences, using an appropriate number of samples from these two regions, while precisely controlling the schedule of decreases of this diameter in the algorithm (in part by increasing the $\Delta$ region as needed to maintain this schedule of diameter decreases). Nevertheless, the essential inspiration and strategy behind these two algorithms are notably related, perhaps indicating that the AVID principle might in fact be a widely useful idea.

# B  Additional Definitions and Notation

We provide additional definitions and notation required for the formal analysis. A fundamental quantity in statistical learning theory is the *VC dimension* (Vapnik and Chervonenkis, 1971), which plays an important role in characterizing the optimal query complexity (and optimal sample complexity of passive learning). It is defined as follows.

**Definition 4.** *For any concept class $\mathbb{C}$, the* VC *dimension of $\mathbb{C}$, denoted by $\mathrm{VC}(\mathbb{C})$, is defined as the supremum $n \in \mathbb{N} \cup \{0\}$ for which there exists a sequence $\{x_1,\ldots,x_n\} \in \mathcal{X}^n$ such that $\{(h(x_1),\ldots,h(x_n)) : h \in \mathbb{C}\} = \{0,1\}^n$ (i.e., all $2^n$ classifications are realizable by $\mathbb{C}$).*

For brevity, in all results, proofs, and discussion below (where $\mathbb{C}$ is clear from the context), we will simply denote by $\mathsf{d} := \mathrm{VC}(\mathbb{C})$. In all statements below, we suppose $\mathsf{d} < \infty$ (see Appendix G). Also note that, by our assumption that $|\mathbb{C}| \geq 3$ (see footnote 1), we always have $\mathsf{d} \geq 1$.

**Additional Notation and Conventions:**  For any distribution $P$ on $\mathcal{X} \times \{0,1\}$, denote by $P_X$ the marginal distribution on $\mathcal{X}$. Throughout, we refer to any sequence $S \in (\mathcal{X} \times \{0,1\})^*$ as a *data set*. For any $x \in \mathbb{R}$, it will be convenient to define $\log(x) = \ln(\max\{x,e\})$, and for $x > 0$ we define $\log(x/0) = x/0 = \infty$ and $0\log(x/0) = 0$. For $a,b \in \mathbb{R} \cup \{\infty\}$, we use $a \wedge b$ or $\min\{a,b\}$ to denote the minimum of $a$ and $b$, and $a \vee b$ or $\max\{a,b\}$ to denote the maximum of $a$ and $b$. We will make use of standard big-$O$ notation ($O$, $\Omega$, $\Theta$ effectively hide universal constant factors, while $\tilde{O}$, $\tilde{\Theta}$ effectively hide log factors) to simplify theorem statements. The precise constant and log factors will always be made explicit in the formal proofs. We also adopt a convention regarding conditional probabilities: all claims involving conditional probabilities given a random variable should be interpreted as holding almost surely (i.e., for a *version* of the conditional probability), such as when claiming that an event holds with conditional probability at least $1 - \delta$ given a random variable $X$. We also continue the notational conventions introduced in Section 4, such as $\mathrm{ER}(h)$, $\mathrm{DIS}(\mathbb{C}')$, $\{f \neq g\}$, overloading set notation to treat $A \subseteq \mathcal{X}$ as notationally interchangeable with its labeled extension $A \times \{0,1\}$, extending notation for set-intersection to allow intersections with sequences, and defining empirical estimates $\hat{P}_S(A) = |S \cap A|/|S|$. See Section 4 for details of these conventions.

**Measurability:**  We remark that, formally speaking, an active learning algorithm can be defined simply as a measurable function $\mathbb{A} : (\mathcal{X} \times \{0,1\})^m \times \mathcal{X} \to \{0,1\}$: that is, taking as input an i.i.d. data set $S = \{(X_i,Y_i)\}_{i\leq m}$ and an independent test point $X$ and evaluating to a prediction $\mathbb{A}(S,X) \in \{0,1\}$. In this view, the number of *queries* is merely bookkeeping, keeping track

of the dependences of this function on the labels $Y_i$. For simplicity of presentation, we have adopted the common colloquialism of referring to the function $\hat{h}$ *returned* by $\mathbb{A}(S)$, which in this view simply refers to the function $\mathbb{A}(S, \cdot)$, so that $\mathrm{er}_P(\hat{h})$ is simply the conditional expectation $\mathbb{E}[\mathbb{1}[\mathbb{A}(S, X) \neq Y]|S]$. The measurability of the algorithms $\mathbb{A}$ defined in this work follows from measurability of the individual operations involved in their execution[10] under the standard measure-theoretic assumptions on $(\mathcal{X}, \mathbb{C})$ specified in footnote 1. To simplify the presentation, we do not explicitly discuss this in the proofs.

## C   The Query Complexity of the AVID Agnostic Algorithm

This section presents a detailed version of Theorem 3, bounding the query complexity of the AVID Agnostic algorithm. Recall the definition of the algorithm and notation from Section 4. Before stating the theorem, we first discuss a few additional technical aspects of the algorithm omitted from the high-level description in Section 4, starting with an explicit specification of the quantities involved. Let $c_0$, $c_1$ be universal constants, defined by Lemmas 7 and 8 of Appendix D. We define[11] $C = \frac{11}{10}$, $C'' = \left(\frac{200C^3}{8-5C^3}\right)^2$, and $C' = \frac{\sqrt{C''}}{16}$. For a given $\varepsilon, \delta \in (0, 1)$ (arguments to $\mathbb{A}_{\mathrm{avid}}$), as in Section 4 we let $N = \left\lceil \log_C\left(\frac{2}{\varepsilon}\right) \right\rceil$, and for $k \in \mathbb{N}$, let $\varepsilon_k = C^{1-k}$, and we then define $m_k := \left\lceil \frac{300C''c_0}{\varepsilon_k} \left( \mathsf{d} \log\left(\frac{C''c_0}{\varepsilon_k}\right) + \log\left(\frac{1}{\delta}\right)\right)\right\rceil$. The algorithm adaptively allocates data subsets $S_k^1$, $S_k^2$, $S_{k,i}^3$, $S_k^4$ during its execution, as described in Section 4.1. Recall that $S_k^1$, $S_{k,i}^3$, and $S_k^4$ are all of size $m_k$ (for any $k, i$ for which they exist). The data subset $S_k^2$ is of size $m_k'$, formally defined as follows. For the value $i_k$ and the set $\Delta_{i_k}$ as defined in the algorithm at the time that $S_k^2$ is allocated (either in Step 2 for some value of $k$, or in Step 9, in which case let $k = N + 1$), letting $\hat{p}_k := 2\hat{P}_{S_k^4}(\Delta_{i_k})$, define $m_k' := \left\lceil \frac{C''c_1^2\hat{p}_k}{\varepsilon_k^2} \left( \mathsf{d} + \log\left(\frac{4(3+N-k)^2}{\delta}\right)\right)\right\rceil$.

We remark that, for simplicity of presentation, we have described the algorithm without explicitly discussing what happens if the algorithm *runs out* of unlabeled examples while allocating examples to subsets $S_k^2$, $S_{k,i}^3$. In this event, the algorithm can simply halt and return an arbitrary predictor $\hat{h}$, as the analysis will account for this event in the $\delta$ failure probability. To avoid excessive clutter, we do not explicitly mention this case in the description of the algorithm or allocation of data subsets used therein (i.e., we explicitly discuss this only in the analysis, and indeed only in the final part of the proof; see the discussion at the start of Appendix E).

The following theorem provides a bound on the query complexity achieved by $\mathbb{A}_{\mathrm{avid}}$ along with a bound on the unlabeled data set size sufficient to achieve it. This result represents a detailed version of the upper bound in Theorem 3 of Section 4 (in particular, Theorems 1 and 3 are immediate implications of this result). The constant factors in the big-$O$ will be made explicit in the formal proof. The proof is given in Appendix E.

**Theorem 5** (Query Complexity of AVID Agnostic). *For any concept class $\mathbb{C}$ with $\mathrm{VC}(\mathbb{C}) < \infty$, letting $\mathsf{d} = \mathrm{VC}(\mathbb{C})$, for every distribution $P$ on $\mathcal{X} \times \{0, 1\}$, letting $\beta = \inf_{h \in \mathbb{C}} \mathrm{er}_P(h)$, for any $\varepsilon, \delta \in (0, 1)$, if the algorithm $\mathbb{A}_{\mathrm{avid}}$ is executed with parameters $(\varepsilon, \delta)$, with any number $m \geq M(\varepsilon, \delta; \beta)$ of*

---

[10]The only part requiring some care in this regard is the definition of $\hat{h}_k$ in (3), where formally we require that, given $V_{k-1}$, $\Delta_{i_k}$, $m_k'$, the function $(S_k^1, S_k^2, x) \mapsto \hat{h}_k(x)$ should be a measurable function; such a measurable function can be shown to exist assuming $\mathbb{C}$ (and therefore $V_{k-1}$) satisfies the conditions of footnote 1, following straightforwardly from arguments of (Dudley, 1999).

[11]For simplicity of presentation, the constant $C$ plays two major roles in the algorithm. First, it controls the schedule of *diameter* guarantees $\varepsilon_k$ in the algorithm. Second, it controls certain constant factors in uniform concentration guarantees employed in the proof (Lemma 10). If we were to separate these roles, into $C_1$ and $C_2$, respectively, the two values exhibit a trade-off. In particular, we can admit a schedule $\varepsilon_k = C_1^{1-k}$ for *any* choice of $1 < C_1 < 2$ by an appropriately large choice of $C''$ (diverging as $C_1 \to 2$) and corresponding $C_2 > 1$ sufficiently close to 1. The source of this 2 limitation is the multiplicative factor in Lemma 20 which, in the limit as $C'' \to \infty$ and $C_2 \to 1$, becomes $\frac{2}{2-C_1}$. We also remark that we have defined constants that enable the cleanest presentation of the algorithm and analysis. We leave the issue of optimizing the constants to minimize the query complexity for future work.

*i.i.d.-P examples, for a value $M(\varepsilon, \delta; \beta)$ (defined in Lemma 24) satisfying*

$$M(\varepsilon, \delta; \beta) = O\left(\frac{\beta + \varepsilon}{\varepsilon^2}\left(\mathsf{d}\log\left(\frac{1}{\varepsilon}\right) + \log\left(\frac{1}{\delta}\right)\right)\right) = \tilde{O}\left(\frac{\beta \mathsf{d}}{\varepsilon^2} + \frac{\mathsf{d}}{\varepsilon}\right),$$

*then with probability at least $1 - \delta$, the returned predictor $\hat{h}$ satisfies $\mathrm{er}_P(\hat{h}) \leq \inf_{h \in \mathbb{C}} \mathrm{er}_P(h) + \varepsilon$ and the algorithm makes a number of queries at most $Q(\varepsilon, \delta; \beta)$ (defined in Lemma 23) satisfying*

$$Q(\varepsilon, \delta; \beta) = O\left(\frac{\beta^2}{\varepsilon^2}\left(\mathsf{d} + \log\left(\frac{1}{\delta}\right)\right) + \min\left\{\mathfrak{s}\log\left(\frac{1}{\varepsilon}\right), \frac{1}{\varepsilon}\right\}\left(\mathsf{d}\log\left(\frac{1}{\varepsilon}\right) + \log\left(\frac{1}{\delta}\right)\right)\right)$$

$$= \tilde{O}\left(\frac{\beta^2 \mathsf{d}}{\varepsilon^2} + \left(\mathfrak{s} \wedge \frac{1}{\varepsilon}\right)\mathsf{d}\right).$$

**Remark on adaptivity to $\beta$:** We emphasize that the algorithm *does not need to know $\beta$* in its execution (i.e., it *adaptively* achieves the above query complexity bound *for all $\beta$*). A more subtle point worth noting is that we can also *run* the algorithm without *ourselves* knowing $\beta$ (to choose $m$), since the guarantee on query complexity holds for *any* unlabeled sample size $m \geq M(\varepsilon, \delta; \beta)$. For instance, if we run the algorithm with a $\beta$-independent number of unlabeled examples $m = \tilde{\Theta}\left(\frac{1}{\varepsilon^2}\left(\mathsf{d}\log\left(\frac{1}{\varepsilon}\right) + \log\left(\frac{1}{\delta}\right)\right)\right)$, the query complexity bound $Q(\varepsilon, \delta; \beta)$ would remain valid as stated in Theorem 5. Additionally, in the proof (see Lemma 24), we show that, in a sense, even the *unlabeled* sample complexity $M(\varepsilon, \delta; \beta)$ is achieved *adaptively*, since the algorithm (with no knowledge of $\beta$) only actually uses the first (at most) $M(\varepsilon, \delta; \beta)$ unlabeled examples in the sequence. This is itself an interesting feature. In particular, if we consider an alternative setting where, rather than getting the unlabeled data altogether at the start, the algorithm can adaptively *sample* new unlabeled examples $X_i \sim P_X$ one-at-a-time during execution (i.e., it has access to an *unlabeled example oracle*, which it can use to construct the data subsets $S_k^1, S_k^2, S_{k,i}^3, S_k^4$, during execution), the analysis establishes that the algorithm will succeed while adaptively sampling at most $M(\varepsilon, \delta; \beta)$ unlabeled examples (and querying at most $Q(\varepsilon, \delta; \beta)$ of them), all *without* knowing $\beta$ (or anything else about $P$).

## D    Concentration Inequalities

This section presents a number of useful concentration inequalities, essential to the analysis. We begin with the classic *multiplicative Chernoff bound* (Chernoff, 1952; Bernstein, 1924). We will find the following particular form to be useful; since this is slightly different from the more-typical statements of Chernoff bounds, we include a brief explanation of how this result is derived from the more-standard exponential form.

**Lemma 6** (Multiplicative Chernoff bound). *Fix any $p \in [0, 1]$ and $n \in \mathbb{N}$, and let $B_1, \ldots, B_n$ be i.i.d.* Bernoulli($p$) *random variables. Let $\bar{B} := \frac{1}{n}\sum_{i=1}^n B_i$. For any $\delta \in (0, 1)$, with probability at least $1 - \delta$, the following both hold:*

$$p \leq \max\left\{2\bar{B}, \frac{8}{n}\ln\left(\frac{2}{\delta}\right)\right\},$$

$$\bar{B} \leq \max\left\{2p, \frac{6}{n}\ln\left(\frac{2}{\delta}\right)\right\}.$$

**Proof.** We include a brief explanation, based on more well-known exponential forms of the Chernoff bound: namely, $\mathbb{P}(\bar{B} < (1/2)p) \leq e^{-np/8}$ and $\mathbb{P}(\bar{B} > 2p) \leq e^{-np/3}$ (see e.g., Zhang, 2023).

For the first inequality in the lemma, we note that it trivially holds if $p < \frac{8}{n}\ln\left(\frac{2}{\delta}\right)$, and otherwise, if $p \geq \frac{8}{n}\ln\left(\frac{2}{\delta}\right)$, then by the above exponential tail bound, we have $\mathbb{P}(\bar{B} < (1/2)p) \leq e^{-np/8} \leq \frac{\delta}{2}$. For the second claimed inequality, note that it trivially holds if $\frac{6}{n}\ln\left(\frac{2}{\delta}\right) \geq 1$, so let us focus on the case $\frac{6}{n}\ln\left(\frac{2}{\delta}\right) < 1$. Note that for $p' \in [0, 1]$ and $B_1', \ldots, B_n'$ i.i.d. Bernoulli($p'$), and $\bar{B}' = \frac{1}{n}\sum_{i=1}^n B_i'$, for any $x \in \mathbb{R}$ the value of $\mathbb{P}(\bar{B}' > x)$ is non-decreasing in $p'$. Thus, letting $p' = \max\{p, \frac{3}{n}\ln\left(\frac{2}{\delta}\right)\} \geq p$, this monotonicity (together with the second exponential tail bound above) implies $\mathbb{P}(\bar{B} > 2p') \leq \mathbb{P}(\bar{B}' > 2p') \leq e^{-np'/3} \leq \frac{\delta}{2}$. The lemma then follows by the union bound, so that both of these inequalities hold simultaneously with probability at least $1 - \delta$. ∎

We will also rely heavily on *uniform* concentration inequalities. Toward stating these, we first introduce additional useful notation.

**VC dimension of collections of sets:** As is standard in the literature, we overload the definition of *VC dimension* (Vapnik and Chervonenkis, 1971) to also allow for collections of sets. Formally, for any non-empty set $\mathcal{Z}$ and any non-empty $\mathcal{A} \subseteq 2^{\mathcal{Z}}$ (i.e., a collection of subsets of $\mathcal{Z}$), the VC dimension of $\mathcal{A}$, denoted by $\mathrm{VC}(\mathcal{A})$, is the supremum $n \in \mathbb{N} \cup \{0\}$ for which there exists $Z \subseteq \mathcal{Z}$ with $|Z| = n$ such that $\{Z \cap A : A \in \mathcal{A}\} = 2^Z$ (i.e., it is possible to pick out any subset of $Z$ by intersection with an appropriate $A \in \mathcal{A}$). Equivalently, $\mathrm{VC}(\mathcal{A})$ is the VC dimension (Definition 4) of the *indicator functions* $\{\mathbb{1}_A : A \in \mathcal{A}\}$.

**Uniform concentration term:** For any non-empty set $\mathcal{Z}$ and any non-empty $\mathcal{A} \subseteq 2^{\mathcal{Z}}$, for any $n \in \mathbb{N}$ and $\delta \in (0,1)$, define (for a universal constant $c_0$ defined by Lemma 7 below)

$$\varepsilon(n,\delta;\mathcal{A}) := \frac{c_0}{n}\left(\mathrm{VC}(\mathcal{A})\log\left(\frac{n}{\mathrm{VC}(\mathcal{A})}\right) + \log\left(\frac{1}{\delta}\right)\right). \tag{7}$$

The following result represents a uniform variant of the classic *Bernstein inequality* (or Bennett inequality) (Bernstein, 1924; Bennett, 1962). It can be derived from results proven by Vapnik and Chervonenkis (1974) (see Hanneke and Kpotufe, 2022 for an explicit derivation, via a layered application of Massart's lemma and Bousquet's inequality). We additionally include implications providing a uniform variant of multiplicative Chernoff bounds, which are easily derived from the stated uniform Bernstein inequality (taking $B = \emptyset$).

**Lemma 7** (Uniform Bernstein and multiplicative Chernoff bounds)**.** *There is a finite universal constant $c_0 > 1$ for which the following holds. Fix any $n \in \mathbb{N}$, $\delta \in (0,1)$, any non-empty set $\mathcal{Z}$, and any set $\mathcal{A} \subseteq 2^{\mathcal{Z}}$ with $\mathrm{VC}(\mathcal{A}) < \infty$.[12] Define $\varepsilon(n,\delta;\mathcal{A})$ as in (7). Fix any distribution $P$ on $\mathcal{Z}$ and let $Z = \{Z_1, \ldots, Z_n\} \sim P^n$ (i.i.d. $P$ random variables). For any measurable set $A \subseteq \mathcal{Z}$, define its* empirical probability $\hat{P}_Z(A) := \frac{1}{n}\sum_{i=1}^n \mathbb{1}[Z_i \in A]$. *With probability at least $1 - \delta$, every $A, B \in \mathcal{A} \cup \{\emptyset\}$ satisfy the following (where $A \oplus B := (A \setminus B) \cup (B \setminus A)$ denotes the symmetric difference)*

$$\left|(\hat{P}_Z(A) - \hat{P}_Z(B)) - (P(A) - P(B))\right|$$
$$\leq \sqrt{\min\{P(A \oplus B), \hat{P}_Z(A \oplus B)\}\varepsilon(n,\delta;\mathcal{A})} + \varepsilon(n,\delta;\mathcal{A}).$$

*Moreover, for any $\varepsilon > 0$ and $\alpha \in (0,1)$ satisfying $\varepsilon(n,\delta;\mathcal{A}) \leq \frac{\alpha^2}{4}\varepsilon$, the above inequality immediately yields the following implications: $\forall A \in \mathcal{A}$,*

$$\hat{P}_Z(A) \geq \varepsilon \implies P(A) > (1-\alpha)\varepsilon, \quad \textit{or equivalently, } P(A) \leq (1-\alpha)\varepsilon \implies \hat{P}_Z(A) < \varepsilon$$
$$P(A) \geq \varepsilon \implies \hat{P}_Z(A) > (1-\alpha)\varepsilon, \quad \textit{or equivalently, } \hat{P}_Z(A) \leq (1-\alpha)\varepsilon \implies P(A) < \varepsilon.$$

We also make use of a uniform concentration inequality which refines the classic uniform convergence bound $\sqrt{\frac{1}{n}\left(\mathrm{VC}(\mathcal{A}) + \log\left(\frac{1}{\delta}\right)\right)}$ of Talagrand (1994) in the case that $\bigcup \mathcal{A}$ has small measure under $P$. The lemma is well-known in the literature, and follows immediately from expectation bounds based on chaining involving an *envelope* function (e.g., Theorem 2.14.1 of van der Vaart and Wellner, 1996) together with Bousquet's inequality (Bousquet, 2002) to achieve high probability. For completeness, we provide a brief direct proof, by simply applying the uniform convergence bound of Talagrand (1994) to the samples from the conditional distribution given a set $D \supseteq \bigcup \mathcal{A}$.

**Lemma 8.** *There is a finite universal constant $c_1 \geq 1$ for which the following holds. Let $\mathcal{A}$ be as in Lemma 7, and suppose $D \subseteq \mathcal{Z}$ is a measurable set such that $\forall A \in \mathcal{A}$, $A \subseteq D$. Then for the same quantities as Lemma 7, if $P(D) \geq \frac{9}{n}\ln\left(\frac{4}{\delta}\right)$, then with probability at least $1 - \delta$, $\forall A \in \mathcal{A}$*

$$\left|\hat{P}_Z(A) - P(A)\right| \leq c_1\sqrt{\frac{P(D)}{n}\left(\mathrm{VC}(\mathcal{A}) + \log\left(\frac{1}{\delta}\right)\right)}.$$

---

[12]We suppose standard mild measure-theoretic restrictions on $\mathcal{A}$ and the $\sigma$-algebra of $\mathcal{Z}$, from empirical process theory: namely, the image-admissible Suslin condition (Dudley, 1999).

**Proof.** Note that the samples in $Z \cap D$ are conditionally i.i.d. $P(\cdot|D)$ given $|Z \cap D|$. For each $A \in \mathcal{A}$, denote by $\hat{P}_Z(A|D) := \hat{P}_{Z \cap D}(A)$ (or $0$ if $|Z \cap D| = 0$). Applying the uniform convergence bound of Talagrand (1994) to the samples in $Z \cap D$ under the conditional distribution given $|Z \cap D|$, together with the law of total probability, yields that, with probability at least $1 - \frac{\delta}{2}$, $\forall A \in \mathcal{A}$,

$$\left| \hat{P}_Z(A|D) - P(A|D) \right| \leq c_1' \sqrt{\frac{1}{|Z \cap D|} \left( \text{VC}(\mathcal{A}) + \log\left(\frac{2}{\delta}\right) \right)}, \tag{8}$$

for a finite universal constant $c_1' \geq 1$. Moreover, by Bernstein's inequality (see Theorem 2.10 of Boucheron, Lugosi, and Massart, 2013), with probability at least $1 - \frac{\delta}{2}$,

$$\left| \hat{P}_Z(D) - P(D) \right| \leq \sqrt{\frac{2P(D)}{n} \ln\left(\frac{4}{\delta}\right)} + \frac{1}{n} \ln\left(\frac{4}{\delta}\right) \leq 2\sqrt{\frac{P(D)}{n} \ln\left(\frac{4}{\delta}\right)}, \tag{9}$$

where the last inequality is due to the assumption that $P(D) \geq \frac{9}{n} \ln\left(\frac{4}{\delta}\right)$. By the union bound, these two events occur simultaneously with probability at least $1 - \delta$. Suppose this occurs. In particular, by the assumption that $P(D) \geq \frac{9}{n} \ln\left(\frac{4}{\delta}\right)$, (9) further implies

$$\frac{1}{n} |Z \cap D| = \hat{P}_Z(D) \geq P(D) - 2\sqrt{\frac{P(D)}{n} \ln\left(\frac{4}{\delta}\right)} \geq \frac{1}{3} P(D),$$

so that the right hand side of (8) is at most

$$c_1' \sqrt{\frac{3}{nP(D)} \left( \text{VC}(\mathcal{A}) + \log\left(\frac{2}{\delta}\right) \right)}.$$

Combining this with (8) and (9) implies that $\forall A \in \mathcal{A}$, since $A \subseteq D$,

$$\left| \hat{P}_Z(A) - P(A) \right| = \left| \hat{P}_Z(D) \hat{P}_Z(A|D) - P(D) P(A|D) \right|$$

$$\leq P(D) \left| \hat{P}_Z(A|D) - P(A|D) \right| + \hat{P}_Z(A|D) 2\sqrt{\frac{P(D)}{n} \ln\left(\frac{4}{\delta}\right)}$$

$$\leq c_1' \sqrt{\frac{3P(D)}{n} \left( \text{VC}(\mathcal{A}) + \log\left(\frac{2}{\delta}\right) \right)} + 2\sqrt{\frac{P(D)}{n} \ln\left(\frac{4}{\delta}\right)}$$

$$\leq c_1 \sqrt{\frac{P(D)}{n} \left( \text{VC}(\mathcal{A}) + \log\left(\frac{1}{\delta}\right) \right)},$$

where $c_1 := c_1' \sqrt{6} + 2\sqrt{\ln(4e)}$ (recalling $\log(x) := \ln(x \vee e)$). ∎

## E   Proof of Theorem 5: Query Complexity of the AVID Agnostic Algorithm

The formal proof of Theorem 5, given at the end of this section, will be built up from a sequence of lemmas, roughly following the outline presented in Section 4.2.

Throughout this section, we fix an arbitrary concept class $\mathbb{C}$ (with $\mathsf{d} := \text{VC}(\mathbb{C}) < \infty$) and distribution $P$ on $\mathcal{X} \times \{0, 1\}$, let $\beta = \inf_{h \in \mathbb{C}} \text{er}_P(h)$, fix any $\varepsilon, \delta \in (0, 1)$ (where $\varepsilon, \delta$ are inputs to the AVID algorithm), let $(X_1, Y_1), (X_2, Y_2), \ldots$ be independent $P$-distributed examples, and we let all values $(N, \varepsilon_k, m_k, m_k'$, etc.) be defined as in Appendix C, based on these values $\varepsilon, \delta$, and the examples $(X_1, Y_1), (X_2, Y_2), \ldots$. Also let $h^\star \in \mathbb{C}$ denote any concept with

$$\text{er}_P(h^\star) < \inf_{h \in \mathbb{C}} \text{er}_P(h) + \frac{\varepsilon}{10^4}. \tag{10}$$

For full generality, we do not assume there exists a minimizer achieving the infimum on the right hand side; rather, any choice of $h^\star$ satisfying this *near*-minimality property will suffice for our purposes in the analysis below.

To simplify the proof, we will establish the sequence of lemmas under a scenario where the algorithm is executed with an *inexhaustible* source of examples (for the adaptive allocation of data subsets): i.e., an infinite sequence $(X_1, Y_1), (X_2, Y_2), \ldots$ of independent $P$-distributed examples. However, it will follow from these lemmas that, with high probability, the algorithm only depends on a *finite* prefix $(X_1, Y_1), \ldots, (X_m, Y_m)$, for a sufficiently large $m = M(\varepsilon, \delta; \beta)$ as in Theorem 5 (see Lemma 24). At the end of the section, when combining the lemmas into a formal proof of Theorem 5, we will return to the standard setting where the algorithm has access *only* to such a finite prefix. In that context, the event that the algorithm attempts to access any examples $(X_t, Y_t)$ with $t > m$ will be accounted for as part of the allowed $\delta$-probability failure event, and thus (as mentioned in Appendix C) in such a case the algorithm can simply halt and return an arbitrary predictor $\hat{h}$. As mentioned in the remark following Theorem 5, the fact that the algorithm *adaptively* decides how many unlabeled examples to use is itself an interesting feature, as it means the algorithm can be considered adaptive to $\beta$ even in its use of unlabeled examples.

Before proceeding with the proof, we first introduce some convenient notation regarding the values of $k$ and $i$ encountered in the algorithm. If the algorithm returns in Step 9, denote by $K := N + 1$, and otherwise, let $K$ be the maximum value of $k$ reached in the '*For*' loop in the algorithm; we argue in Lemma 10 below that the algorithm terminates eventually, with high probability, so that this latter case coincides with the case of returning in Step 4, with $K$ being the value of $k$ on which this occurs. Let $\mathcal{K} := \{1, \ldots, K \wedge N\}$: that is, the set of values of $k$ encountered in the '*For*' loop in the algorithm. Also, for each $k \in \mathcal{K}$, denote by $\mathcal{I}_k$ the values of $i$ encountered by the algorithm on round $k$; in particular, for $k < K$, $\mathcal{I}_k = \{i_k, \ldots, i_{k+1}\}$. In the case $K = N + 1$, for convenience also denote by $\mathcal{I}_{N+1} := \{i_{N+1}\}$.

We begin with a lemma which motivates our choice of sample size $m_k$ for $S_k^1$, $S_{k,i}^3$, $S_k^4$. Recall $m_k := \left\lceil \frac{300 C'' c_0}{\varepsilon_k} \left( \mathsf{d} \log\left( \frac{C'' c_0}{\varepsilon_k} \right) + \log\left( \frac{1}{\delta} \right) \right) \right\rceil$. Also recall our convention (adopted throughout this work) of treating sets $D \subseteq \mathcal{X}$ as notationally interchangeable with their labeled extension $D \times \{0, 1\}$, such as in $A \cap D$ or $A \setminus D$ for $A \subseteq \mathcal{X} \times \{0, 1\}$.

**Lemma 9.** *Fix any set $D \subseteq \mathcal{X}$ and define a family of subsets of $\mathcal{X} \times \{0, 1\}$:*

$$\mathcal{A} = \Big\{ ((\mathrm{ER}(f) \cap \{f = g\}) \cup (\mathrm{ER}(h) \cap \{f \neq g\})) \setminus D : f, g, h \in \mathbb{C} \Big\}$$

$$\cup \Big\{ (\{f \neq g\} \times \{0, 1\}) \setminus D : f, g \in \mathbb{C} \Big\} \cup \Big\{ \mathrm{ER}(h) \setminus D : h \in \mathbb{C} \Big\}.$$

*For any $n \in \mathbb{N}$ and $\delta' \in (0, 1)$, let $\varepsilon(n, \delta'; \mathcal{A})$ be defined as in (7). For each $k \in \{1, \ldots, N + 1\}$, letting $\delta_k := \frac{\delta \varepsilon_{k+3}^2}{72}$, it holds that*

$$\varepsilon(m_k, \delta_k; \mathcal{A}) < \frac{\varepsilon_k}{C''}. \tag{11}$$

**Proof.** We begin by bounding $\mathrm{VC}(\mathcal{A})$, as needed to evaluate $\varepsilon(m_k, \delta_k; \mathcal{A})$. Define the following families of subsets of $\mathcal{X} \times \{0, 1\}$:

$$\mathcal{A}_0 := \{\mathrm{ER}(h) : h \in \mathbb{C}\} \cup \{\emptyset, \mathcal{X} \times \{0, 1\}\},$$
$$\mathcal{A}_1 := \{\{f \neq g\} \times \{0, 1\} : f, g \in \mathbb{C}\} \cup \{\mathcal{X} \times \{0, 1\}\},$$
$$\mathcal{A}_2 := \{((A \setminus C) \cup (B \cap C)) \setminus D : A, B \in \mathcal{A}_0, C \in \mathcal{A}_1\}.$$

First note that $\mathcal{A} \subseteq \mathcal{A}_2$. To see this, note that for any $f, g, h \in \mathbb{C}$, taking $A = \mathrm{ER}(f)$, $B = \mathrm{ER}(h)$, $C = \{f \neq g\} \times \{0, 1\}$, we have that $((\mathrm{ER}(f) \cap \{f = g\}) \cup (\mathrm{ER}(h) \cap \{f \neq g\})) \setminus D = ((A \setminus C) \cup (B \cap C)) \setminus D \in \mathcal{A}_2$. Similarly, for any $f, g \in \mathbb{C}$, taking $A = \emptyset$, $B = \mathcal{X} \times \{0, 1\}$, $C = \{f \neq g\} \times \{0, 1\}$ reveals $(\{f \neq g\} \times \{0, 1\}) \setminus D = ((A \setminus C) \cup (B \cap C)) \setminus D \in \mathcal{A}_2$. Finally, for $h, f \in \mathbb{C}$, taking $A = \mathrm{ER}(h)$, $B = \emptyset$, $C = \{f \neq f\} \times \{0, 1\} = \emptyset$ reveals $\mathrm{ER}(h) \setminus D = ((A \setminus C) \cup (B \cap C)) \setminus D \in \mathcal{A}_2$.

Next we bound $\mathrm{VC}(\mathcal{A}_2)$. It is immediate from the definition that $\mathrm{VC}(\{\mathrm{ER}(h) : h \in \mathbb{C}\}) = \mathsf{d}$. Moreover, this implies $\mathrm{VC}(\mathcal{A}_0) \leq \mathsf{d} + 2$ (Vidyasagar, 2003, Lemma 4.11). Also note that $\mathcal{A}_1 \subseteq \{A \oplus B : A, B \in \mathcal{A}_0\}$, where $A \oplus B := (A \setminus B) \cup (B \setminus A)$ is the symmetric difference: that is, trivially $(\mathcal{X} \times \{0, 1\}) \oplus \emptyset = \mathcal{X} \times \{0, 1\}$, and for any $f, g \in \mathbb{C}$, $\{f \neq g\} \times \{0, 1\} = \mathrm{ER}(f) \oplus \mathrm{ER}(g)$. Thus, any element of $\mathcal{A}_2$ can be expressed as a fixed function of four sets $A, B, A', B' \in \mathcal{A}_0$: namely $(A, B, A', B') \mapsto ((A \setminus (A' \oplus B')) \cup (B \cap (A' \oplus B'))) \setminus D$. Based on this fact, well-known results about the effect of such combinations on the VC dimension imply $\mathrm{VC}(\mathcal{A}_2) = O(\mathrm{VC}(\mathcal{A}_0))$: explicitly, Theorem 4.5 of Vidyasagar (2003) implies $\mathrm{VC}(\mathcal{A}_2) \leq 25 \mathrm{VC}(\mathcal{A}_0) \leq 25(\mathsf{d} + 2)$. By the

assumption that $|\mathbb{C}| \geq 3$ (footnote 1) we know $\mathsf{d} \geq 1$, so that $25(\mathsf{d}+2) \leq 75\mathsf{d}$. Altogether, we have $\mathrm{VC}(\mathcal{A}) \leq 75\mathsf{d}$.

With this in mind, we may note that (also using that $\mathsf{d} \geq 1$ and $C'' \geq 9C^3$)

$$
\begin{aligned}
m_k &\geq \frac{300C''c_0}{\varepsilon_k}\left(\mathsf{d}\log\left(\frac{C''c_0}{\varepsilon_k}\right) + \log\left(\frac{1}{\delta}\right)\right) \\
&\geq \frac{150C''c_0}{\varepsilon_k}\left(\mathsf{d}\log\left(\frac{C''c_0}{\varepsilon_k}\right) + \log\left(\frac{9C^3}{\delta\varepsilon_k}\right)\right) \\
&\geq \frac{2C''c_0}{\varepsilon_k}\left(\mathrm{VC}(\mathcal{A})\log\left(\frac{C''c_0}{\varepsilon_k}\right) + \log\left(\frac{1}{\delta_k}\right)\right). 
\end{aligned} \tag{12}
$$

In particular, if $\mathrm{VC}(\mathcal{A}) \geq 1$, then by Corollary 4.1 of Vidyasagar (2003), (12) implies

$$
m_k > \frac{C''c_0}{\varepsilon_k}\left(\mathrm{VC}(\mathcal{A})\log\left(\frac{m_k}{\mathrm{VC}(\mathcal{A})}\right) + \log\left(\frac{1}{\delta_k}\right)\right). \tag{13}
$$

Moreover, if $\mathrm{VC}(\mathcal{A}) = 0$, then recalling we define $0\log(1/0) = 0$, (12) trivially implies (13) in this case as well. Thus, regardless of the value of $\mathrm{VC}(\mathcal{A})$, by definition of $\varepsilon(m_k, \delta_k; \mathcal{A})$, the claim in (11) follows from (13). ∎

We continue the proof with a lemma conveniently summarizing several uniform concentration bounds which are useful in various places throughout the rest of the proof. In particular, the lemma focuses on concentration inequalities in the $\mathcal{X} \setminus \Delta_{i_k}$ region of focus of the learning algorithm. It will therefore be convenient to explicitly define the portion of the functions in $V_{k-1}^{(4)}$ specific to this region: namely, for every $k \in \{1, \dots, K\}$, define[13]

$$
V_{k-1}^{(3)} := \{f\mathbb{1}_{\{f=g\}} + h\mathbb{1}_{\{f \neq g\}} : f, g \in V_{k-1}, h \in \mathbb{C}\}.
$$

**Lemma 10.** *On an event $E_0$ of probability at least $1 - \frac{\delta}{4}$, for every $k \in \{1, \dots, K\}$, it holds that*

$$
\forall h, h' \in V_{k-1}^{(3)}, \ \left|\left(\hat{P}_{S_k^1}(\mathrm{ER}(h) \cap D_{k-1} \setminus \Delta_{i_k}) - \hat{P}_{S_k^1}(\mathrm{ER}(h') \cap D_{k-1} \setminus \Delta_{i_k})\right)\right.
$$
$$
\left. - (P(\mathrm{ER}(h) \setminus \Delta_{i_k}) - P(\mathrm{ER}(h') \setminus \Delta_{i_k}))\right|
$$
$$
< \sqrt{P_X(\{h \neq h'\} \setminus \Delta_{i_k})\frac{\varepsilon_k}{C''}} + \frac{\varepsilon_k}{C''}, \tag{14}
$$

*and for every $k \in \mathcal{K}$ and every $i \in \mathcal{I}_k$, $\forall f, g \in \mathbb{C}$,*

$$
\hat{P}_{S_{k,i}^3}(\{f \neq g\} \setminus \Delta_i) \geq \varepsilon_{k+2} \implies P_X(\{f \neq g\} \setminus \Delta_i) > \varepsilon_{k+3} \tag{15}
$$
$$
\hat{P}_{S_{k,i}^3}(\{f \neq g\} \setminus \Delta_i) \leq \varepsilon_{k+2} \implies P_X(\{f \neq g\} \setminus \Delta_i) < \varepsilon_{k+1}, \tag{16}
$$

*and moreover, $\max \mathcal{I}_k \leq \frac{1}{\varepsilon_{k+3}}$. In particular, the latter implies the algorithm eventually terminates (in Step 9 if $K = N + 1$, or in Step 4 if $K \leq N$).*

**Proof.** Consider any $k \in \{1, \dots, N+1\}$ having a non-zero probability of $k \leq K$. Let $\delta_k$ be as in Lemma 9. Recall that the data set $S_k^1$ is independent of all data involved in rounds $k' < k$ in the algorithm, whereas the event $k \leq K$ and (in this event) the set $\Delta_{i_k}$ are entirely determined by data involved in rounds $k' < k$. Thus, even conditioned on the event that $k \leq K$ and and the set $\Delta_{i_k}$, the data set $S_k^1$ remains conditionally i.i.d.-$P$. Therefore, letting $\mathcal{A}_k$ denote the set $\mathcal{A}$ as defined in Lemma 9 with $D = \Delta_{i_k}$, applying the uniform Bernstein inequality (Lemma 7 in Appendix D) with this $\mathcal{A}_k$ under the conditional distribution given the event $k \leq K$ and the set $\Delta_{i_k}$ implies that, with conditional probability at least $1 - \delta_k$ given the event $k \leq K$ and the set $\Delta_{i_k}$, it holds that $\forall A, B \in \mathcal{A}_k$,

$$
\left|\left(\hat{P}_{S_k^1}(A) - \hat{P}_{S_k^1}(B)\right) - \left(P(A) - P(B)\right)\right| \leq \sqrt{P(A \oplus B)\varepsilon(m_k, \delta_k; \mathcal{A}_k)} + \varepsilon(m_k, \delta_k; \mathcal{A}_k). \tag{17}
$$

---

[13]Since this work focuses on binary classification, $V_{k-1}^{(3)}$ can equivalently be stated as $\{\mathrm{Maj}(f, g, h) : f, g \in V_{k-1}, h \in \mathbb{C}\}$, where $\mathrm{Maj}(f, g, h)(x) = \mathbb{1}[f(x) + g(x) + h(x) \geq 2]$ is the majority vote function. The definition of $V_{k-1}^{(3)}$ above expresses a more-general form, which, as we discuss in Section G, also extends to *multiclass* classification.

By the law of total probability, on an event $E_{0,k}$ of probability at least $1 - \delta_k$, if $k \leq K$ (and thus $\Delta_{i_k}$ and $\mathcal{A}_k$ are defined) then (17) holds $\forall A, B \in \mathcal{A}_k$.

In particular, on the event $E_{0,k}$, supposing $k \leq K$, if we consider any $h, h' \in V_{k-1}^{(3)}$, then for the sets $A = \mathrm{ER}(h) \setminus \Delta_{i_k} \in \mathcal{A}_k$ and $B = \mathrm{ER}(h') \setminus \Delta_{i_k} \in \mathcal{A}_k$, we may note that the symmetric difference $A \oplus B = (\mathrm{ER}(h) \oplus \mathrm{ER}(h')) \setminus \Delta_{i_k} = (\{h \neq h'\} \times \{0,1\}) \setminus \Delta_{i_k}$, so that together with (11) of Lemma 9, (17) implies

$$\left| \left( \hat{P}_{S_k^1}(\mathrm{ER}(h) \setminus \Delta_{i_k}) - \hat{P}_{S_k^1}(\mathrm{ER}(h') \setminus \Delta_{i_k}) \right) - \left( P(\mathrm{ER}(h) \setminus \Delta_{i_k}) - P(\mathrm{ER}(h') \setminus \Delta_{i_k}) \right) \right|$$
$$< \sqrt{P_X(\{h \neq h'\} \setminus \Delta_{i_k}) \frac{\varepsilon_k}{C''}} + \frac{\varepsilon_k}{C''}. \tag{18}$$

To arrive at the claim in (14), we merely note that for any $f, g, f', g' \in V_{k-1}$ and $h, h' \in \mathbb{C}$, letting $\mathrm{DL}(f, g, h) := f\mathbb{1}_{\{f=g\}} + h\mathbb{1}_{\{f \neq g\}}$ and $\mathrm{DL}(f', g', h') := f'\mathbb{1}_{\{f'=g'\}} + h'\mathbb{1}_{\{f' \neq g'\}}$, for any $x \in \mathcal{X} \setminus D_{k-1}$, we have $g(x) = f(x) = f'(x) = g'(x)$, so that $\mathrm{DL}(f, g, h)(x) = f(x) = f'(x) = \mathrm{DL}(f', g', h')(x)$. Thus, any $h, h' \in V_{k-1}^{(3)}$ have $h(x) = h'(x)$ for all $x \notin D_{k-1}$, and therefore

$$\hat{P}_{S_k^1}(\mathrm{ER}(h) \cap D_{k-1} \setminus \Delta_{i_k}) - \hat{P}_{S_k^1}(\mathrm{ER}(h') \cap D_{k-1} \setminus \Delta_{i_k}) = \hat{P}_{S_k^1}(\mathrm{ER}(h) \setminus \Delta_{i_k}) - \hat{P}_{S_k^1}(\mathrm{ER}(h') \setminus \Delta_{i_k}),$$

so that (14) follows from (18). To unify the discussion below, for any $k \in \{1, \ldots, N+1\}$ with probability zero of $k \leq K$, also denote by $E_{0,k}$ the event (of probability one) that $k > K$.

Turning now to the claims in (15) and (16), consider any $(k, i)$ having non-zero probability that $k \in \mathcal{K}$ and $i \in \mathcal{I}_k$. Note that, since $S_{k,i}^3$ is a data set of size $m_k$, allocated from the remaining *unused* unlabeled data upon reaching Step 5 with values $(k, i)$ (noting this can happen at most once in the algorithm), the samples in $S_{k,i}^3$ are conditionally i.i.d.-$P$ given $k \in \mathcal{K}$ and $i \in \mathcal{I}_k$, and moreover, $S_{k,i}^3$ is conditionally independent of $\Delta_i$ given the events that $k \in \mathcal{K}$ and $i \in \mathcal{I}_k$. In the event that $k \in \mathcal{K}$ and $i \in \mathcal{I}_k$, let $\mathcal{A}_{k,i}$ denote the set $\mathcal{A}$ as defined in Lemma 9 with $D = \Delta_i$. Recalling again our definition of $C = \frac{11}{10}$ and $C'' \geq 32C^5 \left( \frac{C}{C-1} \right)^2$, note that for $\alpha = 1 - \frac{1}{C}$, (11) of Lemma 9 implies $\varepsilon(m_k, \delta_k; \mathcal{A}_{k,i}) < \frac{\varepsilon_k}{C''} < \frac{\alpha^2}{4}\varepsilon_{k+2} < \frac{\alpha^2}{4}\varepsilon_{k+1}$. Therefore, applying Lemma 7 of Appendix D under the conditional distribution given the events that $k \in \mathcal{K}$ and $i \in \mathcal{I}_k$ and the set $\Delta_i$, we have that with conditional probability at least $1 - \delta_k$, $\forall f, g \in \mathbb{C}$, the set $(\{f \neq g\} \setminus \Delta_i) \times \{0,1\} \in \mathcal{A}_{k,i}$ satisfies

$$\hat{P}_{S_{k,i}^3}(\{f \neq g\} \setminus \Delta_i) \geq \varepsilon_{k+2} \implies P_X(\{f \neq g\} \setminus \Delta_i) > (1 - \alpha)\varepsilon_{k+2} = \varepsilon_{k+3}$$

and

$$\hat{P}_{S_{k,i}^3}(\{f \neq g\} \setminus \Delta_i) \leq \varepsilon_{k+2} = (1 - \alpha)\varepsilon_{k+1} \implies P_X(\{f \neq g\} \setminus \Delta_i) < \varepsilon_{k+1}.$$

By the law of total probability, there is an event $E_{0,k,i}$ of probability at least $1 - \delta_k$, on which, if $k \in \mathcal{K}$ and $i \in \mathcal{I}_k$, then the above inequalities hold $\forall f, g \in \mathbb{C}$. To unify cases, for any $(k, i)$ with $k \leq N$ and $i \leq 1/\varepsilon_{k+3}$ having probability zero of satisfying $k \in \mathcal{K}$ and $i \in \mathcal{I}_k$, also define $E_{0,k,i}$ as the event (of probability one) that either $k \notin \mathcal{K}$ or $i \notin \mathcal{I}_k$.

We have thus established (14) for all $k \leq K$ and (15 - 16) for all $k \in \mathcal{K}$ and $i \in \mathcal{I}_k$ with $i \leq 1/\varepsilon_{k+3}$, on the event $E_0 := \left( \bigcap_{k \leq N+1} E_{0,k} \right) \cap \bigcap_{k \leq N} \bigcap_{i \leq 1/\varepsilon_{k+3}} E_{0,k,i}$. By the union bound, $E_0$ fails with probability at most

$$\sum_{k=1}^{N+1} \left( \delta_k + \sum_{i \leq 1/\varepsilon_{k+3}} \delta_k \right) \leq \sum_{k=1}^{N+1} \left( 1 + \frac{1}{\varepsilon_{k+3}} \right) \delta_k \leq \sum_{k=1}^{N+1} \frac{2\delta_k}{\varepsilon_{k+3}} = \sum_{k=1}^{N+1} \frac{\delta}{36}\varepsilon_{k+3} < \frac{\delta}{4},$$

where the equality follows from our definition of $\delta_k = \frac{\delta \varepsilon_{k+3}^2}{72}$ (from Lemma 9) and the last inequality follows from our choice of $C = \frac{11}{10}$.

Finally, we argue that, on the event $E_0$, for any $k \in \mathcal{K}$, the maximum value of $i \in \mathcal{I}_k$ satisfies $i \leq 1/\varepsilon_{k+3}$. We argue this by induction. Specifically, we will argue that, for any $k \in \mathcal{K}$ and $i \in \mathcal{I}_k$, $P_X(\mathcal{X} \setminus \Delta_i) \leq 1 - i\varepsilon_{k+3}$. For the purpose of induction, suppose that for some $k \in \mathcal{K}$, we have $P_X(\mathcal{X} \setminus \Delta_{i_k}) \leq 1 - i_k\varepsilon_{k+3}$ (which is trivially satisfied for $k = 1$, since $i_k = 0$, which can therefore serve as a base case for induction). Taking this $i_k$ as a base case for a further nested

induction on $i \in \mathcal{I}_k$ (noting that $i_k$ is the minimum element of $\mathcal{I}_k$), suppose that for some $i \in \mathcal{I}_k$ we have $P_X(\mathcal{X} \setminus \Delta_i) \leq 1 - i\varepsilon_{k+3}$. Since probabilities are non-negative, this necessarily implies $i \leq 1/\varepsilon_{k+3}$. Then note that, if $i$ is not the maximal element of $\mathcal{I}_k$, the algorithm augments $\Delta_i$ in Step 7, so that $\Delta_{i+1} = \Delta_i \cup \{f \neq g\}$ for $(f, g)$ defined in Step 6. By the criterion in Step 5, we further know that $\hat{P}_{S^3_{k,i}}(\{f \neq g\} \setminus \Delta_i) > \varepsilon_{k+2}$. Since $i \leq 1/\varepsilon_{k+3}$, the event $E_0$ implies (15) holds, which therefore implies $P_X(\Delta_{i+1} \setminus \Delta_i) = P_X(\{f \neq g\} \setminus \Delta_i) > \varepsilon_{k+3}$, so that $P_X(\mathcal{X} \setminus \Delta_{i+1}) = P_X(\mathcal{X} \setminus \Delta_i) - P_X(\Delta_{i+1} \setminus \Delta_i) < 1 - (i+1)\varepsilon_{k+3}$, thus extending the inductive hypothesis. By the principle of induction, this establishes that $P_X(\mathcal{X} \setminus \Delta_i) \leq 1 - i\varepsilon_{k+3}$ for every $i \in \mathcal{I}_k$. In particular, returning to the induction on $k$, in the event that this $k$ is not the maximal element of $\mathcal{K}$, we have $i_{k+1} \in \mathcal{I}_k$, so that $P_X(\mathcal{X} \setminus \Delta_{i_{k+1}}) \leq 1 - i_{k+1}\varepsilon_{k+3} \leq 1 - i_{k+1}\varepsilon_{(k+1)+3}$, which therefore extends the inductive hypothesis for $k$. By the principle of induction, we have thus established that every $k \in \mathcal{K}$ and $i \in \mathcal{I}_k$ satisfy $P_X(\mathcal{X} \setminus \Delta_i) \leq 1 - i\varepsilon_{k+3}$. In particular, since probabilities are non-negative, this immediately implies any such $(k, i)$ satisfy $i \leq 1/\varepsilon_{k+3}$, as claimed. Thus, on the event $E_0$, we have established all of the claimed inequalities: (14) for all $k \in \{1, \ldots, K\}$, and (15 - 16) for all $k \in \mathcal{K}$ and $i \in \mathcal{I}_k$, which further satisfy $\max \mathcal{I}_k \leq \frac{1}{\varepsilon_{k+3}}$. ∎

The following is an obvious implication of Lemma 10, which will be useful to state explicitly for later reference.

**Lemma 11.** *On the event $E_0$, for every $k \in \mathcal{K}$ and $i \in \mathcal{I}_k$, if the algorithm reaches Step 6 with these values $(k, i)$, then for $f, g$ as defined there,*

$$P_X(\{f \neq g\} \setminus \Delta_i) > \varepsilon_{k+3}.$$

*Moreover, on the event $E_0$, every $k \in \{1, \ldots, K\}$ with $\Delta_{i_k} \neq \emptyset$ satisfies $P_X(\Delta_{i_k}) > \varepsilon_{k+2}$.*

**Proof.** By the condition in Step 5, if the algorithm reaches Step 6 then $\hat{P}_{S^3_{k,i}}(\{f \neq g\} \setminus \Delta_i) > \varepsilon_{k+2}$. By (15) of Lemma 10, on the event $E_0$, this implies $P_X(\{f \neq g\} \setminus \Delta_i) > \varepsilon_{k+3}$.

Turning now to the second claim, suppose again that $E_0$ occurs, and first note that this claim is trivially satisfied if $\Delta_{i_K} = \emptyset$. To address the remaining case, suppose $\Delta_{i_K} \neq \emptyset$, and consider the minimum value $k' \in \{1, \ldots, K\}$ for which $\Delta_{i_{k'}} \neq \emptyset$. By definition we have $\Delta_{i_1} = \Delta_0 = \emptyset$, which implies we must have $k' \geq 2$. By minimality of $k'$, we also know that the algorithm reaches Step 6 at least once during round $k = k' - 1$ of the '*For*' loop, in particular with $i = i_k$. Thus, letting $(f, g)$ be as defined in Step 6 for these values $(k, i) = (k' - 1, i_{k'-1})$, by the first claim in the lemma, we have $P_X(\Delta_{i_{k'}}) \geq P_X(f \neq g) = P_X(\{f \neq g\} \setminus \Delta_{i_{k'-1}}) > \varepsilon_{k'+2}$. Thus, since $\Delta_{i_{k''}}$ is non-decreasing in $k''$, and minimality of $k'$ implies all $k''$ with $\Delta_{i_{k''}} \neq \emptyset$ have $k'' \geq k'$, we conclude that every $k'' \in \{1, \ldots, K\}$ with $\Delta_{i_{k''}} \neq \emptyset$ satisfies $P_X(\Delta_{i_{k''}}) \geq P_X(\Delta_{i_{k'}}) > \varepsilon_{k'+2} \geq \varepsilon_{k''+2}$. ∎

Next we state a bound on the diameters of $V_{k-1}$ and $V^{(3)}_{k-1}$, useful for Lemmas 15, 20, and 22.

**Lemma 12.** *On the event $E_0$, for every $k \in \{1, \ldots, K\}$,*

$$\sup_{f,g \in V_{k-1}} P_X(\{f \neq g\} \setminus \Delta_{i_k}) \leq \varepsilon_k \tag{19}$$

*and* $$\sup_{f,g \in V^{(3)}_{k-1}} P_X(\{f \neq g\} \setminus \Delta_{i_k}) \leq 3\varepsilon_k. \tag{20}$$

**Proof.** Throughout this proof, we suppose the event $E_0$ holds. The inequality (19) is trivially satisfied for $k = 1$, recalling that $V_0 = \mathbb{C}$ and $\varepsilon_1 = C^0 = 1$. For the remaining case, fix any $k' \in \{2, \ldots, K\}$ and consider the round $k = k' - 1$ in the '*For*' loop (noting that, by definition of $K$, we have $k = k' - 1 \in \mathcal{K}$ regardless of whether $K = N + 1$ or $K \leq N$). Since $k + 1 = k' \leq K$, we know the algorithm reaches Step 8 in round $k$ (i.e., it does not terminate early in Step 4 during round $k$). In particular, this means the condition in Step 5 fails for the value $i = i_{k+1} = \max \mathcal{I}_k$: that is, $\max_{f,g \in V_k} \hat{P}_{S^3_{k,i}}(\{f \neq g\} \setminus \Delta_{i_{k+1}}) \leq \varepsilon_{k+2}$. By (16) of Lemma 10, this implies

$$\sup_{f,g \in V_{k'-1}} P_X(\{f \neq g\} \setminus \Delta_{i_{k'}}) = \sup_{f,g \in V_k} P_X(\{f \neq g\} \setminus \Delta_{i_{k+1}}) < \varepsilon_{k+1} = \varepsilon_{k'}.$$

This completes the proof of (19) for every $k \in \{1, \ldots, K\}$.

To show (20), let $k \in \{1, \ldots, K\}$, and for any $f, g \in V_{k-1}$ and $h \in \mathbb{C}$, denote by $\mathrm{DL}(f, g, h) := f \mathbb{1}_{\{f=g\}} + h \mathbb{1}_{\{f \neq g\}} \in V_{k-1}^{(3)}$. Note that for any $f, g, f', g' \in V_{k-1}$, $h, h' \in \mathbb{C}$, and $x \in \mathcal{X}$, if $g(x) = f(x) = f'(x) = g'(x)$, then $\mathrm{DL}(f, g, h)(x) = \mathrm{DL}(f', g', h')(x)$. Therefore,

$$P_X(\{\mathrm{DL}(f, g, h) \neq \mathrm{DL}(f', g', h')\} \setminus \Delta_{i_k}) \leq P_X((\{f \neq g\} \cup \{f' \neq g'\} \cup \{f \neq f'\}) \setminus \Delta_{i_k})$$
$$\leq P_X(\{f \neq g\} \setminus \Delta_{i_k}) + P_X(\{f' \neq g'\} \setminus \Delta_{i_k}) + P_X(\{f \neq f'\} \setminus \Delta_{i_k}) \leq 3\varepsilon_k,$$

where the last inequality is by (19). This completes the proof of the lemma. ∎

The following Lemmas 13 and 14 concern concentration of empirical errors in the set $S_k^2 \cap \Delta_{i_k}$, which will be useful in establishing guarantees on the quality of $\hat{h}_k$ (in Lemma 15) and of the functions in $V_k$ (in Lemmas 16 and 17) below. We first need to argue that the $\hat{p}_k$ quantities approximate $P_X(\Delta_{i_k})$, which leads to the data sets $S_k^2$ being of appropriate size for concentration of empirical error rates.

**Lemma 13.** *There is an event $E_1$ of probability at least $1 - \frac{\delta}{4}$ such that, on $E_0 \cap E_1$, $\forall k \in \{1, \ldots, K\}$, the quantity $\hat{p}_k := 2\hat{P}_{S_k^4}(\Delta_{i_k})$ (as defined above) satisfies*

$$P_X(\Delta_{i_k}) \leq \hat{p}_k \leq 4P_X(\Delta_{i_k}). \tag{21}$$

**Proof.** Consider any $k \in \{1, \ldots, N+1\}$ having non-zero probability that $k \leq K$. Note that the execution of the algorithm does not depend on $S_k^4$ at any time prior to Step 2 of round $k$ (or Step 9 if $k = N+1$), supposing this step is even reached in the algorithm (i.e., $k \leq K$). Thus, since the event that $k \leq K$ and the set $\Delta_{i_k}$ are both completely determined by events occurring prior to this first time the examples in $S_k^4$ are used by the algorithm, we have that $S_k^4$ is independent of these. Thus, conditioned on the event that $k \leq K$ and on the random variable $\Delta_{i_k}$, we have that for the sequence of $m_k$ examples $(X_t, Y_t)$ comprising $S_k^4$, the corresponding sequence of indicator random variables $\mathbb{1}[X_t \in \Delta_{i_k}]$ are conditionally independent $\mathrm{Bernoulli}(P_X(\Delta_{i_k}))$ random variables. Therefore, applying a multiplicative Chernoff bound (Lemma 6 of Appendix D) under the conditional distribution given the event $k \leq K$ and the random variable $\Delta_{i_k}$, together with the law of total probability, we have that on an event $E_{1,k}$ of probability at least $1 - \frac{\delta\varepsilon_k}{44}$, if $k \leq K$, then

$$P_X(\Delta_{i_k}) \leq \max\left\{ 2\hat{P}_{S_k^4}(\Delta_{i_k}), \frac{8}{m_k} \ln\left(\frac{88}{\delta\varepsilon_k}\right) \right\}, \tag{22}$$

$$\hat{P}_{S_k^4}(\Delta_{i_k}) \leq \max\left\{ 2P_X(\Delta_{i_k}), \frac{6}{m_k} \ln\left(\frac{88}{\delta\varepsilon_k}\right) \right\}. \tag{23}$$

For simplicity, for any $k \in \{1, \ldots, N+1\}$ having probability zero of $k \leq K$, simply define $E_{1,k}$ as the event of probability one that $k > K$, so that the above claim also holds (vacuously) for such values $k$. Define an event $E_1 = \bigcap_{k=1}^{N+1} E_{1,k}$, and note that, by the union bound, $E_1$ occurs with probability at least $1 - \sum_{k=1}^{N+1} \frac{\delta\varepsilon_k}{44} \geq 1 - \frac{\delta}{4}$.

We now argue these inequalities further imply the simpler inequalities stated in (21), on the additional event $E_0$. Suppose the event $E_0 \cap E_1$ holds, and let $k \in \{1, \ldots, K\}$. If $\Delta_{i_k} = \emptyset$, (21) trivially holds since $\hat{p}_k = 0 = P_X(\Delta_{i_k})$. To address the remaining case, suppose $\Delta_{i_k} \neq \emptyset$. By the final claim in Lemma 11, we have $P_X(\Delta_{i_k}) > \varepsilon_{k+2}$. Also note that, by definition of $m_k$ (and recalling $|\mathbb{C}| \geq 2$, which implies $\mathsf{d} \geq 1$), we have $\frac{8}{m_k} \ln\left(\frac{88}{\delta\varepsilon_k}\right) < \varepsilon_{k+2}$. In particular, these imply $2P_X(\Delta_{i_k}) > 2\varepsilon_{k+2} > \frac{6}{m_k} \ln\left(\frac{88}{\delta\varepsilon_k}\right)$, so that the right hand side of (23) equals $2P_X(\Delta_{i_k})$ and hence $\hat{p}_k \leq 4P_X(\Delta_{i_k})$. Moreover, since $\frac{8}{m_k} \ln\left(\frac{88}{\delta\varepsilon_k}\right) < \varepsilon_{k+2} < P_X(\Delta_{i_k})$, the "max" on the right hand side of (22) cannot be achieved by the second term (as it is smaller than the quantity on the left hand side), so it must be achieved by the first term. Therefore, $P_X(\Delta_{i_k}) \leq 2\hat{P}_{S_k^4}(\Delta_{i_k}) = \hat{p}_k$. ∎

Using Lemma 13 to bound the size of the data set $S_k^2$ (which is based on $\hat{p}_k$), we are now ready to establish a concentration inequality for the error rates in the $\Delta_{i_k}$ region in the following lemma.

**Lemma 14.** *There is an event $E_2$ of probability at least $1 - \frac{\delta}{4}$ such that, on the event $E_0 \cap E_1 \cap E_2$, $\forall k \in \{1, \ldots, K\}$,*

$$\sup_{h \in \mathbb{C}} \left| \hat{P}_{S_k^2}(\mathrm{ER}(h) \cap \Delta_{i_k}) - P(\mathrm{ER}(h) \cap \Delta_{i_k}) \right| \leq \frac{\varepsilon_k}{\sqrt{C''}}. \tag{24}$$

**Proof.** Consider any $k \in \{1, \ldots, N+1\}$ having non-zero probability of $k \leq K$. Supposing $k \leq K$ occurs, define a collection of sets $\mathcal{A}'_k := \{\mathrm{ER}(h) \cap \Delta_{i_k} : h \in \mathbb{C}\}$. Note that $\mathrm{VC}(\mathcal{A}'_k) \leq \mathsf{d}$ (which is immediate from the definition of VC dimension). We aim to apply Lemma 8 of Appendix D, a refinement of the uniform convergence bound of Talagrand (1994), which accounts for an *envelope* set $D \supseteq \bigcup \mathcal{A}'_k$; specifically, we instantiate the various sets and variables in Lemma 8 to be $\mathcal{Z} = \mathcal{X} \times \{0,1\}$, $n = m'_k$, $\mathcal{A} = \mathcal{A}'_k$, envelope set $D = \Delta_{i_k}$, data set $Z = S^2_k$, and confidence parameter $\delta/(4(3+N-k)^2)$, and we apply the lemma under the conditional distribution given the event $k \leq K$ and given the random variables $\Delta_{i_k}$ and $m'_k$. Since the event that $k \leq K$, and the random variables $\Delta_{i_k}$ and $m'_k$, are all completely determined by examples allocated to data sets *before* allocating examples to the data set $S^2_k$, we may note that the $m'_k$ examples comprising $S^2_k$ are conditionally independent $P$-distributed random variables given the event that $k \leq K$ and given the random variables $\Delta_{i_k}$ and $m'_k$. Thus, applying Lemma 8 of Appendix D under the conditional distribution given the event that $k \leq K$ and given the random variables $\Delta_{i_k}$ and $m'_k$, together with the law of total probability, we have that on an event $E_{2,k}$ of probability at least $1 - \frac{\delta}{4(3+N-k)^2}$, if $k \leq K$ and $P_X(\Delta_{i_k}) \geq \frac{9}{m'_k} \ln\left(\frac{16(3+N-k)^2}{\delta}\right)$, then

$$\sup_{h \in \mathbb{C}} \left| \hat{P}_{S^2_k}(\mathrm{ER}(h) \cap \Delta_{i_k}) - P(\mathrm{ER}(h) \cap \Delta_{i_k}) \right| \leq c_1 \sqrt{\frac{P_X(\Delta_{i_k})}{m'_k} \left( \mathsf{d} + \log\left(\frac{4(3+N-k)^2}{\delta}\right) \right)}. \quad (25)$$

For simplicity, for any $k \in \{1, \ldots, N+1\}$ having zero probability of $k \leq K$, let $E_{2,k}$ denote the event of probability one that $k > K$. Finally, define $E_2 = \bigcap_{k=1}^{N+1} E_{2,k}$, and note that, by the union bound, $E_2$ holds with probability at least $1 - \sum_{k=1}^{N+1} \frac{\delta}{4(3+N-k)^2} \geq 1 - \frac{\delta}{4} \sum_{j=2}^{\infty} \frac{1}{j^2} \geq 1 - \frac{\delta}{4}$.

Now suppose the event $E_0 \cap E_1 \cap E_2$ occurs, and consider any $k \in \{1, \ldots, K\}$. If $\Delta_{i_k} = \emptyset$ then (24) holds trivially since the left hand side of (24) is then zero. To address the remaining case, suppose $\Delta_{i_k} \neq \emptyset$. By the final claim in Lemma 11, we have $P_X(\Delta_{i_k}) > \varepsilon_{k+2}$. Moreover, by Lemma 13 we have $\hat{p}_k \geq P_X(\Delta_{i_k})$. Recalling $m'_k := \left\lceil \frac{C'' c_1^2 \hat{p}_k}{\varepsilon_k^2} \left( \mathsf{d} + \log\left(\frac{4(3+N-k)^2}{\delta}\right) \right) \right\rceil$, these imply

$$m'_k > \frac{C'' c_1^2}{C^4 \varepsilon_{k+2}} \ln\left(\frac{4(3+N-k)^2}{\delta}\right) > \frac{9}{\varepsilon_{k+2}} \ln\left(\frac{16(3+N-k)^2}{\delta}\right),$$

where the last inequality follows from $c_1 \geq 1$ and $C'' \geq 18 C^4$. Thus, $\frac{9}{m'_k} \ln\left(\frac{16(3+N-k)^2}{\delta}\right) < \varepsilon_{k+2} < P_X(\Delta_{i_k})$. By the definition of $E_2$, it follows that (25) holds. Moreover, since $\hat{p}_k \geq P_X(\Delta_{i_k})$, we have that $m'_k \geq \frac{C'' c_1^2 P_X(\Delta_{i_k})}{\varepsilon_k^2} \left( \mathsf{d} + \log\left(\frac{4(3+N-k)^2}{\delta}\right) \right)$, so that the right hand side of (25) is at most $\frac{\varepsilon_k}{\sqrt{C''}}$, thus establishing (24). ∎

Combining the concentration inequality from Lemma 14 with (14) of Lemma 10 together with (20) of Lemma 12 yields a concentration inequality for the differences $\hat{\mathrm{er}}^{1,2}_k(h) - \hat{\mathrm{er}}^{1,2}_k(h')$ among $h, h' \in V_{k-1}$, recalling the definition from (1):

$$\hat{\mathrm{er}}^{1,2}_k(h) := \hat{P}_{S^1_k}(\mathrm{ER}(h) \cap D_{k-1} \setminus \Delta_{i_k}) + \hat{P}_{S^2_k}(\mathrm{ER}(h) \cap \Delta_{i_k}).$$

In fact, the implication is stronger than this, admitting functions $h, h' \in V^{(4)}_{k-1}$. In particular, for any $k \in \{1, \ldots, K\}$, note that $V^{(4)}_{k-1}$ in (2) can equivalently be defined as

$$V^{(4)}_{k-1} = \left\{ h_1 \mathbb{1}_{\mathcal{X} \setminus \Delta_{i_k}} + h_2 \mathbb{1}_{\Delta_{i_k}} : h_1 \in V^{(3)}_{k-1}, h_2 \in \mathbb{C} \right\}.$$

The following lemma provides a concentration inequality for $\hat{\mathrm{er}}^{1,2}_k(h) - \hat{\mathrm{er}}^{1,2}_k(h')$ among functions $h, h' \in V^{(4)}_{k-1}$.

**Lemma 15.** *On the event $E_0 \cap E_1 \cap E_2$, for every $k \in \{1, \ldots, K\}$ we have*

$$\sup_{h, h' \in V^{(4)}_{k-1}} \left| \left( \hat{\mathrm{er}}^{l,2}_k(h) - \hat{\mathrm{er}}^{l,2}_k(h') \right) - (\mathrm{er}_P(h) - \mathrm{er}_P(h')) \right| \leq \frac{\varepsilon_k}{4C'}, \quad (26)$$

*recalling that $C' := \frac{\sqrt{C''}}{16}$. Moreover, (26) implies*

$$\mathrm{er}_P(\hat{h}_k) \leq \inf_{h \in V^{(4)}_{k-1}} \mathrm{er}_P(h) + \frac{\varepsilon_k}{4C'}. \quad (27)$$

**Proof.** Suppose the event $E_0 \cap E_1 \cap E_2$ holds and consider any $k \in \{1, \dots, K\}$. Note that for any $h = h_1 \mathbb{1}_{\mathcal{X} \setminus \Delta_{i_k}} + h_2 \mathbb{1}_{\Delta_{i_k}} \in V_{k-1}^{(4)}$ we have $\mathrm{er}_P(h) = P(\mathrm{ER}(h_1) \setminus \Delta_{i_k}) + P(\mathrm{ER}(h_2) \cap \Delta_{i_k})$ and $\hat{\mathrm{er}}_k^{1,2}(h) = \hat{P}_{S_k^1}(\mathrm{ER}(h_1) \cap D_{k-1} \setminus \Delta_{i_k}) + \hat{P}_{S_k^2}(\mathrm{ER}(h_2) \cap \Delta_{i_k})$.

Consider any $h_1, h_1' \in V_{k-1}^{(3)}$ and any $h_2, h_2' \in \mathbb{C}$, and let $h = h_1 \mathbb{1}_{\mathcal{X} \setminus \Delta_{i_k}} + h_2 \mathbb{1}_{\Delta_{i_k}}$ and $h' = h_1' \mathbb{1}_{\mathcal{X} \setminus \Delta_{i_k}} + h_2' \mathbb{1}_{\Delta_{i_k}}$. By Lemma 14, we have $\forall h_2'' \in \{h_2, h_2'\}$,

$$\left| \hat{P}_{S_k^2}(\mathrm{ER}(h_2'') \cap \Delta_{i_k}) - P(\mathrm{ER}(h_2'') \cap \Delta_{i_k}) \right| \leq \frac{\varepsilon_k}{\sqrt{C''}}.$$

Additionally, since (20) of Lemma 12 implies $P_X(\{h_1 \neq h_1'\} \setminus \Delta_{i_k}) \leq 3\varepsilon_k$, the inequality (14) of Lemma 10 implies

$$\left| \hat{P}_{S_k^1}(\mathrm{ER}(h_1) \cap D_{k-1} \setminus \Delta_{i_k}) - \hat{P}_{S_k^1}(\mathrm{ER}(h_1') \cap D_{k-1} \setminus \Delta_{i_k}) - (P(\mathrm{ER}(h_1) \setminus \Delta_{i_k}) - P(\mathrm{ER}(h_1') \setminus \Delta_{i_k})) \right|$$

$$< \sqrt{\frac{3\varepsilon_k^2}{C''}} + \frac{\varepsilon_k}{C''} \leq \frac{2\varepsilon_k}{\sqrt{C''}}, \tag{28}$$

where the last inequality follows from $C'' \geq 14$. Combining these with the triangle inequality (namely, $|((\hat{a} + \hat{b}) - (\hat{a}' + \hat{b}')) - ((a + b) - (a' + b'))| \leq |(\hat{a} - \hat{a}') - (a - a')| + |\hat{b} - b| + |\hat{b}' - b'|$) yields that

$$\left| (\hat{\mathrm{er}}_k^{1,2}(h) - \hat{\mathrm{er}}_k^{1,2}(h')) - (\mathrm{er}_P(h) - \mathrm{er}_P(h')) \right| < \frac{4\varepsilon_k}{\sqrt{C''}} = \frac{\varepsilon_k}{4C'}.$$

To see that (26) implies (27), note that, by the definition of $\hat{h}_k$ in (3), $h = \hat{h}_k$ has minimal $\hat{\mathrm{er}}_k^{1,2}(h)$ among all $h \in V_{k-1}^{(4)}$: that is, $\forall h \in V_{k-1}^{(4)}$, $\hat{\mathrm{er}}_k^{1,2}(\hat{h}_k) - \hat{\mathrm{er}}_k^{1,2}(h) \leq 0$. Together with (26), this implies that $\forall h \in V_{k-1}^{(4)}$, $\mathrm{er}_P(\hat{h}_k) - \mathrm{er}_P(h) \leq \hat{\mathrm{er}}_k^{1,2}(\hat{h}_k) - \hat{\mathrm{er}}_k^{1,2}(h) + \frac{\varepsilon_k}{4C'} \leq \frac{\varepsilon_k}{4C'}$. ∎

In particular, Lemma 15 immediately implies the following lemma concerning the quality of the functions $h$ in $V_k$.

**Lemma 16.** *On the event $E_0 \cap E_1 \cap E_2$, $\forall k \in \mathcal{K}$, $\forall h \in V_{k-1}$, the following implications hold:*

$$h \in V_k \implies \mathrm{er}_P(h) \leq \mathrm{er}_P(\hat{h}_k) + \frac{5\varepsilon_k}{4C'}, \tag{29}$$

$$\mathrm{er}_P(h) \leq \mathrm{er}_P(\hat{h}_k) + \frac{3\varepsilon_k}{4C'} \implies h \in V_k. \tag{30}$$

**Proof.** Suppose the event $E_0 \cap E_1 \cap E_2$ occurs and consider any $k \in \mathcal{K}$ and $h \in V_{k-1}$. In particular, note that we also have $h \in V_{k-1}^{(4)}$, since letting $f = g = h$, we have $h = f \mathbb{1}_{\{f=g\}} + h \mathbb{1}_{\{f \neq g\}} \in V_{k-1}^{(3)}$, and thus $h = h \mathbb{1}_{\mathcal{X} \setminus \Delta_{i_k}} + h \mathbb{1}_{\Delta_{i_k}} \in V_{k-1}^{(4)}$.

If $h \in V_k$, then by definition of $V_k$ in Step 3 we have $\hat{\mathrm{er}}_k^{1,2}(h) - \hat{\mathrm{er}}_k^{1,2}(\hat{h}_k) \leq \frac{\varepsilon_k}{C'}$. Together with (26) of Lemma 15, this implies $\mathrm{er}_P(h) - \mathrm{er}_P(\hat{h}_k) \leq \frac{\varepsilon_k}{C'} + \frac{\varepsilon_k}{4C'} = \frac{5\varepsilon_k}{4C'}$, which establishes (29).

On the other hand, if $\mathrm{er}_P(h) - \mathrm{er}_P(\hat{h}_k) \leq \frac{3\varepsilon_k}{4C'}$, then (26) of Lemma 15 implies $\hat{\mathrm{er}}_k^{1,2}(h) - \hat{\mathrm{er}}_k^{1,2}(\hat{h}_k) \leq \frac{3\varepsilon_k}{4C'} + \frac{\varepsilon_k}{4C'} = \frac{\varepsilon_k}{C'}$. Thus, any such $h$ is retained in $V_k$, which establishes (30). ∎

The main implication of Lemma 16 pertains to the early stopping case in Step 4, which we turn to next. Recall $h^\star$ denotes an (arbitrary) concept in $\mathbb{C}$ with $\mathrm{er}_P(h^\star) < \inf_{h \in \mathbb{C}} \mathrm{er}_P(h) + \frac{\varepsilon}{10^4}$. In the following lemma, in addition to arguing that the predictor $\hat{h}$ returned in Step 4 has low excess error rate compared to $h^\star$ (in fact, *negative*), this lemma also reveals a second major role of this early stopping case: it ensures that on all rounds $k$ in which the algorithm does *not* terminate in Step 4, we retain $h^\star \in V_k$.

**Lemma 17.** *On the event $E_0 \cap E_1 \cap E_2$, the following implications hold for every $k \in \mathcal{K}$:*

- *If $\mathbb{A}_{\mathrm{avid}}$ does* not *return in Step 4 on round $k$, then $h^\star \in V_k$.*

- *If $\mathbb{A}_{\mathrm{avid}}$ returns in Step 4 on round $k$, then $\mathrm{er}_P(\hat{h}_k) < \mathrm{er}_P(h^\star)$.*

**Proof.** Suppose the event $E_0 \cap E_1 \cap E_2$ occurs. We will prove the first claim by induction on $k$. As a base case, we trivially have $h^\star \in \mathbb{C} = V_0$. Now, for the purpose of induction, let $k \in \mathcal{K}$ be such that $h^\star \in V_{k-1}$. Also note (as discussed in the proof of Lemma 16) that this also implies $h^\star \in V_{k-1}^{(4)}$. If $h^\star \notin V_k$, then by (30) of Lemma 16, we have $\mathrm{er}_P(h^\star) - \mathrm{er}_P(\hat{h}_k) > \frac{3\varepsilon_k}{4C'}$. In particular, this implies that if $V_k \neq \emptyset$, then together with (26) of Lemma 15, we have

$$\min_{h \in V_k} \hat{\mathrm{er}}_k^{1,2}(h) - \hat{\mathrm{er}}_k^{1,2}(\hat{h}_k) \geq \min_{h \in V_{k-1}} \hat{\mathrm{er}}_k^{1,2}(h) - \hat{\mathrm{er}}_k^{1,2}(\hat{h}_k) \geq \inf_{h \in V_{k-1}} \mathrm{er}_P(h) - \mathrm{er}_P(\hat{h}_k) - \frac{\varepsilon_k}{4C'}$$

$$> \mathrm{er}_P(h^\star) - \mathrm{er}_P(\hat{h}_k) - \frac{\varepsilon}{10^4} - \frac{\varepsilon_k}{4C'} > \frac{3\varepsilon_k}{4C'} - \frac{\varepsilon}{10^4} - \frac{\varepsilon_k}{4C'} > \frac{\varepsilon_k}{4C'},$$

where the last inequality follows from $\frac{\varepsilon}{10^4} < \frac{2\varepsilon_N}{10^4} \leq \frac{\varepsilon_k}{4C'}$. Thus, either $V_k = \emptyset$ or $\min_{h \in V_k} \hat{\mathrm{er}}_k^{1,2}(h) - \hat{\mathrm{er}}_k^{1,2}(\hat{h}_k) > \frac{\varepsilon_k}{4C'}$, so that either way the algorithm will return in Step 4 in this case. Therefore, if the algorithm does *not* return in Step 4 on round $k$, it must be that $h^\star \in V_k$. This completes the proof of the first claim, by the principle of induction.

Finally, we turn to the second claim. Suppose, for some $k \in \mathcal{K}$, the algorithm returns in Step 4 on round $k$. In particular, either $k = 1$, in which case $h^\star \in \mathbb{C} = V_{k-1}$, or $k > 1$, in which case (since the algorithm did not return in Step 4 on round $k - 1$) the first claim in the lemma implies $h^\star \in V_{k-1}$. Again note that this also implies $h^\star \in V_{k-1}^{(4)}$. If $h^\star \notin V_k$, then (30) of Lemma 16 implies $\mathrm{er}_P(h^\star) > \mathrm{er}_P(\hat{h}_k) + \frac{3\varepsilon_k}{4C'}$. Otherwise, if $h^\star \in V_k$, the condition in Step 4 implies $\hat{\mathrm{er}}_k^{1,2}(h^\star) - \hat{\mathrm{er}}_k^{1,2}(\hat{h}_k) > \frac{\varepsilon_k}{4C'}$. Together with (26) of Lemma 15, this implies

$$\mathrm{er}_P(h^\star) - \mathrm{er}_P(\hat{h}_k) \geq \hat{\mathrm{er}}_k^{1,2}(h^\star) - \hat{\mathrm{er}}_k^{1,2}(\hat{h}_k) - \frac{\varepsilon_k}{4C'} > 0.$$

Thus, in either case, we have $\mathrm{er}_P(h^\star) > \mathrm{er}_P(\hat{h}_k)$, which establishes the second claim. ∎

Lemmas 15, 16, and 17 together have a particularly nice implication, which, although not strictly needed for the proof of Theorem 5, is worth noting (and will be useful in Appendix F). Specifically, we have the following corollary.

**Corollary 18.** *On the event* $E_0 \cap E_1 \cap E_2$, $\forall k \in \mathcal{K}$,

$$V_k \subseteq \left\{ h \in \mathbb{C} : \mathrm{er}_P(h) - \mathrm{er}_P(h^\star) \leq \frac{3\varepsilon_k}{2C'} \right\}.$$

**Proof.** Suppose the event $E_0 \cap E_1 \cap E_2$ occurs and consider any $k \in \mathcal{K}$. Since $k - 1 < K$, Lemma 17 implies $h^\star \in V_{k-1}$ in the case $k \geq 2$, while the case $k = 1$ has $h^\star \in V_0$ by definition of $V_0 = \mathbb{C}$. Together with (27) of Lemma 15, this implies $\mathrm{er}_P(\hat{h}_k) \leq \mathrm{er}_P(h^\star) + \frac{\varepsilon_k}{4C'}$. Combined with (29) of Lemma 16, we have that every $h \in V_k$ satisfies $\mathrm{er}_P(h) \leq \mathrm{er}_P(\hat{h}_k) + \frac{5\varepsilon_k}{4C'} \leq \mathrm{er}_P(h^\star) + \frac{3\varepsilon_k}{2C'}$. ∎

At this point, we may note that Lemmas 15 and 17 together completely address the error guarantee for the $\hat{h}$ returned by $\mathbb{A}_{\mathrm{avid}}$, on the event $E_0 \cap E_1 \cap E_2$. as summarized in the following lemma.

**Lemma 19.** *On the event* $E_0 \cap E_1 \cap E_2$, $\mathbb{A}_{\mathrm{avid}}$ *eventually terminates, and the function* $\hat{h}$ *it returns satisfies* $\mathrm{er}_P(\hat{h}) \leq \inf_{h \in \mathbb{C}} \mathrm{er}_P(h) + \varepsilon$.

**Proof.** Suppose the event $E_0 \cap E_1 \cap E_2$ occurs. If the algorithm terminates in Step 4 in some round $k \in \mathcal{K}$, by definition we have $\hat{h} = \hat{h}_k$, and thus Lemma 17 implies $\mathrm{er}_P(\hat{h}) < \mathrm{er}_P(h^\star) < \inf_{h \in \mathbb{C}} \mathrm{er}_P(h) + \frac{\varepsilon}{10^4}$. On the other hand, if the algorithm does not return in Step 4 on any round $k \in \mathcal{K}$, then we have $K = N + 1$ (recalling Lemma 10 implies the algorithm eventually terminates), so that by definition $\hat{h} = \hat{h}_{N+1}$. Since in this case Lemma 17 implies $h^\star \in V_N$ (and hence $h^\star \in V_N^{(4)}$), (27) of Lemma 15 implies $\mathrm{er}_P(\hat{h}) = \mathrm{er}_P(\hat{h}_{N+1}) \leq \mathrm{er}_P(h^\star) + \frac{\varepsilon_{N+1}}{4C'} < \inf_{h \in \mathbb{C}} \mathrm{er}_P(h) + \frac{\varepsilon}{10^4} + \frac{\varepsilon}{8C'} < \inf_{h \in \mathbb{C}} \mathrm{er}_P(h) + \varepsilon$. ∎

With the analysis of error guarantees complete, we turn now to establishing the bound $Q(\varepsilon, \delta; \beta)$ on the number of queries, as claimed in Theorem 5. This will be comprised of two main parts. First, we

argue that the set $\Delta_{i_k}$ never grows too large: specifically, recalling $\beta := \inf_{h \in \mathbb{C}} \mathrm{er}_P(h)$, Lemma 20 will establish that $P_X(\Delta_{i_k}) = O(\beta)$, which in turn allows us to bound the number of queries in $S_k^2 \cap \Delta_{i_k}$ on each round (in the proof of Lemma 23). Second, in the proof of Lemma 22, we bound $P_X(D_{k-1} \setminus \Delta_{i_k}) \le \mathfrak{s}\varepsilon_k$, by reasoning in terms of the disagreement coefficient (Hanneke, 2007b), relating the latter to the star number via a result of Hanneke and Yang (2015). This in turn allows us to bound the number of queries in $S_k^1 \cap D_{k-1} \setminus \Delta_{i_k}$ on each round (in the proof of Lemma 23).

We begin with the first of these parts, stated in the following lemma. We remark that this lemma plays a special role in constraining the allowed values of the constant $C$, as the argument breaks down if $C$ is taken too large. On the other hand, the proof also reveals that it is possible to decrease the factor "5" in this lemma to *any* value $c > 2$ by taking $C > 1$ appropriately close to 1 and by an appropriately large choice of the constant $C''$ (and hence $C'$). See footnote 11 for further discussion.

**Lemma 20.** *On the event $E_0 \cap E_1 \cap E_2$, for all $k \in \{1, \dots, K\}$ and $i \in \mathcal{I}_k$,*

$$P_X(\Delta_i) \le 5 \inf_{h \in \mathbb{C}} P(\mathrm{ER}(h) \cap \Delta_i) \le 5\beta.$$

**Proof.** We will argue that, on $E_0 \cap E_1 \cap E_2$, for any $h_0 \in \mathbb{C}$, each region $\Delta_{i+1} \setminus \Delta_i$ (defined in Step 7) satisfies

$$P(\mathrm{ER}(h_0) \cap \Delta_{i+1} \setminus \Delta_i) > \left(1 - \frac{C^3}{2} - \frac{5C^3}{4C'}\right) P_X(\Delta_{i+1} \setminus \Delta_i) = \frac{1}{5} P_X(\Delta_{i+1} \setminus \Delta_i), \quad (31)$$

so that each addition to $\Delta_i$ "chops off" a piece of $\mathrm{ER}(h_0)$ of measure proportional to the increase in measure $P_X(\Delta_{i+1}) - P_X(\Delta_i) = P_X(\Delta_{i+1} \setminus \Delta_i)$. The claim in the lemma then follows immediately from (31), since it holds trivially for $i = 0$ (recalling $\Delta_0 = \emptyset$), and if any $k \in \{1, \dots, K\}$ and $i \in \mathcal{I}_k$ has $i \ge 1$, then applying (31) inductively yields

$$P(\mathrm{ER}(h_0) \cap \Delta_i) = \sum_{j=0}^{i-1} P(\mathrm{ER}(h_0) \cap \Delta_{j+1} \setminus \Delta_j) > \sum_{j=0}^{i-1} \frac{1}{5} P_X(\Delta_{j+1} \setminus \Delta_j) = \frac{1}{5} P_X(\Delta_i).$$

Taking the infimum over all $h_0 \in \mathbb{C}$ then implies the lemma.

We proceed now with the formal proof of (31). Suppose the event $E_0 \cap E_1 \cap E_2$ occurs, and for the purpose of analyzing the *increases* $\Delta_{i+1} \setminus \Delta_i$ of the $\Delta_i$ set (which only occur in Step 7), consider any $k \in \mathcal{K}$ and any $i \in \mathcal{I}_k$ with $i < \max \mathcal{I}_k$ (equivalently, the algorithm reaches Step 7 with this $(k, i)$). Let $(f, g)$ be as defined in Step 6 for this $(k, i)$ so that $\Delta_{i+1} \setminus \Delta_i = \{f \ne g\} \setminus \Delta_i$.

Note that $\{f \ne g\} \times \{0, 1\} = (\mathrm{ER}(f) \cap \{f \ne g\}) \cup (\mathrm{ER}(g) \cap \{f \ne g\})$, so that (lower-bounding 'max' by 'average')

$$\max_{f' \in \{f, g\}} P(\mathrm{ER}(f') \cap \{f \ne g\} \setminus \Delta_{i_k})$$

$$\ge \frac{1}{2} P(\mathrm{ER}(f) \cap \{f \ne g\} \setminus \Delta_{i_k}) + \frac{1}{2} P(\mathrm{ER}(g) \cap \{f \ne g\} \setminus \Delta_{i_k}) \ge \frac{1}{2} P_X(\{f \ne g\} \setminus \Delta_{i_k}),$$

where in fact the last inequality holds with equality (since, for $\{0, 1\}$ labels, $(\mathrm{ER}(f) \cap \{f \ne g\})$ and $(\mathrm{ER}(g) \cap \{f \ne g\})$ are disjoint). Thus, $\exists f' \in \{f, g\}$ with

$$P(\mathrm{ER}(f') \cap \{f \ne g\} \setminus \Delta_{i_k}) \ge \frac{1}{2} P_X(\{f \ne g\} \setminus \Delta_{i_k}).$$

Let $h' = f' \mathbb{1}_{\{f = g\} \setminus \Delta_{i_k}} + h_0 \mathbb{1}_{\{f \ne g\} \setminus \Delta_{i_k}} + f' \mathbb{1}_{\Delta_{i_k}}$ and note that $h' \in V_{k-1}^{(4)}$. Also recall that $\hat{\mathrm{er}}_k^{1,2}(\hat{h}_k) = \min_{h \in V_{k-1}^{(4)}} \hat{\mathrm{er}}_k^{1,2}(h)$, and hence $\hat{\mathrm{er}}_k^{1,2}(h') \ge \hat{\mathrm{er}}_k^{1,2}(\hat{h}_k)$. Since we also have $f' \in V_{k-1}^{(4)}$ (as discussed in the proof of Lemma 16), Lemma 15 implies

$$\mathrm{er}_P(f') - \mathrm{er}_P(h') \le \hat{\mathrm{er}}_k^{1,2}(f') - \hat{\mathrm{er}}_k^{1,2}(h') + \frac{\varepsilon_k}{4C'} \le \hat{\mathrm{er}}_k^{1,2}(f') - \hat{\mathrm{er}}_k^{1,2}(\hat{h}_k) + \frac{\varepsilon_k}{4C'} \le \frac{5\varepsilon_k}{4C'}, \quad (32)$$

where the last inequality is due to $f' \in \{f, g\} \subseteq V_k$, recalling the definition of $V_k$ in Step 3.

Moreover, by definition of $f'$ and $h'$, we have

$$\mathrm{er}_P(f') - \mathrm{er}_P(h') = P(\mathrm{ER}(f') \cap \{f \ne g\} \setminus \Delta_{i_k}) - P(\mathrm{ER}(h_0) \cap \{f \ne g\} \setminus \Delta_{i_k})$$

$$\ge \frac{1}{2} P_X(\{f \ne g\} \setminus \Delta_{i_k}) - P(\mathrm{ER}(h_0) \cap \{f \ne g\} \setminus \Delta_{i_k}).$$

Equivalently: $P(\mathrm{ER}(h_0) \cap \{f \neq g\} \setminus \Delta_{i_k}) \geq \frac{1}{2} P_X(\{f \neq g\} \setminus \Delta_{i_k}) - (\mathrm{er}_P(f') - \mathrm{er}_P(h'))$.
Combining this with (32), we conclude that

$$P(\mathrm{ER}(h_0) \cap \{f \neq g\} \setminus \Delta_{i_k}) \geq \frac{1}{2} P_X(\{f \neq g\} \setminus \Delta_{i_k}) - \frac{5\varepsilon_k}{4C'}. \tag{33}$$

Also note that, since $\Delta_i \supseteq \Delta_{i_k}$, we have

$$\Delta_{i+1} \setminus \Delta_i = \{f \neq g\} \setminus \Delta_i = (\{f \neq g\} \setminus \Delta_{i_k}) \setminus (\{f \neq g\} \cap \Delta_i \setminus \Delta_{i_k}),$$

so that

$$P(\mathrm{ER}(h_0) \cap \Delta_{i+1} \setminus \Delta_i) \geq P(\mathrm{ER}(h_0) \cap \{f \neq g\} \setminus \Delta_{i_k}) - P_X(\{f \neq g\} \cap \Delta_i \setminus \Delta_{i_k}). \tag{34}$$

Moreover, again since $\Delta_i \supseteq \Delta_{i_k}$, we have

$$P_X(\{f \neq g\} \cap \Delta_i \setminus \Delta_{i_k}) = P_X(\{f \neq g\} \setminus \Delta_{i_k}) - P_X(\{f \neq g\} \setminus \Delta_i). \tag{35}$$

Combining (34), (35), and (33) yields that

$$\begin{aligned} &P(\mathrm{ER}(h_0) \cap \Delta_{i+1} \setminus \Delta_i) \\ &\geq P(\mathrm{ER}(h_0) \cap \{f \neq g\} \setminus \Delta_{i_k}) - P_X(\{f \neq g\} \setminus \Delta_{i_k}) + P_X(\{f \neq g\} \setminus \Delta_i) \\ &\geq P_X(\{f \neq g\} \setminus \Delta_i) - \frac{1}{2} P_X(\{f \neq g\} \setminus \Delta_{i_k}) - \frac{5\varepsilon_k}{4C'}. \end{aligned} \tag{36}$$

Lemma 12 and the fact that $f, g \in V_k \subseteq V_{k-1}$ imply $P_X(\{f \neq g\} \setminus \Delta_{i_k}) \leq \varepsilon_k$, and Lemma 11 implies $P_X(\{f \neq g\} \setminus \Delta_i) > \varepsilon_{k+3}$. Together, we have

$$P_X(\{f \neq g\} \setminus \Delta_{i_k}) < C^3 P_X(\{f \neq g\} \setminus \Delta_i).$$

Additionally, again since $P_X(\{f \neq g\} \setminus \Delta_i) > \varepsilon_{k+3}$, we have that $\frac{5\varepsilon_k}{4C'} < \frac{5C^3}{4C'} P_X(\{f \neq g\} \setminus \Delta_i)$.
Combining these inequalities with (36), and recalling $\Delta_{i+1} \setminus \Delta_i = \{f \neq g\} \setminus \Delta_i$, yields that

$$P(\mathrm{ER}(h_0) \cap \Delta_{i+1} \setminus \Delta_i) > \left(1 - \frac{C^3}{2} - \frac{5C^3}{4C'}\right) P_X(\Delta_{i+1} \setminus \Delta_i).$$

Recalling that $C' = \frac{\sqrt{C''}}{16} = \frac{25C^3}{16-10C^3}$, we have $\frac{C^3}{2} + \frac{5C^3}{4C'} = \frac{4}{5}$, so that the right hand side above equals $\frac{1}{5} P_X(\Delta_{i+1} \setminus \Delta_i)$, which establishes (31). ∎

Next we turn to the second part of the argument outlined above: bounding the number of queries in the sets $S_k^1 \cap D_{k-1} \setminus \Delta_{i_k}$. We begin by stating a known fact, due to Hanneke and Yang (2015, Theorem 10): namely, that the *disagreement coefficient* (Hanneke, 2007b) is upper bounded by the *star number* (Hanneke and Yang, 2015) (indeed, Theorem 10 of Hanneke and Yang, 2015 shows the relation is even *sharp* in the worst case over $h$ and distributions $P'_X$).

**Lemma 21** (Hanneke and Yang, 2015). *For any measurable $h : \mathcal{X} \to \{0,1\}$, any distribution $P'_X$ on $\mathcal{X}$, and any $r > 0$, defining the $r$-ball centered at $h$ as $\mathrm{B}_{P'_X}(h, r) := \{h' \in \mathbb{C} : P'_X(h' \neq h) \leq r\}$, it holds that*

$$P'_X\big(\mathrm{DIS}\big(\mathrm{B}_{P'_X}(h, r)\big)\big) \leq \mathfrak{s}r.$$

Toward bounding the number of queries in the sets $S_k^1 \cap D_{k-1} \setminus \Delta_{i_k}$ in the algorithm, the following lemma establishes a bound on $P_X(D_{k-1} \setminus \Delta_{i_k})$ by a straightforward application of Lemma 21 to the *conditional* probabilities $P_X(D_{k-1} | \mathcal{X} \setminus \Delta_{i_k})$, in combination with a diameter bound supplied by (19) of Lemma 12.

**Lemma 22.** *On the event $E_0 \cap E_1 \cap E_2$, for every $k \in \{1, \ldots, K\}$, $P_X(D_{k-1} \setminus \Delta_{i_k}) \leq \mathfrak{s}\varepsilon_k$.*

**Proof.** Suppose the event $E_0 \cap E_1 \cap E_2$ holds and consider any $k \in \{1, \ldots, K\}$. If $P_X(\mathcal{X} \setminus \Delta_{i_k}) = 0$, we trivially have that $P_X(D_{k-1} \setminus \Delta_{i_k}) = 0 \leq \mathfrak{s}\varepsilon_k$. To address the remaining case, suppose $P_X(\mathcal{X} \setminus \Delta_{i_k}) > 0$, and denote by $P_k := P_X(\cdot | \mathcal{X} \setminus \Delta_{i_k})$. By (19) of Lemma 12 we have

$$\sup_{f,g \in V_{k-1}} P_k(f \neq g) \leq \frac{\varepsilon_k}{P_X(\mathcal{X} \setminus \Delta_{i_k})}.$$

In particular, since $k - 1 < K$, Lemma 17 implies $h^\star \in V_{k-1}$ in the case $k \geq 2$, while the case $k = 1$ has $h^\star \in V_0$ by definition of $V_0 = \mathbb{C}$. Thus, the above inequality implies

$$V_{k-1} \subseteq \mathrm{B}_{P_k}\left(h^\star, \frac{\varepsilon_k}{P_X(\mathcal{X} \setminus \Delta_{i_k})}\right). \tag{37}$$

Together with Lemma 21, this implies

$$P_k(D_{k-1}) = P_k(\mathrm{DIS}(V_{k-1})) \leq P_k\left(\mathrm{DIS}\left(\mathrm{B}_{P_k}\left(h^\star, \frac{\varepsilon_k}{P_X(\mathcal{X} \setminus \Delta_{i_k})}\right)\right)\right) \leq \mathfrak{s} \frac{\varepsilon_k}{P_X(\mathcal{X} \setminus \Delta_{i_k})}. \tag{38}$$

We therefore have that

$$P_X(D_{k-1} \setminus \Delta_{i_k}) = P_k(D_{k-1})P_X(\mathcal{X} \setminus \Delta_{i_k}) \leq \mathfrak{s}\varepsilon_k.$$

$\blacksquare$

We are now ready to state a lemma bounding the total number of queries in the algorithm, by a combination of Lemmas 13, 20, and 22 together with a multiplicative Chernoff bound argument. For convenience, this lemma also supplies an upper bound on the sizes $m'_k$ of the data sets $S^2_k$, which will be of further use when establishing the bound $M(\varepsilon, \delta; \beta)$ on the total number of unlabeled examples sufficient for the execution of the algorithm (Lemma 24 below). Specifically, in the following lemma, for any $k \in \{1, \ldots, N+1\}$, denote by

$$\overline{m}'_k := \frac{25C''c_1^2\beta}{\varepsilon_k^2}\left(\mathsf{d} + \log\left(\frac{4(3+N-k)^2}{\delta}\right)\right),$$

$$\text{and} \quad M_2 := \frac{700C''c_1^2\beta}{\varepsilon^2}\left(\mathsf{d} + \log\left(\frac{4e^4}{\delta}\right)\right).$$

**Lemma 23.** *There is an event $E_3$ of probability at least $1 - \frac{\delta}{4}$, such that on $\bigcap_{j=0}^3 E_j$, $\forall k \in \{1, \ldots, K\}$, the following claims hold: $m'_k \leq \overline{m}'_k$,*

$$\left|S^1_k \cap D_{k-1} \setminus \Delta_{i_k}\right| \leq 3\mathfrak{s}\varepsilon_k m_k, \tag{39}$$

$$\left|S^2_k \cap \Delta_{i_k}\right| \leq 2P_X(\Delta_{i_k})m'_k \leq 10\beta\overline{m}'_k. \tag{40}$$

*Moreover, we have $\sum_{k=1}^{N+1} \overline{m}'_k \leq M_2$, and the total number of queries by $\mathbb{A}_{\mathrm{avid}}$ is at most $Q(\varepsilon, \delta; \beta)$, where*

$$Q(\varepsilon, \delta; \beta) := 10\beta M_2 + \min\{M_1, (3/2)\mathfrak{s}\varepsilon(N+1)m_{N+1}\}$$

$$= O\left(\frac{\beta^2}{\varepsilon^2}\left(\mathsf{d} + \log\left(\frac{1}{\delta}\right)\right) + \min\left\{\mathfrak{s}\log\left(\frac{1}{\varepsilon}\right), \frac{1}{\varepsilon}\right\}\left(\mathsf{d}\log\left(\frac{1}{\varepsilon}\right) + \log\left(\frac{1}{\delta}\right)\right)\right). \tag{41}$$

**Proof.** By Lemma 13, on $E_0 \cap E_1 \cap E_2$, $\forall k \in \{1, \ldots, K\}$, $P_X(\Delta_{i_k}) \leq \hat{p}_k \leq 4P_X(\Delta_{i_k})$. Recall the definition of $m'_k := \left\lceil \frac{C''c_1^2\hat{p}_k}{\varepsilon_k^2}\left(\mathsf{d} + \log\left(\frac{4(3+N-k)^2}{\delta}\right)\right)\right\rceil$. Thus, if $\Delta_{i_k} = \emptyset$, we have $\hat{p}_k = 0$, hence $m'_k = 0$, and the implication $m'_k \leq \overline{m}'_k$ trivially follows. Otherwise, on $E_0 \cap E_1 \cap E_2$, if $\Delta_{i_k} \neq \emptyset$, the final claim in Lemma 11 implies $P_X(\Delta_{i_k}) > \varepsilon_{k+2}$, so that together $m'_k$ is at most

$$\left\lceil \frac{C''c_1^2 4P_X(\Delta_{i_k})}{\varepsilon_k^2}\left(\mathsf{d} + \log\left(\frac{4(3+N-k)^2}{\delta}\right)\right)\right\rceil \leq \frac{5C''c_1^2P_X(\Delta_{i_k})}{\varepsilon_k^2}\left(\mathsf{d} + \log\left(\frac{4(3+N-k)^2}{\delta}\right)\right).$$

Since Lemma 20 implies $P_X(\Delta_{i_k}) \leq 5\beta$ on $E_0 \cap E_1 \cap E_2$, we conclude that $m'_k \leq \overline{m}'_k$.

We next turn to establishing (39). Consider any $k \in \{1, \ldots, N+1\}$ having non-zero probability that $k \leq K$. Given that $k \leq K$, note that $V_{k-1}$ and $\Delta_{i_k}$ have no dependence on $S^1_k$, so that the samples in $S^1_k$ are conditionally i.i.d.-$P$ given the event that $k \leq K$ and given the random variables $V_{k-1}$ and $\Delta_{i_k}$. Therefore, applying a multiplicative Chernoff bound (Lemma 6 of Appendix D) under the conditional distribution given the event $k \leq K$ and the random variables $V_{k-1}$ and $\Delta_{i_k}$, with conditional probability at least $1 - \frac{\delta}{8k(k+1)}$,

$$\left|S^1_k \cap D_{k-1} \setminus \Delta_{i_k}\right| \leq 2m_k P_X(D_{k-1} \setminus \Delta_{i_k}) + 6\ln\left(\frac{16k(k+1)}{\delta}\right). \tag{42}$$

In particular, by the law of total probability, this implies that for every $k \in \{1, \ldots, N+1\}$, with probability at least $1 - \frac{\delta}{8k(k+1)}$, if $k \leq K$ then (42) holds. Letting $E_3'$ denote the event that (42) holds for every $k \in \{1, \ldots, K\}$, by the union bound, $E_3'$ holds with probability at least $1 - \frac{\delta}{8}$. Combining (42) with Lemma 22, we have that on the event $E_0 \cap E_1 \cap E_2 \cap E_3'$, $\forall k \in \{1, \ldots, K\}$,

$$\left| S_k^1 \cap D_{k-1} \setminus \Delta_{i_k} \right| \leq 2\mathfrak{s}\varepsilon_k m_k + 6\ln\left(\frac{16k(k+1)}{\delta}\right) \leq 3\mathfrak{s}\varepsilon_k m_k,$$

where the rightmost inequality follows from recalling $m_k := \left\lceil \frac{300C''c_0}{\varepsilon_k} \left( \mathsf{d}\log\left(\frac{C''c_0}{\varepsilon_k}\right) + \log\left(\frac{1}{\delta}\right) \right) \right\rceil$, which satisfies $\varepsilon_k m_k \geq 6\ln\left(\frac{16k(k+1)}{\delta}\right)$. Thus, we have established (39).

We argue the left inequality in (40) similarly. Consider any $k \in \{1, \ldots, N+1\}$ having non-zero probability of $k \leq K$. Given $k \leq K$, note that $\Delta_{i_k}$ has no dependence on $S_k^2$ or $m_k'$, so that the $m_k'$ samples in $S_k^2$ are conditionally i.i.d.-$P$ given the event $k \leq K$ and given the random variables $\Delta_{i_k}$ and $m_k'$. Therefore, applying a multiplicative Chernoff bound (Lemma 6 of Appendix D) under the conditional distribution given the event $k \leq K$ and the random variables $\Delta_{i_k}$ and $m_k'$, with conditional probability at least $1 - \frac{\delta}{8(3+N-k)^2}$,

$$\left| S_k^2 \cap \Delta_{i_k} \right| \leq \max\left\{ 2P_X(\Delta_{i_k})m_k', 6\ln\left(\frac{16(3+N-k)^2}{\delta}\right) \right\}. \tag{43}$$

By the law of total probability, we have that for every $k \in \{1, \ldots, N+1\}$, with probability at least $1 - \frac{\delta}{8(3+N-k)^2}$, if $k \leq K$ then (43) holds. Letting $E_3''$ denote the event that (43) holds for every $k \in \{1, \ldots, K\}$, by the union bound, $E_3''$ holds with probability at least $1 - \sum_{k=1}^{N+1} \frac{\delta}{8(3+N-k)^2} \geq 1 - \frac{\delta}{8}$. Let $E_3 = E_3' \cap E_3''$, and note that, by the union bound, $E_3$ holds with probability at least $1 - \frac{\delta}{4}$. For the remainder of the proof, let us suppose the event $\bigcap_{j=0}^3 E_j$ occurs.

To arrive at the simpler claimed inequalities in (40), we follow a similar argument to the final part of the proof of Lemma 14. Explicitly, we first note that for any $k \in \{1, \ldots, K\}$, if $\Delta_{i_k} = \emptyset$, we trivially have $|S_k^2 \cap \Delta_{i_k}| = 0 = 2P_X(\Delta_{i_k})m_k' \leq 10\beta\overline{m}_k'$. On the other hand, if $\Delta_{i_k} \neq \emptyset$, the final claim in Lemma 11 implies $P_X(\Delta_{i_k}) > \varepsilon_{k+2}$, and combined with Lemma 13 this further implies $\hat{p}_k \geq P_X(\Delta_{i_k}) > \varepsilon_{k+2}$. Therefore, in this case,

$$2P_X(\Delta_{i_k})m_k' > \frac{2C''c_1^2}{C^4}\left( \mathsf{d} + \ln\left(\frac{4(3+N-k)^2}{\delta}\right) \right) \geq 6\ln\left(\frac{16(3+N-k)^2}{\delta}\right),$$

where the rightmost inequality follows from $c_1 \geq 1$ and $C'' \geq 6C^4$. Thus, the left inequality in (40) follows from (43). The right inequality in (40) follows immediately from the fact (established above) that $m_k' \leq \overline{m}_k'$, together with the fact (from Lemma 20) that $P_X(\Delta_{i_k}) \leq 5\beta$.

The remaining claims in the lemma follow from reasoning about convergence of the relevant series. Specifically, recalling that $\varepsilon_k = C^{1-k}$, $N = \lceil \log_C\left(\frac{2}{\varepsilon}\right) \rceil$, and $C = \frac{11}{10}$, we note that $\sum_{k=1}^{N+1} \frac{1}{\varepsilon_k^2} = \frac{1}{C^2-1}\left(C^{2(N+1)} - 1\right) \leq \frac{28}{\varepsilon^2}$ and

$$\sum_{k=1}^{N+1} \frac{1}{\varepsilon_k^2} \ln(3+N-k) = C^{2N} \sum_{j=0}^{N} C^{-2j} \ln(2+j) \leq C^{2N} \cdot 10 \leq \frac{49}{\varepsilon^2}.$$

Recalling $\overline{m}_k' = \frac{25C''c_1^2\beta}{\varepsilon_k^2}\left( \mathsf{d} + 2\ln(3+N-k) + \ln\left(\frac{4}{\delta}\right) \right)$, we have

$$\sum_{k=1}^{N+1} \overline{m}_k' \leq \frac{25C''c_1^2\beta}{\varepsilon^2}\left( 28\mathsf{d} + 2 \cdot 49 + 28\ln\left(\frac{4}{\delta}\right) \right) \leq M_2. \tag{44}$$

To obtain the query bound $Q(\varepsilon, \delta; \beta)$ in (41), note that the total number of queries is precisely

$$\left( \sum_{k=1}^K \left| S_k^2 \cap \Delta_{i_k} \right| \right) + \left( \sum_{k=1}^K \left| S_k^1 \cap D_{k-1} \setminus \Delta_{i_k} \right| \right). \tag{45}$$

By (40), the first term in (45) is upper bounded by $10\beta \cdot \sum_{k=1}^{N+1} \overline{m}'_k$, and (44) implies this is at most $10\beta M_2$. The second term in (45) is trivially upper bounded by $M_1 := \sum_{k=1}^{N+1} m_k$. Moreover, noting that $\varepsilon_k m_k$ is increasing in $k$, (39) implies the second term in (45) is also upper bounded by $3\mathfrak{s}\varepsilon_{N+1} \cdot m_{N+1} \cdot (N+1) \leq (3/2)\mathfrak{s}\varepsilon(N+1)m_{N+1}$. Together with the definition of $Q(\varepsilon, \delta; \beta)$ from (41), we have that the total number of queries (45) is at most $Q(\varepsilon, \delta; \beta)$.

The bound on the asymptotic form of $Q(\varepsilon, \delta; \beta)$ in (41) follows immediately from the definitions. Specifically, by definition of $M_2$, we have $10\beta M_2 = O\left(\frac{\beta^2}{\varepsilon^2}\left(\mathsf{d} + \log\left(\frac{1}{\delta}\right)\right)\right)$. Moreover, since $\varepsilon_{N+1} \geq \frac{\varepsilon}{2C}$, we have $(3/2)\mathfrak{s}\varepsilon(N+1)m_{N+1} = O\left(\mathfrak{s}\log\left(\frac{1}{\varepsilon}\right)\left(\mathsf{d}\log\left(\frac{1}{\varepsilon}\right) + \log\left(\frac{1}{\delta}\right)\right)\right)$, while (since each $k \leq N+1$ has $\varepsilon_k \geq \frac{\varepsilon}{2C}$), $M_1 = \sum_{k=1}^{N+1} m_k \leq \sum_{k=1}^{N+1} \frac{301 C'' c_0}{\varepsilon_k}\left(\mathsf{d}\log\left(\frac{2CC'' c_0}{\varepsilon}\right) + \log\left(\frac{1}{\delta}\right)\right) = O\left(\frac{1}{\varepsilon}\left(\mathsf{d}\log\left(\frac{1}{\varepsilon}\right) + \log\left(\frac{1}{\delta}\right)\right)\right)$ by evaluating the geometric series. ∎

As a final step before composing these lemmas into a proof of Theorem 5, we state an explicit bound on the number of unlabeled examples used by the algorithm. Much of this analysis is already implied by the above lemmas: namely, by definition, the number of examples allocated to data sets $S_k^1$ and $S_k^4$ is precisely $2M_1 = 2\sum_{k=1}^{N+1} m_k$, and Lemma 23 implies the number of examples allocated to data sets $S_k^2$ is at most $M_2$. What remains is to bound the number of examples allocated to the data sets $S_{k,i}^3$, which hinges on bounding the number of iterations of the '*While*' loop for each $k$. We have already noted, in Lemma 10, that $\max \mathcal{I}_k \leq \frac{1}{\varepsilon_{k+3}}$ on the event $E_0$, which already suffices to establish a coarse bound $\tilde{O}\left(\frac{\mathsf{d}}{\varepsilon^2}\right)$. However, we will need a slight refinement to obtain the claimed upper bound, which will follow from a combination of Lemmas 11 and 20.

**Lemma 24.** *On the event $\bigcap_{j=0}^3 E_j$, the total number of examples allocated to data sets $S_k^1$, $S_k^4$ ($k \leq N+1$), $S_k^2$ ($k \leq K$), and $S_{k,i}^3$ ($k \in \mathcal{K}$, $i \in \mathcal{I}_k$) is at most*

$$M(\varepsilon, \delta; \beta) := 3M_1 + M_2 + \frac{100C^4 \beta m_N}{\varepsilon} = O\left(\frac{\beta + \varepsilon}{\varepsilon^2}\left(\mathsf{d}\log\left(\frac{1}{\varepsilon}\right) + \log\left(\frac{1}{\delta}\right)\right)\right).$$

**Proof.** Suppose the event $\bigcap_{j=0}^3 E_j$ occurs. By definition, the number of examples allocated to data sets $S_k^1$ and $S_k^4$ is $m_k$ each, for $k \in \{1, \ldots, N+1\}$, so that the total number of such examples is $\sum_{k=1}^{N+1} 2m_k = 2M_1$. Also, by the first claim in Lemma 23, the number $m'_k$ of examples allocated to each $S_k^2$ data set (for $k \in \{1, \ldots, K\}$) satisfies $m'_k \leq \overline{m}'_k$. Moreover, Lemma 23 also establishes that $\sum_{k=1}^{N+1} \overline{m}'_k \leq M_2$. Together, we have that the total number of examples allocated to data sets $S_k^2$ is $\sum_{k=1}^K m'_k \leq M_2$. Thus, to complete the proof of Lemma 24, it suffices to bound the total number of examples allocated to data sets $S_{k,i}^3$ ($k \in \mathcal{K}$, $i \in \mathcal{I}_k$).

Toward this end, recall that for each $k \in \mathcal{K}$, each $S_{k,i}^3$ is of size $m_k$, and is allocated if and when the algorithm reaches Step 5 with values $(k, i)$. Thus, if $k = K$ (which, by the final claim in Lemma 10, occurs only if the algorithm returns in Step 4 in round $k$), then *no* examples are allocated to any $S_{k,i}^3$ sets in round $k$, whereas if $k < K$, then the number of $S_{k,i}^3$ data sets allocated during round $k$ is precisely the number of distinct values of $i$ encountered in round $k$: that is, $|\mathcal{I}_k|$. Moreover, note that since each time through the '*While*' loop increments $i$, each $k \in \mathcal{K}$ with $k < K$ has $|\mathcal{I}_k| = i_{k+1} - i_k + 1$. It follows that the total number of examples allocated to data sets $S_{k,i}^3$ in the algorithm is precisely $\sum_{k \in \mathcal{K}: k < K} m_k(i_{k+1} - i_k + 1)$.

Next we upper bound $i_{k+1} - i_k$ for each $k \in \mathcal{K}$ with $k < K$. Specifically, for any such $k$, note that $\Delta_{i_{k+1}} \setminus \Delta_{i_k} = \bigcup_{i=i_k}^{i_{k+1}-1}(\Delta_{i+1} \setminus \Delta_i)$, and by definition the sets $\Delta_{i+1} \setminus \Delta_i$ are disjoint over $i$. Moreover, by Lemma 11 and the definition of $\Delta_{i+1}$ in Step 7, any $i \in \{i_k, \ldots, i_{k+1} - 1\}$ has $P_X(\Delta_{i+1} \setminus \Delta_i) > \varepsilon_{k+3}$. Therefore,

$$P_X(\Delta_{i_{k+1}} \setminus \Delta_{i_k}) = \sum_{i=i_k}^{i_{k+1}-1} P_X(\Delta_{i+1} \setminus \Delta_i) \geq (i_{k+1} - i_k)\varepsilon_{k+3}.$$

On the other hand, by Lemma 20, $P_X(\Delta_{i_{k+1}} \setminus \Delta_{i_k}) \leq P_X(\Delta_{i_{k+1}}) \leq 5\beta$. Combining these inequalities, we conclude that $(i_{k+1} - i_k) \leq \frac{5C^3 \beta}{\varepsilon_k}$. Combined with the facts that $m_k \leq m_N$ and

$\sum_{k \in \mathcal{K}: k < K} m_k \leq M_1$, altogether we have

$$\sum_{k \in \mathcal{K}: k < K} m_k(i_{k+1} - i_k + 1) \leq M_1 + m_N \sum_{k=1}^{N} \frac{5C^3\beta}{\varepsilon_k} \leq M_1 + \frac{100C^4\beta m_N}{\varepsilon},$$

where the last inequality follows by evaluating the geometric series and recalling $\varepsilon_N \geq \frac{\varepsilon}{2C}$. This completes the proof that the total number of examples allocated to data sets $S_k^1, S_{k,i}^3, S_k^2, S_k^4$ is at most $M(\varepsilon, \delta; \beta)$. The claimed asymptotic form of $M(\varepsilon, \delta; \beta)$ follows immediately from the definitions of the quantities involved: namely, by definition, $3M_1 = \Theta\left(\frac{1}{\varepsilon}\left(\mathsf{d}\log\left(\frac{1}{\varepsilon}\right) + \log\left(\frac{1}{\delta}\right)\right)\right)$, $M_2 = \Theta\left(\frac{\beta}{\varepsilon^2}\left(\mathsf{d} + \log\left(\frac{1}{\delta}\right)\right)\right)$, and (since $\varepsilon_N \geq \frac{\varepsilon}{2C}$) $\frac{100C^4\beta m_N}{\varepsilon} = \Theta\left(\frac{\beta}{\varepsilon^2}\left(\mathsf{d}\log\left(\frac{1}{\varepsilon}\right) + \log\left(\frac{1}{\delta}\right)\right)\right)$. $\blacksquare$

We are now ready to combine the above lemmas into a complete proof of Theorem 5.

**Proof of Theorem 5.** By the union bound, the event $\bigcap_{j=0}^3 E_j$ has probability at least $1 - \delta$. By Lemma 24, on $\bigcap_{j=0}^3 E_j$, $\mathbb{A}_{\mathrm{avid}}$ uses at most $M(\varepsilon, \delta; \beta)$ (as defined in the lemma) of the examples in the sequence; in particular, this means that if we were to run the algorithm with a finite sequence $(X_1, Y_1), \ldots, (X_m, Y_m)$, for any $m \geq M(\varepsilon, \delta; \beta)$, then on the event $\bigcap_{j=0}^3 E_j$, the behavior of the algorithm (e.g., queries, returned $\hat{h}$) is identical to the *idealized* setting the above lemmas were established under (where there is an unlimited supply of examples), and hence the claims in the above lemmas remain valid. Thus, for any sample size $m \geq M(\varepsilon, \delta; \beta)$, on the event $\bigcap_{j=0}^3 E_j$, by Lemma 19 we have $\mathrm{er}_P(\hat{h}) \leq \inf_{h \in \mathbb{C}} \mathrm{er}_P(h) + \varepsilon$, and by Lemma 23 the total number of queries is at most $Q(\varepsilon, \delta; \beta)$ as defined therein. $\blacksquare$

**A remark on intersecting with $D_{k-1}$ in $\Delta_{i_k}$:** We remark that Theorem 5 remains valid if we restrict either (or both) $h_1, h_2$ to be in $V_{k-1}$ in the definition (2) of $V_{k-1}^{(4)}$. The entire proof remains valid (applying the same change to $V_{k-1}^{(3)}$), with the only exception being the first inequality in Lemma 20, which should then replace $\mathbb{C}$ by $V_{k-1}$. This change is of no consequence to the second inequality in the lemma since the proof of Lemma 17 in fact implies that $h^\star \in V_{k-1}$ holds simultaneously (on $E_0 \cap E_1 \cap E_2$) for *all* functions $h^\star \in \mathbb{C}$ satisfying (10) (and hence $\inf_{h \in V_{k-1}} \mathrm{er}_P(h) = \beta$). Moreover, with this restriction to require $h_2 \in V_{k-1}$ in $V_{k-1}^{(4)}$, we can extend the intersection with $D_{k-1}$ to the $\Delta_{i_k}$ region: that is, instead of querying all of $S_k^2 \cap \Delta_{i_k}$ (in Step 2) or $S_{N+1}^2 \cap \Delta_{i_{N+1}}$ (in Step 9), we can instead merely query the subset $S_k^2 \cap D_{k-1} \cap \Delta_{i_k}$ (in Step 2) or $S_{N+1}^2 \cap D_N \cap \Delta_{i_{N+1}}$ (in Step 9). With this change, we must then also modify the definition of $\hat{\mathrm{er}}_k^{1,2}(h)$ in (1) to $\hat{\mathrm{er}}_k^{1,2}(h) := \hat{P}_{S_k^1}(\mathrm{ER}(h) \cap D_{k-1} \setminus \Delta_{i_k}) + \hat{P}_{S_k^2}(\mathrm{ER}(h) \cap D_{k-1} \cap \Delta_{i_k})$. The argument in Lemma 15 extends to this modified definition of $\hat{\mathrm{er}}_k^{1,2}(h)$, since (as in the proof of (14) in Lemma 10) we are only interested in error *differences*, which, for $h, h' \in V_{k-1}$, satisfy $\hat{P}_{S_k^2}(\mathrm{ER}(h) \cap D_{k-1} \cap \Delta_{i_k}) - \hat{P}_{S_k^2}(\mathrm{ER}(h') \cap D_{k-1} \cap \Delta_{i_k}) = \hat{P}_{S_k^2}(\mathrm{ER}(h) \cap \Delta_{i_k}) - \hat{P}_{S_k^2}(\mathrm{ER}(h') \cap \Delta_{i_k})$. Indeed, with additional modifications to the proof, we can then even slightly refine the query complexity analysis, since if we replace the sets $\mathrm{ER}(h) \cap \Delta_{i_k}$ with $\mathrm{ER}(h) \cap D_{k-1} \cap \Delta_{i_k}$ in Lemma 14, the *envelope* set in the application of Lemma 8 in the proof of Lemma 14 can be chosen as $D_{k-1} \cap \Delta_{i_k}$, so that we can refine the definition of $\hat{p}_k$ to $2\hat{P}_{S_k^4}(D_{k-1} \cap \Delta_{i_k}) + O(\varepsilon_k)$. However, since these changes concern only the leading term $\frac{\beta^2}{\varepsilon^2}\left(\mathsf{d} + \log\left(\frac{1}{\delta}\right)\right)$ in Theorem 5, which is already optimal (perfectly matching the lower bounds of Kääriäinen, 2006; Beygelzimer, Dasgupta, and Langford, 2009), they are completely inconsequential to the theorem. We have therefore stated the algorithm without these modifications, for simplicity. However, this modified variant would be interesting in the context of $P$-dependent analysis, where it can lead to refinements to the leading term in the upper bound under certain favorable distributions. We leave the investigation of such refinements as an interesting direction for future work (focusing our $P$-dependent analysis in Appendix F on refining the *lower-order* term).

# F  Distribution-Dependent Analysis

In addition to analysis based on the star number (Hanneke and Yang, 2015), the active learning literature includes a variety of *distribution-dependent* complexity measures which have been used to analyze the query complexity in various contexts (see Appendix A). In this section, we will add to this line of work a distribution-dependent analysis of $\mathbb{A}_{\mathrm{avid}}$ which replaces the star number $\mathfrak{s}$ in Theorem 3 by a (never-larger) distribution-dependent quantity (Theorem 27), which can be further upper-bounded in terms of a simpler and more-familiar quantity: namely, a quadratic $\theta^2$ dependence in the *disagreement coefficient* (Hanneke, 2007b; Definition 25 below). We also show (in Appendix F.1) that it is *not possible* (by *any* algorithm) to obtain a lower-order term which replaces the star number in Theorem 3 with the disagreement coefficient $\theta$ itself, so that the aforementioned $\theta^2$ quadratic dependence generally cannot be reduced to linear (without introducing other factors). We will also present (in Appendix F.3) a slight refinement of $\mathbb{A}_{\mathrm{avid}}$, which replaces the region of disagreement $D_{k-1}$ by a carefully-chosen *subregion*, following the technique of Zhang and Chaudhuri (2014); Balcan, Broder, and Zhang (2007), which yields a corresponding refinement of the distribution-dependent query complexity bound. For instance, in the case of learning homogeneous linear classifiers under a uniform (or isotropic log-concave) distribution, this recovers a known query complexity bound $\tilde{O}\left(\mathsf{d}\frac{\beta^2}{\varepsilon^2} + \mathsf{d}\right)$ (and indeed, improves log factors in the lead term compared to prior works).

**The Disagreement Coefficient:**    In the context of *agnostic* active learning, the most commonly-used $P$-dependent complexity measure is the *disagreement coefficient*, introduced by Hanneke (2007b), defined as follows.

**Definition 25.** *For any concept class $\mathbb{C}$ and distribution $P_X$ on $\mathcal{X}$, for any measurable function $f : \mathcal{X} \to \{0,1\}$, for any $\varepsilon \geq 0$, the* disagreement coefficient, *denoted by $\theta_{P_X,f}(\varepsilon)$, is defined as*

$$\theta_{P_X,f}(\varepsilon) := \sup_{r > \varepsilon} \frac{P_X(\mathrm{DIS}(\mathrm{B}_{P_X}(f,r)))}{r} \vee 1,$$

*where $\mathrm{B}_{P_X}(f,r) := \{h \in \mathbb{C} : P_X(h \neq f) \leq r\}$ denotes the $r$-ball centered at $f$, and $\mathrm{DIS}(\mathbb{C}') := \{x \in \mathcal{X} : \exists h, h' \in \mathbb{C}', h(x) \neq h'(x)\}$ denotes the* region of disagreement *(as in Section 4). For any distribution $P$ on $\mathcal{X} \times \{0,1\}$ and $\varepsilon > 0$, for $h^\star$ as in (10),[14] define $\theta_P(\varepsilon) := \theta_{P_X,h^\star}(\varepsilon)$.*

There are many works establishing bounds on the disagreement coefficient for commonly-studied classes $\mathbb{C}$ under various restrictions on the distribution $P$ (see Hanneke, 2014, for a detailed summary). As discussed in Appendix A, the disagreement coefficient commonly appears in analyses of the query complexity of *disagreement-based* active learning methods (e.g., Hanneke, 2007b, 2009b, 2011, 2014; Dasgupta, Hsu, and Monteleoni, 2007). Since the lower-order term in Theorem 3 arises from the analysis of queries in the region of disagreement $\mathrm{DIS}(V_{k-1})$ of $V_{k-1}$, one might naturally wonder whether we can replace $\mathfrak{s}$ with $\theta_P(\varepsilon)$ in the upper bound in Theorem 3. Hanneke and Yang (2015) have shown that $\sup_P \theta_P(\varepsilon) = \mathfrak{s} \wedge \frac{1}{\varepsilon}$ for $\varepsilon \in (0,1]$, which implies that if we could replace $\mathfrak{s}$ by $\theta_P(\varepsilon)$ it would indeed represent a distribution-dependent refinement of the upper bound in Theorem 3. However, it turns out this is *not possible* (by *any* algorithm) for some classes $\mathbb{C}$, as we demonstrate by an example in Appendix F.1. Following this, in Appendix F.2, we find that it is possible to achieve a lower-order term $\tilde{O}(\mathsf{d}\theta_P(\beta + \varepsilon)^2)$, and indeed this is achieved by $\mathbb{A}_{\mathrm{avid}}$. This quadratic dependence unfortunately means the upper bound is sometimes loose (i.e., sometimes larger than that in Theorem 3). However, as an intermediate step, we also establish a query complexity bound (Theorem 27) expressed in terms of a modified disagreement coefficient which is *never larger* than the $\mathfrak{s}$-dependent query complexity bound in Theorem 3 (though which is more difficult to evaluate due to a more-involved definition).

---

[14]When $h^\star$ is not uniquely defined, in principle we can define $\theta_P(\varepsilon)$ as the *infimum* value among all choices of such $h^\star$. It is also possible to define $h^\star$ as an $\varepsilon$-independent fixed function, even when $\mathrm{er}_P(h)$ does not have a minimizer in $\mathbb{C}$, by choosing it as an element of the $L_1(P_X)$-closure of $\mathbb{C}$ having $\mathrm{er}_P(h^\star) = \inf_{h \in \mathbb{C}} \mathrm{er}_P(h)$: see (Hanneke, 2012) for a proof that such an $h^\star$ always exists when $\mathrm{VC}(\mathbb{C}) < \infty$. In particular, with such an $h^\star$, the limiting value $\theta_P(0) := \theta_{P_X,h^\star}(0)$ is also well-defined.

## F.1 Impossibility of Replacing $\mathfrak{s}$ with $\theta_P(\varepsilon)$

In this section, we present an example demonstrating that *no algorithm* can achieve a $P$-dependent query complexity bound which replaces $\mathfrak{s}$ by $\theta_P(\varepsilon)$ in Theorem 3.

**An Example:** Consider the following concept class (see Hanneke, 2007b, for a related construction). Let $\mathcal{X} = \mathbb{Z}$ (the integers) and define a concept class

$$\mathbb{C}_{\mathrm{ts}} := \{\mathbb{1}_{\{-t\} \cup [t,\infty)} : t \in \mathbb{N}\}.$$

In other words, each $h \in \mathbb{C}_{\mathrm{ts}}$ defines a *threshold* classifier on the *positive* integers and a *singleton* classifier on the *negative* integers, and the position $-t$ of the singleton point mirrors the position $t$ of the threshold boundary point.

Fix any $\varepsilon, \beta \in (0, 1/3)$, denote by $n = \frac{1-2\beta}{2\varepsilon}$, and for simplicity suppose $n \in \mathbb{N}$. Define a marginal distribution $P_X$ on $\mathcal{X}$ as follows: $\forall x \in \{1, \ldots, n\}$, $P_X(\{x\}) = \frac{2\beta}{n}$ and $P_X(\{-x\}) = 2\varepsilon$. Note that this completely specifies $P_X$. Now define a family of probability distributions $P_1, \ldots, P_n$: for each $t \in \{1, \ldots, n\}$, $P_t$ has marginal distribution $P_X$ on $\mathcal{X}$ and conditional distribution $\forall x \in \mathcal{X}$

$$P_t(Y = 1 | X = x) = \begin{cases} 1, & \text{if } x = -t \\ 0, & \text{if } x \in \{-1, \ldots, -n\} \setminus \{-t\} \\ \frac{1}{2}, & \text{otherwise} \end{cases}.$$

In particular, note that each $P_t$ satisfies $\inf_{h \in \mathbb{C}_{\mathrm{ts}}} \mathrm{er}_{P_t}(h) = \beta$ and $h^\star$ is uniquely equal $\mathbb{1}_{\{-t\} \cup [t,\infty)}$.[15]

**Upper-bounding $\theta_{P_t}(\varepsilon)$:** For any $P_t$, we will argue $\theta_{P_t}(\varepsilon) \leq \theta_{P_X, h^\star}(0) = O(\frac{1}{\beta})$. For any $r > 0$, if $h \in \mathrm{B}_{P_X}(h^\star, r)$, then letting $k_h := |\{h \neq h^\star\} \cap \{1, \ldots, n\}|$, since each $x \in \{1, \ldots, n\}$ has $P_X(\{x\}) = \frac{2\beta}{n}$, we must have $k_h \leq k_r := \lfloor \frac{rn}{2\beta} \rfloor$. Since $h$ and $h^\star$ both implement threshold functions in this region $\{1, \ldots, n\}$, the $k_h$ elements in $\{h \neq h^\star\} \cap \{1, \ldots, n\}$ are a contiguous segment: either $\{t, \ldots, t + k_h - 1\}$ or $\{t - k_h, \ldots, t - 1\}$. In either case, we have $\{h \neq h^\star\} \cap \{1, \ldots, n\} \subseteq \{t - k_r, \ldots, t + k_r - 1\} \cap \{1, \ldots, n\}$. Moreover, since $h = \mathbb{1}_{\{-t'\} \cup [t',\infty)}$ for some $t' \in \mathbb{N}$, this further implies $\{h \neq h^\star\} \cap \{-1, \ldots, -n\} \subseteq \{-(t - k_r), \ldots, -(t + k_r)\} \cap \{-1, \ldots, -n\}$. Since $\mathrm{DIS}(\mathrm{B}_{P_X}(h^\star, r))$ is just the union of these $\{h \neq h^\star\}$ regions among all $h \in \mathrm{B}_{P_X}(h^\star, r)$, we have

$$\mathrm{DIS}(\mathrm{B}_{P_X}(h^\star, r)) \cap \{-n, \ldots, -1, 1, \ldots, n\}$$
$$\subseteq (\{-(t - k_r), \ldots, -(t + k_r)\} \cap \{-1, \ldots, -n\}) \cup (\{t - k_r, \ldots, t + k_r - 1\} \cap \{1, \ldots, n\})$$
$$\implies P_X(\mathrm{DIS}(\mathrm{B}_{P_X}(h^\star, r))) \leq (2k_r + 1)2\varepsilon + 2k_r \frac{2\beta}{n} \leq \frac{r(1-2\beta)}{\beta} + 2\varepsilon + 2r.$$

We also note that any $r < 2\varepsilon$ has $\mathrm{B}_{P_X}(h^\star, r) = \{h^\star\}$. Altogether,

$$\theta_{P_t}(\varepsilon) \leq \theta_{P_X, h^\star}(0) \leq \sup_{r \geq 2\varepsilon} \frac{r((1-2\beta)/\beta) + 2\varepsilon + 2r}{r} = \frac{1-2\beta}{\beta} + 3 = O\left(\frac{1}{\beta}\right).$$

**Lower-bounding the query complexity:** On the other hand, we will argue that the query complexity is $\Omega(\frac{1}{\varepsilon})$ under the assumption $P \in \{P_1, \ldots, P_n\}$. Note that every $h : \mathcal{X} \to \{0, 1\}$ has the *same* value of $P_t(\mathrm{ER}(h) \setminus \{-1, \ldots, -n\}) = \beta$. Together with the definition of $P_t$ in $\{-1, \ldots, -n\}$, this implies any $h : \mathcal{X} \to \{0, 1\}$ with $\{h \neq h^\star\} \cap \{-1, \ldots, -n\} \neq \emptyset$ has $\mathrm{er}_{P_t}(h) - \mathrm{er}_{P_t}(h^\star) \geq 2\varepsilon$. Thus, the problem of learning, to an excess error $\varepsilon$ (under the assumption that $P \in \{P_t : t \in \{1, \ldots, n\}\}$) is equivalent to the problem of *identifying* the value $t \in \{1, \ldots, n\}$ for which the distribution $P = P_t$: that is, if an algorithm returns $\hat{h}$ with $\mathrm{er}_{P_t}(\hat{h}) - \mathrm{er}_{P_t}(h^\star) \leq \varepsilon$, the unique $x \in \{1, \ldots, n\}$ for which $\hat{h}(-x) = 1$ satisfies $x = t$. Moreover, in the active learning problem defined by these distributions, for every $x \notin \{-1, \ldots, -n\}$, the conditional distribution $P_t(Y = 1 | X = x)$ of responses to queries for examples at $x$ is *invariant* to $t$, so that such queries reveal no information about which

---

[15]Indeed, these distributions $P_t$ even satisfy the stronger *benign noise* property: i.e., $\inf_{h \in \mathbb{C}_{\mathrm{ts}}} \mathrm{er}_{P_t}(h) =$ Bayes risk (a setting studied by Hanneke, 2009b; Hanneke and Yang, 2015). Thus, the argument in this section further implies the impossibility of using $\theta_P(\varepsilon)$ in the lower-order term under benign noise.

$t$ has $P = P_t$, and hence without loss of generality we can restrict to active learning algorithms that do not query outside $\{-1, \ldots, -n\}$. The unlabeled examples also reveal no such information, since all $P_t$ have the same marginal distribution $P_X$. Altogether, the active learning problem for this set of distributions is information-theoretically no easier (in terms of query complexity) than the problem of actively identifying a *singleton* classifier on $\{-1, \ldots, -n\}$ in the realizable case under marginal Uniform($\{-1, \ldots, -n\}$).[16] It is well known that the minimax query complexity of this latter problem (with confidence parameter $\delta = 1/3$) is $\Omega(n) = \Omega(\frac{1}{\varepsilon})$ (Dasgupta, 2004, 2005; Hanneke, 2014; Hanneke and Yang, 2015), which therefore serves as a lower bound on the minimax query complexity for $P \in \{P_t : t \in \{1, \ldots, n\}\}$: that is, for every active learning algorithm, there exists $P \in \{P_t : t \in \{1, \ldots, n\}\}$ for which, with probability at least $\delta$, it either makes $\Omega(\frac{1}{\varepsilon})$ queries or returns $\hat{h}$ with $\mathrm{er}_P(\hat{h}) > \inf_{h \in \mathbb{C}_{\mathrm{ts}}} \mathrm{er}_P(h) + \varepsilon$.

**Conclusion that $\theta_{P_t}(\varepsilon)$ is not achievable in the lower-order term:** From the above arguments, we can conclude that for the class $\mathbb{C}_{\mathrm{ts}}$, it is *not possible* to replace $\mathfrak{s}$ by $\theta_P(\varepsilon)$ (or indeed $\theta_P(0)$) in the upper bound of Theorem 3 to obtain a $P$-dependent refinement of the upper bound. Formally, for *every* active learning algorithm guaranteeing that, for every $P$ with $\inf_{h \in \mathbb{C}_{\mathrm{ts}}} \mathrm{er}_P(h) \le \beta$, with probability at least $2/3$ (i.e., $\delta = 1/3$), it returns $\hat{h}$ with $\mathrm{er}_P(\hat{h}) \le \inf_{h \in \mathbb{C}_{\mathrm{ts}}} \mathrm{er}_P(h) + \varepsilon$, there exists a distribution $P$ satisfying this for which $\theta_P(\varepsilon) \le \frac{1-2\beta}{\beta} + 3 = O(\frac{1}{\beta})$, yet with probability at least $1/3$, the algorithm makes a number of queries $\Omega(\frac{1}{\varepsilon})$. We have argued this conclusion for any choices of $\varepsilon, \beta \in (0, 1/3)$ (with $n \in \mathbb{N}$ for simplicity). In particular, for $\varepsilon \ll \beta \ll \sqrt{\varepsilon}$ (e.g., $\beta \approx \varepsilon^{2/3}$), such a distribution $P$ has $\frac{\beta^2}{\varepsilon^2} + \theta_P(\varepsilon) = \frac{\beta^2}{\varepsilon^2} + O(\frac{1}{\beta}) \ll \frac{1}{\varepsilon}$, so that replacing $\mathfrak{s}$ with $\theta_P(\varepsilon)$ in Theorem 3 *cannot* yield a valid query complexity bound (holding for all $P$) for *any* active learning algorithm. Indeed, we have established that this conclusion also holds for $\theta_P(0)$ (as defined in footnote 14).

We will see in Corollary 28 of Appendix F.2 that $\mathbb{A}_{\mathrm{avid}}$ *does* achieve an upper-bound $\tilde{O}\left(\mathrm{d}\theta_P(\beta + \varepsilon)^2\right)$ on the lower-order term: a *quadratic* dependence on the disagreement coefficient. This conclusion is compatible with the above scenario, since $\frac{\beta^2}{\varepsilon^2} + \frac{1}{\beta^2} = \Omega\left(\frac{1}{\varepsilon}\right)$ for the full range of $\beta, \varepsilon$.

## F.2 Replacing $\mathfrak{s}$ with $\theta_P(\beta + \varepsilon)^2$

Appendix F.1 implies the disagreement coefficient $\theta_P(\varepsilon)$, as defined in Definition 25, cannot be used as a $P$-dependent substitute for the star number $\mathfrak{s}$ in Theorem 3 (at least, not with a *linear* dependence). In this section, we will argue that the AVID Agnostic algorithm $\mathbb{A}_{\mathrm{avid}}$ *does* achieve a $P$-dependent lower-order term which is at most *quadratic* in the disagreement coefficient: namely, $\tilde{O}(\mathrm{d}\theta_P(\beta + \varepsilon)^2)$. We will argue this by first establishing a $P$-dependent refinement of Theorem 5 based on a modified disagreement coefficient (Definition 26) which is *never larger* than the star number. While this quantity itself is often more-difficult to calculate, compared to the original disagreement coefficient $\theta_P(\varepsilon)$, fortunately it is always upper bounded by $O\left(\frac{\beta^2}{\varepsilon^2} + \theta_P(\beta + \varepsilon)^2\right)$. In particular, this means that for any $P$ with $\theta_P(0) < \infty$, the asymptotic dependence on $\varepsilon, \delta$ in the lower-order term in Theorem 3 can be reduced to $\mathrm{polylog}\left(\frac{1}{\varepsilon\delta}\right)$.

Specifically, the modified disagreement coefficient we consider can be expressed as the value $\theta_{P_\Delta, h^\star}(\varepsilon)$ produced under a *restriction* of $P_X$ to a subregion $\mathcal{X} \setminus \Delta$ of size at least $1 - O(\beta)$. Toward stating the definition, we first extend Definition 25 to allow for general measures $\mu$: that is, for any measure $\mu$ on $\mathcal{X}$ and measurable $f : \mathcal{X} \to \{0, 1\}$, define $\mathrm{B}_\mu(f, r) := \{h \in \mathbb{C} : \mu(h \ne f) \le r\}$,

---

[16]Formally, for any active learning algorithm $\mathbb{A}$, under distributions $P \in \{P_t : t \in \{1, \ldots, n\}\}$, we can convert $\mathbb{A}$ into an active learner $\mathbb{A}'$ for realizable-case singletons under Uniform($\{-1, \ldots, -n\}$) with at most the query complexity of $\mathbb{A}$ under such distributions $P$. Specifically, given any number $m$ of i.i.d. unlabeled examples $X_1, \ldots, X_m \sim$ Uniform($\{-1, \ldots, -n\}$), define independent random variables (also independent of $X_1, \ldots, X_m$) $B_1, \ldots, B_m \sim$ Bernoulli($2\beta$), $X_1', \ldots, X_m' \sim$ Uniform($\{1, \ldots, n\}$), and $Y_1', \ldots, Y_m' \sim$ Bernoulli($\frac{1}{2}$). For each $i \le m$, let $X_i'' = X_i$ if $B_i = 0$ and $X_i'' = X_i'$ if $B_i = 1$. Then $\mathbb{A}'$ runs $\mathbb{A}$ with unlabeled data $X_1'', \ldots, X_m''$; whenever $\mathbb{A}$ queries an $X_i''$ with $B_i = 0$, $\mathbb{A}'$ queries for the label $Y_i$ of $X_i$ and gives this as a response to the query, and whenever $\mathbb{A}$ queries an $X_i''$ with $B_i = 1$, $\mathbb{A}'$ gives $Y_i'$ as a response to the query. Note that the corresponding data sequence and responses observed by $\mathbb{A}$ are indeed identical to running $\mathbb{A}$ under $P = P_t$, where $-t$ is the singleton location for the realizable-case singleton problem $P_t(\cdot|\{-1, \ldots, -n\})$. Thus, the query complexity of $\mathbb{A}'$ identifying the $t$ for the realizable-case singletons distribution $P_t(\cdot|\{-1, \ldots, -n\})$ is at most that of $\mathbb{A}$ identifying this $t$ when $P = P_t$.

and for $\varepsilon \geq 0$ define $\theta_{\mu,f}(\varepsilon) := \sup_{r > \varepsilon} \frac{\mu(\mathrm{DIS}(\mathrm{B}_\mu(f,r)))}{r} \vee 1$. We then consider the following definition: a *region-excluded disagreement coefficient.*

**Definition 26.** *For any distribution $P$ on $\mathcal{X} \times \{0,1\}$ and any measurable $\Delta \subseteq \mathcal{X}$, define a measure $A \mapsto P_\Delta(A) := P_X(A \setminus \Delta)$. For any $\varepsilon, \tau \geq 0$, for $h^\star \in \mathbb{C}$ as in* (10) *(under $P$),*[17] *define*

$$\theta_P(\varepsilon;\tau) := \sup_{\Delta \subseteq \mathcal{X}: P_X(\Delta) \leq \tau} \theta_{P_\Delta, h^\star}(\varepsilon).$$

We can equivalently define $\theta_P(\varepsilon;\tau)$ as the disagreement coefficient under a worst-case *conditional* distribution $P_X(\cdot | \mathcal{X} \setminus \Delta)$: that is,

$$\theta_P(\varepsilon;\tau) = \sup_{\Delta \subseteq \mathcal{X}: P_X(\Delta) \leq \tau} \theta_{P_X(\cdot | \mathcal{X} \setminus \Delta), h^\star}(\varepsilon / P_X(\mathcal{X} \setminus \Delta)), \tag{46}$$

where we define $\theta_{P_X(\cdot | \mathcal{X} \setminus \Delta), h^\star}(\varepsilon / P_X(\mathcal{X} \setminus \Delta)) = 1$ in the case $P_X(\mathcal{X} \setminus \Delta) = 0$ (which coincides with the value $\theta_{P_\Delta, h^\star}(\varepsilon)$ for such $\Delta$).

We may note that $\theta_P(\varepsilon;\tau)$ indeed provides a *refinement* of the star number, in that it is *never larger*. Specifically, since Hanneke and Yang (2015) have shown

$$\sup_{P_X} \sup_{h \in \mathbb{C}} \theta_{P_X, h}(\varepsilon) = \mathfrak{s} \wedge \frac{1}{\varepsilon} \tag{47}$$

for every $\varepsilon \in (0,1]$, the expression in (46) of $\theta_P(\varepsilon;\tau)$ as the disagreement coefficient under *conditional* distributions immediately implies

$$\theta_P(\varepsilon;\tau) \leq \mathfrak{s} \wedge \frac{1}{\varepsilon}. \tag{48}$$

Thus, replacing $\mathfrak{s}$ in Theorem 3 by $\theta_P(\varepsilon;\tau)$ would indeed yield a (never-larger) $P$-dependent refinement.

We give examples below (Appendix F.2.1) of calculating and upper-bounding $\theta_P(\varepsilon;\tau)$ under various scenarios $(\mathbb{C}, P)$. We remark that, due to the supremum over regions $\Delta$, the quantity $\theta_P(\varepsilon;\tau)$ is often *much* more involved to calculate or bound compared to the original disagreement coefficient $\theta_P(\varepsilon)$ in Definition 25. We might therefore think of $\theta_P(\varepsilon;\tau)$ as a kind of *intermediate* complexity measure, which is useful in that it provides a $P$-dependent refinement of $\mathfrak{s}$, while also admitting general upper bounds which are more accessible than directly calculating $\theta_P(\varepsilon;\tau)$. Concretely, there are at least *weak* relations between $\theta_P(\varepsilon;\tau)$ and the more-familiar disagreement coefficient from Definition 25: namely, $\theta_P(\varepsilon) \leq \theta_P(\varepsilon;\tau)$ and

$$\theta_P(\varepsilon;\tau) \leq \sup_{r > \varepsilon} \frac{P_X(\mathrm{DIS}(\mathrm{B}_{P_X}(h^\star, \tau + r)))}{r} \vee 1$$

$$\leq \theta_P(\tau + \varepsilon)\left(\frac{\tau + \varepsilon}{\varepsilon}\right) \leq \theta_P(\tau + \varepsilon)^2 + \left(\frac{\tau + \varepsilon}{\varepsilon}\right)^2. \tag{49}$$

These upper bounds on $\theta_P(\varepsilon;\tau)$ are noteworthy since $\theta_P(\tau + \varepsilon)$ is typically significantly easier to calculate compared to directly calculating $\theta_P(\varepsilon;\tau)$ (and there are already many works deriving bounds on $\theta_P(\tau + \varepsilon)$ for various scenarios; see Hanneke, 2014).

The quantity $\theta_P(\varepsilon;\tau)$ is particularly well-suited for the analysis of $\mathbb{A}_{\mathrm{avid}}$, since the algorithm explicitly maintains low diameter of $V_k$ under a region-excluded measure $A \mapsto P_X(A \setminus \Delta_{i_k})$. Specifically, Lemma 12 implies $V_{k-1} \subseteq \mathrm{B}_{P_{\Delta_{i_k}}}(h^\star, \varepsilon_k)$, while Lemma 20 implies $P_X(\Delta_{i_k}) \leq 5\beta$, so that $P_X(D_{k-1} \setminus \Delta_{i_k}) \leq \theta_P(\varepsilon_k; 5\beta)\varepsilon_k$, and hence the number of queries in $S_k^1 \cap D_{k-1} \setminus \Delta_{i_k}$ is $O(\theta_P(\varepsilon_k; 5\beta)\varepsilon_k m_k) = \tilde{O}(\theta_P(\varepsilon; 5\beta)\mathrm{d})$. Formally, this leads to the following result, which simply replaces $\mathfrak{s}$ with $\theta_P(\varepsilon; 5\beta)$ in the lower-order term compared to Theorem 5. Due to (48), the query complexity bound in this result is *never larger* than that of Theorem 5 (and below we discuss scenarios where it is strictly smaller). We remark that, based on the comment preceding Lemma 20, the factor "5" in $\theta_P(\varepsilon; 5\beta)$ in this theorem can be reduced to any value $c > 2$ by appropriately adjusting the constants $C, C''$ in the algorithm.

---

[17]The remarks concerning the choice of $h^\star$ in footnote 14 also apply here, noting that the lemmas concerning $h^\star$ in Appendix E actually apply simultaneously to all functions $h^\star \in \mathbb{C}$ satisfying (10).

**Theorem 27** (Distribution-dependent Query Complexity of AVID Agnostic). *For any concept class* $\mathbb{C}$ *with* $\mathrm{VC}(\mathbb{C}) < \infty$, *letting* $\mathsf{d} = \mathrm{VC}(\mathbb{C})$, *for every distribution* $P$ *on* $\mathcal{X} \times \{0,1\}$, *letting* $\beta = \inf_{h \in \mathbb{C}} \mathrm{er}_P(h)$, *for any* $\varepsilon, \delta \in (0,1)$, *if the algorithm* $\mathbb{A}_{\mathrm{avid}}$ *is executed with parameters* $(\varepsilon, \delta)$, *with any number* $m \geq M(\varepsilon, \delta; \beta)$ *of i.i.d.-P examples (for* $M(\varepsilon, \delta; \beta)$ *as in Theorem 5, defined in Lemma 24), then with probability at least* $1 - \delta$, *the returned predictor* $\hat{h}$ *satisfies* $\mathrm{er}_P(\hat{h}) \leq \inf_{h \in \mathbb{C}} \mathrm{er}_P(h) + \varepsilon$ *and the algorithm makes a number of queries at most* $Q(\varepsilon, \delta; P)$ *satisfying*

$$
Q(\varepsilon, \delta; P) = O\left( \frac{\beta^2}{\varepsilon^2} \left( \mathsf{d} + \log\left( \frac{1}{\delta} \right) \right) + \min\left\{ \theta_P(\varepsilon; 5\beta) \log\left( \frac{1}{\varepsilon} \right), \frac{1}{\varepsilon} \right\} \left( \mathsf{d} \log\left( \frac{1}{\varepsilon} \right) + \log\left( \frac{1}{\delta} \right) \right) \right)
$$

$$
= \tilde{O}\left( \mathsf{d} \frac{\beta^2}{\varepsilon^2} + \mathsf{d} \theta_P(\varepsilon; 5\beta) \right).
$$

**Proof.** The result follows identically to Theorem 5, with only one minor change: replacing $\mathfrak{s}$ with $2C\theta_P(\varepsilon; 5\beta)$ in Lemma 22. Note that this one change will suffice, since every subsequent appearance of $\mathfrak{s}$ in the proof is due to its appearance in Lemma 22, and hence changing $\mathfrak{s}$ to $2C\theta_P(\varepsilon; 5\beta)$ in this lemma allows us to make the same change in every subsequent appearance of $\mathfrak{s}$ in the proof.

To see why Lemma 22 remains valid with this change, first note that its proof establishes that, on the event $E_0 \cap E_1 \cap E_2$, in the non-trivial case of $P_X(\mathcal{X} \setminus \Delta_{i_k}) \neq 0$, (38) holds. Rather than relaxing the third expression in (38) using the star number, we can instead relax it using $\theta_P(\varepsilon; 5\beta)$: that is, for $P_k$ as defined in that context, (38) implies

$$
P_X(D_{k-1} \setminus \Delta_{i_k}) = P_k(D_{k-1}) P_X(\mathcal{X} \setminus \Delta_{i_k}) \leq P_k\left( \mathrm{DIS}\left( \mathrm{B}_{P_k}\left( h^\star, \frac{\varepsilon_k}{P_X(\mathcal{X} \setminus \Delta_{i_k})} \right) \right) \right) P_X(\mathcal{X} \setminus \Delta_{i_k})
$$

$$
= P_{\Delta_{i_k}}\left( \mathrm{DIS}\left( \mathrm{B}_{P_{\Delta_{i_k}}}(h^\star, \varepsilon_k) \right) \right) \leq \theta_P(\varepsilon/(2C); 5\beta) \varepsilon_k,
$$

where the last inequality follows from Definition 26, the fact that $\varepsilon_k > \frac{\varepsilon}{2C}$, and the fact (from Lemma 20) that $P_X(\Delta_{i_k}) \leq 5\beta$. We then note that (as in Corollary 7.2 of Hanneke, 2014) for any $\Delta \subseteq \mathcal{X}$,

$$
\theta_{P_\Delta, h^\star}(\varepsilon/(2C)) = \sup_{r > \varepsilon} \frac{P_\Delta(\mathrm{DIS}(\mathrm{B}_{P_\Delta}(h^\star, r/(2C))))}{r/(2C)} \vee 1 \leq 2C \sup_{r > \varepsilon} \frac{P_\Delta(\mathrm{DIS}(\mathrm{B}_{P_\Delta}(h^\star, r)))}{r} \vee 1,
$$

and therefore $\theta_P(\varepsilon/(2C); 5\beta) \leq 2C\theta_P(\varepsilon; 5\beta)$. Altogether, we have that Lemma 22 remains valid while replacing $\mathfrak{s}$ with $2C\theta_P(\varepsilon; 5\beta)$. $\blacksquare$

We emphasize that $\mathbb{A}_{\mathrm{avid}}$ *does not need to know* the value $\theta_P(\varepsilon; 5\beta)$ (or anything else about $P$) to achieve this query complexity: that is, it is *adaptive* to the value of $\theta_P(\varepsilon; 5\beta)$.

Together with (49), the above result further implies a (sometimes loose) relaxation, in which the lower-order term has a *quadratic* dependence on $\theta_P(\beta + \varepsilon)$, as formally stated in the following corollary (compare this with Appendix F.1, which showed it is impossible to generally reduce this $\theta_P(\beta + \varepsilon)^2$ term to a linear term $\theta_P(\beta + \varepsilon)$ or even $\theta_P(0)$).

**Corollary 28.** *The query complexity bound* $Q(\varepsilon, \delta; P)$ *in Theorem 27 (achieved by* $\mathbb{A}_{\mathrm{avid}}$*) satisfies*

$$
Q(\varepsilon, \delta; P) = O\left( \frac{\beta^2}{\varepsilon^2} \left( \mathsf{d} + \log\left( \frac{1}{\delta} \right) \right) \right) + \tilde{O}\left( \min\left\{ \mathsf{d} \theta_P(\beta + \varepsilon) \left( \frac{\beta + \varepsilon}{\varepsilon} \right), \frac{\mathsf{d}}{\varepsilon} \right\} \right)
$$

$$
= O\left( \frac{\beta^2}{\varepsilon^2} \left( \mathsf{d} + \log\left( \frac{1}{\delta} \right) \right) \right) + \tilde{O}\left( \min\left\{ \mathsf{d} \theta_P(\beta + \varepsilon)^2, \frac{\mathsf{d}}{\varepsilon} \right\} \right).
$$

**Proof.** Due to the first two inequalities in (49), and $\theta_P(5\beta + \varepsilon) \leq \theta_P(\beta + \varepsilon)$, the second term in the expression of $Q(\varepsilon, \delta; P)$ in Theorem 27 is at most

$$
O\left( \min\left\{ \theta_P(\beta + \varepsilon) \log\left( \frac{1}{\varepsilon} \right) \left( \frac{\beta + \varepsilon}{\varepsilon} \right), \frac{1}{\varepsilon} \right\} \left( \mathsf{d} \log\left( \frac{1}{\varepsilon} \right) + \log\left( \frac{1}{\delta} \right) \right) \right). \tag{50}
$$

Relaxing $\mathsf{d} \log\left( \frac{1}{\varepsilon} \right) + \log\left( \frac{1}{\delta} \right) \leq \log\left( \frac{1}{\varepsilon} \right) \left( \mathsf{d} + \log\left( \frac{1}{\delta} \right) \right)$ and noting that

$$
\theta_P(\beta + \varepsilon) \log^2\left( \frac{1}{\varepsilon} \right) \left( \frac{\beta + \varepsilon}{\varepsilon} \right) \leq \theta_P(\beta + \varepsilon)^2 \log^4\left( \frac{1}{\varepsilon} \right) + \left( \frac{\beta + \varepsilon}{\varepsilon} \right)^2,
$$

and $\left(\frac{\beta+\varepsilon}{\varepsilon}\right)^2 \leq 4\frac{\beta^2}{\varepsilon^2} + 4$, the quantity (50) is at most

$$O\left(\frac{\beta^2}{\varepsilon^2}\left(\mathsf{d} + \log\left(\frac{1}{\delta}\right)\right) + \min\left\{\theta_P(\beta+\varepsilon)^2 \log^4\left(\frac{1}{\varepsilon}\right), \frac{1}{\varepsilon}\log\left(\frac{1}{\varepsilon}\right)\right\}\left(\mathsf{d} + \log\left(\frac{1}{\delta}\right)\right)\right).$$

Adding this to the first term in the expression of $Q(\varepsilon, \delta; P)$, the result follows. ∎

In particular, Corollary 28 implies that, whenever $\theta_P(0) < \infty$, the dependence on $\beta, \varepsilon$ in the query complexity bound in Theorem 27 is of order $\frac{\beta^2}{\varepsilon^2} + \text{polylog}\left(\frac{1}{\varepsilon}\right)$. For instance, see (Hanneke, 2014, Chapter 7) for some general conditions on $(\mathbb{C}, P)$ under which this occurs. We remark that the *first* bound in Corollary 28 is at least never larger than the upper bound in Theorem 1, since we always have $\theta_P(\beta+\varepsilon)\left(\frac{\beta+\varepsilon}{\varepsilon}\right) \leq \frac{1}{\varepsilon}$; however, we note that this is *not* the case for the *second* upper bound in Corollary 28. Beyond these basic observations, there exist scenarios $(\mathbb{C}, P)$ where both upper bounds in Corollary 28 are *loose* compared to Theorem 27, to such an extent that they are sometimes even larger than the $\mathfrak{s}$-dependent bound in Theorem 5 (see Example 6 below). It is for this reason that we have chosen to express Theorem 27 in terms of the more-complicated quantity $\theta_P(\varepsilon; \tau)$, to provide a starting point for $P$-dependent analysis that is at least never worse than Theorem 5.

### F.2.1 Examples

We next present some examples illustrating the values of the lower-order terms in Theorem 27 and Corollary 28 by bounding the quantities $\theta_P(\varepsilon; \tau)$ and $\theta_P(\beta+\varepsilon)^2$. Specifically, Example 1 achieves this via the relation to $\mathfrak{s}$, Example 2 provides a simple scenario with $\mathfrak{s} = \infty$ where it is possible to directly bound $\theta_P(\varepsilon; \tau)$, Example 3 expresses a bound on $\theta_P(\beta+\varepsilon)$ which is known in the literature, but when combined with Corollary 28 provides an improved $P$-dependent query complexity bound compared to previous works. Example 5 revisits the example from Appendix F.1 to illustrate that $\theta_P(\varepsilon; 5\beta)$ provides a valid lower-order term for this example. In Appendix F.4, we will present additional examples of $P$-dependent query complexity bounds, for some classes with $\text{VC}(\mathbb{C}) = \infty$, via $P_X$-dependent *covering numbers*.

**Example 1** (Thresholds). Due to (48), any $\mathbb{C}$ with finite star number $\mathfrak{s}$ admits a bounded $\theta_P(\varepsilon; \tau)$. A simple example of this is *threshold* classifiers: namely, $\mathcal{X} = \mathbb{R}$ and $\mathbb{C} = \{\mathbb{1}_{[t,\infty)} : t \in \mathbb{R}\}$. This class has $\mathfrak{s} = 2$ (Hanneke and Yang, 2015), and hence $\theta_P(\varepsilon; \tau) \leq 2$ for any $P$.

**Example 2** (Linear classifiers under 1-sparse distributions). To illustrate a simple example where $\theta_P(\varepsilon; \tau) = O(1)$ while $\mathfrak{s} = \infty$, consider the class $\mathbb{C}$ of linear classifiers in $\mathcal{X} = \mathbb{R}^p$, $p \geq 2$: that is, $\mathbb{C} = \{x \mapsto \mathbb{1}_{\langle w,x\rangle+b\geq 0} : w \in \mathbb{R}^p, b \in \mathbb{R}\}$. This class has $\mathfrak{s} = \infty$ (Hanneke and Yang, 2015). However, if we consider $P_X$ as a distribution supported entirely on *one axis* (e.g., $\text{Uniform}([0, 1] \times \{0\}^{p-1})$), then it is a simple exercise to show that $\theta_P(\varepsilon; \tau) \leq 2$: the concepts in $\text{B}_{P_\Delta}(h^\star, r)$ are those that disagree with $h^\star$ on at most $r$ measure (under $P_\Delta$) either to the left or right of where the $h^\star$ separator intersects the axis, so that $\text{DIS}(\text{B}_{P_\Delta}(h^\star, r))$ is simply the union of these two (at most) $r$-measure regions, hence has $P_\Delta$ measure at most $2r$.

While the above examples merely recover known results, the following example derives a previously-unknown $P$-dependent query complexity bound, which significantly improves over the best previously-known bound for this scenario.

**Example 3** (Rectangles). Consider the case $\mathcal{X} = \mathbb{R}^p$, $p \geq 1$, and $\mathbb{C} = \{\mathbb{1}_{[a_1,b_1]\times\cdots\times[a_p,b_p]} : a_1 \leq b_1, \ldots, a_p \leq b_p\}$: the class of *axis-aligned rectangles* (Mitchell, 1979). This class is known to have $\mathfrak{s} = \infty$ (Hanneke and Yang, 2015). Consider $P_X = \text{Uniform}([0, 1]^p)$ (the example trivially extends to any product distribution $P_X$ with marginals on each axis having continuous CDFs) and any $P$ with well-defined $h^\star \in \text{argmin}_{h\in\mathbb{C}} \text{er}_P(h)$ satisfying $P_X(\{x : h^\star(x) = 1\}) =: \lambda > 0$. The optimal first-order query complexity under these conditions is not yet precisely known. However, Wiener, Hanneke, and El-Yaniv (2015) have shown that $\theta_P(\beta+\varepsilon) = O\left(\frac{\mathsf{d}}{\lambda}\log(\mathsf{d}) \wedge \frac{1}{\beta+\varepsilon}\right)$ for this scenario, and based on this, the best known query complexity upper bound is of the form $\tilde{O}\left(\min\left\{\frac{\mathsf{d}^2}{\lambda}\frac{(\beta+\varepsilon)^2}{\varepsilon^2}, \mathsf{d}\frac{\beta+\varepsilon}{\varepsilon^2}\right\}\right)$. We can derive a bound which improves over this, as follows. We first recall that Theorem 1 provides a query complexity bound $\tilde{O}\left(\mathsf{d}\frac{\beta^2}{\varepsilon^2} + \frac{\mathsf{d}}{\varepsilon}\right)$, which *already* improves over the query complexity bound

of Wiener, Hanneke, and El-Yaniv (2015) in all regimes with $\varepsilon \ll \beta \ll 1$ (for every $\lambda$). However, we can further refine the lower-order term by introducing a dependence on $\lambda$. Specifically, the first bound in Corollary 28 provides a query complexity bound $\tilde{O}\left(\mathsf{d}\frac{\beta^2}{\varepsilon^2} + \frac{\mathsf{d}^2(\beta+\varepsilon)}{\lambda\varepsilon}\right)$, which offers a refinement over Theorem 1 whenever $\lambda \gg \mathsf{d}(\beta + \varepsilon)$. Moreover, the second bound in Corollary 28 provides a query complexity bound $\tilde{O}\left(\mathsf{d}\frac{\beta^2}{\varepsilon^2} + \frac{\mathsf{d}^3}{\lambda^2}\right)$. In particular, for $\lambda = \Theta(1)$ and $\beta = \tilde{O}(\sqrt{\varepsilon})$, this yields a query complexity bound $\mathrm{poly}(\mathsf{d})\,\mathrm{polylog}\left(\frac{1}{\varepsilon\delta}\right)$, which was only available in the bound of Wiener, Hanneke, and El-Yaniv (2015) in the more-restrictive regime $\beta = \tilde{O}(\varepsilon)$. We leave open the question of identifying the *optimal* query complexity for this scenario. In particular, one concrete technical question toward that end would be to determine whether, for $\lambda > 2\tau$, $\theta_P(\varepsilon; \tau) = \tilde{O}\left(\frac{\mathsf{d}}{\lambda}\right)$.

**Example 4** (Linear Classifiers). Consider the commonly-studied concept class of *linear classifiers*, defined as: $\mathcal{X} = \mathbb{R}^{\mathsf{d}-1}$ ($\mathsf{d} \geq 3$) and $\mathbb{C} = \{h_{w,b} : w \in \mathbb{R}^{\mathsf{d}-1}, b \in \mathbb{R}\}$, where $h_{w,b}(x) = \mathbb{1}[\langle w, x\rangle + b \geq 0]$. This is perhaps the most well-studied concept class in the active learning literature. Its VC dimension satisfies $\mathrm{VC}(\mathbb{C}) = \mathsf{d}$ (Vapnik and Chervonenkis, 1974), and while its star number satisfies $\mathfrak{s} = \infty$ (Hanneke and Yang, 2015), the disagreement coefficient has been shown to be bounded or sublinear under various distributional conditions (Hanneke, 2007b, 2014; Balcan, Hanneke, and Vaughan, 2010; Friedman, 2009; Mahalanabis, 2011; Wiener, Hanneke, and El-Yaniv, 2015). These results compose directly with Corollary 28 to yield previously-unknown bounds on the query complexity under these same conditions. For instance, if $P_X$ is a mixture of a finite number $t$ of multivariate Gaussian distributions with full-rank diagonal covariance matrices, then Wiener, Hanneke, and El-Yaniv (2015) provide a bound $\theta_P(r) \leq c_{\mathsf{d},t} \log^{\mathsf{d}-2}\left(\frac{1}{r}\right)$ for a $(\mathsf{d}, t)$-dependent constant $c_{\mathsf{d},t}$. Plugging into Corollary 28 (or rather, the explicit bounds in the proof thereof) yields a novel query complexity bound of order $\frac{\beta^2}{\varepsilon^2}\left(\mathsf{d} + \log\left(\frac{1}{\delta}\right)\right) + c_{\mathsf{d},t}^2 \log^{2(\mathsf{d}-2)}\left(\frac{1}{\beta+\varepsilon}\right) \log^4\left(\frac{1}{\varepsilon}\right)\left(\mathsf{d} + \log\left(\frac{1}{\delta}\right)\right)$. More generally, if $P_X$ admits a density with respect to the Lebesgue measure on $\mathbb{R}^{\mathsf{d}-1}$, then (taking $h^\star$ as in footnote 14) Hanneke (2014) argues that $\theta_P(r) = o\left(\frac{1}{r}\right)$ (where the specific form of this function $\theta_P(r)$ varies depending on $P$). Recalling that (as $\varepsilon \to 0$) the lower-order term becomes relevant only in the regime $\beta \ll \sqrt{\varepsilon}$, combining this with Corollary 28 yields a query complexity bound which often provides refinements over Theorem 1. In particular, under sufficient regularity conditions on $P_X$ (see the proof of Hanneke, 2014) to ensure this $o\left(\frac{1}{r}\right)$ function further satisfies $\theta_P(r) \log^2\left(\frac{1}{r}\right) = o\left(\frac{1}{\varepsilon}\right)$, the resulting asymptotic dependence on $(\varepsilon, \beta)$ is of the form $\frac{\beta^2}{\varepsilon^2} + o\left(\frac{1}{\varepsilon}\right)$. Moreover, if additionally the density of $P_X$ is bounded and has finite-diameter support, and if the hyperplane boundary corresponding to $h^\star$ passes through a continuity point of this density in its support, then Hanneke (2014) argues $\theta_P(r) = O(1)$, so that Corollary 28 yields a query complexity bound with asymptotic dependence on $(\varepsilon, \beta)$ of the form $\frac{\beta^2}{\varepsilon^2} + \log^4\left(\frac{1}{\varepsilon}\right)$. Moreover, under the further restrictions (density bounded away from 0, compactness of the support), Friedman (2009); Mahalanabis (2011) argue $\theta_P(r)$ is asymptotically bounded by $O(\mathsf{d})$ (for the precise statement, see the original works, or discussion thereof by Hanneke, 2014).

**Example 5** (Coupled thresholds and singletons). Let us revisit the example from Appendix F.1, for which we argued that $\theta_P(\varepsilon)$ cannot itself be used to replace the star number in Theorem 3 (for *any* algorithm). We will here explain how the region-excluded disagreement coefficient $\theta_P(\varepsilon; 5\beta)$ explicitly corrects for the issue with $\theta_P(\varepsilon)$ in this example. Specifically, consider again $\mathcal{X} = \mathbb{Z}$, and $\mathbb{C} = \mathbb{C}_{\mathsf{ts}} := \{\mathbb{1}_{\{-t\}\cup[t,\infty)} : t \in \mathbb{N}\}$, the class of *coupled* thresholds and singletons. Let $\varepsilon, \beta \in (0, 1/3)$, let $n = \frac{1-2\beta}{2\varepsilon}$ (and assume $n \in \mathbb{N}$), and consider again the distributions $P_t$, $t \in \{1, \ldots, n\}$, as defined in Appendix F.1: that is, all $P_t$ have marginal $P_X$ on $\mathcal{X}$, where for $x \in \{1, \ldots, n\}$, $P_X(\{x\}) = \frac{2\beta}{n}$, $P_X(\{-x\}) = 2\varepsilon$, and for $x \in \{-1, \ldots, -n\}$, $P_t(Y = 1|X = x) = \mathbb{1}[x = -t]$, while every $x \notin \{-1, \ldots, -n\}$ has $P_t(Y = 1|X = x) = \frac{1}{2}$. Note that, for $\Delta = [0, \infty)$, we have $P_X(\Delta) = 2\beta$. Moreover, for $h^\star$ as defined under $P_t$, we have $\mathrm{B}_{P_\Delta}(h^\star, 4\varepsilon) = \mathbb{C}_{\mathsf{ts}}$ (since only the disagreements on the *singleton* part are measured by $P_\Delta$). This implies $\mathrm{DIS}(\mathrm{B}_{P_\Delta}(h^\star, 4\varepsilon)) = \mathbb{Z} \setminus \{0\}$, so that $P_\Delta(\mathrm{DIS}(\mathrm{B}_{P_\Delta}(h^\star, 4\varepsilon))) = 1 - 2\beta$. Therefore, $\theta_P(\varepsilon; 5\beta) \geq \theta_P(\varepsilon; 2\beta) \geq \frac{P_\Delta(\mathrm{DIS}(\mathrm{B}_{P_\Delta}(h^\star, 4\varepsilon)))}{4\varepsilon} = \frac{1-2\beta}{4\varepsilon}$. Since we always have $\theta_P(\varepsilon; 5\beta) \leq \frac{1}{\varepsilon}$, we conclude that $\theta_P(\varepsilon; 5\beta) = \Theta\left(\frac{1}{\varepsilon}\right)$. As argued in Appendix F.1, the minimax optimal query complexity (constraining to $P \in \{P_t : t\{1, \ldots, n\}\}$) is $\Omega\left(\frac{1}{\varepsilon}\right)$, so that, unlike $\theta_P(\varepsilon)$, the quantity $\theta_P(\varepsilon; 5\beta)$ is an appropriate replacement for $\mathfrak{s}$ in Theorem 3. Of course, Theorem 27 shows this replacement is *always* valid, so the point here is merely to illustrate how the exclusion of the $\Delta$ region in the definition of

$\theta_P(\varepsilon; 5\beta)$ is precisely the right type of correction, compared to $\theta_P(\varepsilon)$, for this example, as it explicitly removes the issue underlying the failure of $\theta_P(\varepsilon)$: namely, the fact that the *threshold* portion of the concepts $\mathbb{1}_{\{-t'\} \cup [t', \infty)}$ is irrelevant to the learning problem inherent in the $P_t$ distributions. It is also worth noting that the *first* upper bound $\theta_P(\varepsilon; 5\beta) \leq \theta_P(5\beta + \varepsilon)\left(\frac{5\beta + \varepsilon}{\varepsilon}\right)$ from (49) also yields a value $\Theta\left(\frac{1}{\varepsilon}\right)$ (since this upper bound is *never* larger than $\frac{1}{\varepsilon}$). However, the *second* upper bound $\theta_P(5\beta + \varepsilon)^2 + \left(\frac{5\beta + \varepsilon}{\varepsilon}\right)^2$ can be significantly looser for this example, in most regimes of $\varepsilon, \beta$ (namely, $\beta \neq \Theta(\sqrt{\varepsilon})$).

### F.2.2 The Error Disagreement Coefficient

It is also possible to derive Corollary 28 via another intermediate $P$-dependent variant of the disagreement coefficient: namely, the *error disagreement coefficient*, defined as follows.

**Definition 29.** *For any probability distribution $P$ on $\mathcal{X} \times \{0, 1\}$, for any $\varepsilon \geq 0$, define*

$$\theta_P^{\mathrm{er}}(\varepsilon) := \sup_{r > \varepsilon} \frac{P_X(\mathrm{DIS}(\mathbb{C}_P(r)))}{r} \vee 1,$$

*where $\mathbb{C}_P(r) := \{h \in \mathbb{C} : \mathrm{er}_P(h) - \inf_{h' \in \mathbb{C}} \mathrm{er}_P(h') \leq r\}$ is known as the $r$-minimal set.*

Similarly to $\theta_P(\varepsilon; \tau)$, the quantity $\theta_P^{\mathrm{er}}(\varepsilon)$ has direct relations to the original disagreement coefficient from Definition 25. Specifically, for $h^\star$ as in (10),[18] since $\mathrm{B}_{P_X}(h^\star, r/2) \subseteq \mathbb{C}_P(r) \subseteq \mathrm{B}_{P_X}(h^\star, 2(\beta + r))$ for any $r > \varepsilon$, we immediately have

$$\frac{1}{2}\theta_P(\varepsilon/2) \leq \theta_P^{\mathrm{er}}(\varepsilon) \leq 2\theta_P(2(\beta + \varepsilon))\left(\frac{\beta + \varepsilon}{\varepsilon}\right) \leq \theta_P(2(\beta + \varepsilon))^2 + 4\left(\frac{\beta + \varepsilon}{\varepsilon}\right)^2. \qquad (51)$$

By definition, we always have $\theta_P^{\mathrm{er}}(\varepsilon) \leq \frac{1}{\varepsilon}$. However, unlike $\theta_P(\varepsilon; \tau)$ in (48), the quantity $\theta_P^{\mathrm{er}}(\varepsilon)$ is *not* always upper-bounded by the star number $\mathfrak{s}$ (see Example 6 below), so that we need be careful when replacing $\mathfrak{s}$ by $\theta_P^{\mathrm{er}}(\varepsilon)$ in Theorem 3.

It is also worth noting that $\theta_P^{\mathrm{er}}(\varepsilon)$ is often not as easy to use for studying specific scenarios, compared to $\theta_P(\varepsilon)$, due to the dependence on the conditional distribution $Y|X$ (whereas $\theta_P(\varepsilon)$ depends only on $P_X$ and $h^\star$). Nonetheless, below we will state a query complexity bound in terms of $\theta_P^{\mathrm{er}}(\varepsilon)$ (Theorem 30) which is sometimes smaller than that in Theorem 27 (as we illustrate in examples below), and moreover (together with (51)) provides another route to proving the query complexity bound in Corollary 28.

The quantity $\theta_P^{\mathrm{er}}(\varepsilon)$ essentially arises naturally in many existing analyses of disagreement-based active learning (e.g., Hanneke, 2009b, 2011, 2014; Koltchinskii, 2010; Foster, Rakhlin, Simchi-Levi, and Xu, 2021), wherein certain algorithms are shown to makes queries in a subset of $\mathrm{DIS}(\mathbb{C}_P(\varepsilon'))$ for an appropriate $\varepsilon' \geq \varepsilon$ (decreasing as the algorithm runs). In those contexts, it is traditional to upper bound $P_X(\mathrm{DIS}(\mathbb{C}_P(\varepsilon')))$ by $\theta_P(r(\varepsilon'))r(\varepsilon')$, where $r(\varepsilon') \geq \sup_{h \in \mathbb{C}_P(\varepsilon')} P_X(h \neq h^\star)$: for instance, $r(\varepsilon') = 2(\beta + \varepsilon')$ suffices in the agnostic setting. However, one can alternatively upper bound $P_X(\mathrm{DIS}(\mathbb{C}_P(\varepsilon')))$ by $\theta_P^{\mathrm{er}}(\varepsilon')\varepsilon'$. Such arguments are also valid in the context of $\mathbb{A}_{\mathrm{avid}}$, since Corollary 18 implies $P_X(D_{k-1}) \leq P_X(\mathrm{DIS}(\mathbb{C}_P(\varepsilon_k))) \leq \theta_P^{\mathrm{er}}(\varepsilon_k)\varepsilon_k$, so that the number of queries in $S_k^1 \cap D_{k-1}$ in round $k$ is of order $\theta_P^{\mathrm{er}}(\varepsilon_k)\varepsilon_k m_k = \tilde{O}(\theta_P^{\mathrm{er}}(\varepsilon_k)\mathsf{d})$, which will lead to a lower-order term $\tilde{O}(\theta_P^{\mathrm{er}}(\varepsilon)\mathsf{d})$. Together with reasoning similar to the proof of Theorem 27, this implies the following.

**Theorem 30.** *Under the same conditions as Theorem 27, with probability at least $1 - \delta$ the predictor $\hat{h}$ returned by $\mathbb{A}_{\mathrm{avid}}$ satisfies $\mathrm{er}_P(\hat{h}) \leq \inf_{h \in \mathbb{C}} \mathrm{er}_P(h) + \varepsilon$ and the algorithm makes a number of queries at most $Q(\varepsilon, \delta; P)$ satisfying*

$$Q(\varepsilon, \delta; P) = O\left(\frac{\beta^2}{\varepsilon^2}\left(\mathsf{d} + \log\left(\frac{1}{\delta}\right)\right) + \min\left\{\theta_P^{\mathrm{er}}(\varepsilon)\log\left(\frac{1}{\varepsilon}\right), \frac{1}{\varepsilon}\right\}\left(\mathsf{d}\log\left(\frac{1}{\varepsilon}\right) + \log\left(\frac{1}{\delta}\right)\right)\right)$$

$$= \tilde{O}\left(\mathsf{d}\frac{\beta^2}{\varepsilon^2} + \mathsf{d}\theta_P^{\mathrm{er}}(\varepsilon)\right).$$

---

[18]The relation sharpens to $\theta_P(\varepsilon) \leq \theta_P^{\mathrm{er}}(\varepsilon) \leq \theta_P(2\beta + \varepsilon)\left(\frac{2\beta + \varepsilon}{\varepsilon}\right)$ if we take $h^\star \in \operatorname{argmin}_{h \in \mathbb{C}} \mathrm{er}_P(h)$, supposing this exists (or otherwise, taking $h^\star$ as discussed in footnote 14).

Since $\theta_P^{\mathrm{er}}(\varepsilon) \le \frac{1}{\varepsilon}$, the query complexity bound in Theorem 30 is never larger than that in Theorem 1. However, unlike Theorem 27, since the quantity $\theta_P^{\mathrm{er}}(\varepsilon)$ is *not* always upper-bounded by the star number $\mathfrak{s}$, the query complexity bound in Theorem 30 is sometimes *larger* than that in Theorem 3 (see Example 6 below). That said, the quantities $\theta_P^{\mathrm{er}}(\varepsilon)$ and $\theta_P(\varepsilon; 5\beta)$ are generally incomparable (see Examples 6 and 7), so that either bound may be useful depending on the scenario being studied. Moreover, in light of (51), Theorem 30 is also useful for providing another route to establishing Corollary 28, which is therefore an immediate corollary of *either* Theorem 27 or Theorem 30.

As mentioned, depending on $(\mathbb{C}, P)$, the quantitative difference between $\theta_P^{\mathrm{er}}(\varepsilon)$ and $\theta_P(\varepsilon; 5\beta)$ can be better or worse. We illustrate this in the following two examples.

**Example 6** ($\theta_P^{\mathrm{er}}(\varepsilon) \gg \mathfrak{s} \ge \theta_P(\varepsilon; 5\beta)$)**.** As mentioned, for some scenarios, $\theta_P^{\mathrm{er}}(\varepsilon)$ can be quite large, even larger than the *star number* $\mathfrak{s}$, so that the bound in Theorem 30 becomes even worse than the $P$-independent bound in Theorem 3 (in contrast to Theorem 27, which is never worse than Theorem 3). For instance, consider a *singletons* class: $\mathcal{X} = \{1, \ldots, \frac{2}{\beta}\}$, $\mathbb{C} = \{\mathbb{1}_{\{t\}} : t \in \mathcal{X}\}$, where $\beta \in (0, 1/2)$ satisfies $\frac{2}{\beta} \in \mathbb{N}$ for simplicity. Let $P_X = \mathrm{Uniform}(\mathcal{X})$, $P(Y = 1 | X = x) = \frac{\beta}{2(1-\beta)}$ for all $x \in \mathcal{X}$. Note that every $h \in \mathbb{C}$ has $\mathrm{er}_P(h) = \beta$. Moreover, $\mathfrak{s} = |\mathcal{X}| - 1 = \frac{2-\beta}{\beta}$. However, consider $0 < \varepsilon \ll \beta$. Since $\mathbb{C}_P(\varepsilon) = \mathbb{C}$ and $\mathrm{DIS}(\mathbb{C}) = \mathcal{X}$, we have $\theta_P^{\mathrm{er}}(\varepsilon) = \frac{1}{\varepsilon} = \Theta\left(\mathfrak{s}\frac{\beta}{\varepsilon}\right)$. For instance, for $\beta = \varepsilon^{2/3}$, the bound in Theorem 3 is (ignoring logs) of order $\frac{\beta^2}{\varepsilon^2} + \frac{1}{\beta} = \frac{2}{\varepsilon^{2/3}} \ll \frac{1}{\varepsilon} = \theta_P^{\mathrm{er}}(\varepsilon)$. In light of (51), this example also witnesses a scenario where both of the query complexity bounds in Corollary 28 are worse than the $\mathfrak{s}$-dependent bound in Theorem 3; more directly, in this example, we have $\theta_P(\beta + \varepsilon)\left(\frac{\beta+\varepsilon}{\varepsilon}\right) = \frac{1}{\varepsilon} \gg \mathfrak{s}$. In contrast, for any $r < \frac{\beta}{2}$ and $\Delta \subset \mathcal{X}$, we have $\mathrm{DIS}(\mathrm{B}_{P_\Delta}(h^\star, r)) \in \{\emptyset, \Delta\}$ (depending whether the $x$ with $h^\star(x) = 1$ is in $\Delta$ or not), so that $P_\Delta(\mathrm{DIS}(\mathrm{B}_{P_\Delta}(h^\star, r))) = 0$; this immediately implies $\theta_P(\varepsilon; \tau) \le \frac{2}{\beta}$ (indeed, by careful reasoning, we can observe that any $\tau \ge \frac{\beta}{2}$ has $\theta_P(\varepsilon; \tau) = \frac{2}{\beta} - 1 = \mathfrak{s}$). More generally, by (48), the bound in Theorem 27 is never worse than that in Theorem 3.

On the other hand, there are scenarios $(\mathbb{C}, P)$ where the opposite occurs, so that in general neither quantity $\theta_P(\varepsilon; 5\beta)$ nor $\theta_P^{\mathrm{er}}(\varepsilon)$ dominates the other. This is illustrated in the following example.

**Example 7** ($\theta_P(\varepsilon; 5\beta) \gg \theta_P^{\mathrm{er}}(\varepsilon)$)**.** Consider again the class from Appendix F.1 (and Example 5): that is, $\mathcal{X} = \mathbb{Z}$ and $\mathbb{C} = \mathbb{C}_{\mathrm{ts}} := \{\mathbb{1}_{\{-t\} \cup [t,\infty)} : t \in \mathbb{N}\}$. Let $\varepsilon, \beta \in (0, 1/3)$ with $\varepsilon \ll \beta$, and define $P$ with marginal $P_X$ on $\mathcal{X}$ as defined in Appendix F.1: that is, $n = \frac{1-2\beta}{2\varepsilon}$, and for $x \in \{1, \ldots, n\}$, $P_X(\{x\}) = \frac{2\beta}{n}$, $P_X(\{-x\}) = 2\varepsilon$. However, rather than the distributions $P_t$ described there, consider the family $P_t'$, $t \in \{1, \ldots, n\}$, with $P_t'(Y = 1 | X = x) = \beta + (1 - 2\beta)\mathbb{1}_{\{-t\} \cup [t,\infty)}(x)$ for every $x \in \mathcal{X}$. These distributions represent a scenario with *uniform classification noise*. For $P = P_t'$ for any $t \in \{1, \ldots, n\}$, letting $h^\star = \mathbb{1}_{\{-t\} \cup [t,\infty)}$, it is easy to see that $\mathrm{er}_P(h^\star) = \inf_{h \in \mathbb{C}} \mathrm{er}_P(h) = \beta$. Moreover, $\theta_P^{\mathrm{er}}(\varepsilon) = \theta_P(\varepsilon/(1-2\beta)) \le \theta_P(0) \le \frac{1-2\beta}{\beta} + 3$, where the last inequality was established in Appendix F.1. In contrast, we argued in Example 5 that $\theta_P(\varepsilon; 5\beta) \ge \frac{1-2\beta}{4\varepsilon} \gg \frac{1-2\beta}{\beta} + 3 \ge \theta_P^{\mathrm{er}}(\varepsilon)$. Thus, in this scenario, $\theta_P(\varepsilon; 5\beta)$ is larger than $\theta_P^{\mathrm{er}}(\varepsilon)$.

A natural question is whether the gaps between $\theta_P(\varepsilon; 5\beta)$ and $\theta_P^{\mathrm{er}}(\varepsilon)$ in Examples 6 and 7 can also arise in cases where the smaller of the two corresponding query complexity bounds (either Theorem 27 or 30) is actually nearly-*sharp* (in a minimax analysis over a family of distributions). This is straightforward to obtain, by defining a family of possible distributions $P$ each obtained as a uniform *mixture* of one of the above two scenarios and the simple 2-point construction of Kääriäinen (2006) giving rise to the $\frac{\beta^2}{\varepsilon^2}$ lower bound. For brevity, we omit the details of this.

The fact that Theorem 27 at least provides a starting point for $P$-dependent analysis which is never worse than the $P$-independent bound in Theorem 3 is a desirable feature. In contrast, the bound in Theorem 30 is sometimes better and sometimes worse than that in Theorem 3, so that one should be careful when using Theorem 30. Nonetheless, as illustrated in Example 7, there are at least some scenarios where $\theta_P^{\mathrm{er}}(\varepsilon)$ may be useful for describing favorable scenarios (particularly concerning the $Y|X$ conditional distribution).

### F.3 Querying in Subregions of the Region of Disagreement

In the active learning literature, one technique for going beyond disagreement-based queries is to query examples in a carefully selected *subregion* $R \subseteq \mathrm{DIS}(V)$ of the region of disagreement of the set $V$ of surviving concepts. This idea originates in the work of Balcan, Broder, and Zhang (2007) on *margin-based* active learning of homogeneous linear classifiers under certain marginals $P_X$ in realizable and Tsybakov-noise scenarios, and was extended to a technique for general concept classes and the agnostic case by Zhang and Chaudhuri (2014) (see Appendix A for further discussion of the history). For instance, the most well-known case of this technique providing improvements over disagreement-based queries (see Example 8 below) is homogeneous linear classifiers under a uniform distribution on a sphere (alternatively, any isotropic log-concave distribution), where the query complexity of this technique is $\tilde{O}\left(\mathsf{d}\frac{(\beta+\varepsilon)^2}{\varepsilon^2}\right)$ (Zhang and Chaudhuri, 2014) (minimax optimal up to log factors), compared to disagreement-based active learning for which the best known bound is $\tilde{O}\left(\mathsf{d}^{3/2} \cdot \frac{(\beta+\varepsilon)^2}{\varepsilon^2}\right)$ (Dasgupta, Hsu, and Monteleoni, 2007).

In this section, we show this technique is also compatible with the AVID principle, and propose a refinement of the AVID Agnostic algorithm which replaces $D_{k-1} = \mathrm{DIS}(V_{k-1})$ in Steps 2 and 9 with a well-chosen *subregion* $R_{k-1} \subseteq D_{k-1}$. We argue that this change does not affect the validity of Theorem 5, and admits refined $P$-dependent query complexity bounds compared to those presented in Appendix F.2. In particular, this shows that the AVID principle can recover the optimal query complexity of homogeneous linear classifiers under the uniform distribution, and generally any isotropic log-concave distribution (indeed, with improved log factors compared to prior works).

The basic argument (building from the original ideas of Balcan, Broder, and Zhang, 2007, and Zhang and Chaudhuri, 2014) is that, rather than querying all examples in $S_k^1 \cap D_{k-1} \setminus \Delta_{i_k}$ in Step 2 of $\mathbb{A}_{\mathrm{avid}}$, the algorithm identifies a *subset* of these examples $Q_k \subseteq S_k^1 \cap D_{k-1} \setminus \Delta_{i_k}$ which suffices for the purpose of updating $V_k$ in Step 3. Specifically, we aim to identify a subset $Q_k \subseteq S_k^1 \cap D_{k-1} \setminus \Delta_{i_k}$ for which, for any $h, h' \in V_{k-1}$,

$$\left| \hat{P}_{S_k^1}(\{h \neq h'\} \cap Q_k) - \hat{P}_{S_k^1}(\{h \neq h'\} \cap D_{k-1} \setminus \Delta_{i_k}) \right| \leq \frac{\varepsilon_k}{3C''}.$$

In other words, most of the significant disagreements in $\mathcal{X} \setminus \Delta_{i_k}$ among concepts in $V_{k-1}$ are captured in the $Q_k$ set. In particular, this retains the guarantees of Lemmas 15 and 16 with only minor adjustments to the constants in the bounds (accounting for the potential $\frac{\varepsilon_k}{8C'}$ probability disagreements that are lost).

Formally, consider the algorithm $\mathbb{A}_{\mathrm{avid}}^{\mathrm{sub}}$ stated in Figure 2, where the $Q_k$ data subset is defined below. The values $C, C', C'', N, m_k$ and data subsets $S_k^1, S_k^4$ are all as defined in $\mathbb{A}_{\mathrm{avid}}$. The data subsets $S_k^2, S_{k,i}^3$ are defined analogously to $\mathbb{A}_{\mathrm{avid}}$ except allocated in the corresponding steps of $\mathbb{A}_{\mathrm{avid}}^{\mathrm{sub}}$: that is, if and when the algorithm reaches Step 2 with a value $k$, or reaches Step 9 (in which case let $k = N + 1$), then for the value $i_k$ and the set $\Delta_{i_k}$ as defined at that time in the algorithm, letting $\hat{p}_k := 2\hat{P}_{S_k^4}(\Delta_{i_k})$, the algorithm allocates to $S_k^2$ the next $m_k' := \left\lceil \frac{C'' c_1^2 \hat{p}_k}{\varepsilon_k^2} \left( \mathsf{d} + \log\left( \frac{4(3+N-k)^2}{\delta} \right) \right) \right\rceil$ consecutive examples not previously allocated to any data subset, and likewise, if and when the algorithm reaches Step 5 with values $(k, i)$, it allocates to $S_{k,i}^3$ the next $m_k$ consecutive examples which have not yet been allocated to any data subset.

We define the $Q_k$ data subset via a technique analogous to the work of Zhang and Chaudhuri (2014), specified via a discrete *linear program* with a finite number of constraints imposed by the set of realizable classifications of $S_k^1$. Let $t_k = \sum_{k'=1}^{k-1} m_{k'}$, and recall $S_k^1 := \{(X_{t_k+1}, Y_{t_k+1}), \ldots, (X_{t_k+m_k}, Y_{t_k+m_k})\}$. The algorithm inductively constructs sets $V_k \subseteq \mathbb{C}$ in Step 3 (analogous to the $V_k$ sets in $\mathbb{A}_{\mathrm{avid}}$). For any given $k \in \{1, \ldots, N+1\}$, denote by

$$V_{k-1}(S_k^1) := \{(h(X_{t_k+1}), \ldots, h(X_{t_k+m_k})) : h \in V_{k-1}\} \subseteq \{0,1\}^{m_k},$$

the set of $V_{k-1}$-realizable classifications of $S_k^1$. For the set $\Delta_{i_k}$ (which is defined inductively based on previous rounds of the algorithm, analogously to the set $\Delta_{i_k}$ in $\mathbb{A}_{\mathrm{avid}}$), consider the following integer linear program with binary variables[19] $\zeta_{1,0}, \zeta_{1,1}, q_1, \ldots, \zeta_{m_k,0}, \zeta_{m_k,1}, q_{m_k}$.

---

[19]For simplicity, in this work, we present a technique based on an *integer* linear program, to arrive at a deterministic querying strategy. It is straightforward to extend the result to allow for non-integer solutions

Input: Error parameter $\varepsilon$, Confidence parameter $\delta$, Unlabeled data $X_1, \ldots, X_m$

Output: Classifier $\hat{h}$

0.  Initialize $i = i_1 = 0$, $\Delta_0 = \emptyset$, $V_0 = \mathbb{C}$
1.  For $k = 1, \ldots, N$
2.      Query all examples in $Q_k$ and $S_k^2 \cap \Delta_{i_k}$
3.      $V_k \leftarrow \left\{ h \in V_{k-1} : \hat{\text{er}}_k^{1,2}(h) \leq \hat{\text{er}}_k^{1,2}(\hat{h}_k) + \frac{\varepsilon_k}{C'} \right\}$
4.      If $V_k = \emptyset$ or $\hat{\text{er}}_k^{1,2}(\hat{h}_k) < \min_{h \in V_k} \hat{\text{er}}_k^{1,2}(h) - \frac{\varepsilon_k}{4C'}$, Then Return $\hat{h} := \hat{h}_k$
5.      While $\max_{f,g \in V_k} \hat{P}_{S_{k,i}^3}(\{f \neq g\} \setminus \Delta_i) > \varepsilon_{k+2}$
6.          $(f, g) \leftarrow \text{argmax}_{(f',g') \in V_k^2} \hat{P}_{S_{k,i}^3}(\{f' \neq g'\} \setminus \Delta_i)$
7.          $\Delta_{i+1} \leftarrow \Delta_i \cup \{f \neq g\}$, and update $i \leftarrow i + 1$
8.      $i_{k+1} \leftarrow i$
9.  Query all examples in $Q_{N+1}$ and $S_{N+1}^2 \cap \Delta_{i_{N+1}}$ and Return $\hat{h} := \hat{h}_{N+1}$

Figure 2: The Subregion-AVID Agnostic algorithm.

$\text{LP}_k$:

minimize $\qquad \sum_{t=1}^{m_k} q_t$

subject to $\qquad \forall (y_1, \ldots, y_{m_k}) \in V_{k-1}(S_k^1), \ \frac{1}{m_k} \sum_{t=1}^{m_k} \zeta_{t, 1-y_t} \mathbb{1}[X_{t_k+t} \notin \Delta_{i_k}] \leq \frac{\varepsilon_k}{6C''}$

$\qquad \forall t \in \{1, \ldots, m_k\}, \ \zeta_{t,0} + \zeta_{t,1} + q_t = 1$

$\qquad \zeta_{1,0}, \zeta_{1,1}, q_1, \ldots, \zeta_{m_k,0}, \zeta_{m_k,1}, q_{m_k} \in \{0, 1\}$

In particular, note that the solution only depends on the *unlabeled* examples $X_{t_k+1}, \ldots, X_{t_k+m_k}$, and thus the algorithm may use the solution of this optimization problem when determining an appropriate set $Q_k$ of queries in Steps 2 and 9. Denote by $q_1^k, \ldots, q_{m_k}^k$ the respective values of the $q_1, \ldots, q_{m_k}$ variables at the solution found by $\text{LP}_k$. Then define $Q_k$ as a subsequence of $S_k^1$:

$$Q_k := \{(X_{t_k+t}, Y_{t_k+t}) : 1 \leq t \leq m_k, q_t^k = 1\}.$$

Let us generalize the definition of $\hat{P}_{S_k^1}$ to involve intersections with the subsequence $Q_k$: for any set $A \subseteq \mathcal{X} \times \{0,1\}$, $\hat{P}_{S_k^1}(A \cap Q_k) := \frac{1}{m_k} \sum_{t=1}^{m_k} q_t^k \cdot \mathbb{1}[(X_{t_k+t}, Y_{t_k+t}) \in A]$, and $\hat{P}_{S_k^1}(\text{ER}(f) \setminus Q_k) := \frac{1}{m_k} \sum_{t=1}^{m_k} (1 - q_t^k) \cdot \mathbb{1}[(X_{t_k+t}, Y_{t_k+t}) \in A]$. As usual, we also overload this notation for $A \subseteq \mathcal{X}$, such as sets $\{f \neq g\}$, interpreting such sets $A$ as synonymous with their labeled extension $A \times \{0,1\}$.

The algorithm also relies on the following modifications to the definition of $\hat{\text{er}}_k^{1,2}$:

$$\forall h, \hat{\text{er}}_k^{1,2}(h) := \hat{P}_{S_k^1}(\text{ER}(h) \cap Q_k) + \hat{P}_{S_k^2}(\text{ER}(h) \cap \Delta_{i_k}). \tag{52}$$

The definitions of $V_{k-1}^{(4)}$ and $\hat{h}_k$ are then defined as in (2) and (3) based on the set $V_{k-1}$ defined in $\mathbb{A}_{\text{avid}}^{\text{sub}}$ and the modified definition of $\hat{\text{er}}_k^{1,2}$ in (52). This completes the specification of the $\mathbb{A}_{\text{avid}}^{\text{sub}}$ algorithm.

We state a query complexity guarantee for this algorithm, phrased in terms of a variant of a *subregion disagreement coefficient*. As in Appendix F.2, we first present the known definition from the literature, which serves both as a starting point for the modified version and as a more-accessible quantity useful for upper-bounding the new quantity. Specifically, the following definition (a refinement of the disagreement coefficient from Definition 25) was proposed by Zhang and Chaudhuri (2014) (see also Hanneke, 2016b).[20]

---

$\zeta_{t,0}, \zeta_{t,1} \in [0,1]$ to the LP, resulting in a *randomized* querying strategy (see Zhang and Chaudhuri, 2014). This makes no significant difference to the query complexity bound (see Hanneke, 2016b, for a related discussion), but may be more attractive from a computational perspective.

[20]The variant stated here is phrased slightly differently, to simplify the definition. In particular, $\varphi_P(\varepsilon, 0)$ is equivalent to a quantity $\varphi_c^{01}(\varepsilon)$ studied by Hanneke (2016b), which is only slightly different than the original

**Definition 31.** *For any measure $\mu$ on $\mathcal{X}$, any $V \subseteq \mathbb{C}$, and any $\eta \geq 0$, define*

$$\Phi_\mu(V, \eta) := \inf\left\{ \mu(R) : \sup_{g \in V} \mu(\{g \neq f\} \setminus R) \leq \eta, \text{ measurable } R \subseteq \mathcal{X} \text{ and } f : \mathcal{X} \to \{0,1\} \right\}.$$

*For any distribution $P_X$ on $\mathcal{X}$ and any measurable $h : \mathcal{X} \to \{0,1\}$, for any $\varepsilon, \alpha \geq 0$, define*

$$\varphi_{P_X,h}(\varepsilon, \alpha) := \sup_{r > \alpha + \varepsilon} \frac{\Phi_{P_X}(\mathrm{B}_{P_X}(h,r), (r-\alpha)/(36CC''))}{r} \vee 1.$$

*In particular, for any distribution $P$ on $\mathcal{X} \times \{0,1\}$, letting $h^\star$ be as in (10) (or see footnote 14), define $\varphi_P(\varepsilon, \alpha) := \varphi_{P_X,h^\star}(\varepsilon, \alpha)$.*

The quantity $\Phi_{P_X}(V, \eta)$ identifies the smallest $P_X(R)$ among regions $R \subseteq \mathcal{X}$ for which functions $g \in V$ do not disagree much outside the region $R$ (i.e., they have at most $\eta$ disagreement with a fixed function $f$ on $\mathcal{X} \setminus R$). In particular, we can upper bound $\Phi_{P_X}(V, \eta)$ by taking $R = \mathrm{DIS}(V)$ and any $f \in V$, which satisfies $\sup_{g \in V} P_X(\{g \neq f\} \setminus R) = 0 \leq \eta$, so that $\Phi_{P_X}(V, \eta) \leq P_X(\mathrm{DIS}(V))$. It immediately follows that the quantity $\varphi_{P_X,h}(\varepsilon, \alpha)$ is never larger than the disagreement coefficient:

$$\varphi_{P_X,h}(\varepsilon, \alpha) \leq \theta_{P_X,h}(\alpha + \varepsilon). \tag{53}$$

Indeed, there are several known examples of scenarios $(\mathbb{C}, P_X, h^\star)$ where $\varphi_P(\varepsilon, \alpha)$ is substantially smaller than $\theta_P(\alpha + \varepsilon)$ (Zhang and Chaudhuri, 2014). One example (discussed formally in Example 8 below) is the class $\mathbb{C}$ of *homogeneous linear classifiers* in $\mathbb{R}^\mathsf{d}$ under $P_X$ a uniform distribution on an origin-centered sphere, where $\theta_P(0) = \Theta\left(\sqrt{\mathsf{d}}\right)$ and $\varphi_P(\varepsilon, \alpha) = O\left(\log\left(\frac{\alpha+\varepsilon}{\varepsilon}\right)\right)$ (Hanneke, 2007b; Balcan, Broder, and Zhang, 2007; Zhang and Chaudhuri, 2014).

By (53) and (47), we also always have $\varphi_P(\varepsilon, \alpha) \leq \mathfrak{s} \wedge \frac{1}{\alpha+\varepsilon}$ for any $\varepsilon, \alpha \geq 0$ with $\alpha + \varepsilon \leq 1$. Indeed, as with $\theta_P(\varepsilon)$ in (47), this inequality turns out to be sharp in the worst case. Specifically, Hanneke (2016b) has shown that $\sup_{P_X} \sup_{h \in \mathbb{C}} \varphi_{P_X,h}(\varepsilon, 0) = \mathfrak{s} \wedge \frac{1}{\varepsilon}$ for $\varepsilon \in (0,1]$. Additionally, by definition we have $\varphi_{P_X,h}(\varepsilon, \alpha) \geq \varphi_{P_X,h}(\alpha + \varepsilon, 0)$. Since (53) implies $\varphi_{P_X,h}(\varepsilon, \alpha) \leq \theta_{P_X,h}(\alpha + \varepsilon)$, it immediately follows from combining this result of Hanneke (2016b) with (47) that for any $\varepsilon, \alpha \geq 0$ with $\alpha + \varepsilon \leq 1$,

$$\sup_{P_X} \sup_{h \in \mathbb{C}} \varphi_{P_X,h}(\varepsilon, \alpha) = \mathfrak{s} \wedge \frac{1}{\alpha+\varepsilon}. \tag{54}$$

Due to (53) and the example in Appendix F.1, we know it is not possible to replace $\mathfrak{s}$ with $\varphi_P(\varepsilon, \alpha)$ in Theorem 3 for any $\alpha \geq 0$ (for any algorithm). However, similarly to the modification $\theta_P(\varepsilon; \tau)$ of $\theta_P(\varepsilon)$ presented in Appendix F.2, we can modify Definition 31 appropriately to provide a quantity suitable for developing a query complexity bound for $\mathbb{A}_{\mathrm{avid}}^{\mathrm{sub}}$. Specifically, as in Definition 26, let us first generalize the definition of $\varphi_{P_X,h}(\varepsilon, \alpha)$ to general *measures* $\mu$: that is, for any measure $\mu$ on $\mathcal{X}$ and measurable function $h : \mathcal{X} \to \{0,1\}$, for any $\varepsilon, \alpha \geq 0$, define $\varphi_{\mu,h}(\varepsilon, \alpha) := \sup_{r > \alpha + \varepsilon} \frac{\Phi_\mu(\mathrm{B}_\mu(h,r),(r-\alpha)/(36CC''))}{r} \vee 1$. Then consider the following definition, representing a *region-excluded subregion disagreement coefficient*.

**Definition 32.** *For any distribution $P$ on $\mathcal{X} \times \{0,1\}$ and any measurable $\Delta \subseteq \mathcal{X}$, define a measure $A \mapsto P_\Delta(A) := P_X(A \setminus \Delta)$. For any $\varepsilon > 0$ and $\alpha, \tau \geq 0$, for $h^\star \in \mathbb{C}$ as in (10) (or see footnotes 14, 17), define*

$$\varphi_P(\varepsilon, \alpha; \tau) := \sup_{\Delta \subseteq \mathcal{X} : P_X(\Delta) \leq \tau} \varphi_{P_\Delta, h^\star}(\varepsilon, \alpha).$$

As was the case for $\theta_P(\varepsilon; \tau)$, we can equivalently define $\varphi_P(\varepsilon, \alpha; \tau)$ as the subregion disagreement coefficient under a worst-case *conditional* distribution $P_X(\cdot | \mathcal{X} \setminus \Delta)$: that is,

$$\varphi_P(\varepsilon, \alpha; \tau) = \sup_{\Delta \subseteq \mathcal{X} : P_X(\Delta) \leq \tau} \varphi_{P_X(\cdot | \mathcal{X} \setminus \Delta), h^\star}(\varepsilon/P_X(\mathcal{X} \setminus \Delta), \alpha/P_X(\mathcal{X} \setminus \Delta)), \tag{55}$$

where we define $\varphi_{P_X(\cdot | \mathcal{X} \setminus \Delta), h^\star}(\varepsilon/P_X(\mathcal{X} \setminus \Delta), \alpha/P_X(\mathcal{X} \setminus \Delta)) = 1$ in the case $P_X(\mathcal{X} \setminus \Delta) = 0$ (which coincides with the value $\varphi_{P_\Delta, h^\star}(\varepsilon, \alpha)$ for such $\Delta$). In particular, combining this equivalent definition with (54) yields that

$$\varphi_P(\varepsilon, \alpha; \tau) \leq \mathfrak{s} \wedge \frac{1}{\alpha+\varepsilon} \vee 1. \tag{56}$$

---

quantity studied by Zhang and Chaudhuri (2014) in that it considers *binary* functions rather than fractional values in $[0,1]$. Hanneke (2016b) has shown this change to binary values makes little quantitative difference compared to the quantity of Zhang and Chaudhuri (2014).

Thus, replacing $\mathfrak{s}$ in Theorem 3 by $\varphi_P(\varepsilon, \alpha; \tau)$ would yield a (never-larger) $P$-dependent refinement.

As with $\theta_P(\varepsilon; \tau)$, the quantity $\varphi_P(\varepsilon, \alpha; \tau)$ itself may often be challenging to calculate. Fortunately, again as with $\theta_P(\varepsilon; \tau)$, it can be upper-bounded by expressions that are more-easily calculated (though at the expense of some slack, so that they are no longer upper-bounded by $\mathfrak{s}$). We might therefore think of $\varphi_P(\varepsilon, \alpha; \tau)$ as an intermediate complexity measure (analogous to $\theta_P(\varepsilon; \tau)$), which is useful in providing a starting point for a $P$-dependent refinement of $\mathfrak{s}$ which is never larger than $\mathfrak{s}$, and which admits general upper bounds which are more accessible than directly calculating $\varphi_P(\varepsilon, \alpha; \tau)$. Specifically, it follows immediately from Definition 32 that we always have a lower bound $\varphi_P(\varepsilon, \alpha) \le \varphi_P(\varepsilon, \alpha; \tau)$, and an upper bound

$$\varphi_P(\varepsilon, \alpha; \tau) \le \sup_{r > \alpha + \varepsilon} \frac{\Phi_{P_X}(B_{P_X}(h^\star, \tau + r), (r - \alpha)/(36CC''))}{r} \vee 1$$

$$\le \varphi_P(\varepsilon, \alpha + \tau) \left( \frac{\alpha + \tau + \varepsilon}{\alpha + \varepsilon} \right) \le \varphi_P(\varepsilon, \alpha + \tau)^2 + \left( \frac{\alpha + \tau + \varepsilon}{\alpha + \varepsilon} \right)^2. \quad (57)$$

Making use of the quantity $\varphi_P(\varepsilon, \alpha; \tau)$ to analyze $\mathbb{A}_{\mathrm{avid}}^{\mathrm{sub}}$ analogously to the analysis of $\mathbb{A}_{\mathrm{avid}}$ based on $\theta_P(\varepsilon; \tau)$ in Theorem 27, we arrive at the following theorem.

**Theorem 33** (Distribution-dependent Query Complexity of Subregion AVID Agnostic). *For any concept class $\mathbb{C}$ with $\mathrm{VC}(\mathbb{C}) < \infty$, letting $\mathsf{d} = \mathrm{VC}(\mathbb{C})$, for every distribution $P$ on $\mathcal{X} \times \{0, 1\}$, letting $\beta = \inf_{h \in \mathbb{C}} \mathrm{er}_P(h)$, for any $\varepsilon, \delta \in (0, 1)$, if the algorithm $\mathbb{A}_{\mathrm{avid}}^{\mathrm{sub}}$ is executed with parameters $(\varepsilon, \delta)$, with any number $m \ge M(\varepsilon, \delta; \beta)$ of i.i.d.-$P$ examples (for $M(\varepsilon, \delta; \beta)$ as in Theorem 5, defined in Lemma 24), then with probability at least $1 - \delta$, the returned predictor $\hat{h}$ satisfies $\mathrm{er}_P(\hat{h}) \le \inf_{h \in \mathbb{C}} \mathrm{er}_P(h) + \varepsilon$ and the algorithm makes a number of queries at most $Q(\varepsilon, \delta; P)$ satisfying*

$$Q(\varepsilon, \delta; P) = O\left( \frac{\beta^2}{\varepsilon^2} \left( \mathsf{d} + \log\left(\frac{1}{\delta}\right) \right) + \min\left\{ \varphi_P(\varepsilon, 0; 5\beta) \log\left(\frac{1}{\varepsilon}\right), \frac{1}{\varepsilon} \right\} \left( \mathsf{d} \log\left(\frac{1}{\varepsilon}\right) + \log\left(\frac{1}{\delta}\right) \right) \right)$$

$$= \tilde{O}\left( \mathsf{d} \frac{\beta^2}{\varepsilon^2} + \mathsf{d}\varphi_P(\varepsilon, 0; 5\beta) \right).$$

**Proof Sketch.** The proof of this theorem follows nearly identically to the proof of Theorem 5. We will merely highlight the changes compared to the original proof. Specifically, throughout the proof, we first replace all definitions from $\mathbb{A}_{\mathrm{avid}}$ with the corresponding definitions from $\mathbb{A}_{\mathrm{avid}}^{\mathrm{sub}}$ (e.g., $\Delta_i$, $i_k$, $\hat{h}_k$, $V_k$, $D_{k-1} = \mathrm{DIS}(V_{k-1})$, $S_k^2$, $S_{k,i}^3$, $K$, $\mathcal{K}$, etc.) so that all definitions in the proof refer to the respective quantities in the $\mathbb{A}_{\mathrm{avid}}^{\mathrm{sub}}$ algorithm. Since the only definitional change in $\mathbb{A}_{\mathrm{avid}}^{\mathrm{sub}}$ compared to $\mathbb{A}_{\mathrm{avid}}$ is in the use of $Q_k$ rather than $S_k^1 \cap D_{k-1} \setminus \Delta_{i_k}$, to provide the $\varepsilon$ error guarantee it will suffice to argue that the inequality (14) of Lemma 10 remains valid (only slightly larger) with this change: namely, on the event $E_0$,

$$\forall h, h' \in V_{k-1}^{(3)}, \ \left| \left( \hat{P}_{S_k^1}(\mathrm{ER}(h) \cap Q_k) - \hat{P}_{S_k^1}(\mathrm{ER}(h') \cap Q_k) \right) \right.$$

$$\left. - (P(\mathrm{ER}(h) \setminus \Delta_{i_k}) - P(\mathrm{ER}(h') \setminus \Delta_{i_k})) \right|$$

$$< \sqrt{P_X(\{h \ne h'\} \setminus \Delta_{i_k}) \frac{\varepsilon_k}{C''}} + \frac{2\varepsilon_k}{C''}. \quad (58)$$

Note that this is only larger than the bound in (14) by an additive $\frac{\varepsilon_k}{C''}$ (which we will argue below is inconsequential to the proof). As was true of (14), we argue that (58) in fact follows immediately from (18), as follows. Consider the values $\zeta_{1,0}^k, \zeta_{1,1}^k, q_1^k, \ldots, \zeta_{m_k,0}^k, \zeta_{m_k,1}^k, q_{m_k}^k$ at the solution of the $\mathrm{LP}_k$ optimization. Due to the first constraint in $\mathrm{LP}_k$, we know every $f \in V_{k-1}$ has

$$\frac{1}{m_k} \sum_{t=1}^{m_k} \zeta_{t,1-f(X_{t_k+t})}^k \mathbb{1}[X_{t_k+t} \notin \Delta_{i_k}] \le \frac{\varepsilon_k}{6C''}.$$

Moreover, due to the second constraint in $\mathrm{LP}_k$, for every $f, g \in V_{k-1}$, any $X_{t_k+t} \in \{f \ne g\}$ has $\zeta_{t,1-f(X_{t_k+t})}^k + \zeta_{t,1-g(X_{t_k+t})}^k + q_t^k = 1$, so that $q_t^k = 0 \implies \zeta_{t,1-f(X_{t_k+t})}^k + \zeta_{t,1-g(X_{t_k+t})}^k = 1$. Together, we have

$$\hat{P}_{S_k^1}((\{f \ne g\} \setminus \Delta_{i_k}) \setminus Q_k) \le \frac{1}{m_k} \sum_{t=1}^{m_k} \left( \zeta_{t,1-f(X_{t_k+t})}^k + \zeta_{t,1-g(X_{t_k+t})}^k \right) \mathbb{1}[X_{t_k+t} \notin \Delta_{i_k}] \le \frac{\varepsilon_k}{3C''}.$$

Recall that every $h \in V_{k-1}^{(3)}$ is of the form $h = \mathrm{DL}(f', g', h') := f'\mathbb{1}_{\{f'=g'\}} + h'\mathbb{1}_{\{f'\neq g'\}}$ for some $f', g' \in V_{k-1}$ and $h' \in \mathbb{C}$. Consider any two such functions $h, h' \in V_{k-1}^{(3)}$, where $h = \mathrm{DL}(f_1, g_1, h_1)$ and $h' = \mathrm{DL}(f_2, g_2, h_2)$ for $f_1, g_1, f_2, g_2 \in V_{k-1}$ and $h_1, h_2 \in \mathbb{C}$. Note that

$$\{h \neq h'\} \subseteq \{f_1 \neq f_2\} \cup \{g_1 \neq g_2\} \cup \{f_1 \neq g_1\}.$$

Therefore, the union bound implies

$\hat{P}_{S_k^1}((\{h \neq h'\} \setminus \Delta_{i_k}) \setminus Q_k)$

$\leq \hat{P}_{S_k^1}((\{f_1 \neq f_2\} \setminus \Delta_{i_k}) \setminus Q_k) + \hat{P}_{S_k^1}((\{g_1 \neq g_2\} \setminus \Delta_{i_k}) \setminus Q_k) + \hat{P}_{S_k^1}((\{f_1 \neq g_1\} \setminus \Delta_{i_k}) \setminus Q_k)$

$\leq \dfrac{\varepsilon_k}{C''}.$

Also note that, due to the indicator $\mathbb{1}[X_{t_k+t} \notin \Delta_{i_k}]$ in the first constraint of $\mathrm{LP}_k$, at the solution to $\mathrm{LP}_k$, every $X_{t_k+t} \notin \Delta_{i_k}$ has $q_t^k = 0$, so that any $h \in V_{k-1}^{(3)}$ has $\mathrm{ER}(h) \cap Q_k = (\mathrm{ER}(h) \setminus \Delta_{i_k}) \cap Q_k$. Altogether, we have that every $h, h' \in V_{k-1}^{(3)}$ satisfy

$$\left| \left( \hat{P}_{S_k^1}(\mathrm{ER}(h) \cap Q_k) - \hat{P}_{S_k^1}(\mathrm{ER}(h') \cap Q_k) \right) - \left( \hat{P}_{S_k^1}(\mathrm{ER}(h) \setminus \Delta_{i_k}) - \hat{P}_{S_k^1}(\mathrm{ER}(h') \setminus \Delta_{i_k}) \right) \right|$$

$$= \left| \hat{P}_{S_k^1}((\mathrm{ER}(h') \setminus \Delta_{i_k}) \setminus Q_k) - \hat{P}_{S_k^1}((\mathrm{ER}(h) \setminus \Delta_{i_k}) \setminus Q_k) \right|$$

$$\leq \hat{P}_{S_k^1}((\{h \neq h'\} \setminus \Delta_{i_k}) \setminus Q_k) \leq \dfrac{\varepsilon_k}{C''}. \tag{59}$$

Together with (18) we arrive at the claimed inequality (58).

In the context of the rest of the proof of Theorem 5, the only place (14) is used is in (28) in the proof of Lemma 15. In that context, substituting (58) yields the same conclusion: namely, for $h_1, h_1' \in V_{k-1}^{(3)}$, since (20) of Lemma 12 implies $P_X(\{h_1 \neq h_1'\} \setminus \Delta_{i_k}) \leq 3\varepsilon_k$, (58) implies

$$\left| \hat{P}_{S_k^1}(\mathrm{ER}(h_1) \cap Q_k) - \hat{P}_{S_k^1}(\mathrm{ER}(h_1') \cap Q_k) - (P(\mathrm{ER}(h_1) \setminus \Delta_{i_k}) - P(\mathrm{ER}(h_1') \setminus \Delta_{i_k})) \right|$$

$$\leq \sqrt{\dfrac{3\varepsilon_k^2}{C''}} + \dfrac{2\varepsilon_k}{C''} \leq \dfrac{2\varepsilon_k}{\sqrt{C''}},$$

where the last inequality follows from $C'' \geq 100$. Therefore, the conclusion of Lemma 15 remains valid (with the modified definition of $\hat{\mathrm{er}}_k^{1,2}$ from (52)). The rest of the proof of the error bound (Lemma 19), and unlabeled sample size $M(\varepsilon, \delta; \beta)$ (Lemma 24), and size of $P_X(\Delta_{i_k})$ (Lemma 20) follow verbatim from this fact.

It remains only to establish the claimed bound $Q(\varepsilon, \delta; P)$ on the number of queries. In the context of the proof of Theorem 5, this effectively means replacing (39) of Lemma 23 with a bound on $|Q_k|$ based on $\varphi_P(\varepsilon, 0; 5\beta)$, on an event $E_3'$ of probability at least $1 - \frac{\delta}{8}$ (which replaces the event $E_3'$ defined in the proof of Lemma 23).

Toward this end, consider any $k \in \{1, \ldots, N+1\}$ having non-zero probability of $k \leq K$. Given the event that $k \leq K$ and the random variables $V_{k-1}$ and $\Delta_{i_k}$, fix a measurable function $h_k : \mathcal{X} \to \{0, 1\}$ and a measurable set $R_k \subseteq \mathcal{X}$ (dependent on $V_{k-1}$ and $\Delta_{i_k}$ but *not* on $S_k^1$) such that

$$\sup_{g \in V_{k-1}} P_X((\{g \neq h_k\} \setminus R_k) \setminus \Delta_{i_k}) \leq \dfrac{\varepsilon_k}{18C''} \tag{60}$$

$$\text{and} \quad P_X(R_k \setminus \Delta_{i_k}) \leq \Phi_{P_{\Delta_{i_k}}}\left(V_{k-1}, \dfrac{\varepsilon_k}{18C''}\right) + \dfrac{\varepsilon_k}{2}. \tag{61}$$

Such a pair $(h_k, R_k)$ is guaranteed to exist by the definition of $\Phi_\mu(\cdot, \cdot)$ in Definition 31.

We aim to argue that the constraints in $\mathrm{LP}_k$ are satisfied by taking $\zeta_{t,h_k(X_{t_k+t})} = \mathbb{1}[X_{t_k+t} \notin R_k]$ and $q_t = \mathbb{1}[X_{t_k+t} \in R_k]$, via a uniform multiplicative Chernoff bound (Lemma 7 of Appendix D). Toward this end, define a collection $\tilde{\mathcal{A}}_k$ of subsets of $\mathcal{X}$:

$$\tilde{\mathcal{A}}_k := \{(\{g \neq h_k\} \setminus R_k) \setminus \Delta_{i_k} : g \in V_{k-1}\}.$$

Note that $\mathrm{VC}(\tilde{\mathcal{A}}_k) \leq \mathrm{d}$. Let $\tilde{\delta}_k := \frac{\delta\varepsilon_{k+3}}{144}$. We bound $\varepsilon(m_k, \tilde{\delta}_k; \tilde{\mathcal{A}}_k)$ by reasoning similar to the proof of Lemma 9. Specifically, we have

$$m_k \geq \dfrac{150C''c_0}{\varepsilon_k}\left(\mathrm{d}\log\left(\dfrac{C''c_0}{\varepsilon_k}\right) + \log\left(\dfrac{C''c_0}{\delta\varepsilon_k}\right)\right) > \dfrac{108C''c_0}{\varepsilon_k}\left(\mathrm{d}\log\left(\dfrac{54C''c_0}{\varepsilon_k}\right) + \log\left(\dfrac{1}{\tilde{\delta}_k}\right)\right),$$

where the last inequality is by $c_0 \geq 1$, $C'' > 144C^3$, and $(C'')^{150/108} > 54C''$. By Corollary 4.1 of Vidyasagar (2003), this implies

$$m_k > \frac{54C''c_0}{\varepsilon_k}\left(\mathsf{d}\log\left(\frac{m_k}{\mathsf{d}}\right) + \log\left(\frac{1}{\tilde{\delta}_k}\right)\right),$$

so that

$$\varepsilon\left(m_k, \tilde{\delta}_k; \tilde{\mathcal{A}}_k\right) < \frac{\varepsilon_k}{54C''}. \tag{62}$$

Letting $\alpha = \frac{2}{3}$, we therefore have $\varepsilon\left(m_k, \tilde{\delta}_k; \tilde{\mathcal{A}}_k\right) < \frac{\alpha^2}{4}\frac{\varepsilon_k}{6C''}$. Together with (60) and Lemma 7 of Appendix D, we have that with conditional probability at least $1 - \tilde{\delta}_k$ given the event that $k \leq K$ and the random variables $V_{k-1}$, $R_k$, and $\Delta_{i_k}$,

$$\sup_{g \in V_{k-1}} \hat{P}_{S_k^1}((\{g \neq h_k\} \setminus R_k) \setminus \Delta_{i_k}) < \frac{\varepsilon_k}{6C''}. \tag{63}$$

By the law of total probability, there is an event $E'_{3,k}$ of probability at least $1 - \tilde{\delta}_k$ such that, on $E'_{3,k}$, if $k \leq K$, then (63) holds. To unify notation, for any $k \in \{1, \ldots, N+1\}$ having probability zero of $k \leq K$, define $E'_{3,k}$ as the event (of probability one) that $k > K$, so that this conclusion also vacuously holds for such values $k$.

In particular, for any $k \in \{1, \ldots, N+1\}$, suppose the events $E'_{3,k}$ and $k \leq K$ occur. For each $t \in \{1, \ldots, m_k\}$, let $\zeta'_{t,0} = \mathbb{1}[h_k(X_{t_k+t}) = 0]\mathbb{1}[X_{t_k+t} \notin (R_k \setminus \Delta_{i_k})]$, $\zeta'_{t,1} = \mathbb{1}[h_k(X_{t_k+t}) = 1]\mathbb{1}[X_{t_k+t} \notin (R_k \setminus \Delta_{i_k})]$, $q'_t = \mathbb{1}[X_{t_k+t} \in R_k \setminus \Delta_{i_k}]$. Note that these values satisfy the second and third constraints on $\zeta_{t,0}, \zeta_{t,1}, q_t$ in $\mathrm{LP}_k$. Moreover, (63) implies that $\forall g \in V_{k-1}$,

$$\frac{1}{m_k}\sum_{t=1}^{m_k} \zeta'_{t,1-g(X_{t_k+t})}\mathbb{1}[X_{t_k+t} \notin \Delta_{i_k}] = \hat{P}_{S_k^1}((\{g \neq h_k\} \setminus R_k) \setminus \Delta_{i_k}) < \frac{\varepsilon_k}{6C''},$$

so that the first constraint in $\mathrm{LP}_k$ is also satisfied by this choice of $\zeta_{t,0}, \zeta_{t,1}, q_t$. Since the values $\zeta^k_{t,0}, \zeta^k_{t,1}, q^k_t$ at the solution of $\mathrm{LP}_k$ *minimize* $\sum_{t=1}^{m_k} q_t$ among all choices of $\zeta_{t,0}, \zeta_{t,1}, q_t$ satisfying the constraints, we conclude that the above values of $q'_t$ satisfy

$$|Q_k| = \sum_{t=1}^{m_k} q^k_t \leq \sum_{t=1}^{m_k} q'_t = m_k\hat{P}_{S_k^1}(R_k \setminus \Delta_{i_k}). \tag{64}$$

Next we upper bound the right hand side of (64) via a multiplicative Chernoff bound (Lemma 6 of Appendix D). Consider again any $k \in \{1, \ldots, N+1\}$ having non-zero probability of $k \leq K$. Given the event $k \leq K$ and the random variables $R_k$ and $\Delta_{k-1}$, Lemma 6 of Appendix D implies that, with conditional probability at least $1 - \tilde{\delta}_k$,

$$\hat{P}_{S_k^1}(R_k \setminus \Delta_{i_k}) \leq \max\left\{2P_X(R_k \setminus \Delta_{i_k}), \frac{6}{m_k}\ln\left(\frac{2}{\tilde{\delta}_k}\right)\right\} \leq \max\{2P_X(R_k \setminus \Delta_{i_k}), \varepsilon_k\}, \tag{65}$$

where the last inequality follows from (62) and straightforward reasoning about numerical constant factors. By the law of total probability, there is an event $E''_{3,k}$ of probability at least $1 - \tilde{\delta}_k$ on which, if $k \leq K$, then (65) holds. To unify notation, for any $k \in \{1, \ldots, N+1\}$ having probability zero of $k \leq K$, also define $E''_{3,k}$ as the event (of probability one) that $k > K$, so that this conclusion also vacuously holds for such values $k$.

Define $E'_3 = \bigcap_{k=1}^{N+1} E'_{3,k} \cap E''_{3,k}$. By the union bound, the event $E'_3$ fails with probability at most

$$\sum_{k=1}^{N+1} 2\tilde{\delta}_k = \sum_{k=1}^{N+1} \frac{\delta\varepsilon_{k+3}}{72} < \frac{\delta}{8},$$

where the last inequality follows from our choice of $C = \frac{11}{10}$.

Altogether, on the event $E'_3$, for every $k \in \{1, \ldots, K\}$, (64), (65), and (61) together imply

$$|Q_k| \leq 2m_k\Phi_{P_{\Delta_{i_k}}}\left(V_{k-1}, \frac{\varepsilon_k}{18C''}\right) + m_k\varepsilon_k. \tag{66}$$

It remains to relate the right hand side of (66) to the quantity $\varphi_P(\varepsilon, 0; 5\beta)$. For the remainder of the proof, suppose the event $E_0 \cap E_1 \cap E_2 \cap E_3$ holds (with $E_3'$ in the definition of $E_3$ from the proof of Lemma 23 replaced by the above definition of $E_3'$). Consider any $k \in \{1, \ldots, K\}$. Recall that Lemma 17 implies $h^\star \in V_{k-1}$, which together with Lemma 12 implies $V_{k-1} \subseteq \mathrm{B}_{P_{\Delta_{i_k}}}(h^\star, \varepsilon_k)$. Thus, since the definition of $\Phi_\mu(\cdot, \cdot)$ is non-decreasing in its first argument, we have

$$\Phi_{P_{\Delta_{i_k}}}\left(V_{k-1}, \frac{\varepsilon_k}{18C''}\right) \le \Phi_{P_{\Delta_{i_k}}}\left(\mathrm{B}_{P_{\Delta_{i_k}}}(h^\star, \varepsilon_k), \frac{\varepsilon_k}{18C''}\right).$$

Also recall that Lemma 20 implies $P_X(\Delta_{i_k}) \le 5\beta$, so that $\Delta_{i_k}$ is among the sets $\Delta$ considered in the supremum in the definition of $\varphi_P(\varepsilon_k, 0; 5\beta)$. Additionally, note that $\varepsilon_k \ge \varepsilon_{N+1} > \frac{\varepsilon}{2C}$. It follows that

$$\Phi_{P_{\Delta_{i_k}}}\left(\mathrm{B}_{P_{\Delta_{i_k}}}(h^\star, \varepsilon_k), \frac{\varepsilon_k}{18C''}\right) = \frac{\Phi_{P_{\Delta_{i_k}}}\left(\mathrm{B}_{P_{\Delta_{i_k}}}(h^\star, \varepsilon_k), \frac{\varepsilon_k}{18C''}\right)}{\varepsilon_k}\varepsilon_k$$

$$\le \sup_{r > \varepsilon/2C} \frac{\Phi_{P_{\Delta_{i_k}}}\left(\mathrm{B}_{P_{\Delta_{i_k}}}(h^\star, r), \frac{r}{18C''}\right)}{r}\varepsilon_k = \sup_{r > \varepsilon} \frac{\Phi_{P_{\Delta_{i_k}}}\left(\mathrm{B}_{P_{\Delta_{i_k}}}\left(h^\star, \frac{r}{2C}\right), \frac{r}{36CC''}\right)}{r/(2C)}\varepsilon_k$$

$$\le 2C \sup_{r > \varepsilon} \frac{\Phi_{P_{\Delta_{i_k}}}\left(\mathrm{B}_{P_{\Delta_{i_k}}}(h^\star, r), \frac{r}{36CC''}\right)}{r}\varepsilon_k \le 2C\varphi_P(\varepsilon, 0; 5\beta)\varepsilon_k.$$

Altogether, we have that every $k \in \{1, \ldots, K\}$ satisfies

$$|Q_k| \le (4C\varphi_P(\varepsilon, 0; 5\beta) + 1)\, m_k\varepsilon_k \le (4C + 1)\varphi_P(\varepsilon, 0; 5\beta)m_k\varepsilon_k.$$

Substituting $|Q_k|$ in place of $|S_k^1 \cap D_{k-1} \setminus \Delta_{i_k}|$ in the proof of Lemma 23, and using the above bound on $|Q_k|$ in place of (39), we arrive at a bound $Q(\varepsilon, \delta; P)$ on the total number of queries

$$Q(\varepsilon, \delta; P) := 10\beta M_2 + \min\left\{M_1, \frac{4C+1}{2}\varphi_P(\varepsilon, 0; 5\beta)\varepsilon(N+1)m_{N+1}\right\}$$

$$= O\left(\frac{\beta^2}{\varepsilon^2}\left(\mathsf{d} + \log\left(\frac{1}{\delta}\right)\right)\right) + \min\left\{\varphi_P(\varepsilon, 0; 5\beta)\log\left(\frac{1}{\varepsilon}\right), \frac{1}{\varepsilon}\right\}\left(\mathsf{d}\log\left(\frac{1}{\varepsilon}\right) + \log\left(\frac{1}{\delta}\right)\right).$$

■

As with $\mathbb{A}_{\mathrm{avid}}$, the algorithm $\mathbb{A}_{\mathrm{avid}}^{\mathrm{sub}}$ *does not need to know* the value $\varphi_P(\varepsilon, 0; 5\beta)$ (or anything else about $P$) to achieve this query complexity: that is, it is *adaptive* to the value $\varphi_P(\varepsilon, 0; 5\beta)$.

Together with (57), Theorem 33 further implies a (sometimes loose) relaxation in terms of $\varphi_P(\varepsilon, 5\beta)$, which is often easier to evaluate for given scenarios $(\mathbb{C}, P)$. This is stated formally in the following corollary. As mentioned above, the example in Appendix F.1 shows that it is not generally possible to reduce the $\varphi_P(\varepsilon, 5\beta)^2$ dependence to a linear $\varphi_P(\varepsilon, 5\beta)$ (or even any $\varphi_P(0, \alpha)$).

**Corollary 34.** *The query complexity bound $Q(\varepsilon, \delta; P)$ in Theorem 33 (achieved by $\mathbb{A}_{\mathrm{avid}}^{\mathrm{sub}}$) satisfies*

$$Q(\varepsilon, \delta; P) = O\left(\frac{\beta^2}{\varepsilon^2}\left(\mathsf{d} + \log\left(\frac{1}{\delta}\right)\right)\right) + \tilde{O}\left(\min\left\{\mathsf{d}\varphi_P(\varepsilon, 5\beta)\left(\frac{\beta + \varepsilon}{\varepsilon}\right), \frac{\mathsf{d}}{\varepsilon}\right\}\right)$$

$$= O\left(\frac{\beta^2}{\varepsilon^2}\left(\mathsf{d} + \log\left(\frac{1}{\delta}\right)\right)\right) + \tilde{O}\left(\min\left\{\mathsf{d}\varphi_P(\varepsilon, 5\beta)^2, \frac{\mathsf{d}}{\varepsilon}\right\}\right).$$

**Proof.** Due to the first two inequalities in (57), the second term in the expression of $Q(\varepsilon, \delta; P)$ in Theorem 27 is at most

$$O\left(\min\left\{\varphi_P(\varepsilon, 5\beta)\log\left(\frac{1}{\varepsilon}\right)\left(\frac{\beta + \varepsilon}{\varepsilon}\right), \frac{1}{\varepsilon}\right\}\left(\mathsf{d}\log\left(\frac{1}{\varepsilon}\right) + \log\left(\frac{1}{\delta}\right)\right)\right). \tag{67}$$

Relaxing $\mathsf{d}\log\left(\frac{1}{\varepsilon}\right) + \log\left(\frac{1}{\delta}\right) \le \log\left(\frac{1}{\varepsilon}\right)\left(\mathsf{d} + \log\left(\frac{1}{\delta}\right)\right)$ and noting that

$$\varphi_P(\varepsilon, 5\beta)\log^2\left(\frac{1}{\varepsilon}\right)\left(\frac{\beta + \varepsilon}{\varepsilon}\right) \le \varphi_P(\varepsilon, 5\beta)^2\log^4\left(\frac{1}{\varepsilon}\right) + \left(\frac{\beta + \varepsilon}{\varepsilon}\right)^2,$$

and $\left(\frac{\beta+\varepsilon}{\varepsilon}\right)^2 \leq 4\frac{\beta^2}{\varepsilon^2} + 4$, the quantity (67) is at most

$$O\left(\frac{\beta^2}{\varepsilon^2}\left(\mathsf{d} + \log\left(\frac{1}{\delta}\right)\right) + \min\left\{\varphi_P(\varepsilon, 5\beta)^2 \log^4\left(\frac{1}{\varepsilon}\right), \frac{1}{\varepsilon}\log\left(\frac{1}{\varepsilon}\right)\right\}\left(\mathsf{d} + \log\left(\frac{1}{\delta}\right)\right)\right).$$

Adding this to the first term in the expression of $Q(\varepsilon, \delta; P)$, the result follows. ∎

An immediate consequence of Corollary 34 is that, whenever $\sup_{\alpha \in [0,5]} \varphi_P(\varepsilon, \alpha) = \mathrm{polylog}\left(\frac{1}{\varepsilon}\right)$, the dependence on $\beta, \varepsilon$ in the query complexity bound in Theorem 33 is of order $\frac{\beta^2}{\varepsilon^2} + \mathrm{polylog}\left(\frac{1}{\varepsilon}\right)$. As was true of Corollary 28, (56) implies the first upper bound in Corollary 34 is never larger than the upper bound in Theorem 1; however, this is not always the case for the second upper bound in Corollary 34. Moreover, unlike Theorem 33, *both* upper bounds in Corollary 34 can sometimes be loose compared to the ꜱ-dependent bound in Theorem 5 (e.g., Example 6 in Appendix F.2.2). For this reason, as with Theorem 27, Theorem 33 is useful despite having a quantity $\varphi_P(\varepsilon, 0; 5\beta)$ that is more challenging to calculate, as it provides a starting point for $P$-dependent analysis that is at least never worse than Theorem 5.

To illustrate a well-known scenario where the technique presented in this subsection provides improvements over the basic $\mathbb{A}_{\mathrm{avid}}$ algorithm, consider the following example.

**Example 8** (Homogeneous linear classifiers, uniform distribution)**.** As an implication of Corollary 34, we find that $\mathbb{A}_{\mathrm{avid}}^{\mathrm{sub}}$ recovers a near-optimal query complexity bound for learning homogeneous linear classifiers under any marginal $P_X$ that is isotropic log-concave. Let $d \geq 2$. For any $x, w \in \mathbb{R}^d$, denote by $h_w(x) = \mathbb{1}[\langle w, x \rangle \geq 0]$. In this scenario, we suppose $\mathcal{X} = \mathbb{R}^d$, $\mathbb{C} = \{h_w : w \in \mathbb{R}^d, \|w\| = 1\}$ (for which $\mathrm{VC}(\mathbb{C}) = d$), and $P_X$ is any isotropic log-concave distribution (Balcan and Long, 2013) (for instance, $P_X = \mathrm{Uniform}(\{x : \|x\| = 1\})$ is one such distribution). In other words, $\mathbb{C}$ is the class of linear classifiers whose hyperplane decision boundary passes through the origin. This scenario has a long history of interest in the active learning literature (see Section A), featuring prominently (with $P_X$ a uniform distribution) in the original $A^2$ paper of Balcan, Beygelzimer, and Langford (2005, 2006, 2009), which studied the case $\beta \lesssim \varepsilon/\sqrt{d}$ and showed a query complexity bound $\tilde{O}\left(d^2 \log\left(\frac{1}{\varepsilon}\right)\log\left(\frac{1}{\delta}\right)\right)$ in this regime. Later works refined this, via subregion-based techniques. Building on the works of Balcan, Broder, and Zhang (2007); Balcan and Long (2013) (which studied more-restrictive noise models), Zhang and Chaudhuri (2014) obtain a query complexity bound $\tilde{O}\left(d\frac{\beta^2}{\varepsilon^2} + d\right)$. Here we argue this query complexity bound can be recovered from Corollary 34 (indeed, with improvements by log factors in the lead term). Specifically, Zhang and Chaudhuri (2014) show (based on results of Balcan and Long, 2013) that $\varphi_P(\varepsilon, 5\beta) = O\left(\log\left(\frac{\beta}{\varepsilon}\right)\right)$. Plugging into Corollary 34 (rather, the expression obtained in the proof thereof), we obtain a query complexity bound

$$O\left(\frac{\beta^2}{\varepsilon^2}\left(d + \log\left(\frac{1}{\delta}\right)\right) + \log^2\left(\frac{\beta}{\varepsilon}\right)\log^4\left(\frac{1}{\varepsilon}\right)\left(d + \log\left(\frac{1}{\delta}\right)\right)\right).$$

Compared to the result of Zhang and Chaudhuri (2014), this improves the lead term by a factor $\log^2\left(\frac{\beta}{\varepsilon}\right)$ (though at the expense of additional log factors in the lower-order term). We also note that this query complexity bound represents a refinement of what would be obtained from Corollary 28, since even for the special case of $P_X$ uniform on an origin-centered sphere, $\theta_P(\beta + \varepsilon) = \Theta\left(\sqrt{\mathsf{d}} \wedge \frac{1}{\beta+\varepsilon}\right)$ (Hanneke, 2007b).

### F.4  Classes with Infinite VC Dimension via Covering Numbers

As one final remark about $P$-dependent query complexity bounds, we note that it is also possible to derive interesting query complexity improvements over passive learning even for classes with $\mathrm{VC}(\mathbb{C}) = \infty$ under conditions on $P$ commonly studied in the *nonparametric* passive learning literature: namely, bounded *covering numbers*.

Denote by $\mathcal{N}(\varepsilon, \mathbb{C}, \mathrm{L}_1(P_X))$ the minimal size of a proper $\varepsilon$-cover: that is, the size of the smallest $\mathbb{C}' \subseteq \mathbb{C}$ for which $\sup_{h \in \mathbb{C}} \min_{h' \in \mathbb{C}'} P_X(h \neq h') \leq \varepsilon$. Being able to construct such a cover from unlabeled examples requires some additional structure beyond finite covering numbers under $P_X$ (e.g.,

finite expected empirical covering numbers suffices; see e.g., van der Vaart and Wellner, 1996). Let us suppose such conditions are satisfied by $(\mathbb{C}, P_X)$, so that (since an active learner can be assumed to have access to an abundant supply of *unlabeled* examples) we may assume we have access to a valid $(\varepsilon/2)$-proper-cover $\mathbb{C}_{\varepsilon/2}$ under $\mathrm{L}_1(P_X)$ of size $O(\mathcal{N}(\varepsilon/2, \mathbb{C}, \mathrm{L}_1(P_X)))$. Constructing this cover $\mathbb{C}_{\varepsilon/2}$ does not affect the query complexity, since it only requires the use of unlabeled examples.

We can then run $\mathbb{A}_{\mathrm{avid}}$ using $\mathbb{C}_{\varepsilon/2}$ in place of $\mathbb{C}$. Since $\mathrm{VC}(\mathbb{C}_{\varepsilon/2}) = O(\log(\mathcal{N}(\varepsilon/2, \mathbb{C}, \mathrm{L}_1(P_X))))$, and $\inf_{h \in \mathbb{C}_{\varepsilon/2}} \mathrm{er}_P(h) \leq \inf_{h \in \mathbb{C}} \mathrm{er}_P(h) + \frac{\varepsilon}{2} =: \beta + \frac{\varepsilon}{2}$, we thereby obtain from Theorem 1 a $P_X$-dependent query complexity bound

$$O\left( \frac{\beta^2}{\varepsilon^2} \log(\mathcal{N}(\varepsilon/2, \mathbb{C}, \mathrm{L}_1(P_X))/\delta) \right) + \tilde{O}\left( \frac{1}{\varepsilon} \log(\mathcal{N}(\varepsilon/2, \mathbb{C}, \mathrm{L}_1(P_X))) \right).$$

This result can then be composed with bounds on the covering numbers $\mathcal{N}(\varepsilon/2, \mathbb{C}, \mathrm{L}_1(P_X))$ of various classes $\mathbb{C}$ under various conditions on $P_X$ known from the literature. For instance, this provides an improved query complexity for *boundary fragment* classes (a class defined by smoothness conditions on the decision boundaries of concepts in $\mathbb{C}$) under near-uniform distributions $P_X$ on $[0,1]^{k+1}$ compared to the results established by Wang (2011) (see Wang, 2011; Tsybakov, 2004 for the precise definitions and covering numbers).

# G    Extensions and Future Directions

We conclude with some extensions and several interesting open questions and future directions.

**Extension to Multiclass Classification:**    We can easily generalize the result to hold for *multiclass classification*: that is, where $\mathcal{Y}$ is a general label space, $\mathbb{C}$ is a family of measurable functions $h : \mathcal{X} \to \mathcal{Y}$, $P$ is a distribution on $\mathcal{X} \times \mathcal{Y}$, and we still define $\mathrm{er}_P(h) := P((x,y) : h(x) \neq y) = P(\mathrm{ER}(h))$. The exact same upper bound extends to this setting if we replace d with $\max\{\mathrm{VC}(\mathcal{A}), \mathrm{d}_G\}$ where $\mathcal{A}$ is as in Lemma 9 (replacing $\{0,1\}$ with $\mathcal{Y}$ there) and $\mathrm{d}_G$ denotes the *graph dimension* of $\mathbb{C}$ (Natarajan, 1989). The star number $\mathfrak{s}$ is still defined as in Definition 2 (see Hanneke, 2024). The proof holds with only superficial modifications to rely solely on $\mathrm{VC}(\mathcal{A})$ (for the $\mathcal{X} \setminus \Delta_{i_k}$ concentration) and $\mathrm{d}_G$ (for concentration in $\Delta_{i_k}$). We further note that this dimension $\max\{\mathrm{VC}(\mathcal{A}), \mathrm{d}_G\}$ is at most $O(\mathrm{d}_N(\mathbb{C}) \log(|\mathcal{Y}|))$, where $\mathrm{d}_N(\mathbb{C})$ is the *Natarajan dimension* of $\mathbb{C}$ (Natarajan, 1989); this follows by a similar argument as used to bound $\mathrm{VC}(\mathcal{A})$ in the proof of Lemma 9, using a generalization of Sauer's lemma for the multiclass setting proven by Haussler and Long (1995).

For a bounded number of labels $|\mathcal{Y}|$, this again leads to essentially optimal query complexity, as a lower bound $\Omega\left( \frac{\mathrm{d}_N(\mathbb{C})\beta^2}{\varepsilon^2} \right)$ based on the Natarajan dimension $\mathrm{d}_N(\mathbb{C})$ can be shown (similarly to the lower bound for binary classification).

However, for unbounded label spaces ($|\mathcal{Y}| = \infty$) the learnability and optimal sample complexity of passive learning in the realizable case are known to depend on a dimension called the *DS dimension* (Brukhim, Carmon, Dinur, Moran, and Yehudayoff, 2022; Daniely and Shalev-Shwartz, 2014) which is between the Natarajan dimension and graph dimension. This raises an important question: What is the optimal query complexity for multiclass agnostic active learning?

**Extension to Stream-based Active Learning:**    For simplicity, we have defined the learning model as so-called *pool-based* active learning, in that the learning algorithm was given the entire sequence $X_1, \ldots, X_m$ of unlabeled examples as input, and can query any example, in any order. However, it is also common to consider an alternative protocol called *stream-based* active learning (or *selective sampling*): namely, where the active learner observes the unlabeled examples $X_t$ one-at-a-time *in sequence*, and for each, decides whether or not to query, and can never revisit that decision later. In the literature on stream-based active learning, it is common to express the guarantees of the active learning in two parts: (1) a bound on the error guarantee expressed as a function of the number $m$ of unlabeled examples processed, and (2) a bound on the number of queries it makes among the first $m$ examples (e.g., Dasgupta, Hsu, and Monteleoni, 2007).

We note that $\mathbb{A}_{\mathrm{avid}}$ can easily be re-expressed as a stream-based active learner. Specifically, rather than limiting the '*For*' loop in Step 1 to $N = O\left( \log\left( \frac{1}{\varepsilon} \right) \right)$ rounds, we can simply let the algorithm run until it has allocated as many unlabeled examples $m$ as we wish. Rather than allocating all of the

$S_k^1, S_k^4$ data subsets at the start, we can simply allocate these sets if and when the algorithm reaches the $k^{\text{th}}$ iteration of the '*For*' loop, at which point the algorithm collects the next $m_k$ examples to allocate to $S_k^1$, querying each of these examples $X_t$ iff $X_t \in D_{k-1} \setminus \Delta_{i_k}$. Likewise, it then collects the next $m_k$ examples to allocate to $S_k^4$ (without making any queries), to calculate the value $m_k'$ (where, in this case, we should suitably replace the value $3 + N - k$ in the log term in $m_k'$ to remove the dependence on $\varepsilon$: for instance, replacing it with $k + 2$ would suffice for the present discussion). It then collects the next $m_k'$ examples to allocate to the data subset $S_k^2$, querying each of these examples $X_t$ iff $X_t \in \Delta_{i_k}$. It then moves on to execute Steps 3-4. Similarly, upon each time it reaches Step 5, it simply collects the next $m_k$ unlabeled examples to construct $S_{k,i}^3$ (without making any queries), which then enables it to execute Steps 5-7. We can execute this until any number $m$ of unlabeled examples have been processed, and define the predictor at such a time as the $\hat{h}_k$ for the last iteration $k$ for which Step 2 was able to completely execute. If the algorithm ever satisfies the early stopping criterion in Step 4 for some iteration $k$, we can simply take $\hat{h}_k$ as its final predictor. We can then derive the corresponding excess error bound and query bound from the above analysis of the query complexity and unlabeled sample complexity: namely, with probability at least $1 - \delta$, the predictor $\hat{h}$ produced after $m$ unlabeled examples satisfies

$$\mathrm{er}_P(\hat{h}) - \inf_{h \in \mathbb{C}} \mathrm{er}_P(h) = O\left(\sqrt{\beta\left(\mathsf{d}\log\left(\frac{m}{\mathsf{d}}\right) + \log\left(\frac{1}{\delta}\right)\right)} + \frac{1}{m}\left(\mathsf{d}\log\left(\frac{m}{\mathsf{d}}\right) + \log\left(\frac{1}{\delta}\right)\right)\right)$$

and its number of queries is bounded by

$$O\left(\beta m + \min\left\{\mathsf{s}\log\left(\frac{m}{\mathsf{d}}\right)\left(\mathsf{d}\log\left(\frac{m}{\mathsf{d}}\right) + \log\left(\frac{1}{\delta}\right)\right), \sqrt{\frac{m}{\beta}\left(\mathsf{d}\log\left(\frac{m}{\mathsf{d}}\right) + \log\left(\frac{1}{\delta}\right)\right)}, m\right\}\right)$$

$$= O(\beta m) + \tilde{O}\left(\min\left\{\mathsf{s}\mathsf{d}, \sqrt{\frac{m\mathsf{d}}{\beta}}, m\right\}\right).$$

Here the $\beta m$ term is where the improvements over passive learning provided by the AVID principle are reflected in the query bound (as the above excess error bound is nearly as small as the best achievable excess error guarantees for passive learning with $m$ labeled examples; Vapnik and Chervonenkis, 1974; Devroye and Lugosi, 1995; Hanneke, Larsen, and Zhivotovskiy, 2024b). In particular, the above guarantees compare favorably to previous analyses of stream-based active learning (e.g., Dasgupta, Hsu, and Monteleoni, 2007) in the regime of moderate-size $\beta$, where, for the same excess error guarantee, the bounds on the number of queries include a term such as $\tilde{O}(\theta_P(\beta)\beta m)$, which becomes of order $\left(\mathsf{s} \wedge \frac{1}{\beta}\right)\beta m$ in the worst case over $P$, and hence is no better than $m$ when $\mathsf{s} = \infty$. In contrast, in this regime of moderate-size $\beta$, where the $\beta m$ term dominates, we obtain a factor $\beta$ improvement in the number of queries.

We also remark that the analysis above also supplies an "anytime" guarantee, where the algorithm can simply be executed indefinitely, and the above excess error bound and query bound hold simultaneously for every $m$ (where, again, if the algorithm ever satisfies the condition in Step 4, its predictor should simply be defined as the corresponding $\hat{h}_k$ forevermore, and it need not query any further examples in the sequence).

**The Optimal Lower-Order Term:** As discussed above, while the leading term in Theorem 3 is exacty optimal (perfectly matching a lower bound), the lower-order term in the upper bound in Theorem 3 presents a small gap (in the dependence on d) compared to the best known lower bound (Hanneke and Yang, 2015). As discussed, some aspects of this gap (concerning $\frac{1}{\varepsilon} + \mathsf{d}$ vs $\frac{\mathsf{d}}{\varepsilon}$) *cannot* be improved if the dependence on $\mathbb{C}$ is only expressed via d and $\mathsf{s}$: that is, without introducing new complexity measures. We leave open the question of formulating such an always-sharp complexity measure, that is, the question: What is the optimal form of the query complexity $\Theta(\mathrm{QC}_a(\varepsilon, \delta; \beta, \mathbb{C}))$ for all classes $\mathbb{C}$? However, aside from this gap, there is a gap which might be improvable even in expressions of the bound purely in terms of d and $\mathsf{s}$: namely, the term $\mathsf{s}\mathsf{d}$ in the upper bound. I conjecture this can be reduced to simply $\mathsf{s}$: that is, $\mathrm{QC}_a(\varepsilon, \delta; \beta, \mathbb{C}) = O\left(\frac{\beta^2}{\varepsilon^2}\left(\mathsf{d} + \log\left(\frac{1}{\delta}\right)\right)\right) + \tilde{O}\left(\min\left\{\mathsf{s}, \frac{\mathsf{d}}{\varepsilon}\right\}\right)$ for every concept class $\mathbb{C}$.

**Proper Learning:** As noted above, the $\mathbb{A}_{\mathrm{avid}}$ algorithm is an *improper* learner, meaning its returned predictor $\hat{h}$ might not be an element of the concept class $\mathbb{C}$ (rather, it is a shallow decision list built from concepts in $\mathbb{C}$). It is an interesting open question to determine whether there exist *proper* active learners achieving the query complexity bound in either Theorem 1 or 3 for every concept class $\mathbb{C}$. It follows from Corollary 18 that, in the return case in Step 9, it would suffice to return $\hat{h}$ equal any element of $V_N$. Thus, the main challenge in obtaining a proper learner is in the early-stopping case in Step 4. In this return case, we have effectively verified that $\mathrm{er}_P\big(\hat{h}_k\big)$ is *better* than $\mathrm{er}_P\big(h^\star\big)$ (Lemma 17). However, the resolution of the error estimates $\hat{\mathrm{er}}_k^{1,2}$ at this stage might not yet be sufficient to find an $h \in V_{k-1}$ nearly as good. Indeed, for this reason, any such early return case in an active learning algorithm may be problematic for proper learning.

On the other hand, we remark that, for all previous known separations between proper and improper sample complexities, the respective proofs break down if the learner is given access to the marginal distribution $P_X$ or a sufficiently large unlabeled data set (Bousquet, Hanneke, Moran, and Zhivotovskiy, 2020; Hanneke, Larsen, and Zhivotovskiy, 2024b; Daniely and Shalev-Shwartz, 2014; Montasser, Hanneke, and Srebro, 2019; Asilis, Devic, Sharan, and Teng, 2025a; Asilis, Høgsgaard, and Velegkas, 2025b). Since, for the purpose of merely bounding the *query complexity*, we may suppose an active learner has access to a large unlabeled data set, this hints that such improvements might indeed be achievable by proper active learners, or otherwise, a novel technique is needed for establishing such a separation between proper and improper active learning.

**Computational Efficiency:** The focus of this work has been solely on the information-theoretic query complexity of agnostic active learning, without any computational or resource constraints beyond the number of queries and unlabeled examples. However, computational considerations are of course also important to consider. To actually achieve the agnostic learning guarantee of $\varepsilon$ excess error is typically thought to be computationally intractable for many concept classes, without distribution restrictions. Nonetheless, it would be interesting to determine whether, at least at some level, the improvements in the leading term reflected in Theorems 1 and 3 might also be reflected in a computationally efficient method, for some classes $\mathbb{C}$ (e.g., linear classifiers) under some restrictions on the distribution $P$ which enable computational tractability yet for which such query complexity bounds are not captured by prior results (e.g., by $\theta_P(\varepsilon)$).

Beyond this, a classical approach to obtaining computationally efficient algorithms in *practice* is to introduce *convex relaxations* of the various optimization problems involved in a given algorithm. In the literature on passive learning, the theory of error bounds for empirical risk minimization has been extended to allow for convex relaxations of the 0-1 loss, called a *surrogate loss*, while still guaranteeing bounds on the excess error rate under appropriate assumptions on $P$ relating excess surrogate risks to excess error rates (Bartlett, Jordan, and McAuliffe, 2006; Zhang, 2004). Prior work on *disagreement-based* active learning has been found to compose well with this theory of surrogate losses. Specifically, Hanneke and Yang (2019); Hanneke (2014) express disagreement-based active learning algorithms, in which the optimization problems defining the query criterion and the learner's final predictor are relaxed to convex programs expressed in terms of any given surrogate loss. For such algorithms, they derive query complexity bounds (based on the disagreement coefficient $\theta_P(\varepsilon)$) holding under the same conditions studied by the passive learning works (Bartlett, Jordan, and McAuliffe, 2006; Zhang, 2004). It is thus a natural question to determine whether such a theory can be made to work for the algorithmic principles underlying $\mathbb{A}_{\mathrm{avid}}$ (i.e., the AVID principle), leading to an algorithm only requiring computationally tractable convex optimization problems based on a given surrogate loss, and expressing query complexity improvements over passive learning (of the type found in Theorems 1 and 3) under these same conditions on $P$ relating excess surrogate risks to the excess error rates. This approach is made challenging in the context of $\mathbb{A}_{\mathrm{avid}}$, due to its use of improper predictors $\hat{h}_k$, and even more-so due to the maximization in Steps 5 and 6 (whereas convex surrogate losses would typically only allow tractability of *minimization* problems).

As a step toward such a technique, an interesting intermediate question is whether Theorems 1 and 3 can be achieved by an active learning algorithm expressed as a *reduction to an empirical risk minimization (ERM) oracle*: that is, where the access to the concept class $\mathbb{C}$ is restricted to solving optimization problems of the form $\mathrm{argmin}_{h \in \mathbb{C}} \hat{\mathrm{er}}_S(h)$ for data sets $S$ (or possibly a *weighted* ERM). This would be particularly interesting if these data sets $S$ are only constructed from subsets of the labeled examples $(X_t, Y_t)$ queried by the algorithm (perhaps plus one additional example $(X_t, y)$ with an artificial label $y$, which may be needed when deciding whether to query $X_t$). Previous works

by Beygelzimer, Hsu, Langford, and Zhang (2010); Hsu (2010) have expressed *disagreement-based* active learning algorithms as reductions to such ERM oracles. It is therefore a natural question to consider whether the AVID principle can also be implemented based only on such oracles (and such an implementation could also be an important step toward enabling the above composition with the theory of surrogate losses).

**Unlabeled Sample Complexity:**   Theorem 5 reveals that, to achieve the stated query complexity bound with $\mathbb{A}_{\mathrm{avid}}$, it suffices to have access to a number of *unlabeled* examples $M(\varepsilon, \delta; \beta) = O\left(\frac{\beta+\varepsilon}{\varepsilon^2}\left(\mathsf{d}\log\left(\frac{1}{\varepsilon}\right) + \log\left(\frac{1}{\delta}\right)\right)\right)$. In comparison, we can obtain an obvious lower bound on the number of unlabeled examples necessary to achieve any query complexity bound by a lower bound on the sample complexity of fully-supervised *passive* learning (Devroye and Lugosi, 1995): i.e., $\Omega\left(\frac{\beta+\varepsilon}{\varepsilon^2}\left(\mathsf{d} + \log\left(\frac{1}{\delta}\right)\right)\right)$. Thus, the upper bound $M(\varepsilon, \delta; \beta)$ in Theorem 5 can be improved by at most a $\log\left(\frac{1}{\varepsilon}\right)$ factor. This naturally raises the question: Is it possible to achieve a near-optimal query complexity $\Theta(\mathrm{QC}_a(\varepsilon, \delta; \beta, \mathbb{C}))$ with an algorithm which uses a number of unlabeled examples $O\left(\frac{\beta+\varepsilon}{\varepsilon^2}\left(\mathsf{d} + \log\left(\frac{1}{\delta}\right)\right)\right)$? Such a result would then be optimal simultaneously in *both* the number of queries *and* the number of unlabeled examples. To date, this is not even known to be achievable by fully-supervised *passive* learning, the best known upper bound having an additive $\tilde{O}\left(\frac{\mathsf{d}}{\varepsilon}\right)$ term (Hanneke, Larsen, and Zhivotovskiy, 2024b). Thus, for now, a more-approachable question would be whether it is possible to match the query complexity bound in Theorem 3 using a number of unlabeled examples suboptimal only in log factors in the lower-order term, that is: Is there an algorithm achieving a query complexity upper bound $O\left(\frac{\beta^2}{\varepsilon^2}\left(\mathsf{d} + \log\left(\frac{1}{\delta}\right)\right)\right) + \tilde{O}\left(\left(\mathfrak{s} \wedge \frac{1}{\varepsilon}\right)\mathsf{d}\right)$ which uses a number of unlabeled examples at most $O\left(\frac{\beta}{\varepsilon^2}\left(\mathsf{d} + \log\left(\frac{1}{\delta}\right)\right)\right) + \tilde{O}\left(\frac{\mathsf{d}}{\varepsilon}\right)$? As an intermediate step, it would already be interesting to determine whether this many unlabeled examples suffices to achieve the query complexity bound in Theorem 1.

**Tsybakov Noise:**   Beyond the above directions, there are a number of further extensions of this work that seem ripe for exploration. One natural direction is extending the techniques in this work to the case of *Tsybakov noise* (Mammen and Tsybakov, 1999; Tsybakov, 2004; Massart and Nédélec, 2006). The optimal query complexity under Tsybakov noise was already identified by Hanneke and Yang (2015) (aside from similar gaps to the $\frac{\mathsf{d}}{\varepsilon}$ vs $\frac{1}{\varepsilon} + \mathsf{d}$ issue discussed above, which require introducing a new complexity measure to resolve). However, the algorithmic techniques in the present work are significantly simpler, and moreover, have the potential to dramatically reduce the number of *unlabeled* examples required for learning, compared to the technique of Hanneke and Yang (2015). I conjecture that the AVID principle is capable of yielding near-optimal query complexity guarantees under Tsybakov noise (with a number of unlabeled examples of the same order as the sample complexity of supervised learning, up to log factors); however, obtaining such guarantees may require a more-sophisticated usage of the principle, such as by the creation of multiple different regions $\Delta$, coinciding with different levels of variance of excess error estimates. Indeed, an analogous *tiered* allocation of queries was key to the original analysis of the query complexity under Tsybakov noise by Hanneke and Yang (2015).

