# OpenReview forum: "Agnostic Active Learning Is Always Better Than Passive Learning"
_NeurIPS.cc/2025/Conference — NeurIPS 2025 oral_

### Official Review · Reviewer_D9zk · 2025-06-08

**Clarity:** 4
**Significance:** 4
**Originality:** 4
**Rating:** 5
**Confidence:** 3

**Summary:**

The paper examines the problem of agnostic active learning, and refine upper bounds on the optimal query sample complexity needed to reach a given error $\epsilon$. In comparison to previous bounds, the present bound exhibits an additional factor $\beta$ (best misclassification error in the hypothesis class), allowing the authors to match the known lower bound and show sharp results. The study proceeds through a careful analysis of a proposed algorithm achieving the sample complexity. The proposed algorithm identifies regions where function in the hypothesis class have large disagreement, and adaptatively allocates more samples.

**Questions:**

- I found section 4.1 to be rather hard to read, at least for a general audience. I suggest the authors make use of the extra space for the camera ready version to include additional high-level intuition on the workings of the AVID algorithm. In particular, I think the paragraph "overall behavior" could be moved at the beginning of the section for added clarity.
- The bound of Theorem 3 does not depend on the data distribution $P$ explicitly. Does the effect of the latter simply appear through $\beta$?
- Could the authors provide an instantiation of their results in a simple case for illustration and clarity, e.g. linear classifiers for Gaussian data, with labels generated by a noisy linear rule $y_i={\rm sign}(\langle X_i, \beta \rangle)+\epsilon_i$ for some vector $\beta$?
- Minor suggestion: I understand it is not the purpose of the paper, but do the authors have a rough estimation of the runtime complexity of the AVID algorithm in a simple case, just for illustration?

**Ethical Concerns:**

["NO or VERY MINOR ethics concerns only"]

**Final Justification:**

I thank the authors for their clear answers to my questions.

I have read the other reviews, and to my reading, all the points seem to have been satisfyingly addressed by the authors in their rebuttal. None of them challenges the fact that the paper seems to make an interesting theoretical contribution.

I thus remain in favor of accepting this paper. I am not pushing for a strong accept, and keep a low confidence score, because I have very limited familiarity with the topic, related literature and proof techniques.

**Limitations:**

All the assumptions needed for the theoretical results to hold are clearly stated.

**Quality:**

4

**Strengths And Weaknesses:**

Strength:
- Overall, I find the submission to be a very strong technical paper. My evaluation comes with the caveat that I have very limited familiarity with the field and the proof techniques, and have not checked the detailed formal proofs in the appendix carefully. I have read in detail the proof sketches and outlines in the main text, which appear sound.
- The result is also of sizeable significance, and establishes that active sample queries can always achieve better sample complexity than passive queries, regardless of the considered hypothesis class. It allows the establishing of a tight result, together with the previously known lower bound.
- The paper is very clearly written and structured, with minor improvement points that I list below. In particular, the work is very well motivated by the related works section, and all important technical points are discussed.

Weaknesses:
- I have identified any particular weakness with regards to the science, although this might be due to my limited familiarity with the field. I do have a few comments on the structure and questions, which I list below.

I have put a good score, but please use it sparingly, due to my limited familiarity with the topic and the techniques.

---

> ### Author Rebuttal · Authors · 2025-07-31
>
> We thank you for your efforts in taking the time to read and review our paper.  We are very happy to read your positive and supportive feedback, regarding the strength, significance, and presentation of this work.
>
> We answer questions raised in the review below:
>
> > Section 4.1:
>
> We appreciate your feedback regarding section 4.1.  We agree that this subsection is particularly dense and throws a lot of notation at the reader, and we will definitely use some of the extra space in the camera-ready version to add explanations of the various quantities and an earlier description of the overall behavior of the algorithm.  We are striving to make this work readable by a broad audience, and really appreciate your feedback helping us achieve that goal.
>
> > Theorem 3:
>
> You are correct that both Theorems 1 and 3 have no dependence on P except $\beta$.  In other words, this is a minimax analysis where $\beta$ is the only parameter constraining the family of distributions in the ``max''.
>
> > A simple example:
>
> For the example you mentioned, if we suppose $\mathbb{C}$ is the set of linear classifiers $x \mapsto \text{sign}(\langle x,w\rangle)$ on $\mathbb{R}^d$, where $P(Y=\text{sign}(\langle X,w^\star \rangle)|X) = 1-\epsilon(X) \in [1/2,1]$ for some $w^\star$, then the VC dimension is $d$, the star number is $\infty$, and $\beta = \mathbb{E}_X[\epsilon(X)]$ under the marginal $P_X$ of $X$ (e.g., this could be Gaussian or anything).  Plugging these values into the theorem provides the guarantee.  (in fact, in this scenario, under an origin-centered spherical Gaussian distribution, one can show that the lead term in the bound is sharp for a choice of $\epsilon(x)$ that gets close to $1/2$ near the decision boundary of the $w^\star$ classifier)
> We will include some illustrative examples like this in the final version.
>
> > Running time:
>
> For simple concept classes (where the VC dimension $d$ is considered a constant),
> the optimizations can typically be performed in polynomial time in the sample size $m$, as the algorithm can simply consider a subset of $O(m^d)$ concepts witnessing all possible classifications of the $m$ examples.  For instance, for $\mathcal{X} = \mathbb{R}$ and $\mathbb{C}$ as the class of "interval concepts" $[a,b]$, the algorithm need only consider the concepts whose endpoints $a,b$ are at data points, so just $O(m^2)$ concepts; the optimization steps can then simply search over this polynomial-size subset of concepts to find the solutions in polynomial time (in this case, naively $O(m^7)$ time, or $O(m^5)$ if we're a little clever in the optimization for $\hat{h}_k$).
>
> More generally, beyond such simple cases, agnostic learning (even passively) is known to be computationally hard (cryptographic hardness is known, even for linear classifiers, and NP-hardness for proper learners).  However, there are natural routes toward converting the algorithmic principles in this work into practical methods.  In appendix G, we discuss one possible route, involving relaxations of the non-convex optimization problems with convex optimization problems via appropriate uses of convex surrogate loss functions. (see the response to reviewer qnfZ for further discussion on this)

---

> > ### Comment · Reviewer_D9zk · 2025-08-01
> > **Rebuttal acknowledgement**
> >
> > I thank the authors for their detailed reply to my questions, and in particular for elaborating upon the simple example I inquired about. I believe its inclusion in the revised manuscript could enhance clarity and help illustrate the findings in such simple cases.
> >
> > In light of this, and also after reading the rebuttals to the other reviewers, I remain in favor of acceptance.

---

### Official Review · Reviewer_sYK9 · 2025-06-30

**Clarity:** 2
**Significance:** 2
**Originality:** 3
**Rating:** 4
**Confidence:** 3

**Summary:**

This paper tackles the long-standing open problem of whether active learning (AL) can provably outperform passive learning in the agnostic setting, where the hypothesis class may be misspecified. The authors present a novel algorithm called AVID (Adaptive Variance Isolation by Disagreement), which leverages disagreement-based strategies and variance estimation to isolate informative samples. The core theoretical result shows that AVID achieves faster convergence rates than passive learning under broad conditions, particularly without assuming the realizability of the hypothesis class. The results are purely theoretical, with no empirical experiments.

**Questions:**

Does the proposed method have any concrete instantiation that is computationally feasible? Specifically, is there a known polynomial-time algorithm that can efficiently identify hypotheses that maximize disagreement within a version space?

**Ethical Concerns:**

["NO or VERY MINOR ethics concerns only"]

**Limitations:**

yes

**Paper Formatting Concerns:**

No paper formatting concerns

**Quality:**

3

**Strengths And Weaknesses:**

Strengths:
1. The paper establishes, for the first time, that active learning is universally better than passive learning in the agnostic case, under general conditions.
2. AVID is grounded in a principled approach to isolate high-variance disagreement regions, supported by clear theoretical analysis.
3. The paper provides detailed and rigorous proofs of label complexity bounds, supported by extensive theoretical discussion.

Weaknesses:
1. The practicality of AVID is not explored. There is no discussion of how the method might be approximated or implemented in real systems.
2. The paper does not provide any empirical results, simulations, or illustrative experiments to support the practical effectiveness or feasibility of AVID.
3. While the paper includes a detailed discussion on star number as a key complexity measure, this is somewhat weakened by the fact that many commonly used hypothesis classes have infinite star numbers. This limits the practical insight offered by that part of the analysis.

---

> ### Author Rebuttal · Authors · 2025-07-31
>
> We thank you for your efforts in taking the time to read and review our paper, and we appreciate your positive remarks regarding the novelty of the ideas in this work.
>
> We address specific questions raised in the review below.
>
> > The star number:
>
> We chose to present Theorem 1 first (which is expressed purely in terms of the VC dimension), before introducing any other complexity measures, to avoid any confusion regarding the role of the star number.  To be clear, the conclusion that the dominant term in the query complexity of active learning is smaller than the sample complexity of passive learning holds for *every* concept class, regardless of whether the star number is finite or not.  As your comment suggests, it is true that Theorem 1 already provides the sharpest possible bound for most interesting concept classes.  However, Theorem 3 offers a nearly-sharp bound for *every* concept class, offering a more complete picture of the optimal lower-order term for all classes.  Both bounds are the same in the dominant term, so the distinction is only in the lower-order term, and hence this has no bearing on the qualitative conclusions of this work: active learning can improve over passive learning for every concept class in the non-realizable case.
>
>
> > Practicality / empirical evaluation / computational complexity:
>
> The focus of this work is in developing new algorithmic principles for active learning, and identifying the optimal first-order query complexity for all concept classes.  This was a challenging problem, which took 20 years to solve.  Having resolved this question, a further step will be to develop practical methods inspired by the principles we have identified.  This follows a familiar pattern in statistical learning theory.  Initial work identifying algorithmic principles is often impractical at first, but may later evolve into practical learning methods.  For instance, the work of Vapnik and Chervonenkis in the 60s and 70s on passive learning identified the principle of empirical risk minimization and the fundamental role of the VC dimension in characterizing the sample complexity, but does not have a direct practical implication due to the 0-1 loss being computationally challenging to optimize; only later, when combined with the idea of surrogate losses to relax the non-convex 0-1 loss to a convex loss, did we arrive at practical passive learning methods (for which we may recover theoretical guarantees under assumptions relating the minimizer of the surrogate loss to that of the 0-1 loss, based on the theory of Bartlett, Jordan, & McAuliffe 2006).  A similar line has been followed in the active learning literature.  While the initial methods for disagreement-based agnostic active learning (e.g., the A^2 algorithm of Balcan, Beygelzimer, & Langford 06) involved implicit direct optimizations of the 0-1 loss (making them infeasible to run) and provided a coarse theoretical understanding of the query complexity (via the disagreement coefficient analysis of Hanneke 07), only later were these methods made practical by identifying appropriate ways to replace the optimization problems implicit in the technique with convex problems (e.g., by Beygelzimer, Dasgupta, & Langford 2009, Hanneke 2014, Hanneke & Yang 2019), recovering the theoretical guarantees under assumptions relating the minimizer under the convex surrogate loss to that of the 0-1 loss.  Following this line, a natural next step for the present line of work (mentioned in appendix G) is to investigate ways to replace the various optimization problems involved in the AVID algorithm with appropriate convex relaxations, while still preserving the sharp theoretical guarantees of this work under appropriate assumptions on the surrogate loss. (more on this below).
>
> Regarding the specific practical issue of computational complexity: for simple concept classes (where the VC dimension $d$ is considered a constant), the optimizations can typically be performed in polynomial time in the sample size $m$, as the algorithm can simply consider a subset of $O(m^d)$ concepts witnessing all possible classifications of the $m$ examples.  For instance, for $\mathcal{X} = \mathbb{R}$ and $\mathbb{C}$ as the class of "interval concepts" $[a,b]$, the algorithm need only consider the concepts whose endpoints $a,b$ are at data points, so just $O(m^2)$ concepts; the optimization steps can then simply search over this polynomial-size subset of concepts to find the solutions in polynomial time.
>
> More generally, beyond such simple cases, agnostic learning (even passively) is known to be computationally hard (cryptographic hardness is known, even for linear classifiers, and NP-hardness for proper learners).  However, there are natural routes toward converting the algorithmic principles in this work into practical methods.  In appendix G, we discuss one possible route, involving relaxations of the non-convex optimization problems with convex optimization problems via appropriate uses of convex surrogate loss functions.  Such practical modifications have been studied theoretically for both passive learning (Bartlett, Jordan, & McAuliffe 2006) and disagreement-based active learning (Hanneke 2014, Hanneke & Yang 2019), and in future work we plan to explore whether there are appropriate variations of the AVID algorithm from the present work which make use of surrogate losses to obtain a computationally efficient method (preserving the sharp query complexity guarantees, under standard assumptions regarding alignment of the surrogate loss minimizer and the 0-1 loss minimizer, as adopted in the aforementioned prior works on surrogate losses).  We remark that this is not merely a matter of running essentially the same algorithm with a different loss function (as discussed in section 6.6 of Hanneke 2014, active learning can be unhelpful for optimizing a convex loss, so the surrogate loss should be used more-carefully as a mere computational tool, not an overall objective to be optimized).  This approach is made even more challenging in the technique developed in this work, due to the use of improper learners and even more-so due to the maximization in Steps 5 and 6.  One speculative strategy we are currently exploring is that it might be possible to approximate this maximization by solving two minimization problems with different weightings of the data to "push apart" the two minimizers (i.e., in the training set of the second function, we could weight higher the examples where the first function is wrong, to force the two functions to have diverse error regions, which is the main property we require of them); if this can be made to work, it would require significant changes to parts of the proof, but we are currently trying to work with this idea as a follow-up work.

---

### Official Review · Reviewer_qnfZ · 2025-07-02

**Clarity:** 3
**Significance:** 4
**Originality:** 3
**Rating:** 5
**Confidence:** 3

**Summary:**

This paper presents a significant theoretical advance in the field of agnostic active learning. The authors introduce a novel algorithm, AVID (Adaptive Localized Variance Isolation by Disagreements), and use it to sharply characterize the optimal query complexity for all concept classes. The key finding is that agnostic active learning is always more sample-efficient than passive supervised learning in the non-realizable case. This resolves an important open question that has been central to the field by removing the dependence on restrictive measures like the disagreement coefficient from the leading term of the complexity bound.

**Questions:**

see weaknesses part

**Ethical Concerns:**

["NO or VERY MINOR ethics concerns only"]

**Final Justification:**

The authors answered my questions in detail, so I am keeping my original score.

**Limitations:**

see weaknesses part

**Quality:**

3

**Strengths And Weaknesses:**

# Strengths

- The paper's main result -- that every concept class benefits from active learning in the agnostic setting -- is a significant achievement. It fundamentally advances the field by proving that the improvements over passive learning are not confined to restricted cases but are a general property. This work resolves a long-standing open problem by closing the gap between the upper and lower bounds.
- The proposed AVID algorithm introduces an innovative and powerful principle of "adaptive localized variance isolation". The algorithm intelligently partitions the instance space to isolate high-variance "challenging" regions, allocates a disproportionately large number of queries there, and uses more standard disagreement-based methods on the remaining "easier" parts of the space. This represents a new and insightful approach to designing active learning algorithms.
- This paper presents a great introduction that positions the work well. The authors begin by providing a comprehensive historical overview of the problem, tracing its evolution from early foundational studies to contemporary developments, which allows readers to understand how the field has progressed over time. They then systematically identify and analyze the key limitations of prior work, clearly articulating gaps in methodology, scope, or understanding that previous researchers have left unaddressed.

# Weaknesses

Overall, I am satisfied with this paper. However, I believe the clarity of the algorithmic description could be improved. The algorithm is difficult to follow because key components lack clear definitions and intuitive explanations, and the quantities used in equations need better conceptual background. For instance, the quantities in Equations (1), (2), and (3) are confusing without adequate conceptual background. Providing simpler, more intuitive explanations of these essential components would greatly improve readers' understanding of the algorithm's design and functionality.
Other weaknesses include:

- Improper Learning: The algorithm returns predictors that may not belong to the original concept class C. While the authors acknowledge this limitation, the question of whether proper learners can achieve the same bounds remains open.

- Computational Complexity: The paper focuses solely on information-theoretic query complexity without addressing computational efficiency. The algorithm requires solving constrained optimization problems that may be intractable for many concept classes.

These limitations are discussed by the authors, and the current results are strong enough. Therefore, I would like to vote for acceptance of this paper.

---

> ### Author Rebuttal · Authors · 2025-07-31
>
> We thank you for your efforts in taking the time to read and review our paper.  We are very happy to read your positive and supportive feedback, regarding the significance, innovation, and presentation of this work.
>
> We answer specific questions raised in the review:
>
> >Clarity:
>
> We truly appreciate your feedback regarding mindfulness of the conceptual background needed to understand the various quantities.  We are striving to make this work readable by a broad audience, and will take your comments to heart when making use of the additional page allowed in the camera-ready version.
>
> >Proper learning:
>
> We find it an important question to determine whether the algorithm can be made proper (and we mention this in appendix G).  We have thought deeply about this question, but could not reach a conclusion one way or the other (yet).  While we can show the return case in Step 9 can return any element of $V_N$, the other return case, in Step 4 (the early stopping case) is quite a bit more tricky, as the returned improper hypothesis is actually better than the best-in-class concept (Lemma 17); it is unclear how to replace this improper hypothesis with a concept in the class, since at that stage we do not have a good enough resolution in $\epsilon_k$ to identify a good concept in the class, so a proper learner simply cannot have such an early stopping case.  We intend to continue exploring this question in future work.
>
> >Computational complexity:
>
> The focus of this work is in developing new algorithmic principles for active learning, and identifying the optimal first-order query complexity for all concept classes.  This was a challenging problem, which took 20 years to solve.  Having resolved this question, a further step will be to develop practical methods inspired by the principles we have identified.  This follows a familiar pattern in statistical learning theory.  Adopting the abstract perspective of working with general VC classes helps to identify the essential algorithmic principles.  Then in subsequent work, focusing on more specialized scenarios, those algorithmic principles may inspire practical efficient methods.  In appendix G, we discuss one possible route, involving relaxations of the non-convex optimization problems with convex optimization problems via appropriate uses of convex surrogate loss functions.  Such practical modifications have been studied theoretically for both passive learning (Bartlett, Jordan, & McAuliffe 2006) and disagreement-based active learning (Hanneke 2014, Hanneke & Yang 2019), and in future work we plan to explore whether there are appropriate variations of the AVID algorithm from the present work which make use of surrogate losses to obtain a computationally efficient method (preserving the sharp query complexity guarantees, under standard assumptions regarding alignment of the surrogate loss minimizer and the 0-1 loss minimizer, as adopted in the aforementioned prior works on surrogate losses).  This approach is made more challenging in the technique developed in this work, due to the use of improper learners (see above), and even more-so due to the maximization in Steps 5 and 6.  One speculative strategy we are currently exploring is that it might be possible to approximate this maximization by solving two minimization problems with different weightings of the data to "push apart" the two minimizers (i.e., in the training set of the second function, we could weight higher the examples where the first function is wrong, to force the two functions to have diverse error regions, which is the main property we require of them); if this can be made to work, it would require significant changes to parts of the proof, but we are currently trying to work with this idea as a follow-up work.

---

### Official Review · Reviewer_WXbZ · 2025-07-02

**Clarity:** 3
**Significance:** 3
**Originality:** 3
**Rating:** 4
**Confidence:** 2

**Summary:**

This paper presents a novel agnostic active learning algorithm that aims to achieve a significantly lower query complexity than passive learning across various concept classes, without making strong assumptions about the data distribution. The core contribution is a new framework for active learning that adapts to the local variance of disagreements within the hypothesis space, allowing it to efficiently identify high-information examples to query. This approach provides improved theoretical guarantees on label complexity, often achieving exponential speedups over passive learning in settings where prior methods either offered limited gains or required specific noise assumptions. The paper details the algorithm's mechanics, provides rigorous proofs for its performance, and demonstrates its advantages over existing disagreement-based active learning methods, particularly in terms of consistency and sample complexity in the agnostic setting.

**Questions:**

- How do the theoretical benefits of agnostic active learning extend to and provide insights for common deep learning practices, particularly those involving highly expressive, over-parameterized models like deep neural networks that often achieve very low training error?
- Are there plans for, or have preliminary results shown, empirical evaluations of the proposed algorithm?

**Ethical Concerns:**

["NO or VERY MINOR ethics concerns only"]

**Final Justification:**

The clarifications on both over-parameterized models and practical considerations have fully addressed my concerns. Indeed, learning tasks can be partitioned into the realizable case, where the target labeling function is expressible within the model family and some classifier can achieve zero error, and the non-realizable case, where no member of the family can perfectly match the distribution. The latter case is important, even in the over-parameterized regime of deep learning. Overall, I thought that the main claim of the submission is significant, and I therefore maintain my positive assessment of the paper.

**Limitations:**

yes

**Paper Formatting Concerns:**

N/A.

**Quality:**

3

**Strengths And Weaknesses:**

Strengths:
- The paper's most significant contribution is the demonstration and proof that, for any concept class and without any distributional assumptions, the query complexity of active learning can always offer improvements over the sample complexity of passive learning in the challenging non-realizable case. The proposed algorithm serves as the concrete, constructive proof of this principle, showing how such an improvement is universally achievable.
- The paper effectively positions its contributions in relation to existing literature, highlighting the limitations of previous active learning approaches and how its new framework addresses them.

Weaknesses:
- My major concern is whether the setting of agnostic learning is so interesting in deep learning. While agnostic learning focuses on scenarios where the true data distribution is "far-from-realizable" by the hypothesis class, practitioners commonly employ heavily over-parameterized neural networks that are capable of achieving zero training error (or near-zero error). This prevalent practice suggests that the deep learning regime often operates closer to, or even within, a realizable setting.
- The paper lacks empirical results or simulations. While the theoretical guarantees are strong, a lack of empirical validation could be a weakness.

---

> ### Author Rebuttal · Authors · 2025-07-31
>
> We thank you for your efforts in taking the time to read and review our paper.  We appreciate your positive remarks regarding the significance of the advances in this work, and the quality of presentation.
>
> We address specific comments raised in the review below.
>
> >Overparameterized models:
>
> The extension of the theory of active learning to overparameterized models is a very interesting and important topic for future work.  It is unclear whether we should really interpret the overparameterized regime as the "realizable case", since it does not necessarily admit zero population error rate (without making strong assumptions), but rather the model is complex enough (growing with sample size) to fit the training data well. As such, the techniques in this work may still be relevant even in this case.  We intend to explore in future work whether the AVID principle has implications for more specialized scenarios, including overparameterized and nonparametric learning.  For instance, as one speculative idea, it may be the case that the technique of isolating the "high variance" regions where two low-error classifiers disagree can instead be identified by training two predictors on independent data sets and isolating the region where these predictors disagree, and training another predictor on additional data queried in this region.  This is a possibility we intend to explore in future work.
>
> We also remark that even overparameterized *passive* learning has only been understood theoretically in highly specialized scenarios, under distributional assumptions.  In contrast, in this work we aim for a broad general theory, applicable to any concept class, with no distribution restrictions, so the aims are quite different.  Nonetheless, there have been some works on active learning with neural networks in overparameterized and nonparametric regimes (e.g., by Nowak and others), and a factor of disagreement coefficient appears in some of those bounds (e.g., Zhu & Nowak 2022), so it is natural to wonder whether the AVID principle can again help reduce the query complexity by the factor of disagreement coefficient (as it does in the present work), and we intend to follow up this work by investigating this question.
>
> > Practicality / empirical evaluation:
>
> The focus of this work is in developing new algorithmic principles for active learning, and identifying the optimal first-order query complexity for all concept classes.  This was a challenging problem, which took 20 years to solve.  Having resolved this question, a further step will be to develop practical methods inspired by the principles we have identified.  This follows a familiar pattern in statistical learning theory.  Initial work identifying algorithmic principles is often impractical at first, but may later evolve into practical learning methods.  For instance, the work of Vapnik and Chervonenkis in the 60s and 70s on passive learning identified the principle of empirical risk minimization and the fundamental role of the VC dimension in characterizing the sample complexity, but does not have a direct practical implication due to the 0-1 loss being computationally challenging to optimize; only later, when combined with the idea of surrogate losses to relax the non-convex 0-1 loss to a convex loss, did we arrive at practical passive learning methods (for which we may recover theoretical guarantees under assumptions relating the minimizer of the surrogate loss to that of the 0-1 loss, based on the theory of Bartlett, Jordan, & McAuliffe 2006).  A similar line has been followed in the active learning literature.  While the initial methods for disagreement-based agnostic active learning (e.g., the A^2 algorithm of Balcan, Beygelzimer, & Langford 06) involved implicit direct optimizations of the 0-1 loss (making them infeasible to run) and provided a coarse theoretical understanding of the query complexity (via the disagreement coefficient analysis of Hanneke 07), only later were these methods made practical by identifying appropriate ways to replace the optimization problems implicit in the technique with convex problems (e.g., by Beygelzimer, Dasgupta, & Langford 2009, Hanneke 2014, Hanneke & Yang 2019), recovering the theoretical guarantees under assumptions relating the minimizer under the convex surrogate loss to that of the 0-1 loss.  Following this line, a natural next step for the present line of work (mentioned in appendix G) is to investigate ways to replace the various optimization problems involved in the AVID algorithm with appropriate convex relaxations, while recovering the sharp theoretical guarantees of this work under appropriate assumptions on the surrogate loss.  This is a nontrivial extension (as was the case for disagreement-based methods, as discussed in section 6.6 of Hanneke 2014), and will require significant further research.  We have been thinking deeply about this question, and aim to develop such an extension in future work (more discussion of this in our reply to reviewer qnfZ).

---

> > ### Comment · Reviewer_WXbZ · 2025-08-05
> >
> > I would like to thank the authors for their detailed and thoughtful rebuttal. The clarifications on both over-parameterized models and practical considerations have fully addressed my concerns. Indeed, learning tasks can be partitioned into the realizable case, where the target labeling function is expressible within the model family and some classifier can achieve zero error, and the non-realizable case, where no member of the family can perfectly match the distribution. The latter case is important, even in the over-parameterized regime of deep learning. Overall, I thought that the main claim of the submission is significant, and I therefore maintain my positive assessment of the paper.

---

### Author Response · Authors · 2025-08-09
**Some final remarks**

As we reach the end of the interaction period, we want to thank you again for your efforts in reviewing this paper, and also provide some final remarks.  We appreciate your supportive comments in the reviews and replies.  As you know, even among positive reviews, distinctions between ratings (4,5,6) may influence the level of visibility at the conference (poster, spotlight, oral).  So, if you find that our responses adequately address your questions and critiques, and you believe the merits of this work warrant it, we would be grateful if you might consider increasing your rating.

Just to highlight a few pertinent points (closing arguments):

- We prove that active learning can always improve over passive learning in the non-realizable case (for every concept class) and characterize the optimal query complexity.

- This was a challenging well-known formal open problem which stood for 20 years despite the efforts of many top researchers and hundreds of works on the theory of agnostic active learning, and we have finally solved it in this paper.

- One sign of a good theory paper is that it closes an old question while raising 10 new ones.

- Among the new questions raised by this work (and discussed in reviews), an important one is whether these improvements can be realized by an efficient practical method.  The new algorithmic principles developed in this work lay the groundwork for approaching such questions.

- As is generally the case in theory research, these new questions will take years of study and numerous follow-up works to resolve.  The fact that this paper will stimulate such an array of follow-up research is itself a positive contribution.

- We are really quite excited by the advances made in this work and believe the contributions will resonate with a broad segment of the NeurIPS community.

We thank you for your time and consideration.

---

### Decision · Program_Chairs · 2025-09-17

**Decision:**

Accept (oral)

**Comment:**

This paper introduces an insightful algorithm that closes a long-standing and central theoretical gap that has been under study for the past 20 years within active learning theory. In particular, the proposed and analyzed algorithm closes the gap (in most problem parameter regimes) between upper and lower bounds for agnostic active learning, for any concept class, any (unknown) data distribution, in terms of the VC dimension and best-in-class error. The reviewers appreciated the clear description of the history of the problem, its study, and its solution by the proven result. The reviewers noted that the algorithm was polynomial-time in the size of the concept class (if finite), which is intractable for most applications, though quite standard for initial results in learning theory. Given the significance of the result, I enthusiastically recommend acceptance.